palaeontology

Middle Miocene, extant genus, *Berardius kobayashii* sp. nov., western North Pacific

**Author for correspondence:**
Ayako Kawatani
e-mail: kawatani@geol.tsukuba.ac.jp

# The oldest fossil record of the extant genus *Berardius* (Odontoceti, Ziphiidae) from the Middle to Late Miocene boundary of the western North Pacific

Ayako Kawatani[1] and Naoki Kohno[1,2]

[1]Graduate School of Life and Environmental Sciences, University of Tsukuba, 1-1-1, Tennoudai, Tsukuba, Ibaraki 305-8577, Japan
[2]National Museum of Nature and Science, 4-1-1, Amakubo, Tsukuba, Ibaraki 305-0005, Japan

AK, 0000-0002-0792-6813

A new species of a beaked whale that belongs to the extant genus *Berardius* is described from the Middle to Late Miocene boundary age Tsurushi Formation (*ca* 12.3–11.5 Ma) on the Sado Island, Niigata Prefecture, Japan. The new species, *Berardius kobayashii* sp. nov. represents the oldest record of this genus and provides a minimum age for the emergence of this extant genus. *Berardius kobayashii* sp. nov. has the following generic characters: the ratio between the width of the premaxillary crests and the width of the premaxillary sac fossae is 1.0–1.25, nodular frontals make isolated protuberance on the posterior part of the vertex. Among the species within the genus, *B. kobayashii* sp. nov. shares a unique character with *B. minimus*: the apices of the left and right hamular processes of the pterygoids contact medially, forming together a posteriorly directed medial point. In addition, *B. kobayashii* sp. nov. displays a unique combination of the following characters: it is extremely small in size, and the nasals are short, the ratio between the length of the medial suture of nasals on the vertex and the maximum width of nasals is less than 0.4. *Berardius kobayashii* sp. nov. fills the gap between the origin of the genus and later diversifications of the extant species. This discovery is also key to elucidate the process of the emergence and dispersal of the genus during the Middle to Late Miocene. Based on the distributional patterns of the fossil and extant species of the genus, the western North Pacific including the Sea of Japan may have been one of the areas for the evolution and radiation of this genus at the time before 11 Ma.

# 1. Introduction

The family Ziphiidae is a clade among Odontoceti (Cetartiodactyla, Cetacea) represented by 23 extant species [1,2] and almost the same number of extinct species [3,4]. Although extant ziphiids have a cosmopolitan distribution, they are among the most mysterious groups of whales because of their deep-diving behaviour. Ziphiidae are thought to have originated and diversified in the Southern Hemisphere, and the ancestors of these deep-diving whales should have initially lived in epipelagic and/or neritic oceans [5]; however, their adaptation to and diversification in the deep sea remain uncertain.

It is extremely difficult to know what happened at the palaeobiogeographic centre of the evolution for the cetaceans, because most of their fossils, with some exceptions, are known in general from shallow marine sediments on and around land areas. Accordingly, it is important to investigate fossils of cetaceans from the pelagic or deep-sea sediments to elucidate their evolutionary histories more precisely.

In the case of the Sado Island in the western margin of the North Pacific (figure 1a), the deep-sea sediments are distributed on and around this island [6,7] as a result of relatively strong tectonic movements around this area after the Late Miocene [8], and thus, it is easy to observe such sediments and obtain fossils of the deep sea and/or pelagic organisms from there. Based on foraminiferal, radiolarian and diatom fossils, the Lower to Upper Miocene formations distributed on the Sado Island are thought to be deposited just after the Sea of Japan expanded and became a deep sea [7,9,10]. A lot of marine mammal fossils (i.e. cetaceans, pinnipeds and desmostylians) have also been obtained from the same formations [11]. In particular, many whale fossils including several ziphiids [12–14] have been found from the Middle to Upper Miocene Tsurushi Formation distributed on the Kosado mountain range of the Sado Island (figure 1b). Thus, the Sado Island is suitable for examining the evolution of pelagic and/or deep-diving whales during the periods spanning the latest Early to earliest Late Miocene. However, studies of fossil whales from the Sado Island have been stagnating despite the accumulation of fossils that followed the preliminary studies by Sado Research Group for Marine Mammalian Fossils [11] and Takahashi et al. [14], and only a few additional studies have been published on the marine mammal fossils from the Tsurushi Formation until now [15].

In this study, we describe one such fossil whale, based on an incomplete skull with ear bones and a partial mandible, from the Middle to Upper Miocene boundary Tsurushi Formation on the Sado Island, which was initially reported by Takahashi et al. [14] as an indeterminate species 'for the time being' in the genus Berardius, and we identify it specifically to elucidate its phylogenetic relationships and the related phylogeographic diversifications of this genus.

# 2. Geological setting

The Sado Island, which is located 35 km northwest of Niigata Prefecture on the Honshu Island, is the largest island in the Sea of Japan (figure 1a). This island is subdivided into the northern half, the Osado mountain range and the southern half, the Kosado mountain range and the Kuninaka plain field that expands between them (figure 1b). The Sado Island mostly consists of Cenozoic formations. The Oligocene to Lower Miocene Green Tuff layer unconformably overlies the basement rocks consisting of Palaeozoic and Mesozoic granite.

The Neogene stratigraphy of the Sado Island has been studied by several researchers (e.g. [16] for the Osado mountain range; [17,18] for the Kosado mountain range). According to Yanagisawa [19], the Neogene stratigraphy of the Osado mountain range is divided into the Orito, Hanyugawa, Tsurushi, Nakayama, Nozaka, Kawachi, Kaidate and Shichiba formations in the ascending order (figure 2a). Recently, the Tsurushi Formation was integrated into the overlying Nakayama Formation as its basal layer on the basis of the revised lithostratigraphy and biochronology of diatom fossils [10]. On the other hand, according to the Ogi Collaborative Research Group [24], the Neogene stratigraphy of the Kosado mountain range is divided into the Aikawa, Sanze, Kyozukayama, Orito, Tsurushi, Notayama and Yamadagawa formations in ascending order (figure 2b). At present, the correlation between the stratigraphic sequences of the Osado and Kosado mountain ranges are still unclear because of some uncertainties for the stratigraphic sequences in the Kosado mountain range [10]. Accordingly, we follow the temporary moratorium by Yanagisawa & Watanabe ([10]; figure 2) and retain the name 'Tsurushi Formation' for the hard siliceous mudstone layer on the Kosado mountain range until the comparison between the stratigraphic sequences of the Osado and Kosado mountain ranges is completed. In fact, until now whale fossils have been discovered only from the Tsurushi Formation on the Kosado mountain range.

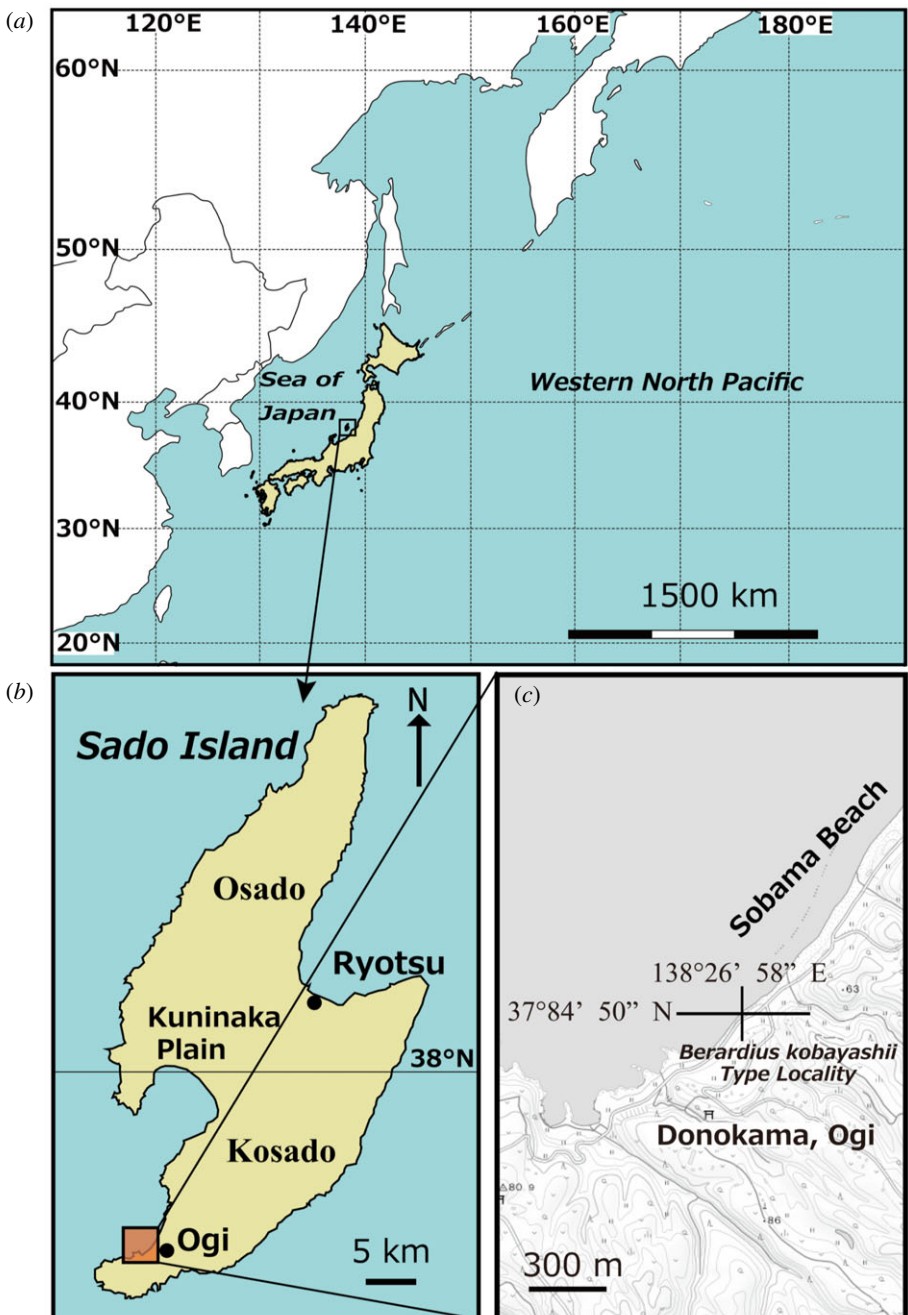

**Figure 1.** Index maps of the Pacific Northwest: (*a*) location of the Japanese Islands, showing Sado Island; (*b*) map of Sado Island, showing some major place names mentioned in text, and location of the enlarged area including the type locality for *Berardius kobayashii* sp. nov.; (*c*) detail of Sobama beach in Donokama, Ogi Town, showing the exact locality for the holotype of *Berardius kobayashii* sp. nov., SCM 5530-6. Based on the GIS map (https://maps.gsi.go.jp/#15/37.844139/138.272681/&base=std&ls= std&disp=1&vs=c0j0h0k0l0u0t0z0r0s0m0f1) of the Geospatial Information Authority of Japan.

The Aikawa, Sanze and Kyozukayama formations, which are the stratigraphically lower units for the Neogene of the Kosado mountain range, are terrestrial formations and mainly consist of andesites and dacite lavas [17,24,25]. The Lower Miocene Sanze and Kyozukayama formations are unconformably overlain by the Orito Formation [24]. The Orito Formation is composed of shallow marine sediments, including conglomerates, sandstone and mudstone [10,24]. The Orito Formation yielded fossils of large foraminifers, echinoderms, molluscs and marine mammals [9,26], and its age is estimated to be late Early Miocene, *ca* 17.5–17.3 Ma, based on marine diatom zonations [10,19]. The Tsurushi Formation is composed of deep-sea sediments overlying the Orito Formation, and it consists mainly of hard siliceous mudstone. The Notayama and Yamadagawa formations consist of diatomaceous mudstones. According to the diatom biostratigraphy, these two formations are Late Miocene in age [19].

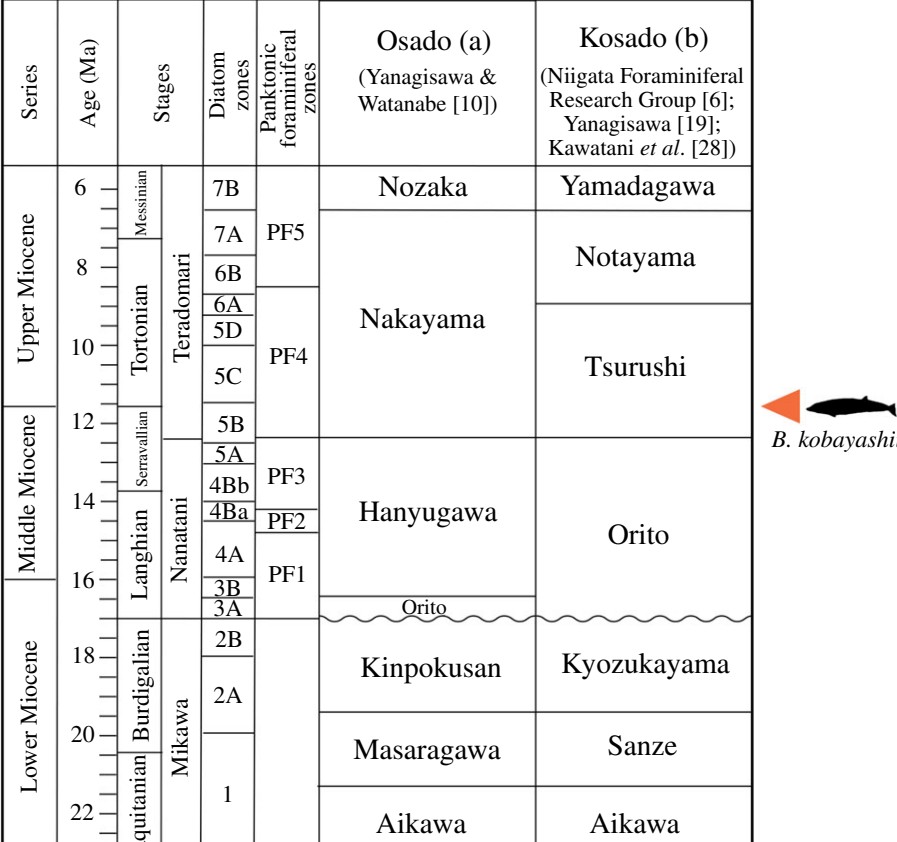

**Figure 2.** Correlation of formations distributed on the Osado (*a*) and Kosado (*b*) mountain ranges containing whale fossil horizon with relevant systems of chronology. Wave lines indicate unconformities. After [6,10,19]. Diatom zones: [10,20,21]; Planktonic foraminiferal zones: [22,23].

The Middle to Upper Miocene Tsurushi Formation was originally defined by Watanabe [25] based on the hard siliceous mudstone exposed at Tsurushi in Sawada Town on the Osado mountain range. The same or similar rock facies has been distributed also on the Kosado mountain range and was recognized as the same formation [27]. On the Kosado mountain range, this formation is composed of hard siliceous mudstone and contemporaneous basalt (i.e. Ogi Basalt Member) distributed regionally on the Ogi Peninsula [19]. The geologic age of the Ogi Basalt Member is estimated to 14.1–11.7 Ma [10]. Watanabe [7] identified foraminiferans from the Tsurushi Formation on the Osado mountain range, which indicate the PF4 Zone of Maiya [22], and correspond to a 12.3–11.5 Ma interval [10,19]. Kawatani *et al.* [28] also confirmed the age of the Tsurushi Formation on the Kosado mountain range based on radiolarians, yielding a 15.3–9.1 Ma interval. Based on this chronological information, the geologic age of the Tsurushi Formation is estimated to approximately 12.3–11.5 Ma (Late Serravallian to Early Tortonian, latest Middle to earliest Late Miocene). Studies of foraminiferal fossils have suggested that the Tsurushi Formation was deposited in a deep-sea environment [7,9].

As mentioned above, various megafossils have been reported from the Tsurushi Formation, for example, molluscs, fish, a sea turtle, birds and marine mammals [9,11,29]. In particular, a lot of marine mammal fossils including whales, pinnipeds and desmostylians have been found in calcareous nodules from the Tsurushi Formation on the Kosado mountain range, and these fossils were briefly reported and summarized by the Sado Research Group for Marine Mammalian Fossils [11], Tazaki *et al.* [13], Horikawa *et al.* [12], Takahashi *et al.* [14] and Barnes & Hirota [15].

## 3. Materials and methods

The specimen described here was initially reported by Takahashi *et al.* [14]. At that time, the specimen was temporarily deposited at Niigata University; now, it is permanently stored at the Sado Museum, Sado City, Niigata Prefecture, Japan, where it bears the catalogue number SCM 5530-6. This specimen was enclosed in a calcareous hard mudstone matrix. Accordingly, it was prepared by acid etching,

using formic acid diluted to a concentration of about 4.5–5.0%, which allowed the extraction of fragile bone elements of the skull, partial mandible and ear bones from the matrix. However, the periotic is fused to the skull by its fan-shaped posterior process, and the dorsal surface of the periotic is entirely invisible from the ventral aspect. So we reconstructed the three-dimensional image of the periotic using the micro-computed tomographic (μCT) scanner Microfocus CT, TXS320-ACTIS, at the National Museum of Nature and Science, Tokyo, Japan.

We then compared the specimen osteologically and taxonomically using both extinct and extant species of ziphiids stored at the institutions listed below and data from the literature. Definitions of measurements and morphological terms follow Mead & Fordyce [30] and Kasuya [31]. We performed a cladistic analysis to clarify the phylogenetic relationship of the new fossil described here to the previously named ziphiids using the characters and data matrix prepared by Bianucci et al. [3]; the matrix was slightly modified with the addition of the extant *Berardius bairdii*, the recently described extant species *Berardius minimus* [2] and the new species described here. Character and character states were managed using Mesquite 3.61 [32]. The phylogenetic analysis based on 31 extinct and extant ziphiid species with six outgroup taxa as operational taxonomic units (OTUs) and 51 morphological characters was performed with TNT 1.5 using the 'New Technology Search' tasked to find minimum length trees 1000 times [33]. As mentioned by Bianucci et al. [3], our preliminary analyses also led to poorly resolved trees. So we down-weighted homoplastic characters in Bianucci et al. [3] using the default value of 3 for the constant $K$ of the method formulated by Goloboff [34]. Multi-state characters were treated as ordered for 18 characters (i.e. Chars. 1, 3, 7, 8, 9, 11, 13, 14, 15, 16, 22, 23, 30, 34, 36, 39, 47 and 51) and unordered for six characters (i.e. Chars. 2, 4, 10, 28, 34 and 45), also following Bianucci et al. [3].

Institutional abbreviations are the following: IRSNB, Institut royal des Sciences naturelles de Belgique, Brussels, Belgium; MNHN: Muséum National d'Histoire Naturelle, Paris, France; NMNS-PV, Department of Geology and Palaeontology, National Museum of Nature and Science, Tsukuba, Japan; NSMT-M, Department of Zoology, National Museum of Nature and Science, Tsukuba, Japan; SAM, Iziko South African Museum, Cape Town, South Africa; SCM, Sado Museum, Sado City, Niigata, Japan.

# 4. Systematics

CETARTIODACTYLA Montgelard, Catzefis and Douzery, 1997
CETACEA Brisson, 1762
ODONTOCETI Flower, 1867
ZIPHIIDAE Gray, 1850

BERARDIINAE Moore, 1968

*Emended diagnosis of subfamily:* Members of the subfamily Berardiinae consisting of the genera *Microberardius*, *Archaeoziphius* and *Berardius* differ from all other ziphiids by the following unique combination of characters: in lateral view, the ascending process of the premaxilla is slightly concave (Char. 7, reversal); weak constriction of the ascending process of the right premaxilla, with a ratio greater than 0.8 between the minimal width of the right ascending process of the premaxilla and the width of the right premaxillary crest (Char. 8, reversal); no anteromedial excavation of the dorsal surface of the nasal (Char. 14, reversal); the nodular protuberance on the posterior part of the vertex is formed by the interparietal and/or the frontals (Char. 17, 0 to greater than 1 and 2); and the anteromedial margin of the supraoccipital is distinctly lower than the dorsal margin of the vertex (Char. 18).

Genus *Berardius* Duvernoy, 1851

*Emended diagnosis of genus:* The genus differs from *Microberardius* with the following unique character: the ratio between the width of the premaxillary crests and the width of the premaxillary sac fossae is 1.0–1.25 (Char. 11). The following characters shared with *Berardius* species, but uncertain in *Microberardius* and *Archaeoziphius* because of the lack of such a portion, are potentially diagnostic for the genus: anterior spine of the tympanic with a more or less rectilinear anterior margin (Char. 22); the posteroventral corner of the sigmoid process of the tympanic that is posteriorly projected in lateral view (Char. 23); the dorsal margin of the involucrum of the tympanic cut by an indentation that is present and visible in medial and/or dorsal view (Char. 24). Although not included in our phylogenetic analysis, the genus differs from *Archaeoziphius* with the following unique combination of

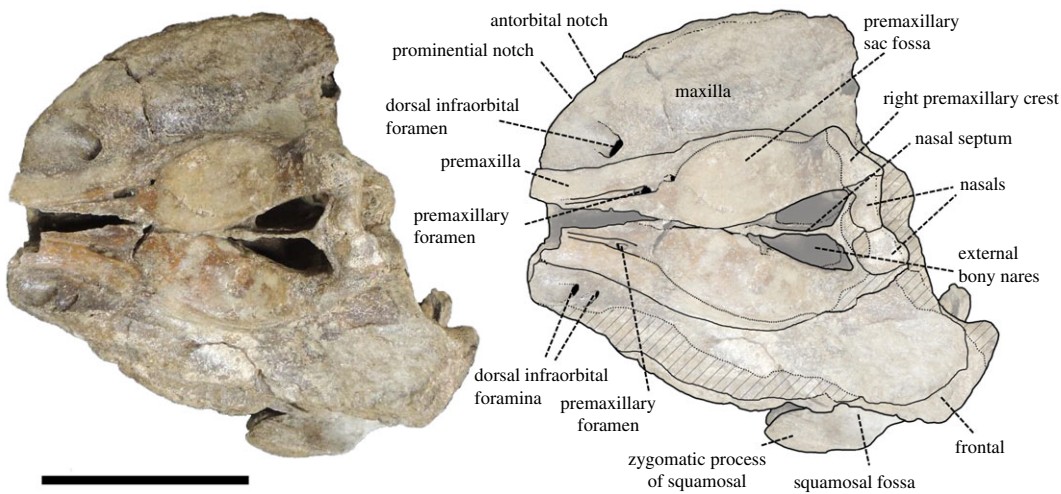

**Figure 3.** Dorsal view of the cranium of *Berardius kobayashii* sp. nov., holotype, SCM 5530-6. (Scale bar, 10 cm).

characters: anterodorsally well-developed robust zygomatic process of the squamosal, dorsoventrally long and laterally wide glenoid fossa, anteroventrally projected large postglenoid process and anteroposteriorly well-developed post-tympanic process.

*Berardius kobayashii* sp. nov.

(Skull: figures 3–8; Ear bone: figures 9–11)
URN: lsid:zoobank.org:act: 317C1452-AE2E-4A08-A473-3B507B452D2C
*Diagnosis of species:* The species differs from all other *Berardius* species (i.e. *B. arnuxii*, *B. bairdii* and *B. minimus*) by the following unique combination of characters: short nasals, with a ratio between the length of the medial suture of nasals on the vertex and the maximum width of nasals less than 0.4 (Char. 13, reversal). Although it is potentially a shared primitive trait with *Archaeoziphius microglenoideus* and *Microberardius africanus*, an extremely small size compared with the three other species (60–70% smaller than *B. bairdii* and *B. arnuxii*, 50% smaller than *B. minimus*).

*Holotype:* SCM 5530-6, incomplete skull including the left tympanic bulla and its corresponding periotic, lacking the rostral portion; the posterior fourth of the horizontal ramus of the left dentary with the coronoid process and mandibular condyle. Collected by Kiyoei Kaneko in or before 1969.

*Type Locality:* Sobama Beach, Ogi Town, Sado City (Sado Island), Niigata Prefecture, Japan. Geographic coordinates: 37°84′50″ N, 138°26′58″ E.

*Formation and age of holotype:* SCM 5530-6 was collected from the latest Middle to earliest Late Miocene age Tsurushi Formation. As discussed above, the geologic age of the Tsurushi Formation is estimated to be approximately 12.3–11.5 Ma (Late Serravallian to Early Tortonian).

*Etymology:* The species is named in honour of the late Dr Iwao Kobayashi, a professor emeritus at Niigata University, for his longstanding contributions to the geology and palaeontology of the northern Fossa Magna region including Niigata Prefecture, and in gratitude for his permission to re-study the marine mammal fossils collected by him and his colleagues from the Sado Island and encouragement to both of us throughout this study.

### Description

*Cranium (general morphology):* The cranium is relatively small (table 1) and slightly asymmetric (figure 3). The ratio between the height and width of the skull is 1 : 1.19. The rostral and occipital parts are mostly broken and missing. A small part of the left mandible was originally articulated to the cranium. The vertex is moderately elevated and slightly skewed towards the left, with its posteriormost portion broken. The temporal fossa has an isosceles triangle outline with long sides that converge anteriorly in the lateral view, and the posterior margin is weakly rounded. The temporal fossa is shallow and its lateral surface is transversely convex. The sutures of the skull are still visible and not fused, but these are tightly closed, indicating that the individual is an adult.

*Premaxilla:* The premaxilla is well preserved posterior to the antorbital notch, and its dorsal surface is almost flat in the lateral view. The premaxillary crest is present on the vertex. The medial margins of the left and right premaxillae are very close to each other at the level just anterior to the bony nares. Contact

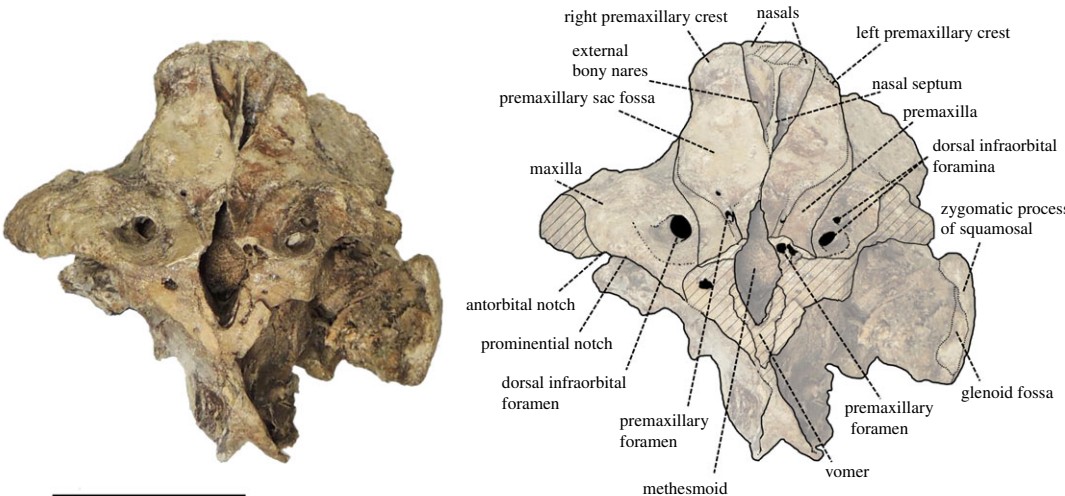

**Figure 4.** Lateral view of the cranium of *Berardius kobayashii* sp. nov., holotype, SCM 5530-6. (*a,b*) Left side and (*c,d*) right side. (Scale bar, 10 cm).

**Figure 5.** Anterior view of the cranium of *Berardius kobayashii* sp. nov., holotype, SCM 5530-6. (Scale bar, 10 cm).

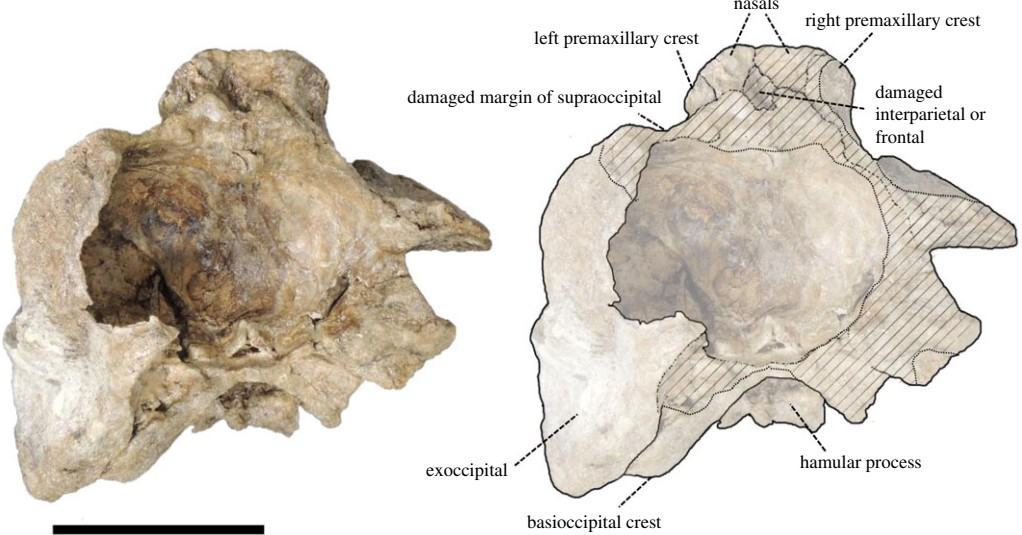

**Figure 6.** Posterior view of the cranium of *Berardius kobayashii* sp. nov., holotype, SCM 5530-6. (Scale bar, 10 cm).

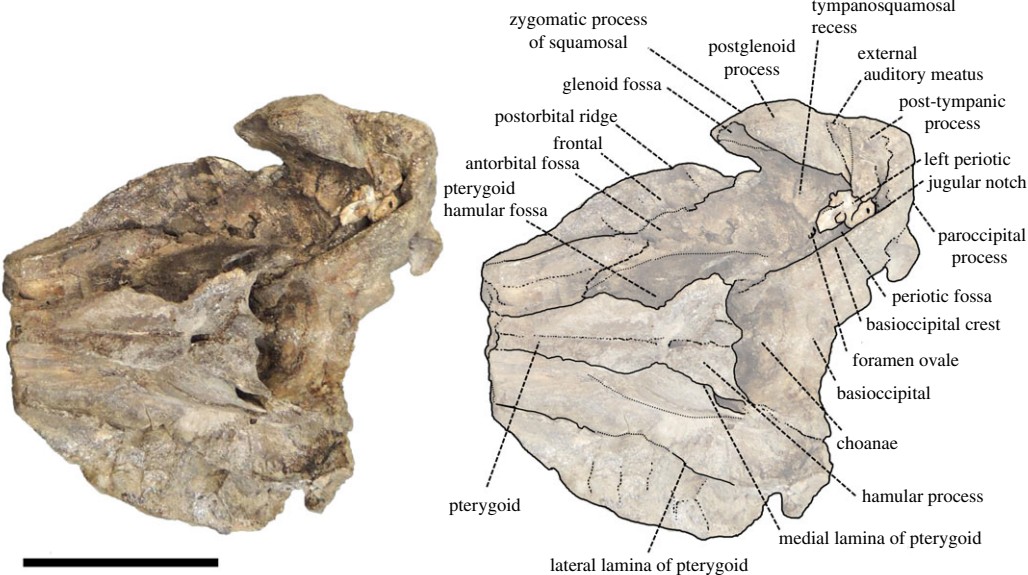

**Figure 7.** Ventral view of the cranium of *Berardius kobayashii* sp. nov., holotype, SCM 5530-6. (Scale bar, 10 cm).

between the nasal and the premaxillary crest is extended along almost the whole length of the nasal. The premaxillary sac fossae are weakly asymmetrical, and the right one is wider than the left one. It does not overhang the maxillae laterally. The premaxillary sac fossa is almost flat and not deeply excavated. The ascending process of the premaxilla gradually rises towards the vertex. The angle between the dorsal surface of the rostrum and the ascending process of the premaxilla is 150°. The surface of the ascending process of the premaxilla is slightly concave in the lateral view. The ascending process of the left premaxilla is somewhat damaged, but that of the right premaxilla is well preserved, and its lateral margin reaches medially a level that is in line with the mid-length of the premaxillary crest. The vertex is gently elevated. In the anterior view, lateral expansion of the premaxillae at the level of the premaxillary sac fossae is prominent. The premaxillary crest is transversely oriented, and it is narrow in transverse width. The distance between the premaxillary crests is wide. The anterior margin of the bony nares is V-shaped and the overall outline of the bony nares is triangular. The anteroposterior diameter (67 mm) of the bony nares is greater than its transverse diameter (39 mm). The small premaxillary foramina is located roughly at the level of the antorbital notch and slightly

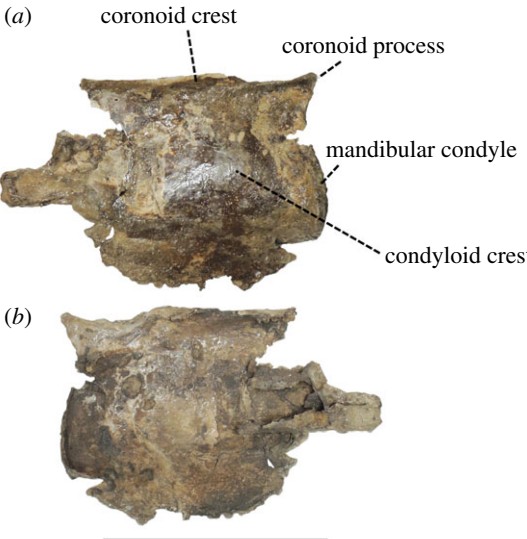

**Figure 8.** Posterior fragment of the left mandible of *Berardius kobayashii* sp. nov., holotype, SCM 5530-6. (*a*) Lateral view and (*b*) medial view. (Scale bar, 10 cm).

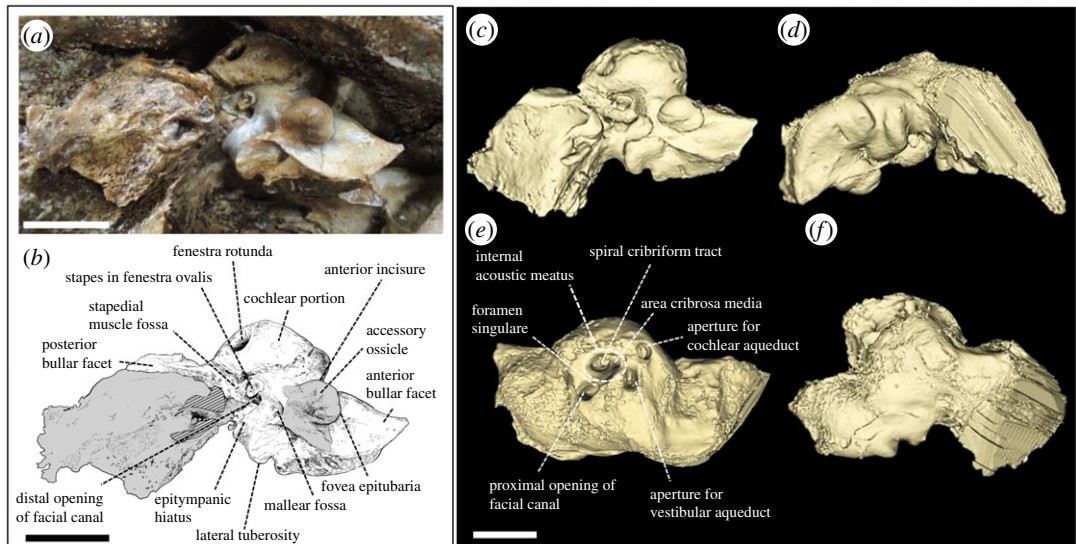

**Figure 9.** Left periotic of *Berardius kobayashii* sp. nov., holotype, SCM 5530-6. (*a*) Ventral view; (*b*) corresponding line drawing with anatomical interpretations (grey hatches indicate parts of the tympanic bulla); (*c–f*) the reconstructed CT image from three-dimensional surface rendering, with anatomical interpretations (*c*, ventral view; *d*, lateral view; *e*, medial view; *f*, dorsal view). (Scale bar, 1 cm).

posterior to the dorsal infraorbital foramina on the maxillae. There are distinct anteromedial sulci anterior to the foramina.

*Maxilla:* The maxilla is broad, but its lateral margin is damaged on both sides posterior to the antorbital notch. In the anterior view, the rostral maxillary crest is present and moderately elevated on the maxilla. The dorsal infraorbital foramina open anterodorsally on the maxillae near the base of the rostrum, and the largest foramen is located on the right maxilla. There are two dorsal infraorbital foramina on the left maxilla with diameters of 13 and 7 mm. There is one large dorsal infraorbital foramen on the right maxilla that reaches a maximum diameter of 16 mm. The maxilla is slightly elevated posterior to these foramina. The region of the prominential notch and related structures are somewhat weathered away, but the dorsal surface of the maxilla is low and flat in that area.

*Nasals:* The nasal is short, and its dorsal outline is trapezoidal. Although the posterior margin of the right nasal is broken, it is slightly broader than the left one. The dorsal surface of the nasal is roughly flat,

(a)

(c)

(b)

(d)

(e)

(f)

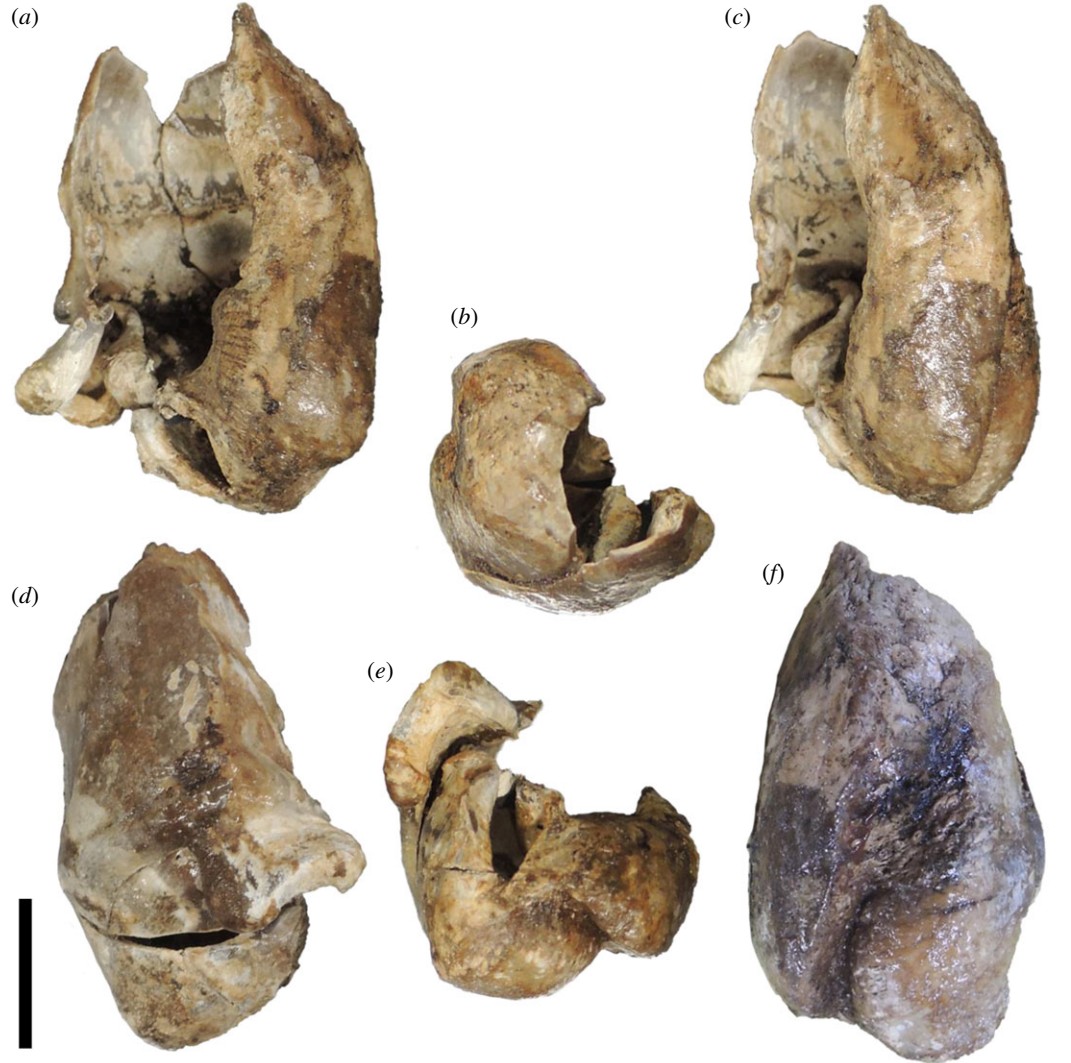

**Figure 10.** The left tympanic bulla of *Berardius kobayashii* sp. nov., holotype, SCM 5530-6. (*a*) Dorsal view, (*b*) anterior view, (*c*) medial view, (*d*) lateral view, (*e*) posterior view and (*f*) ventral view (Scale bar, 1 cm).

and there is a weak anteromedial excavation. The nasal is not included in the premaxillary crest. The posterior nasal-frontal suture is only preserved on the left nasal and is posteriorly convex.

*Frontal:* The frontal is fully covered by the maxilla in the supraorbital region and only visible as a vertical section beneath the overlapping maxilla in the orbital region and temporal fossa in the lateral view. It is dorsoventrally thick and relatively dense but not osteosclerotic. There is a slight ventral flange at the level of the preorbital process that anteriorly defines the orbit. The antorbital fossa is shallow and flares posteriorly to the slightly thickening and anterolaterally extended infratemporal crest. The posterodorsal part of the frontal around the posterior part of the vertex is broken away.

*Occipitals:* Although occipitals are almost broken away, parts of the supraoccipital, the left exoccipital and an anterior part of the basioccipital are preserved. The supraoccipital is mostly broken, but its anteromedial margin is distinctly lower than the dorsal margin of the vertex inferred from the damaged surface of the supraoccipital in the posterior view (figure 6). The left exoccipital is preserved and subsquare in outline. It is laterally narrow and ventrally projected in the posterior view (figure 6). The portion of the temporal crest extends almost vertically and makes the paroccipital process at the ventral tip of the crest. The paroccipital process is triangular in shape in the posterior view (figure 6). The anterior portion of the basioccipital is preserved. The basioccipital basin is concave, and it is posteriorly broadened in the ventral view (figure 7). The basioccipital crests are keel-like and diverged posterolaterally. The angle formed by the crests is approximately 60°.

*Pterygoid:* The pterygoid is dorsoventrally deep and excavated by the anteroposteriorly and dorsoventrally wide hamular fossa. The lateral lamina is preserved on the right pterygoid. It is thin

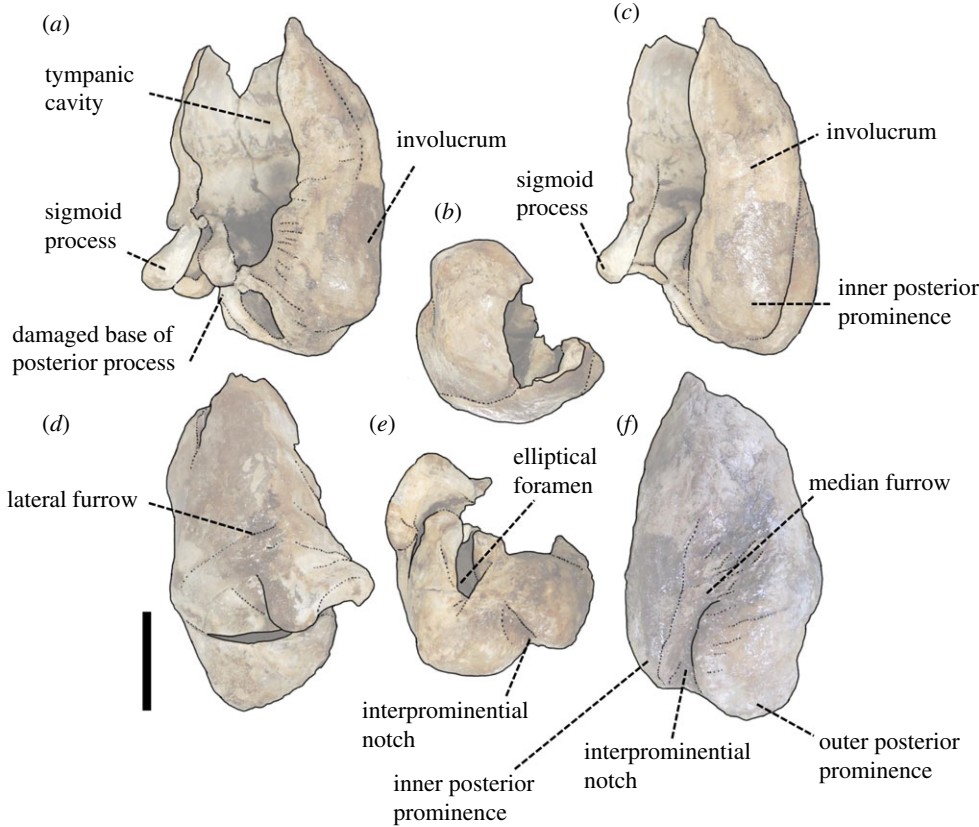

**Figure 11.** Key features of the left tympanic bulla of *Berardius kobayashii* sp. nov., holotype, SCM 5530-6. (*a*) Dorsal view, (*b*) anterior view, (*c*) medial view, (*d*) lateral view, (*e*) posterior view and (*f*) ventral view. (Scale bar, 1 cm).

**Table 1.** Measurements (in mm) of the skull of *Berardius kobayashii* sp. nov., holotype, SCM 5530-6. '*' indicates estimated transverse measurements that are half-skull measurements multiplied by two and 'e' indicates estimation.

| measurements (in mm) of the skull of *Berardius kobayashii* | |
|---|---|
| skull | |
| preserved length from the tip of broken rostrum to the hindmost margin of occipital | 236 |
| greatest postorbital width | *e219 |
| greatest bizygomatic width | *e245 |
| greatest width of bony nares | 38 |
| maximum width of premaxillary sac fossa | 87 |
| maximum width of right premaxillary sac fossa | 43 |
| maximum width of left premaxillary sac fossa | 38 |
| maximum width of nasals | 40 |
| greatest length of left temporal fossa, measured to external margin of raised suture | 87 |
| greatest width of left temporal fossa at right angles to greatest length | 61 |
| major diameter of left temporal fossa proper | 38 |
| minor diameter of left temporal fossa proper | 33 |
| distance from the foremost end of junction between nasals to the hindmost point of margin of supraoccipital crest | 33 |
| length of left orbit from the apex of preorbital process of frontal to the apex of postorbital process | 51 |
| greatest width of internal nares | 91 |
| greatest length of left pterygoid | 109 |

and low at its base. There are transverse crests on the surface of the medial lamina. The hamular fossa of the pterygoid sinus is extending anteriorly over the palatal surface of the rostrum. The apices of the left and right hamular processes are incomplete, but both processes contact medially and form together with a posteriorly directed medial point. Each process sends a posterolateral point, so the posterior margin of the joined hamular processes draws a W shape (figure 6).

*Alisphenoid:* The alisphenoid is well preserved on the right side, posterior to the pterygoid sinus and ventrally exposed as a distinct concavity, directed anterolaterally. A thin plate-like crest runs in a converging direction, which is thought to be the alisphenoid–squamosal suture, which separates this concavity posteriorly from the tympanosquamosal recess. The concavity on the alisphenoid continues posteromedially to the large foramen ovale.

*Squamosal:* The left squamosal is well preserved. The squamosal that makes the floor of the temporal fossa is anteroposteriorly long and dorsoventrally shallow, and its surface is slightly concave. The zygomatic process of the left squamosal is short anteroposteriorly compared with its dorsoventral height. It is well developed and anterodorsally tapered with a triangular outline, but its anterior margin is crescentic. The glenoid fossa is wide and shallow, but its medial part deeps in front of the obliquely extended postglenoid process. The glenoid fossa is margined medially by the tympanosquamosal recess. The surface of the posterior part of the tympanosquamosal recess is rugose just in front of the periotic including numerous pits and crests. The falciform process is located anterior to the anterior process of the periotic, and it is short and thin. The ventral margin of the postglenoid process is more dorsally located than the ventral margin of the paroccipital process of the exoccipital (figure 4).

*Mandible:* The mandibles are almost broken away, but the ascending ramus of the left mandible with the coronoid process and the mandibular condyle is partially preserved (figure 7). The posterior margin of the angular process is also broken away. The mandibular body is very thin and fragile in the portion of the lateral wall of the mandibular foramen. Its outer or lateral surface is smooth and slightly swollen laterally, forming the condyloid crest. Its inner or medial surface (or the lateral wall of the mandibular foramen) is relatively rough. The mandibular foramen is anteroposteriorly long and dorsoventrally high. The coronoid crest is well developed as an anteroposteriorly extended ridge and continues to the coronoid process at the posterodorsal end of the mandible. The ventral margin of the coronoid crest is weakly concave along the crest. The mandibular condyle protrudes posteriorly and is located almost at the same anteroposterior level as the end of the coronoid process. It is dorsoventrally long and laterally crescentic, and it is slightly swollen posteriorly. There is a notch between the coronoid process and the mandibular condyle.

*Periotic:* The left periotic is preserved (figure 9, table 2). It is relatively slender in shape. The dorsal surface of the body is smooth. The anterior and posterior processes are almost straight anteroposteriorly in the ventral view. The anterior process is laterally thick, but its apex is slender and pointed (figure 9a). It is weakly bent ventrally. The anterior bullar facet is weakly excavated anteroposteriorly. A rounded accessory ossicle of the tympanic bulla is still articulated with the fovea epitubaria at the posteromedial base of the anterior process. The mallear fossa is excavated posterior to the accessory ossicle and opens posteriorly. It is laterally delimited by a mediolaterally elongated lateral tuberosity. The cochlear portion (=pars cochlearis) is rounded in outline, relatively low, and it is slightly bent anteroventrally with a weak depression at the anteromedial surface, so the anterior incisure is narrow and slightly covered with the anterior edge of the cochlear portion. The epitympanic hiatus is relatively deep and wide. The internal acoustic meatus contains the spiral cribriform tract. The spiral cribriform tract is relatively small and almost circular, slightly compressed transversely. The posterolateral margin of the internal acoustic meatus is elevated and reaches almost the top of the cochlear portion. The foramen singulare opens in the same aperture for the facial canal (figure 9b). The proximal opening of the facial canal is relatively large and elliptical in shape. The aperture for the cochlear aqueduct is large and semicircular. It is located close to the posterior margin of the internal auditory meatus. It is also relatively close to the aperture for the vestibular aqueduct (=endolymphatic duct) that is elliptical in shape. The fenestra rotunda is relatively large and semicircular in outline. The posterior process is relatively long posteriorly and inflated mediolaterally. It is fused to the skull in the mediolaterally narrow periotic fossa of the skull. The posterior bullar facet is covered by the fused fragment of the posterior process of the tympanic bulla. It is slightly bent posterolaterally at the base just posterior to the cochlear portion. Because of this fusion with the posterior process of the tympanic bulla and its posterior expansion, the posterior bullar facet is posteriorly flared and fan shaped in the ventral view (figure 9a). Although mostly covered by the remnant of the posterior process of the tympanic bulla, the surface of the

**Table 2.** Measurements (in mm) of the ear bones of *Berardius kobayashii* sp. nov., holotype, SCM 5530-6.

| measurements (in mm) of the ear bones of *Berardius kobayashii* | |
| --- | --- |
| **tympanic bulla** | |
| standard length of tympanic bulla, distance from the anterior tip to the posterior end of outer posterior prominence | 37 |
| distance from the anterior tip to the posterior end of inner posterior prominence | 34 |
| distance from the posteroventral tip of outer posterior prominence to the tip of sigmoid process | 24 |
| distance from the posteroventral tip of outer posterior prominence to the tip of conical process | 17.5 |
| width of tympanic bulla at the level of sigmoid process | 20 |
| height of tympanic bulla, from the tip of sigmoid process to ventral keel | 23 |
| width across inner and outer posterior prominence | 18 |
| greatest depth of interprominential notch | 5 |
| width of the upper border of sigmoid process | 11 |
| **periotic** | |
| standard (maximum anteroposterior) length, from the tip of anterior process to the posterior end of posterior process, measured on a straight line parallel to the cerebral border | 39.7 |
| standard (maximum mediolateral) width, from the medial edge of cochlear potion to the apex of lateral tuberosity | 23 |
| maximum dorsoventral depth of anterior process perpendicular to the axis of periotic | 13.6 |
| maximum mediolateral width of anterior process at the base | 17 |
| length of anterior process from the anterior apex to the posterior border of mallear fossa | 16.2 |
| length of anterior process from the anterior apex to anterior incisure | 14.3 |
| anteroposterior length of cochlear potion | 17.3 |
| dorsoventral depth of cochlear potion | 18.7 |
| mediolateral width of cochlear potion from medial edge to fenestra ovalis | 10.8 |
| maximum diameter of fenestra rotunda | 2.7 |
| maximum diameter of the internal acoustic meatus | 4.7 |
| maximum diameter of the cochlear aqueduct | 1.9 |
| maximum diameter of the aperture for the vestibular aqueduct | 3.6 |
| anteroposterior diameter of proximal opening of facial canal | 3.4 |
| anteroposterior diameter of the distal opening of facial canal | 1.9 |
| length of the posterior process (posterior bullar facet between anteroposterior tips) | 17.3 |
| width of the posterior process (posterior bullar facet between mediolateral tips) | 12.1 |

narrow part of posterior bullar facet that is not covered by the posterior process of the tympanic is smooth.

*Tympanic bulla:* The left tympanic bulla is preserved (figures 10 and 11, table 2). The anterior part of the ventral wall of the tympanic bulla is cylindrical, and its anterior margin is transversely rectilinear in outline. The involucrum is shorter anteriorly than the lateral wall. The dorsal margin of the involucrum is cut by an indentation. The inner and outer posterior prominences are thick and posteriorly short, and they are ventrally bulging. The outer posterior prominence bulges more ventrally than the inner posterior prominence. The median furrow runs anteroposteriorly on the ventral surface, and it is transversely narrow and anteriorly shallow. The interprominential notch is relatively deep at the posterior end of the median furrow. The lateral furrow is shallow, and its posterior ridge is prominent and twisted on the lateral wall of the tympanic bulla. The sigmoid process is square in the anterior view, and it is anteroposteriorly thin. The posteroventral corner of the sigmoid process is projected

posteriorly. A damaged base of the posterior process lies on the posterolateral end of the sigmoidal process. The elliptical foramen is present (figures 10 and 11).

# 5. Results of the phylogenetic analysis

We compared the characteristics of SCM 5530-6 with both extant and fossil Ziphiidae including 27 genera and 55 species based mainly on Bianucci et al. [3] and Lambert et al. [35]. SCM 5530-6 was originally reported and described as a ziphiid by Takahashi et al. [14] based on the posteriorly located bony nares with a gently elevated ascending process of the premaxilla and the posteroventrally steeply sloped supraoccipital. In addition to the above characters, SCM 5530-6 displays diagnostic characters of the family Ziphiidae such as the presence of the premaxillary crests on the gently elevated vertex, and wide and anteriorly extended hamular fossa of the pterygoid sinus [3,35]. We analysed its phylogenetic relationships based on the data matrix and parsimony settings by Bianucci et al. [3].

As a result of the analysis with down-weighted homoplastic characters, five most parsimonious trees were obtained with a tree length of 177 steps, Goloboff fit ($k = 3$) −58.44286, a consistency index (CI) of 0.475 and a retention index (RI) of 0.767. The strict consensus tree is shown in figure 12. The 50% majority-rule consensus tree is also shown in the electronic supplementary material, S4. Our results suggest that Chavinziphius from the Late Miocene of Peru (eastern South Pacific) that was recognized as the earliest diverging stem ziphiid by Bianucci et al. [3] is the sister taxon of the clade only including the extant subfamilies, i.e. the Berardiinae, Ziphiinae and Hyperoodontinae. Instead, Ninoziphius also from the Miocene of Peru is recognized as the earliest diverging stem ziphiid. As for SCM 5530-6, it is nested in the monophyletic subfamily Berardiinae consisting of the genera Berardius, Archaeoziphius and Microberardius ([3,36]; figure 12).

The Berardiinae is supported by five synapomorphies: a slightly concave ascending process of the premaxilla (Char. 7: as reversal); a weak constriction on the ascending process of the right premaxilla (between premaxillary sac fossa and premaxillary crest) and the ratio between the minimal width of the ascending process of the premaxilla and the width of the right premaxillary crest is greater than 0.80 (Char. 8: as reversal); no anteromedial excavation of the dorsal surface of the nasal (Char. 14: as reversal); an isolated rounded protuberance formed by the interparietal and/or frontals on the posterior part of the vertex (Char. 17) and the anteromedial margin of the supraoccipital that is lower than the frontals on the vertex (Char. 18) (see also [3,32,37]). Although not well-preserved on SCM 5530-6, the following character is also considered to be a potentially unique character for the subfamily: the posteromedial narrowing of nasals and frontals at the vertex, which is narrower than the anteriormost transverse width of nasals (Char. 50). Among those characters, quite a few synapomorphies of the subfamily are interpreted as reversals from derived to secondarily primitive character states. A low elevation of the ascending process of the premaxilla is observed in Berardius and related fossil species such as Archaeoziphius microglenoideus and Microberardius africanus [3,36,38], though this condition is also seen in Chavinziphius maxillocristatus, one of the stem ziphiids [3]. Another example is a cluster of dorsal infraorbital foramina, which is present on the base of the rostrum in the early diverging ziphiids [3]. On the other hand, only a large foramen is present in most crown ziphiids [3]. SCM 5530-6 has a large foramen on the right maxilla, and two relatively large foramina on the left maxilla, and therefore, the condition of these foramina is thought to be the condition seen in the later diverging, extant ziphiids. However, Ninoziphius platyrostris from the Miocene of Peru, which is considered to be the earliest branching ziphiid (figure 12), has a relatively large dorsal infraorbital foramen [35], suggesting that this character state might have evolved in two distantly related clades. This case also suggests that some characters seen in the berardiines that are considered to be synplesiomorphies may not be primitive character states that were conservatively retained in them.

Among the species within the Berardiinae, Microberardius africanus was recognized as the first split with the rest of genera and species including SCM 5530-6, but no autapomorphies were recognized to characterize Microberardius itself in our analysis. In this regard, Microberardius seems to be 'ancestral' to the later diverging berardiines. However, Bianucci et al. [3] suggest that Microberardius is distinguishable from other berardiines (though not preserved in SCM 5530-6) by having the following characters: the rostrum is higher than wide along most of its length, the rostral base is narrower and the maxillary crest is not extended on the rostral base.

By contrast, the monophyly of the species belonging in the genera Berardius and Archaeoziphius, including SCM 5530-6, is supported by the following unique suite of characters: the ratio between the

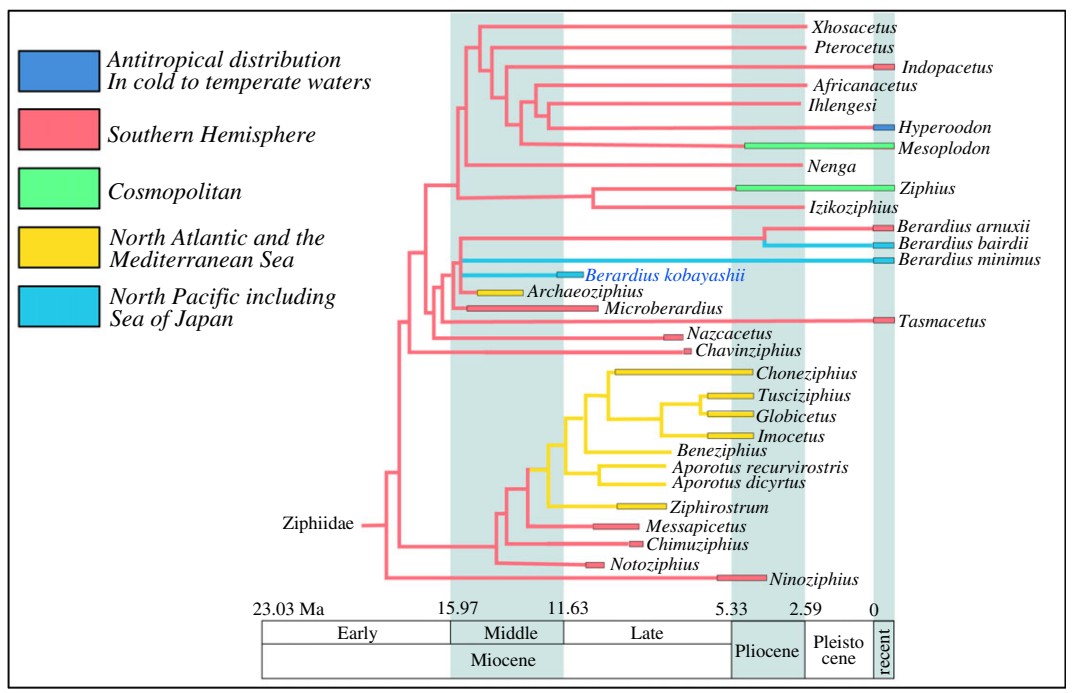

**Figure 12.** Phylogenetic relationship and palaeobiogeographic distribution of *Berardius kobayashii* sp. nov, within the Ziphiidae based on Bianucci *et al*. [3] and our re-analysis. The strict consensus tree resulting from five most parsimonious trees, 177 steps long, with the consistency index = 0.475 and the retention index = 0.767.

width of the premaxillary crest and that of the premaxillary sac fossa of SCM 5530-6 is moderate, from 1.0 to 1.25 (Char. 11); an isolated rounded protuberance is formed by the frontals, not the interparietal, on the posterior part of the vertex (Char. 17, 0 to greater than 2).

Although the basic framework of the relationships among the genera of the Berardiinae (i.e. [*Microberardius*, [*Archaeoziphius*, *Berardius*]]), which was originally demonstrated by Bianucci *et al.* [3], is also supported by our analysis, *Archaeoziphius microglenoideus* was included in an unresolved politomy of the extant *Berardius* species plus SCM 5530-6 in our strict consensus tree or bracketed in between the clade containing two other extant species of *Berardius* (i.e. *B. bairdii* and *B. arnuxii*) and the clade containing newly added *Berardius minimus* plus SCM 5530-6 in our 50% majority-rule consensus tree (figure 12 and electronic supplementary material, S4). However, 27 out of 51 (i.e. 53%) characters used in Bianucci *et al.* [3] and in our analysis are missing and uncertain in *A. microglenoideus*, and therefore, its phylogenetic position is still unstable (see also the electronic supplementary material, S4). In addition, because of other observable characters that were not included in our phylogenetic analysis such as small semicircular glenoid fossa, small postglenoid process and short and anteroposteriorly flattened zygomatic process of the squamosal in *Archaeoziphius*, the position of this genus is separated from the extant species of *Berardius* plus SCM 5530-6 and is anticipated to be 'outside' the clade of the genus *Berardius* as a sister taxon to the latter. In this regard, the monophyly of the genus *Berardius* including SCM 5530-6 is highly considerable, and the position of SCM 5530-6 as a sister taxon to *B. minimus* in our 50% majority-rule consensus tree also makes SCM 5530-6 a species of *Berardius* in the Berardiinae. As for the specific relationship of SCM 5530-6 to other species in the genus, SCM 5530-6 is more closely related to *B. minimus*, as was mentioned above, in having the apices of the left and right hamular processes of the pterygoids that contact medially, forming together a posteriorly directed medial point (Char. 36). In this regard, it should be noted that although this posterior projection of the hamular processes on the holotype of *B. minimus* is weak, referred specimens such as NSMT-M 35206 and 42000 have a distinctive posterior projection at that portion. In addition, the temporal fossa of SCM 5530-6 is shallow and its lateral surface is convex also as in *B. minimus.* By contrast, it is definitely concave in *B. bairdii* and *B. arnuxii.* As for skull sutures, *B. minimus* has much tighter sutures as pointed out by Yamada *et al.* [2], SCM 5530-6 also has much tighter sutures compared with those in *B. bairdii* and *B. arnuxii.* In fact, three out of five most parsimonious trees recover SCM 5530-6 as the sister taxon of *B. minimus* (electronic supplementary material, S4). Finally, SCM 5530-6 is distinguishable from *B. minimus* in having the

shorter nasals, the ratio between the length of the medial suture of nasals on the vertex and the maximum width of nasals less than 0.4 (Char. 13: as reversal). Furthermore, the skull size, based on the bizygomatic width, of SCM 5530-6 is 50% smaller than *B. minimus*, as was mentioned above. Consequently, the identification of SCM 5530-6 as a distinctive species within the genus *Berardius* is warranted. Thus, we propose *B. kobayashii* sp. nov., based on SCM 5530-6 from the Middle to Late Miocene boundary age Tsurushi Formation.

## 6. Discussion

In previous studies, Takahashi *et al.* [14] identified SCM 5530-6, now the holotype of *B. kobayashii* sp. nov. as 'provisionally *Berardius* sp. indet.' based on the following characteristics: the external bony nares are triangular in outline; the vertex is gently elevated and the posterior part of the pterygoid (though it was misidentified as median part of the palate in [14, p. 104]) is ventrally well developed. However, they compared SCM 5530-6 only with some extant genera of the Ziphiidae; i.e. *Ziphius*, *Berardius* and *Mesoplodon*, and they also pointed out that the transverse width of the premaxilla of SCM 5530-6 is greater relative to the width of the maxilla at the level of the antorbital notch compared with that of then extant two species of *Berardius*, and that the lateral margin of the same portion is curved laterally in contrast to the relatively straight condition of this portion in both species of *Berardius*. In addition, they also pointed out that the zygomatic process of the squamosal is shaped like an anterodorsally tapered triangle in contrast to an anteriorly sub-squared rectangle in both species of *Berardius*. In this regard, both inter- and intra-generic comparisons were quite limited to their comparable taxa among the extant ziphiids, and the generality of their taxonomical differentiation of characters were somewhat arbitrary. Furthermore, two new genera of the Berardiinae were described [36,38], and the third extant species of *Berardius* was also added after Takahashi *et al.* [14] as mentioned above [3]. Accordingly, the comprehensive comparisons among both extinct and extant species of *Berardius* and of the Berardiinae in the Ziphiidae were necessary to locate SCM 5530-6 more precisely in the phylogenetic framework. As a result of our cladistic analysis modified from Bianucci *et al.* [38], *B. kobayashii* sp. nov. was recognized as the closest to *B. minimus* as discussed above. It confirms that the new species based on SCM 5530-6 as the holotype actually belongs in the genus *Berardius*, and it also indicates that although *B. minimus* is known only by the extant individuals in the northern North Pacific, the generic emergence could have been much earlier than the emergence and diversification of the other previously known two antitropical species; i.e. the northern North Pacific *B. bairdii* and the Antarctic *B. arnuxii*. Consequently, the emergence and diversification of the genus *Berardius* must have occurred in the late Middle or early Late Miocene and potentially in the northern North Pacific based on the new fossil from the Middle to Late Miocene boundary age Tsurushi Formation.

For now, three genera are recognized in the subfamily Berardiinae; i.e. *Microberardius* from the Neogene (tentatively Middle to Late Miocene) off South Africa [38], *Archaeoziphius* from the Middle Miocene of Belgium [36] and *Berardius* from the Middle to Late Miocene boundary of Japan nowadays in the North Pacific and the southern oceans including the South Atlantic and South Pacific north to the limit of the Antarctic ([1,2]; this study). Previously, there was no fossil record of the berardiines in the North Pacific Ocean. So, the new fossil described here reveals that berardiine ziphiids already dispersed to the western North Pacific at latest in the early Late Miocene (*ca* 11 Ma) and that the emergence of the extant genus *Berardius* goes back to the Middle to Late Miocene boundary age of the North Pacific, then they might have diversified with or just after the age of their closely related *Archaeoziphius* in the early Middle Miocene of the North Atlantic [36].

At the moment, it is difficult to infer a more precise origin of the subfamily Berardiinae and the genus *Berardius* based on the present fossil evidence. However, the potential origin of the subfamily must have been in the Southern Hemisphere on the basis of their phylogenetic relationships and phylogeographic distributions of their sister and outgroups ([3,5,36–38]; figure 12). In addition, the potentially close relationship between *B. kobayashii* sp. nov. and *B. minimus* suggests an emergence of the genus in the North Pacific. And that the ancestor of *B. bairdii* and *B. arnuxii* may have evolved somewhere from a *B. kobayashii*-like berardiine in the Northern Hemisphere or somewhere in between the Northern and Southern Hemispheres by the end of the Miocene [39]. It could have been followed by an antitropical dispersal (with gigantism) [40]. In any case, the two currently sympatric forms, the smaller *B. minimus* and the larger *B. bairdii* in the North Pacific could correspond to a secondary contact later than the Pliocene [40]. However, all this evidence suggests the rapid diversification of taxa in the subfamily

Berardiinae in the initial stage of their evolution during the Middle and/or Late Miocene in the Northern Hemisphere (figure 12).

It is difficult to establish the generic emergence of extant taxa based only on morphological data without genetic information allowing for divergence time estimations. In the case of the genus *Berardius*, its emergence time estimation depends on the divergence time of the Berardiinae and other ziphiid subfamilies because the Berardiinae is at present monogeneric. In this regard, McGowen *et al.* [41], for instance, estimated the divergence time of Berardiinae from other subfamilies to be 21.98 Ma (95% CI: 17.11–29 Ma). They also estimated the split of the two species of *Berardius* known at that time (i.e. *B. arnuxii* and *B. bairdii*) to have occurred about 2.9 Ma (95% CI: 0.68–5.81 Ma). It means that the expected divergence time of the genus *Berardius* still spans a 17.11–5.81 Ma interval in minimum or 15–5.81 Ma based on the geologic age of the most closely related genus *Archaeoziphius* [36]. The divergence time of *B. minimus* from the ancestor of the two other species (i.e. *B. bairdii* and *B. arnuxii*) within the genus is still uncertain [2,40,42]. However, the minimum estimate of the divergence time could be dated back to at least the early Late Miocene in age based on our phylogenetic analysis that suggested a potential sister group relationship of *B. minimus* with the Middle/Late Miocene boundary age *B. kobayashii* sp. nov.

Accordingly, it is possible to interpret that the genus *Berardius* began to diversify at this period, which is known to be the period of global cooling and of coastline regression [43–46]. It is important to be noted that the somewhat speciose extant genus *Mesoplodon*, which includes at least 15 extant species, had already appeared and diversified by the late Middle or early Late Miocene [39,41,47]. This suggests that the much earlier and less diversified appearance of the genus *Berardius* might also have proceeded their generic emergence, and *B. kobayashii* sp. nov. could be interpreted as one of such representatives of the earlier generic emergence and diversification. It will be further considered based on molecular studies for *B. minimus* and also morphological studies for other fossil berardiines to be described in the future. Furthermore, some other ziphiid fossils including ziphiine and hyperoodontine have also been known from the Tsurushi Formation [11–13]. As mentioned above, the Tsurushi Formation deposited in deep-sea environment, and it should be emphasized that all the extant ziphiids are deep-diving whales having adapted to deep sea with considerable ecomorphological specializations during their adaptation and diversification in each clade. Some extant genera of ziphiids including the berardiines such as *Berardius* as exemplified by *B. kobayashii* sp. nov. might have emerged with adaptation to such a harsh environment during this period. These hypotheses could be further tested in the future with the discovery of more complete specimens.

## 7. Conclusion

A new species of beaked whale in the genus *Berardius* from the Middle to Late Miocene boundary age Tsurushi Formation (*ca* 12.3–11.5 Ma) in the Sado Island, Niigata Prefecture, Japan, was described on the basis of SCM 5530-6. *Berardius kobayashii* sp. nov. is distinguishable from three other species within *Berardius* by having the following characters: the skull is extremely small in size (60–70% smaller than *B. bairdii* and *B. arnuxii*, 50% smaller than *B. minimus*); the transverse diameter of the premaxilla is relatively wide compared with the width of the rostrum at the level of the anterior notch; the zygomatic process of the squamosal is anterodorsally tapered and triangular in shape in lateral view; and the nasals are short. *Berardius kobayashii* sp. nov. is the fourth species of the genus. Therefore, the lineage leading to the extant species of *Berardius* already lived in the western North Pacific at the Middle to Late Miocene boundary, about 11 Ma. The western North Pacific, including the Sea of Japan, may have been one of the areas of the evolution and radiation of *Berardius* species.

Data accessibility. Additional data are available as the electronic supplementary materials (Supplementary files S1–S7: S1, list of the taxa and specimens of the berardiine ziphiids used for this study; S2, list of characters used for our cladistic analysis; S3, character/taxon matrix; S4, resultant cladograms including the unweighted strict and 50% majority-rule consensus cladograms, five equally most parsimonious cladograms with implied weighting of $K = 3$; S5, references for character collections; S6, Nexus file; S7, TNT file). The new species has been registered in Zoobank. The LSID for this publication is: urn: lsid: zoobank. org: act: 317C1452-AE2E-4A08-A473-3B507B452D2C.

Authors' contributions. A.K. and N.K. conceived and designed the project, performed the experiment and analysed the data. A.K. prepared SCM 5530-6. N.K. contributed reagents/materials/preparation tools/analysis tools. A.K. and N.K. wrote the paper.

Competing interests. We declare we have no competing interests.

Funding. A.K. has funded the travel grant for the 2017 fiscal year from the Regional Promotion Division of Sado City.
Acknowledgements. We thank the late I. Kobayashi (Professor Emeritus at Niigata University), A. Matsuoka (Niigata University), M. Shimizu (Asahimachi Museum, Niigata University), K. Takigawa and M. Ikarashi (Sado Museum), M. Aida and Y. Ichihashi (Sado Geopark Promotion Office and Sado Museum), K. Narita (Nagano City Museum), M. Kato (Nagaoka Municipal Science Museum), Y. Tajima and T.K. Yamada (NMNS), for allowing access to the specimens and giving us the opportunity to observe specimens at their respective museums. We thank Y. Yanagisawa (Geological Survey of Japan), K. Takahashi (Biwako Museum), H. Ichishima (Fukui Prefectural Dinosaur Museum), K. Nagasawa (formerly Yamagata Prefectural Museum), T. Kimura (Gunma Museum of Natural History), for providing useful discussion and advice with the literature on fossil ziphiids. We thank C. Sakata (NMNS) for CT scanning the periotic for this study. Finally, the manuscript was greatly improved by careful attention to detail and extensive comments from O. Lambert (Institut royal des Sciences naturelles de Belgique) and A. Benites-Palomino (University of Zurich), to which we greatly appreciate it.

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
