## [Peer Review File · Royal Society Open Science]

Review History

RSOS-201152.R0 (Original submission)

Review form: Reviewer 1 (Olivier Lambert)

Is the manuscript scientifically sound in its present form?

No

Are the interpretations and conclusions justified by the results?

No

Is the language acceptable?

Yes

Do you have any ethical concerns with this paper?

No

Have you any concerns about statistical analyses in this paper?

No

Recommendation?

Major revision is needed (please make suggestions in comments)

Comments to the Author(s)

The present work deals with a very informative ziphiid fossil skull, including the highly informative ear bones. Those elements are only rarely found associated to other cranial material. Furthermore, the stratigraphic context is relatively well constrained, providing a geological age that is rather exciting (most extinct ziphiids are found in younger deposits). Finally, the specimen displays similarities with members of the extant family Berardiinae, potentially confirming the antiquity of this clade. All together, these elements make this study important and worth publishing.

The text is well organized and the bones are finely illustrated (see a few comments below and in the annotated pdf (Appendix A)). I found the descriptions clear and concise (maybe a bit too short), although a few anatomical interpretations may ask for a second check.

I made a series of comments in the annotated pdf, and those mostly deal with English and typos, which should be easily fixed. My main concerns and suggestions are listed below, and those may request some more work. I hope that those will prove useful, as this very fine, scientifically informative specimen definitely deserves to be published.

1- Concerning the stratigraphic context, you reach the conclusion that the formation from which the studied fossil originates can be dated from an interval spanning 12.3–11.5 Ma. This means that this unit may extend in the earliest Tortonian (starting at 11.63 Ma). If this is correct it may be difficult to exclude an early late Miocene age, and I would suggest modifying the text (and the title + abstract) accordingly. Maybe 'from the latest middle to earliest late Miocene', or 'from the middle to late Miocene boundary'. This would remove any possible confusion, I think. Indeed, many ziphiids have been dated from the Tortonian, whereas Serravallian records are much less frequent.

2- The emended diagnosis of *Berardius* is long, and it is not easy to understand how it was compiled, for several reasons. First, some characters are also listed for the subfamily Berardiinae. Second, differences are not associated to other genera. It may be ok to keep such a list, but then it should start with something like 'The genus *Berardius* differs from all other ziphiids with the following unique combination of characters: ...'. However, because you provide a diagnosis for Berardiinae just above, it may be sufficient to provide here differences with other Berardiinae (and maybe expand somewhat the diagnosis for Berardiinae, including all characters cited here that also apply to other berardiines).

It would certainly be useful to list, in the diagnosis of *Berardius*, specific differences with *Archaeoziphius* and *Microberardius*.

3- For the very informative periotic I would suggest also providing lateral and medial views, based on the 3D model.

A minor issue: the photo of the ventral view in fig. 8a is not in the same orientation as the line drawing in fig. 8b. If not too complicated I would suggest correcting this. Another probably easier solution would be to provide a true ventral view based on the 3D model.

4- I fully agree that this specimen differs from *Archaeoziphius* and *Microberardius*. However, are these differences greater than the ones separating this specimen from extant *Berardius* species? You mention potential synapomorphies of *Berardius* at the level of the periotic, but unfortunately the periotic is unknown in *Archaeoziphius* and *Microberardius*, meaning that these characters could possibly represent synapomorphies for Berardiinae instead of *Berardius*, making the

attribution of that genus not strongly supported. Furthermore, other similarities with *Berardius* may correspond to plesiomorphic features.

I would suggest reorganizing the comparison, in a way to first provide characters of the family Ziphiidae that can be observed in the new species, then provide characters of the subfamily Berardiinae, and finally derived characters of the genus *Berardius*. This last part may prove more difficult, as sister groups *Archaeoziphius* and *Microberardius* unfortunately lack ear bones.

5- A phylogenetic tree is provided, but it is not made clear how the relationships of the new species with other ziphiids were investigated. My impression is that you used a tree from a previous analysis, before manually adding the new species. This should probably be explained. If this is the case, I would tend to think that testing the affinities of the new species with a proper phylogenetic analysis would provide a more convenient base for the systematic and taxonomic results, as well as for the discussion on palaeobiogeography and the evolution of the genus *Berardius*. As currently presented, the discussion may sound a bit too speculative. In other words, I think that the attribution of the new species to the genus *Berardius* should be backed up by a phylogeny.

6- It may be interesting to note that divergence date estimates for the genus *Berardius* (at that time only including *B. bairdii* and *B. arnuxii*) are considerably younger (mean = 2.9 Ma) than the age of the new species (see McGowen et al. 2009).

McGowen, M.R., Spaulding, M., and Gatesy, J. 2009. Divergence date estimation and a comprehensive molecular tree of extant cetaceans. *Molecular Phylogenetics and Evolution* 53 (3): 891-906.

O Lambert

Review form: Reviewer 2 (Aldo Benítez)

Is the manuscript scientifically sound in its present form?

No

Are the interpretations and conclusions justified by the results?

No

Is the language acceptable?

No

Do you have any ethical concerns with this paper?

No

Have you any concerns about statistical analyses in this paper?

No

Recommendation?

Major revision is needed (please make suggestions in comments)

Comments to the Author(s)

Tuesday, 14.07.2020

Dear

Kawatani, Ayako; Kohno, Naoki

I have done an extensive revision of your work and found the overall purpose of the manuscript interesting and useful. Despite this, I do believe that the work in its current state will need a substantial revision in order to be publishable. Below you'll find my comments:

1) I found the manuscript inconsistently written and difficult to read. I would strongly advise to send the manuscript to a native English speaker to help with the drafting of the paper. As a non-native English speaker, I completely understand the fact that the authors could have made more mistakes than usual, as this is the case with me. There are several parts where the inconsistency of the manuscript (grammar, bad translations or changes between tenses) makes it mostly difficult to read, or the idea in the sentences is lost.

2) Proposing a new species, specially for an extant genus, must have a strong phylogenetic framework in order to support this assignation. Since this is a work on Systematics it should be the center of the manuscript. I did not find major references to it, but a figure and with some lines in text:

a. There is no description of the trees, statistical support, consensus, or even references to the methodology undertaken. A simple statement as 'it follows Bianucci et al. 2016' is not enough.
 b. No list of plesiomorphies, synapomorphies or apomorphies that support each of the Systematical assignations is specified, but some references in the 'Comparisons' section. If this is not provided, the support of the new species cannot be easily corroborated or compared, either in reviewing process or by other researchers interested in this work.

3) A differential diagnosis of the new species is missing. This section is key to check both the apomorphies that support the new species, and the synapomorphies that it shares with other closely related taxa. Furthermore, only a single apomorphy (provided in the abstract) supports the new species? (Being the 'smallest' is not a diagnostic feature; it could be important to identify it, but how can it be cladistical coded?). References to this should be in the abstract, some features can be found on the 'diagnosis of the species' section, but not clear.

4) The description is mostly ambiguous and difficult to read. I would highly advise to rewrite the descriptions based on regions or views. Please see comments in text (Appendix B). Squamosal is almost 100% complete and has several major characters of odontocetes, a few lines to describe it is not enough.

5) The comparisons section is okay, as it emphasizes the comparison between taxa. However, I would encourage the authors to discuss the specific variations among taxa in the Discussion. A suitable way to reorder both the Comparisons and Discussion could be to briefly present the phylogenetic synapomorphies + how this support the assignation to: Ziphiida, Berardiinae, Berardius in the Comparison section, with references to other taxa; and emphasize the most important characters in the Discussion section. Furthermore, key characteristics that delimit the genus Berardius should be referenced in the Discussion.

Finally, I wish to emphasize that the material presented in this manuscript could be easily It has a lot of potential, but several deficiencies still need to be corrected.

Best regards,

Decision letter (RSOS-201152.R0)

Dear Ms Kawatani,

The editors assigned to your paper ("The oldest fossil record of the extant genus *Berardius* (Odontoceti, Ziphiidae) from the middle Miocene of the western North Pacific") have now received comments from reviewers. We would like you to revise your paper in accordance with the referee and Associate Editor suggestions which can be found below (not including confidential reports to the Editor). Please note this decision does not guarantee eventual acceptance.

Please submit a copy of your revised paper before 13-Aug-2020. Please note that the revision deadline will expire at 00.00am on this date. If we do not hear from you within this time then it will be assumed that the paper has been withdrawn. In exceptional circumstances, extensions may be possible if agreed with the Editorial Office in advance. We do not allow multiple rounds of revision so we urge you to make every effort to fully address all of the comments at this stage. If deemed necessary by the Editors, your manuscript will be sent back to one or more of the original reviewers for assessment. If the original reviewers are not available, we may invite new reviewers.

- Data accessibility

It is a condition of publication that all supporting data are made available either as supplementary information or preferably in a suitable permanent repository. The data accessibility section should state where the article's supporting data can be accessed. This section should also include details, where possible of where to access other relevant research materials such as statistical tools, protocols, software etc can be accessed. If the data have been deposited in an external repository this section should list the database, accession number and link to the DOI for all data from the article that have been made publicly available. Data sets that have been

deposited in an external repository and have a DOI should also be appropriately cited in the manuscript and included in the reference list.

If you wish to submit your supporting data or code to Dryad (<http://datadryad.org/>), or modify your current submission to dryad, please use the following link:
<http://datadryad.org/submit?journalID=RSOS&manu=RSOS-201152>

- **Competing interests**

- **Authors' contributions**

- **Acknowledgements**

- **Funding statement**

on behalf of Professor Marcelo Sanchez (Associate Editor) and Kevin Padian (Subject Editor)
openscience@royalsociety.org

Subject Editor's comments (Professor Kevin Padian):

Comments to the Author:

Both reviewers see much value in this work should it be thoroughly revised. Our Associate Editor very much agrees with their assessment. This can become a very good paper and the

reviews provide specific and good suggestions how to improve it. I hope the comments are useful to you and we look forward to your revision.

Reviewers' Comments to Author:

Reviewer: 1

Comments to the Author(s)

The present work deals with a very informative ziphiid fossil skull, including the highly informative ear bones. Those elements are only rarely found associated to other cranial material. Furthermore, the stratigraphic context is relatively well constrained, providing a geological age that is rather exciting (most extinct ziphiids are found in younger deposits). Finally, the specimen displays similarities with members of the extant family Berardiinae, potentially confirming the antiquity of this clade. All together, these elements make this study important and worth publishing.

The text is well organized and the bones are finely illustrated (see a few comments below and in the annotated pdf). I found the descriptions clear and concise (maybe a bit too short), although a few anatomical interpretations may ask for a second check.

I made a series of comments in the annotated pdf, and those mostly deal with English and typos, which should be easily fixed. My main concerns and suggestions are listed below, and those may request some more work. I hope that those will prove useful, as this very fine, scientifically informative specimen definitely deserves to be published.

1- Concerning the stratigraphic context, you reach the conclusion that the formation from which the studied fossil originates can be dated from an interval spanning 12.3–11.5 Ma. This means that this unit may extend in the earliest Tortonian (starting at 11.63 Ma). If this is correct it may be difficult to exclude an early late Miocene age, and I would suggest modifying the text (and the title + abstract) accordingly. Maybe 'from the latest middle to earliest late Miocene', or 'from the middle to late Miocene boundary'. This would remove any possible confusion, I think. Indeed, many ziphiids have been dated from the Tortonian, whereas Serravallian records are much less frequent.

2- The emended diagnosis of *Berardius* is long, and it is not easy to understand how it was compiled, for several reasons. First, some characters are also listed for the subfamily Berardiinae. Second, differences are not associated to other genera. It may be ok to keep such a list, but then it should start with something like 'The genus *Berardius* differs from all other ziphiids with the following unique combination of characters: ...'. However, because you provide a diagnosis for Berardiinae just above, it may be sufficient to provide here differences with other Berardiinae (and maybe expand somewhat the diagnosis for Berardiinae, including all characters cited here that also apply to other berardiines).

It would certainly be useful to list, in the diagnosis of *Berardius*, specific differences with *Archaeoziphius* and *Microberardius*.

3- For the very informative periotic I would suggest also providing lateral and medial views, based on the 3D model.

A minor issue: the photo of the ventral view in fig. 8a is not in the same orientation as the line drawing in fig. 8b. If not too complicated I would suggest correcting this. Another probably easier solution would be to provide a true ventral view based on the 3D model.

4- I fully agree that this specimen differs from *Archaeoziphius* and *Microberardius*. However, are these differences greater than the ones separating this specimen from extant *Berardius* species? You mention potential synapomorphies of *Berardius* at the level of the periotic, but unfortunately the periotic is unknown in *Archaeoziphius* and *Microberardius*, meaning that these characters

could possibly represent synapomorphies for Berardiinae instead of Berardius, making the attribution of that genus not strongly supported. Furthermore, other similarities with Berardius may correspond to plesiomorphic features.

I would suggest reorganizing the comparison, in a way to first provide characters of the family Ziphiidae that can be observed in the new species, then provide characters of the subfamily Berardiinae, and finally derived characters of the genus Berardius. This last part may prove more difficult, as sister groups *Archaeoziphius* and *Microberardius* unfortunately lack ear bones.

5- A phylogenetic tree is provided, but it is not made clear how the relationships of the new species with other ziphiids were investigated. My impression is that you used a tree from a previous analysis, before manually adding the new species. This should probably be explained. If this is the case, I would tend to think that testing the affinities of the new species with a proper phylogenetic analysis would provide a more convenient base for the systematic and taxonomic results, as well as for the discussion on palaeobiogeography and the evolution of the genus *Berardius*. As currently presented, the discussion may sound a bit too speculative. In other words, I think that the attribution of the new species to the genus *Berardius* should be backed up by a phylogeny.

6- It may be interesting to note that divergence date estimates for the genus *Berardius* (at that time only including *B. bairdii* and *B. arnuxii*) are considerably younger (mean = 2.9 Ma) than the age of the new species (see McGowen et al. 2009).

McGowen, M.R., Spaulding, M., and Gatesy, J. 2009. Divergence date estimation and a comprehensive molecular tree of extant cetaceans. *Molecular Phylogenetics and Evolution* 53 (3): 891-906.

O Lambert

Reviewer: 2

Comments to the Author(s)
Tuesday, 14.07.2020

Dear

Kawatani, Ayako; Kohno, Naoki

I have done an extensive revision of your work and found the overall purpose of the manuscript interesting and useful. Despite this, I do believe that the work in its current state will need a substantial revision in order to be publishable. Below you'll find my comments:

1) I found the manuscript inconsistently written and difficult to read. I would strongly advise to send the manuscript to a native English speaker to help with the drafting of the paper. As a non-native English speaker, I completely understand the fact that the authors could have made more mistakes than usual, as this is the case with me. There are several parts where the inconsistency of the manuscript (grammar, bad translations or changes between tenses) makes it mostly difficult to read, or the idea in the sentences is lost.

2) Proposing a new species, specially for an extant genus, must have a strong phylogenetic framework in order to support this assignation. Since this is a work on Systematics it should be the center of the manuscript. I did not find major references to it, but a figure and with some lines in text:

a. There is no description of the trees, statistical support, consensus, or even references to the methodology undertaken. A simple statement as 'it follows Bianucci et al. 2016' is not enough.
 b. No list of plesiomorphies, synapomorphies or apomorphies that support each of the Systematical assignments is specified, but some references in the 'Comparisons' section. If this is not provided, the support of the new species cannot be easily corroborated or compared, either in reviewing process or by other researchers interested in this work.

3) A differential diagnosis of the new species is missing. This section is key to check both the apomorphies that support the new species, and the synapomorphies that it shares with other closely related taxa. Furthermore, only a single apomorphy (provided in the abstract) supports the new species? (Being the 'smallest' is not a diagnostic feature; it could be important to identify it, but how can it be cladistical coded?). References to this should be in the abstract, some features can be found on the 'diagnosis of the species' section, but not clear.

4) The description is mostly ambiguous and difficult to read. I would highly advise to rewrite the descriptions based on regions or views. Please see comments in text. Squamosal is almost 100% complete and has several major characters of odontocetes, a few lines to describe it is not enough.

5) The comparisons section is okay, as it emphasizes the comparison between taxa. However, I would encourage the authors to discuss the specific variations among taxa in the Discussion. A suitable way to reorder both the Comparisons and Discussion could be to briefly present the phylogenetic synapomorphies + how this support the assignation to: Ziphiida, Berardiinae, Berardius in the Comparison section, with references to other taxa; and emphasize the most important characters in the Discussion section. Furthermore, key characteristics that delimit the genus Berardius should be referenced in the Discussion.

Finally, I wish to emphasize that the material presented in this manuscript could be easily It has a lot of potential, but several deficiencies still need to be corrected.

Best regards,

Author's Response to Decision Letter for (RSOS-201152.R0)

See Appendix C.

RSOS-201152.R1 (Revision)

Review form: Reviewer 1 (Olivier Lambert)

Is the manuscript scientifically sound in its present form?

Yes

Are the interpretations and conclusions justified by the results?

Yes

Is the language acceptable?

Yes

Do you have any ethical concerns with this paper?

No

Have you any concerns about statistical analyses in this paper?

No

Recommendation?

Accept with minor revision (please list in comments)

Comments to the Author(s)

I would like to congratulate the authors for the many improvements they made to their interesting work. The addition of a phylogenetic analysis is certainly an asset, further supporting the referral of the new species to the subfamily Berardiinae. At this point, I still noted a number of minor and moderate issues that should, in my opinion, be addressed before publication. I noted most of my minor questions directly in the pdf ((text with highlighted changes, figures and figure captions) see Appendix D). My main concerns are listed below:

- Diagnosis of the genus *Berardius*: As it is, there is no difference with *Archaeoziphius* in this diagnosis, meaning that one won't be able to distinguish members of these two genera. To better support the attribution to the genus *Berardius*, characters shared with extant *Berardius* species and not with *Archaeoziphius* should be provided. Considering the greater age proximity with *Archaeoziphius* and the relatively poorly resolved phylogenetic relationships among berardiines (see below), the reader may find it more appropriate to exclude the new species from the genus *Berardius*. Placing a new species in a pre-existing genus or in a new genus is at least partly arbitrary, and although I would personally not refer the new species to the genus *Berardius* (due to the relatively limited support) I understand that the authors may prefer this solution. If they keep that hypothesis, I strongly recommend to somewhat rework the diagnosis, in a way to make the case more convincing.

- Another way to make your hypothesis better supported would be to prepare a separate section, before or after the phylogeny, where a detailed morphological comparison is made between the new species and all the other berardiines. Providing additional similarities and differences in a clearly organized section will facilitate the reading, and I suspect that other differences, for example with *Archaeoziphius*, could be found (for example at the level of the glenoid fossa).

- The tree that is illustrated in figure 12 is not the consensus tree, contrary to what is mentioned in the figure caption. I guess that this is the 50% consensus tree. Though you may have a favoured topology, represented by this tree, I would recommend figuring the strict consensus too. This will better inform on the support of the proposed relationships.

- As a consequence of the relatively low support, I would suggest somewhat toning down the conclusions regarding the placement of the new species in the genus *Berardius*. Maybe something like: 'we tentatively refer the new species to the genus *Berardius*, a hypothesis that should be further tested in the future with the discovery of more complete specimens'. This is just a suggestion.

- I found some last problematic interpretations of morphological features in the description (for example labels for the posterolateral sinus fossa in several figures), but I think those will be easily solved using recent literature on the group.

- In the comparison, you note an interesting similarity of the new species with *Berardius minimus*, at the level of the hamular processes. Looking at the illustrations of the holotype of the

latter species in Yamada et al. (2019), I could not find that feature. If you observed other specimens, then this could be mentioned. Another possibility is that I did not understand well the morphological feature, and then this part could be slightly revised.

- In some places the use of geographic concepts is not made consistently (see comments in the text and figures about the northern North Pacific, the Southern Ocean...). The geographic distribution of the extant species of *Berardius* could also be somewhat more precisely described.

O Lambert

Review form: Reviewer 2 (Aldo Benites)

Is the manuscript scientifically sound in its present form?

Yes

Are the interpretations and conclusions justified by the results?

Yes

Is the language acceptable?

Yes

Do you have any ethical concerns with this paper?

No

Have you any concerns about statistical analyses in this paper?

No

Recommendation?

Accept with minor revision (please list in comments)

Comments to the Author(s)

Dear Ayako Kawatani and Naoki Kohno,

I found really nice the improvements that you performed into this research piece. Because the specimen will not only be of interest to palaeontologists, but also to modern marine mammalogists. I think that some minor improvements could help doing so. Nevertheless, these improvements are just small corrections that deal mostly with structure or phrasing some not so clear statements in the manuscript. Please refer to the attached PDF (Appendix E) to see in detail what I do refer to: as there are sometimes really long sentences of four or even five lines which are difficult to follow, or other time some sentences really short.

Hope you can cope with these small corrections which should not be a problem, and I'm really hoping to see this paper published soon.

All the best,

Aldo Benites-Palomino
Palaeontological Museum & Institute
University of Zurich

Decision letter (RSOS-201152.R1)

Dear Ms Kawatani

On behalf of the Editors, we are pleased to inform you that your Manuscript RSOS-201152.R1 "The oldest fossil record of the extant genus *Berardius* (Odontoceti, Ziphiidae) from the middle to late Miocene boundary of the western North Pacific" has been accepted for publication in Royal Society Open Science subject to minor revision in accordance with the referees' reports. Please find the referees' comments along with any feedback from the Editors below my signature.

Please submit your revised manuscript and required files (see below) no later than 7 days from today's (ie 25-Jan-2021) date. Note: the ScholarOne system will 'lock' if submission of the revision is attempted 7 or more days after the deadline. If you do not think you will be able to meet this deadline please contact the editorial office immediately.

on behalf of Professor Marcelo Sanchez (Associate Editor) and Kevin Padian (Subject Editor)
openscience@royalsociety.org

Reviewer comments to Author:

Reviewer: 1

Comments to the Author(s)

I would like to congratulate the authors for the many improvements they made to their interesting work. The addition of a phylogenetic analysis is certainly an asset, further supporting the referral of the new species to the subfamily Berardiinae. At this point, I still noted a number of minor and moderate issues that should, in my opinion, be addressed before publication. I noted most of my minor questions directly in the pdf (text with highlighted changes, figures and figure captions). My main concerns are listed below:

- Diagnosis of the genus *Berardius*: As it is, there is no difference with *Archaeoziphius* in this diagnosis, meaning that one won't be able to distinguish members of these two genera. To better support the attribution to the genus *Berardius*, characters shared with extant *Berardius* species and not with *Archaeoziphius* should be provided. Considering the greater age proximity with *Archaeoziphius* and the relatively poorly resolved phylogenetic relationships among berardiines (see below), the reader may find it more appropriate to exclude the new species from the genus *Berardius*. Placing a new species in a pre-existing genus or in a new genus is at least partly arbitrary, and although I would personally not refer the new species to the genus *Berardius* (due to the relatively limited support) I understand that the authors may prefer this solution. If they keep that hypothesis, I strongly recommend to somewhat rework the diagnosis, in a way to make the case more convincing.

- Another way to make your hypothesis better supported would be to prepare a separate section, before or after the phylogeny, where a detailed morphological comparison is made between the new species and all the other berardiines. Providing additional similarities and differences in a clearly organized section will facilitate the reading, and I suspect that other differences, for example with *Archaeoziphius*, could be found (for example at the level of the glenoid fossa).

- The tree that is illustrated in figure 12 is not the consensus tree, contrary to what is mentioned in the figure caption. I guess that this is the 50% consensus tree. Though you may have a favoured topology, represented by this tree, I would recommend figuring the strict consensus too. This will better inform on the support of the proposed relationships.

- As a consequence of the relatively low support, I would suggest somewhat toning down the conclusions regarding the placement of the new species in the genus *Berardius*. Maybe something like: 'we tentatively refer the new species to the genus *Berardius*, a hypothesis that should be further tested in the future with the discovery of more complete specimens'. This is just a suggestion.

- I found some last problematic interpretations of morphological features in the description (for example labels for the posterolateral sinus fossa in several figures), but I think those will be easily solved using recent literature on the group.

- In the comparison, you note an interesting similarity of the new species with *Berardius minimus*, at the level of the hamular processes. Looking at the illustrations of the holotype of the latter species in Yamada et al. (2019), I could not find that feature. If you observed other specimens, then this could be mentioned. Another possibility is that I did not understand well the morphological feature, and then this part could be slightly revised.

- In some places the use of geographic concepts is not made consistently (see comments in the text and figures about the northern North Pacific, the Southern Ocean...). The geographic distribution of the extant species of *Berardius* could also be somewhat more precisely described.

O Lambert

Reviewer: 2

Comments to the Author(s)

Dear Ayako Kawatani and Naoki Kohno,

I found really nice the improvements that you performed into this research piece. Because the specimen will not only be of interest to palaeontologists, but also to modern marine mammologists. I think that some minor improvements could help doing so. Nevertheless, these

improvements are just small corrections that deal mostly with structure or phrasing some not so clear statements in the manuscript. Please refer to the attached PDF to see in detail what I do refer to: as there are sometimes really long sentences of four or even five lines which are difficult to follow, or other time some sentences really short.

Hope you can cope with these small corrections which should not be a problem, and I'm really hoping to see this paper published soon.

All the best,

Aldo Benites-Palomino
Palaeontological Museum & Institute
University of Zurich

===PREPARING YOUR MANUSCRIPT===

Your revised paper should include the changes requested by the referees and Editors of your manuscript. You should provide two versions of this manuscript and both versions must be provided in an editable format:
one version identifying all the changes that have been made (for instance, in coloured highlight, in bold text, or tracked changes);
a 'clean' version of the new manuscript that incorporates the changes made, but does not highlight them. This version will be used for typesetting.
Please ensure that any equations included in the paper are editable text and not embedded images.

===PREPARING YOUR REVISION IN SCHOLARONE===

Author's Response to Decision Letter for (RSOS-201152.R1)

See Appendix F.

Decision letter (RSOS-201152.R2)

Dear Ms Kawatani,

It is a pleasure to accept your manuscript entitled "The oldest fossil record of the extant genus *Berardius* (Odontoceti, Ziphiidae) from the middle to late Miocene boundary of the western North Pacific" in its current form for publication in Royal Society Open Science. The comments of the reviewer(s) who reviewed your manuscript are included at the foot of this letter.

You can expect to receive a proof of your article in the near future. Please contact the editorial office (openscience@royalsociety.org) and the production office (openscience_proofs@royalsociety.org) to let us know if you are likely to be away from e-mail contact – if you are going to be away, please nominate a co-author (if available) to manage the proofing process, and ensure they are copied into your email to the journal.

on behalf of Professor Marcelo Sanchez (Associate Editor) and Kevin Padian (Subject Editor)
openscience@royalsociety.org

Appendix A**ROYAL SOCIETY
OPEN SCIENCE****The oldest fossil record of the extant genus *Berardius*
(Odontoceti, Ziphiidae) from the middle Miocene of the
western North Pacific**

Journal:	Royal Society Open Science
Manuscript ID	RSOS-201152
Article Type:	Research
Date Submitted by the Author:	02-Jul-2020
Complete List of Authors:	Kawatani, Ayako; University of Tsukuba Graduate School of Life and Environmental Sciences Kohno, Naoki; University of Tsukuba Graduate School of Life and Environmental Sciences; National Museum of Nature and Science, Geology and Paleontology
Subject:	palaeontology < BIOLOGY, Palaeontology < EARTH SCIENCES
Keywords:	middle Miocene, extant genus, Berardius kobayashii , sp. nov., western North Pacific
Subject Category:	Organismal and Evolutionary Biology

Author-supplied statements

Relevant information will appear here if provided.

Ethics

Does your article include research that required ethical approval or permits?:

This article does not present research with ethical considerations

Statement (if applicable):

CUST_IF_YES_ETHICS :No data available.

Data

It is a condition of publication that data, code and materials supporting your paper are made publicly available. Does your paper present new data?:

Yes

Statement (if applicable):

All the new data including materials, measurements photographs and necessary text data are provided in our manuscript.

Conflict of interest

I/We declare we have no competing interests

Statement (if applicable):

CUST_STATE_CONFLICT :No data available.

Authors' contributions

This paper has multiple authors and our individual contributions were as below

Statement (if applicable):

A.K. and N.K. conceived and designed the project, performed the experiment and analysed the data.

A.K. prepared SCM 5530-6. N.K. contributed reagents/materials/preparation tools/analysis tools.

A.K. and N.K. wrote the paper.

The oldest fossil record of the extant genus *Berardius* (Odontoceti, Ziphiidae) from the middle Miocene of the western North Pacific

Ayako Kawatani¹ and Naoki Kohno^{1,2}

¹Graduate School of Life and Environmental Sciences, University of Tsukuba, 1-1-1, Tennoudai, Tsukuba, Ibaraki, 305-8577, Japan

²National Museum of Nature and Science, 4-1-1, Amakubo, Tsukuba, Ibaraki, 305-0005, Japan

Keywords: middle Miocene, extant genus, *Berardius kobayashii*, sp. nov., western North Pacific.

1. Summary

A new species of a beaked whale that belongs to the extant genus *Berardius* is described from the late middle Miocene Tsurushi Formation (ca. 12 Ma) on the Sado Island, Niigata Prefecture, Japan. The new species, *Berardius kobayashii*, represents the oldest record of the genus and also of the extant ziphiid genera in the world. *Berardius kobayashii*, sp. nov., has following generic characters of *Berardius* such as the gently elevated vertex, the long and slender anterior process of the periotic. *Berardius kobayashii*, sp. nov., also has following unique characters: it is extremely small in size, apices of the left and right hamular process of the pterygoids contact medially, forming together a posteriorly directed medial point. *Berardius kobayashii*, sp. nov. fills the gap between the origin of the genus and later diversifications of the extant species in the genus. Furthermore, it suggests that *Berardius* species had originated in the North Pacific including the Sea of Japan at the time at latest 12 Ma. The discovery will also be a key to elucidate the process of their habitat expansion as the cosmopolitan genus among the Ziphiidae. Based on the distributional patterns of the fossil and extant species of the genus suggest that the western North Pacific including the Sea of Japan may have been one of the areas for the evolution and radiation of the extant genera among the crown Ziphiidae.

2. Introduction

The family Ziphiidae of the cetartiodactyl clade Odontoceti (Cetacea) are represented by 23 extant species [1, 2] and almost the same number of extinct species [3, 4]. Although the extant ziphiids have cosmopolitan distribution, they are one of the most mysterious groups of whales because of their deep-diving behavior. The Ziphiidae is thought to be originated and diversified in the Southern Ocean, and the ancestors of these deep-diving whales initially lived in the epipelagic and/or neritic oceans [5]. But, their adaptation to and diversification in the deep-sea have still been uncertain.

[revised manuscript text omitted]

The Aikawa, Sanze and Kyozyukasan Formations, which are stratigraphically lower units in the Neogene of the Kosado mountain range, are terrestrial formations and mainly consisted of andesites and dacite lavas [17, 20, 21]. The early Miocene Sanze and Kyozyukayama Formations are unconformably overlain by the Orito Formation [20]. The Orito Formation is the shallow marine sediments composed of conglomerates, sandstone and mudstone [10, 20]. The Orito Formation yields fossils of large foraminifers, echinoderms, mollusks and marine mammals [9, 22], and its age is estimated to be late early Miocene, ca. 17.5-17.3 Ma, based on marine diatom fossils [10, 19]. The Tsurushi Formation is deep sea sediments overlying the Orito Formation, and it consists of mainly hard siliceous mudstone. The Notayama and Yamadagawa Formations consist of diatomaceous mudstones. According to the diatom biostratigraphy, these two formations are late Miocene in age [19].

2.2 The Tsurushi Formation

The Middle Miocene Tsurushi Formation was originally defined by Watanabe [21] based on the hard siliceous mudstone exposed at Tsurushi in Sawada Town on the Osado mountain range. The same or similar rock facies has been distributed also on the Kosado mountain range and was recognized as the same Formation [23]. The Tsurushi Formation distributed on the Kosado mountain range is composed of hard siliceous mudstone and the contemporaneous basalt (i.e., Ogi Basalt Member) distributed regionally on the Ogi Peninsula [19]. The geologic age of the Ogi Basalt Member is estimated to be 14.1–11.7 Ma [10]. Watanabe [7] identified the foraminiferal fossils from the Tsurushi Formation on the Osado mountain range, which indicate the PF4 Zone of Maiya [24], and it corresponds to 12.3–11.5 Ma [10, 19]. Kawatani et al. [25] also confirmed the age of the Tsurushi Formation on the Kosado mountain range that produced whale fossils based on the radiolarian fossils, which corresponds to 15.3–9.1 Ma. Based on these chronological information, the geologic age of the Tsurushi Formation is estimated to be approximately 12.3–11.5 Ma. Studies of foraminiferal fossils have suggested that the Tsurushi Formation was deposited in the deep-sea environment [7, 9].

As mentioned above, various megafossils have been reported from the Tsurushi Formation; for example, mollusks, fish, a sea turtle, birds, and marine mammals [9, 11, 26]. In particular, a lot of marine mammal fossils such as whales, pinnipeds and desmostylians have been found in calcareous nodules from the Tsurushi Formation on the Kosado mountain range, and these fossils were briefly reported and summarized by Sado Research Group for Marine Mammalian Fossils [11], Tazaki et al. [13], Horikawa et al. [12], Takahashi et al. [14] and Barnes and Hirota [15].

4. Materials and Methods

The specimen described here was initially reported by Takahashi et al. [14]. At that time the specimen was temporarily deposited at Niigata University, and now it is permanently stored at the Sado Museum, Sado City, Niigata Prefecture, Japan, where it bears the catalog number SCM 5530-6. This specimen had been enclosed in calcareous hard mud stone matrices. Accordingly, it was prepared by acid etching technique using formic acid diluted with concentration of about 4.5–5.0% in water, which allowed the extraction of fragile bone elements of the skull, partial mandible and ear bones from hard matrices. However, the periotic has been fused to the skull loosely by its fan-shaped posterior process, and the dorsal surface of the periotic is invisible entirely from the ventral aspect. So we reconstructed the 3D image of the periotic by the micro-Computed Tomographic (μ CT) scanner of the Microfocus CT, TXS320-ACTIS, at the National Museum of Nature and Science, Tokyo, Japan.

We then compared them osteologically and taxonomically with both fossil and extant species of ziphiids stored at the following institutions and the literature cited. Definitions of measured portions and morphological terms follow Mead and Fordyce [27] and Kasuya [28]. The characters for comparisons and the phylogenetic analysis follow Bianucci et al. [3].

Institutional abbreviations are the following: NMNS-PV, Department of Geology and Paleontology, National Museum of Nature and Science, Tsukuba, Japan; NSMT-M, Department of Zoology, National Museum of Nature and Science, Tsukuba, Japan; SCM, Sado Museum, Sado City, Niigata, Japan; TMNHN, Togakushi Museum of Natural History, Nagano, Japan.

5. Systematics

Order CETARTIODACTYLA Montgelard, Catzefis and Douzery, 1997

 Infraorder CETACEA Brisson, 1762

 Parvorder ODONTOCETI Flower, 1867

 Family ZIPHIIDAE Gray, 1850

 Subfamily BERARDIINAE Moore, 1968

Emended Diagnosis of subfamily: Two pairs of well-developed teeth on the mandible; the nodular protuberance formed by the interparietal or the frontals on the vertex; the narrow and thin premaxillary crest; the supraoccipital is lower than the frontals on the vertex; the part between dorsal

surfaces of the nasal bones and ends of the premaxillaries rises above the supraoccipital (modified from [29]).

Genus *Berardius* Duvernoy, 1851

Emended diagnosis of genus: The ratio between the rostral length and the condylobasal length is between 0.63–0.70; the mesorostral groove is empty; the mesorostral groove opens for the proximal third of the rostrum; the premaxillary basin is absent; asymmetry of the premaxillary sac fossae absent or weak; the premaxillary sac fossa is not laterally overhanged on the maxilla; deep excavation of the premaxillary sac fossa is absent; the ascending process of the premaxilla in lateral view is slightly concave; constriction on the ascending process of the right premaxilla is roughly absent; the vertex is gently elevated; the premaxillary crest is transversely directed; the ratio between the width of the premaxillary crests and the width of the premaxillary sac fossae is 1.0 to 1.25; the distance between the left and right premaxillary crests is large; the nasal elongation ratio between the length of the medial suture of the nasals and the maximum width of the nasals is large; the anteromedial excavation of the dorsal surface of the nasal is absent; the premaxillary crest present on the vertex; inclusion of the nasal in the premaxillary crest is absent; contact between the nasal and the premaxillary crest is extended more than half the length of the nasal to almost the whole length; the premaxillary foramen is located roughly at the level of the antorbital notch; there is no excrescences on the dorsal surface of the maxilla at the posterior half of the rostrum; the interparietal or the frontal is present as an isolated rounded protuberance on the posterior part of the vertex; the posterolateral narrowing of the nasals and frontals is present at the vertex that is narrower than the nasals; the anteromedial margin of the supraoccipital is distinctly lower than the dorsal margin of the vertex; the extremely ossified trapezoidal vertex is absent; the angle formed by the basioccipital crests is larger than 50° in ventral view; the hamular fossa of the pterygoid sinus is wide, extending anteriorly on the palatal surface of the rostrum; the apices of the left and right hamular processes of the pterygoids contact medially; the apex of the hamular process is excavated by the fossa for the hamular lobe of the pterygoid sinus; the anteroventral wall of the tympanic bulla is narrow and triangular; the anterior spine of the tympanic with a more or less rectilinear anterior margin; the posteroventral corner of the sigmoid process of the tympanic is posteriorly projected in lateral view; the dorsal margin of the involucrum of the tympanic that is cut by an indentation is present and visible in medial and/or dorsal view; the periotic has long and slender anterior and posterior processes; the posterior bullar facet of the periotic is fan-shaped (modified from [3], [29], [30], [31]).

Berardius kobayashii, sp. nov.

(Skull: Figure 3-7; Ear bone: Figure 8-10)

Berardius sp. indet. Takahashi et al. 1989

URN: lsid:zoobank.org:act: 317C1452-AE2E-4A08-A473-3B507B452D2C

Diagnosis of species: Extremely small in size among the four species of *Berardius* (60–70 % smaller than *B. bairdii* and *B. anuxii*, 50% smaller than *B. minimus*); the transverse diameter of the premaxilla is relatively wide; the lateral margin of the premaxilla is curved; the zygomatic process of the squamosal is anterodorsally tapered and triangle in shape; the nasals are short; there is a tiny anteromedial excavation on the dorsal surface of the nasals.; the apices of the left and right hamular process of the pterygoids contact medially, forming together a posteriorly directed medial point.

Holotype: SCM 5530-6, incomplete skull with a left tympanic bulla and periotic, lacking the rostral portion; a posterior one fourth of the horizontal ramus of the left dentary with the condyloid process and mandibular condyle. Collected by Kiyoei Kaneko in or before 1969.

Type Locality: Sobama Beach, Ogi Town, Sado Island, Niigata Prefecture, Japan. Geographic coordinates: 37° 84'50" N, 138° 26'58" E.

Formation and Age of Holotype: SCM 5530-6 was collected from the middle to late Miocene Tsurushi Formation. As discussed above, the geologic age of the Tsurushi Formation is estimated to be approximately 12.3–11.5 Ma.

Etymology: The species is named in honor of the late Dr. Iwao Kobayashi, a professor emeritus at Niigata University, for his longstanding contributions to the geology and paleontology of the northern Fossa Magna region including Niigata Prefecture, and in gratitude for his permission to re-study the marine mammal fossils collected by him and his colleagues from the Sado Island and encouragement to both of us throughout this study.

Description

Cranium (general morphology)

The cranium is relatively small (Table 1) and **less asymmetry** (Figure 3,5). The ratio between the height and width of the skull is 1:1.19. The rostral and the occipital parts are **almost broken** and missing ~~away~~. The small part of the left mandible ~~had~~ attached to the cranium. The vertex is moderately elevated and slightly skewed toward the left. **The extremely ossified trapezoidal vertex** is almost broken. The **temporal fossa is isosceles triangle in shape** that converges anteriorly, and the posterior margin is weakly rounded. The temporal fossa is shallow and **concave**. The sutures of the skull are still visible but tightly closed, indicating that the individual is an adult.

Premaxilla

The premaxilla is **almost flat**. The premaxillary crest is present on the vertex. The medial margin between the left and right premaxillae **is very close**. Contact between the nasal and the premaxillary crest is extended along ~~the almost~~ whole length of the nasal. The premaxillary sac fossae are weakly asymmetrical and do not overhang the maxillae laterally. The premaxillary sac fossa is almost flat and not deeply excavated. The ascending process of the premaxilla gradually rises toward the vertex. It is slightly concave in lateral view. The ascending process of the left premaxilla is somewhat damaged, but that of the right premaxilla is well preserved and **moderately constricted just in front of the posterior end of the premaxillary crest**. The vertex is gently elevated. In anterior view, lateral expansion of the premaxillae at the **portion** of the premaxillary sac fossae is prominent. The premaxillary crest is transversely oriented, and it is narrow in **width**. The distance between the premaxillary crests is wide. The **anterolateral margin** of the bony nares is V-shaped. The small premaxillary foramina are located roughly at the level of the antorbital notch and slightly posterior to the **infraorbital foramina on the premaxillae**. There are distinct grooves anterior to the foramina. **The angle between the dorsal surface of the rostrum and the ascending process of the premaxilla is 150°**.

Maxilla

The maxilla is broad and its **margin** is curved. The **rostral maxillary crest** is present and moderately elevated on the maxilla. The **infraorbital foramina** open anterodorsally on the maxillae near the base of the rostrum, and the largest foramen is located on the right maxilla. There are two **maxillary foramina** on the left maxilla ~~and are~~ 13 mm and 7 mm ~~in diameter, respectively~~. There is one large **maxillary foramen** on the right maxilla ~~and has~~ a diameter ~~reaching~~ 16 mm. The maxilla is slightly elevated at the posterior side of these foramina. ~~The~~ **excrescence** on the dorsal surface of the maxilla ~~is absent~~ on the ~~half~~ of the rostrum. **The prominent notch and related structure are low**.

Nasals

The nasal is short, and its dorsal outline is trapezoidal. **The shape of the bony nares is triangular in outline, and the anteroposterior diameter (67 mm) of the bony nares is wider** than the transverse diameter (39 mm). Although the posterior margin of the right nasal is broken, it is slightly broader than the left one. The dorsal surface of the nasal is roughly flat, but there is a weak anteromedial excavation ~~on its dorsal surface~~. The nasal is not included in the premaxillary crest. **The nasal contacts the premaxilla laterally for almost the whole length**. The posterior nasal-frontal suture is posteriorly convex.

Frontal

The frontal is fully covered by the maxilla in dorsal view and only visible its vertical section beneath
the overlapping maxilla at the portion of the orbital and temporal fossae in lateral view. It is
dorsoventrally thick and relatively dense but not osteosclerotic nor pachyosteosclerotic.

**Supraoccipital**

The supraoccipital is almost broken away. The anteromedial margin of the supraoccipital is
distinctly lower than the dorsal margin of the vertex. The supraoccipital protuberance protrudes
posteriorly.

**Pterygoid**

The pterygoid is dorsoventrally deep and excavated by the anteroposteriorly and dorsoventrally
wide hamular fossa. There are distinct transverse crests on the pterygoid. The hamular fossa of the
pterygoid sinus is wide, extending anteriorly on the palatal surface of the rostrum. Apices of the left
and right hamular processes are incomplete, but both processes contact medially and form together a
posteriorly directed medial point.

**Squamosal**

The left zygomatic process of the squamosal is preserved. It is short anteroposteriorly compared to
its dorsoventral height. The zygomatic process of the squamosal is well developed and anterodorsally
tapered with triangle outline, but its anterior margin is crescent in shape.

**Mandible**

The mandibles are almost broken away, but the ascending ramus of the left mandible with the
coronoid process and the mandibular condyle is partially preserved (Figure 7). The posterior margin
of the angular process is also broken away. The mandibular body is very thin and fragile at the
portion of the lateral wall of the mandibular foramen. Its outer or lateral surface is smooth and
slightly swelled laterally, forming the condyloid crest. Its inner or medial surface (or the lateral wall
of the mandibular foramen) is relatively rough. The mandibular foramen is anteroposteriorly long and
dorsoventrally wide. The coronoid crest is well developed as an anteroposteriorly extended ridge and
continues to the coronoid process at the posterodorsal end of the mandible. The ventral margin of the
coronoid crest is weakly concave along the crest. The mandibular condyle protrudes posteriorly and
locates almost the same position at the end of the coronoid process. It is dorsoventrally long and
laterally crescentic, and it is slightly swelled posteriorly. There is a notch between the coronoid
process and the mandibular condyle but is absent between the mandibular condyle and the angular
process, which is almost broken though.

**Tympanic Bulla**

The left tympanic bulla is preserved (Table 2). The anterior part of the ventral wall of the tympanic
bulla is cylindrical and is lacking its anterior spine. The involucrum is shorter than the lateral wall.
The dorsal margin of the involucrum is cut by an indentation. The inner and outer posterior
prominences are thick and short, and they swell well. The outer posterior prominence is much
swelled than the inner one. The median furrow runs anteroposteriorly on the ventral wall, and it is
transversely narrow and anteriorly shallow. The interprominental notch is relatively deep at the
posterior end of the median furrow. The lateral furrow is shallow, and its posterior ridge is prominent
and twisted on the lateral wall of the tympanic bulla. The sigmoid process is square and rather thin.
The lateral border of the sigmoid process is twisted posteriorly. A damaged base of the posterior
process lies on the posterolateral end of the sigmoidal process. The elliptical foramen is present
(Figure 9,10).

**Periotic**

The left periotic is preserved (Table 2). It is relatively slender in shape. The dorsal surface of the
body is smooth. The anterior and posterior processes are almost strait anteroposteriorly in ventral
view. The anterior process is long and slender, and its apex is pointed (Figure 8A). It is weakly bent
ventrally. The anterior bullar facet is weakly excavated anteroposteriorly. A rounded accessory

ossicle of the tympanic bulla is still articulated with the fovea epitubaria at the posteromedial base of the anterior process. The malleolar fossa is excavated posterior to the accessory ossicle (~~in the fovea epitubaria~~) and opens posteriorly. It is laterally delimited by a mediolaterally elongated lateral tuberosity. The cochlear portion (=pars cochlearis) is rounded in outline, relatively low, and it is slightly bent anteroventrally with a weak depression at the anteromedial surface, so the anterior incisure is narrow and slightly covered with the posterior edge of the cochlear portion. The epitympanic hiatus is wide. The internal acoustic meatus containing the central foramen (=spiral cribriform tract) and the foramen singulare is separated by a developed transverse crest from the proximal opening of the facial canal. The central foramen is relatively small and almost circular, slightly compressed transversely. Its posterolateral margin is elevated and reaches almost the top of the cochlear portion. The foramen singulare opens near the aperture for the vestibular aqueduct (Figure 8B). The proximal opening of the facial canal is relatively large and elliptical in shape. The aperture for the cochlear aqueduct is large and semicircular. It is located close to the posterior margin of the internal auditory meatus. It is also relatively close to the rounded aperture for the vestibular aqueduct (=endolymphatic duct) that is elliptical in shape. The fenestra rotunda is relatively large and semicircular in outline. The posterior process is relatively long posteriorly and inflated mediolaterally. It is fused to the skull by the mediolaterally narrow periotic fossa of the skull. The posterior bullar surface is covered by the fused fragment of the posterior process of the tympanic bulla. It is slightly bent posterolaterally at the base just posterior to the cochlear portion. Because of this fusion of the posterior process of the tympanic bulla and its posterior expansion, the portion of the posterior bullar facet is posteriorly flared and fan-shape in ventral view (Figure 8A). Although mostly covered by the remnant of the posterior process of the tympanic bulla, the surface of the posterior bullar facet is smooth. The epitympanic hiatus is relatively deep.

6. Comparisons

Takahashi et al. [14] identified SCM 5530-6 as belonging to the Zhiphiidae based on the following characters: the bony nares are located posteriorly; the rostrum is moderately wide; the occipital slopes relatively steeply posterodorsally; and the cranium is not strongly asymmetrical. Furthermore, they compared SCM 5530-6 with extant genera of the Zhiphiidae; *Zhiphius*, *Berardius*, and *Mesoplodon*, and identified SCM 5530-6 as a species of the genus *Berardius* ~~among them~~ based on the following characteristics: the bony nares is triangular in outline; the vertex is elevated gently; and the median part of the palatine is convex on the ventral side ([14], p.104). But they also pointed out that the transverse diameter of the premaxilla of SCM 5530-6 is relatively wider than the then extant two species of *Berardius*, and the lateral margin of the same portion is curved in contrast to the straight condition of this portion in the extant species of *Berardius*. In addition, the morphology of the zygomatic process of the squamosal is shaped like an anterodorsally tapered triangle in contrast to an anteriorly squared rectangle in the extant species of *Berardius*.

We farther compared the characteristics of SCM 5530-6 with both extant and fossil Zhiphiidae including 28 genera and 55 species based mainly on Bianucci et al. [3] and Lambert et al. [31]. Although the left side of the premaxilla of SCM 5530-6 is broken, the ratio between the width of the premaxillary crest and that of the premaxillary sac fossa of SCM 5530-6 is moderate, from 1.0 to 1.25 (Char. 11 of [3]). The infraorbital foramina open anterodorsally on the maxilla near the base of the rostrum. In general, a cluster of the infraorbital foramina is present on the base of the rostrum in the early diverging zhiphiids [3]. On the other hand, only a large foramen is present in the crown zhiphiids [3]. SCM 5530-6 has a large foramen on the right maxilla, and two relatively large foramina on the left maxilla, and therefore, the condition of these foramina is thought to be the condition seen in the later diverging, extant zhiphiids.

The supraoccipital of SCM 5530-6 is lower than the frontals on the vertex, and the vertex is gently elevated (Chars. 9 and 18 of [3]). A similar low elevation of the ascending process is observed in *Berardius* and related fossil species such as *Archaeozhiphius microglenoideus* and *Microberardius africanus* [3, 29, 33].

Archaeoziphius differs from *Berardius* by the following characters: the overall size of the skull is much smaller; the nasal is included in the premaxillary crest for a short distance along the posteromedial angle of the crest [33]; Chars. 15 and 51 of [3]. However, the inclusion of the nasal in the premaxillary crest is absent on SCM 5530-6 as in the species of *Berardius*.

Microberardius differs from *Berardius* by the following characters: the size of the skull is much smaller; the premaxillary crest is much narrower; the vomer at the anterior portion of the mesorostral groove is thickening; the rostrum is higher than wider along the most of its length; the base of the rostrum is narrower and the maxillary crest is not extended at the base of the rostrum [29]. By contrast, SCM 5530-6 does not have the narrow premaxillary crest.

Consequently, SCM 5530-6 differs from both *Archaeoziphius* and *Microberardius*. In addition, the anterior process of the periotic is slender, and its apex is long and pointed, and the posterior process is long and slender. These characters are potential synapomorphies of the genus *Berardius* (see also [28, 34]). Therefore, SCM 5530-6 is definitely identified as a species of the genus *Berardius* as was already pointed out by Takahashi et al. [14].

However, SCM 5530-6 has some additional characters differing from the extant, now three species of *Berardius*, such as *Berardius bairdii*, *B. arnuxii* and *B. minimus*, which was not mentioned by Takahashi et al. [14]. For instance, the nasal of SCM 5530-6 is shorter than *B. bairdii*, *B. arnuxii* and *B. minimus*. Also, there is a tiny anteromedial excavation on the dorsal surface of the nasal, which is not observed in other three species of *Berardius*. The apices of the pterygoids contact medially in all the species of *Berardius*, and these are forming together a posteriorly directed medial point in SCM 5530-6, but these are less excavated and U-shaped posterior margin in other three species of *Berardius*. The last character is not seen in any species of both fossil nor extant ziphiids including the three extant species of *Berardius*. But, the hamular fossa of the pterygoid sinus is wide, extending anteriorly on the palatal surface of the rostrum as seen in the extant *Berardius* and deep diving whales [35, 36]. The height of the skull relative to the width of SCM 5530-6 is 84% and slightly higher than the extant species of *Berardius*; i.e., 83% in *B. minimus* and 77% in *B. bairdii*. The skull of SCM 5530-6 has much tighter sutures compared with those of extant *Berardius bairdii* and *B. arnuxii*. As has pointed out by Yamada et al. [2], the skull of *B. minimus* also has much tighter sutures, but SCM 5530-6 has much tighter sutures than that of *B. minimus*. The temporal fossa of SCM 5530-6 is shallow like *B. minimus*, but the lateral wall of the fossa is concave like *B. bairdii* and *B. arnuxii*. The lateral expansion of the premaxillae at the posterior halves is prominent like *B. minimus*. The prominent notch and related structure are much lower, less distinct and less rugged than *B. minimus*. The greatest postorbital width of SCM 5530-6 is almost 60 to 70% smaller than the extant *B. bairdii* and *B. arnuxii* and 50% smaller than *B. minimus*. The tympanic bulla and the periotic of SCM 5530-6 closely resembles that of *B. bairdii*. But, the total length of the tympano-periotic bone is 40 to 50% smaller than that of *B. bairdii*, the total length of tympanic bulla is 30% smaller than that of *B. minimus*. Consequently, the length of the tympano-periotic bone relative to the skull is larger than other *Berardius* species. As results of the above morphological comparisons, the new ziphiid fossil SCM 5530-6 from the middle Miocene Tsurushi Formation is undoubtedly identifiable as a species of the genus *Berardius*. In addition, comparisons with all the known species within the genus; i.e., the extant three species that are presently distributed in the northern North Pacific (i.e., *B. bairdii* and *B. minimus*) and the Antarctic (i.e., *B. arnuxii*), SCM 5530-6 is distinguishable from all the three extant species based on the above mentioned cranial characters: i.e., it is extremely small in size; the nasal is short; and the apices of the left and right hamular process of the pterygoids contact medially, forming together a posteriorly directed medial point. Furthermore, the size of the new fossil is extreme among all the known species of the genus; i.e., 50% smaller than *B. minimus* and surprisingly 60 to 70% smaller than both *B. bairdii* and *B. arnuxii*. Consequently, identification of SCM 5530-6 as a new species within the genus *Berardius*, i.e., *Berardius kobayashii*, is warranted.

7. Discussion

Interestingly, despite its extremely small size in *B. kobayashii* that is smaller than the smallest extant species *B. minimus*, the cranial morphologies of *B. kobayashii* are much closer to the larger

species, *B. bairdii* and *B. arnuxii* as discussed in the “comparisons” section. It suggests that although *B. minimus* is represented only by the extant species, its emergence could have been much earlier than the emergence and diversification of the rest of three species including the northern North Pacific middle Miocene extinct *B. kobayashii* sp. nov. and the extant *B. bairdii* and also the Antarctic *B. arnuxii*. Consequently, their emergence and diversification must have occurred in or by the middle Miocene and potentially in the western North Pacific based on the new fossil from the middle Miocene Tsurushi Formation described here and the present distribution of two extant species of *Berardius* in the North Pacific (see also [2]).

For now, three genera are recognized in the subfamily Berardiinae; i.e., *Microberardius* from the late Miocene of South Africa [29], *Archaeoziphius* from the middle Miocene of Belgium [33] and *Berardius* from the middle Miocene to the Recent of the northern North Pacific and the Antarctic oceans [1, 2, this study]. Previously, there was no fossil record of the berardiines in the North Pacific Ocean. So, the new fossil described here reveals that the berardiine ziphid was already existed at least in the western North Pacific at latest in the late middle Miocene (ca. 12 Ma). It also suggests that the emergence of the berardiine ziphids goes back to the early middle Miocene or much earlier because their sister taxon *Archaeoziphius* in the North Atlantic was already existed in the early middle Miocene [33]. At the moment, it is difficult to consider more precise origin of the Berardiinae based on the present fossil evidence, but its potential origin might have been in the Southern Hemisphere on the basis of recent studies on their phylogenetic relationships and phylogeographic distributions of sister and out groups for the Berardiine ziphiids [3, 5, 33]. However, all these evidence suggest the rapid diversification of the subfamily Berardiinae in the middle Miocene in the Northern Hemisphere (Figure 11).

8. Conclusions

An extinct beaked whale from the late middle Miocene Tsurushi Formation (ca. 12 Ma) in the Sado Island, Niigata Prefecture, Japan, was described as a new species of the extant genus *Berardius*; i.e., *B. kobayashii*. *Berardius kobayashii*, sp. nov., is the fourth species of the genus. Namely, the lineage leading to the extant genus of the ziphiids had already lived in the western North Pacific at the time of at latest 12 Ma. The western North Pacific including the Sea of Japan may, therefore, have been one of the areas for the evolution and radiation of *Berardius* species. The existence of a species of the extant ziphiid genus that have been known as one of the deep-diving whales from the middle Miocene deep-sea deposits will become the key to elucidate the process of their habitat expansions as deep-sea dwellers among the Ziphiidae.

Acknowledgments

We would like to thank the late I. Kobayashi (Professor Emeritus at Niigata University), A. Matsuoka (Niigata University), M. Shimizu (Asahimachi Museum, Niigata University), M. Ikarashi (Sado Museum), M. Aida (Sado Geopark Promotion Office and Sado Museum), K. Narita (Nagano City Museum), M. Kato (Nagaoka Municipal Science Museum), Y. Tajima and T.K. Yamada (NMNS), for allowing access to the specimens and giving me the opportunities to observe specimens at respective museums; K. Takahashi (Biwako Museum), H. Ichishima (Fukui Prefectural Dinosaur Museum), K. Nagasawa (formerly Yamagata Prefectural Museum), T. Kimura (Gunma Museum of Natural History), for providing useful discussion and advice with the literature on fossil ziphiids.

Funding Statement

A.K. was funded by the travel fund for the 2017 fiscal year by the Regional Promotion Division of Sado City.

Data Accessibility

This study and the nomenclatural acts it contains have been registered in Zoobank. The LSID for this publication is: urn:lsid:zoobank.org:pub: 7BEDEBA0-46B9-4125-A7F8-2169EB016C29.

Competing Interests

We declare we have no competing interests.

Authors' Contributions

A.K. and N.K. conceived and designed the project, performed the experiment and analysed the data. A.K. prepared SCM 5530-6. N.K. contributed reagents/materials/preparation tools/analysis tools. A.K. and N.K. wrote the paper.

References

1. Rosel P E, Archer FI, Baker CS, Boness DJ, Brownell RL, Churchill M Jr., Costa AP, Domning DP, Fordyce RE, Jefferson TA, Kinze C, Oliveira LR, Perrin WF, Wang JY, Yamada TK. 2019 List of marine mammal species and subspecies. *Committee on Taxonomy, The Society for Marine Mammalogy*.
2. Yamada TK, Kitamura S, Abe S, Tajima Y, Matsuda A, Mead JG, Matsuishi TF. 2019 Description of a new species of beaked whale (Berardius) found in the North Pacific. *Scientific reports* **9(1)**, 1-14.
3. Bianucci G, Di Celma C, Urbina M, Lambert O. 2016 New beaked whales from the late Miocene of Peru and evidence for convergent evolution in stem and crown Ziphiidae (Cetacea, Odontoceti). *PeerJ*. **4**, e2479.
4. Lambert O, de Muizon C, Duhamel G, Van Der Plicht J. 2018 Neogene and Quaternary fossil remains of beaked whales (Cetacea, Odontoceti, Ziphiidae) from deep-sea deposits off Crozet and Kerguelen islands, Southern Ocean. *Geodiversitas* **40(2)**, 135-140.
5. Lambert O, Collareta A, Landini W, Post K, Ramassamy B, Di Celma C, Urbina M, Bianucci G. 2015 No deep diving: evidence of predation on epipelagic fish for a stem beaked whale from the Late Miocene of Peru. *Proc. R. Soc. B*, 282.
6. Niigata Foraminiferal Research Group. 1967 Some foraminiferal assemblages from the Sawane area, Sado Island (Preliminary report): Restudy on the stratigraphy of the "Sawane Formation" based on the foraminiferal fossils. *Commemorative Volume for Mr. Y. Hiramatsu*, 115-119.
7. Watanabe K. 1987 Tertiary Foraminifera and Radiolaria Fossils from Sado Island, Central Japan. *Publications from the Sado Museum* **9**, 127-156.
8. Okamoto Y, Satou M, Watanabe M, Yamamoto H. 1992 Inversion Tectonics in the southeastern part of the Japan Sea. *Jour. Struct. Geol. Japan* **38**, 47-58.
9. Kobayashi I. 2001 The Cenozoic of Niigata prefecture, the case of Sado Island (Niigata no shinseikai Sado-hen). *Niigata Geotechnology Society* **57**, 9-21.
10. Yanagisawa Y, Watanabe M. 2017 Revised lithostratigraphy of the Neogene sedimentary sequence in the southern part of the Osado Mountain area, Sado Island, Niigata Prefecture, Japan. *Bull. Geol. Surv. Japan* **68 (6)**, 259-285.
11. Sado Research Group for Marine Mammalian Fossils. 1987 Fossil Cetacean from Sado Island, Central Japan. *Publications from the Sado Museum* **9**, 211-217.
12. Horikawa H, Tazaki K, Kanno T. 1987 Fossil Ziphiidae from Koshiji-Syo off Sado Island, Central Japan. *Publications from the Sado Museum* **9**, 225-230.
13. Tazaki K, Horikawa H, Miyazaki S. 1987 Fossil Ziphiidae from Hyotan-Guri off Sado Island, Central Japan. *Publications from the Sado Museum* **9**, 219-223.
14. Takahashi K, Nomura M, Kobayashi I. 1989 A fossil cetacean skull (Berardius sp. indet.) from Dogama, Ogi-machi, Sado Island, Central Japan. *Earth Science; Chikyu Kagaku* **43**, 102-105.
15. Barnes LG, Hirota K. 1995 Miocene pinnipeds of the otariid subfamily Allodesminae in the North Pacific Ocean: Systematics and relationships. *The Island Arc* **3**, 329-360.
16. Utashiro T. 1950 The stratigraphy of Sawane-Aikawa region in Southwest Sado Island (Osado Mountain range). *Journal of the Geological Society of Japan* **56**, 302-303.
17. Shimazu M, Kanai Y, Toyama T, Ichihashi K, Minakawa J, Takahama N. 1973 Structural development and igneous activity in the Sado island. *Memoirs of the Geological Society of Japan* **9**, 147-157.
18. Shimazu M, Toyama T. 1982 Neogene volcanic rocks of the Sado islands. *Journal of the Geological Society of Japan* **88**, 381-400.
19. Yanagisawa Y. 2012 Late Miocene diatoms in the Hamochi area, Sado Island, Niigata Prefecture, Japan. *Open-File Report of the Geological Survey of Japan, AIST* **568**, 1-19.
20. Ogi Collaborative Research Group. 1986 Late Cenozoic Group in the Southern part of Kosado Mountains, Niigata prefecture. *Earth Science* **40**, 417-436.
21. Takeuchi K, Ozaki M, Komatsubara T. 2011 Explanatory notes of 1:200,000 geological map of the coastal zone around Niigata. Digital Geoscience Map, Seamless Geoinformation of Coastal Zone "Coastal Zone around Niigata".
22. Tokunaga S. 1939 A New Fossil Mammal Belonging to Desmostyliidae. *Jub. Pub. Com., Professor H. Yabe, M.I.A. Sixtieth Birthday*. Sendai: Institute of Geology and Paleontology, Tohoku Imperial University. 289-299.
23. Watanabe H. 1932 Neogene (Shin Daisankei). *Nihon Chishitsu kousan shi* 92-152.
24. Maiya S. 1978 Late Cenozoic planktonic foraminiferal biostratigraphy of the oil-field region of Northeast Japan. *Cenozoic Geology of Japan*, 35-60.
25. Kawatani A, Sashida K, Sachiko A, Kohno N. 2019 Radiolarian fossils and estimated depositional age of the Miocene Tsurushi Formation distributed in Sado Island, Niigata Prefecture, Japan. *Bull. Geol. Surv. Japan* **70(1-2)**, 91-99. (doi: <https://doi.org/10.9795/bullgsj.70.91>)
26. Ono K, Ueno T. 1985 Tertiary Vertebrates from Sado Island, Niigata Prefecture, Central Japan. *Memoirs of the National Science Museum* **18**, 65-71.
27. Mead JG, Fordyce RE. 2009 The therian skull: a lexicon with emphasis on the odontocetes. *Smithsonian Contributions to Zoology*, 1-249.
28. Kasuya T. 1973 Systematic consideration of recent toothed whales based on the morphology of tympano-periotic bone. *Scientific Reports of the Whales Research Institute* **25**, 1-103.
29. Bianucci G, Lambert O, Post K. 2007 A high diversity in fossil beaked whales (Mammalia, Odontoceti, Ziphiidae) recovered by trawling from the sea floor off South Africa. *Geodiversitas* **29(4)**, 561-618.
30. Bianucci G, Lambert O, Post K. 2010 High concentration of long - snouted beaked whales (genus Messapicetus) from the Miocene of Peru. *Palaeontology* **53(5)**, 1077-1098.
31. Lambert O, De Muizon C, Bianucci G. 2013 The most basal beaked whale *Ninoziphius platyrostris* Muizon, 1983: clues on the evolutionary history of the family Ziphiidae (Cetacea: Odontoceti). *Zoological Journal of the Linnean Society* **167(4)**, 569-598.
32. Miwa M, Yanagisawa Y, Yamada K, Irizuki T, Shoji M, Tanaka Y. 2004 Planktonic foraminiferal biostratigraphy of the Pliocene Kuwae Formation in the Tainai River section, Niigata Prefecture and the age of the base of the No. 3 Globorotalia inflata bed. *Journal of the Japanese Association for Petroleum Technology* **69(3)**, 272-283.
33. Lambert O, Louwye S. 2006 *Archaeoziphius microglenoideus*, a new primitive beaked whale (Mammalia, Cetacea, Odontoceti) from the Middle Miocene of Belgium. *Journal of Vertebrate Paleontology* **26(1)**, 182-191.
34. Bianucci G. 1997 The Odontoceti (Mammalia Cetacea) from Italian Pliocene. The Ziphiidae. *Palaeontographia italica* **84**, 163-192.
35. Reidenberg JS, Laitman JT. 2008 Sisters of the sinuses: cetacean air sacs. *The Anatomical Record: Advances in Integrative Anatomy and Evolutionary Biology* **291(11)**, 1389-1396.
36. Ramassamy B, Lambert O, Collareta A, Urbina M, Bianucci G. 2018 Description of the skeleton of the fossil beaked whale *Messapicetus gregarius*: searching potential proxies for deep-diving abilities. *Mitteilungen aus dem Museum für Naturkunde in Berlin. Fossil Record* **21(1)**, 11-32.
37. Akiba F. 1986 Middle Miocene to Quaternary diatom biostratigraphy in the Nankai Trough and Japan Trench, and modified Lower Miocene through Quaternary diatom zones for middle-to-high latitudes of the North Pacific. In Kagami, H., Karig, D. E., Coulbourn, W. T., et al., *Initial Reports of the Deep Sea Drilling Project, U. S. Government*

Printing Office, Washington D. C. **87**,
 393–480.

38. Yanagisawa Y, Akiba F. 1998 Refined
 Neogene diatom biostratigraphy for the
 northwest Pacific around Japan, with an

introduction of code numbers for
 selected diatom biohorizons. *Jour. Geol.
 Soc. Japan* **104**, 395–414.

Tables

Measurements (in mm) of the skull of *Berardius kobayashii*

Skull

Condylobasal length from tip of rostrum to hindmost margin of occipital condyles.	236
Greatest postorbital width.	245
Greatest width of bony nares.	38
Maximum width of premaxillary sac fossae.	87
Maximum width of right premaxillary sac fossa.	43
Maximum width of left premaxillary sac fossa.	38
Maximum width of nasals.	40
Greatest length of left posttemporal fossa, measured to external margin of raised suture.	87
Greatest width of left posttemporal fossa at right angles to greatest length.	61
Major diameter of left temporal fossa proper.	38
Minor diameter of left temporal fossa proper.	33
Distance from foremost end of junction between nasals to hindmost point of margin of supraoccipital crest.	33
Length of left orbit from apex of preorbital process of frontal to apex of postorbital process.	51
Greatest width of internal flares.	91
Greatest length of left pterygoid.	109

Measurements (in mm) of the ear bone of *Berardius kobayashii*

Tympanic bulla

Standard length of tympanic bulla, distance from anterior tip to posterior end of outer posterior prominence.	37
Distance from anterior tip to posterior end of inner posterior prominence.	34
Distance postero-ventral tip of outer posterior prominence to tip of sigmoid process.	24
Distance postero-ventral tip of outer posterior prominence to tip of conical process.	17.5
Width of tympanic bulla at the level of sigmoid process.	20
Height of tympanic bulla, from tip of sigmoid process to ventral keel.	23
Width across inner and outer posterior prominence.	18
Greatest depth of interprominential notch.	5
Width of upper border of sigmoid process.	11

Periotic

Standard (maximum anteroposterior) length, from tip of anterior process to posterior end of posterior process, measured on a straight line parallel to cerebral border	39.7
Standard (maximum mediolateral) width, from medial edge of cochlear portion to apex of lateral tuberosity	23
Maximum dorsoventral depth of anterior process perpendicular to axis of periotic	13.6

Maximum mediolateral width of anterior process at base	17
Length of anterior process from anterior apex to posterior border of mallear fossa	16.2
Length of anterior process from anterior apex to anterior incisure	14.3
Anteroposterior length of cochlear potion	17.3
Dorsoventral depth of cochlear potion	18.7
Mediolateral width of cochlear potion from medial edge to fenestra ovalis	10.8
Maximum diameter of fenestra rotunda	2.7
Maximum diameter of internal acoustic meatus	4.7
Maximum diameter of cochlear aqueduct	1.9
Maximum diameter of aperture for the vestibular aqueduct	3.6
Anteroposterior diameter of proximal opening of facial canal	3.4
Anteroposterior diameter of distal opening of facial canal	1.9
Length of posterior process (posterior bullar facet between anteroposterior tips)	17.3
Width of posterior process (posterior bullar facet between mediolateral tips)	12.1

Figure and table captions

Figure 1. Index maps of the Pacific Northwest: A, locations of the Japanese Islands, showing Sado Island; B, map of Sado Island, showing some major place names mentioned in text, and location of enlarged area including the type locality for *Berardius kobayashii*, sp. nov.; C, detail of Sobama beach in Donokama, Ogi Town, showing the exact locality for the holotype of *Berardius kobayashii*, SCM 5530-6. Based on the GIS map (<https://maps.gsi.go.jp/#15/37.844139/138.272681/&base=std&ls=std&disp=1&vs=c0j0h0k0l0u0t0z0r0s0m0f1>) of the Geospatial Information Authority of Japan.

Figure 2. Correlation of formations distributed on the Osado (a) and Kosado (b) mountain ranges containing whale fossil horizon with relevant systems of chronology. Wave lines indicate unconformities. After [6, 19, 10]. Diatom zones: [10, 37, 38]; Planktonic foraminiferal zones:[24, 32].

Figure 3. Dorsal view of the cranium of *Berardius kobayashii*, sp. nov., holotype, SCM 5530-6. (Scale bar = 10 cm)

Figure 4. Lateral view of the cranium of *Berardius kobayashii*, sp. nov., holotype, SCM 5530-6. (Scale bar = 10 cm)

Figure 5. Anterior view of the cranium of *Berardius kobayashii*, sp. nov., holotype, SCM 5530-6. (Scale bar = 10 cm)

Figure 6. Ventral view of the cranium of *Berardius kobayashii*, sp. nov., holotype, SCM 5530-6. (Scale bar = 10 cm)

Figure 7. Posterior fragment of the left mandible of *Berardius kobayashii*, sp. nov., holotype, SCM 5530-6. A, lateral view; B, medial view. (Scale bar = 10 cm)

Figure 8. Left periotic of *Berardius kobayashii*, sp. nov., holotype, SCM 5530-6. A, ventral view; B, corresponding line drawing with anatomical interpretations (gray hatches indicate parts of the tympanic bulla); C, dorsal view of the reconstructed CT image from 3D surface rendering, with anatomical interpretations; D, comparisons of the lateral view of anterior process of periotic among the genera of Ziphiidae (modified from Bianucci, 1997). (Scale bar = 1 cm)

Figure 9. The left tympanic bulla A, dorsal view; B, anterior view; C, medial view; D, lateral view; E, posterior
view; F, ventral view (Scale bar = 1 cm)

Figure 10. Key features of the left tympanic bulla of *Berardius kobayashii*, sp. nov., holotype, SCM 5530-6. A,
dorsal view; B, anterior view; C, medial view; D, lateral view; E, posterior view; F, ventral view. (Scale bar = 1
8 cm)

Figure 11. Phylogenetic relationship and paleobiogeographic distribution of the Ziphiidae based on Bianucci
et al. [3]

Figure 12. Distribution and potential roots for the dispersal of the known extinct species of the Berardiinae in
the middle Miocene.

Table 1. Measurements (in mm) of the skull of *Berardius kobayashii*, sp. nov., holotype, SCM 5530-6.

Table 2. Measurements (in mm) of the ear bones of *Berardius kobayashii*, sp. nov., holotype, SCM 5530-6.

Figure 1. Index maps of the Pacific Northwest: A, locations of the Japanese Islands, showing Sado Island; B, map of Sado Island, showing some major place names mentioned in text, and location of enlarged area including the type locality for *Berardius kobayashii*, sp. nov.; C, detail of Sobama beach in Donokama, Ogi Town, showing the exact locality for the holotype of *Berardius kobayashii*, SCM 5530-6. Based on the GIS map (<https://maps.gsi.go.jp/#15/37.844139/138.272681/&base=std&ls=std&disp=1&vs=c0j0h0k0l0u0t0z0r0s0m0f1>) of the Geospatial Information Authority of Japan.

514x745mm (96 x 96 DPI)

Series	Age (Ma)	Stages	Diatom Zones	Planktonic foraminiferal zones	Osado (a) (Yanagisawa & Watanabe, 2017)	Kosado (b) (Niigata Foraminiferal Research Group, 1967; Yanagisawa, 2012; Kawatani et al., 2019)	
Upper Miocene	6	Messinian	7B	PF5	Nozaka	Yamadagawa	
		Tortonian	7A				PF4
	6B		Nakayama	Notayama			
	6A						
	5D						
	5C						
	10		Tortonian	5B	PF3	Tsurushi	
	12			5A			
	Middle Miocene		14	Serravallian	4Bb	PF2	Hanyugawa
		4Ba					
Langhian		Nanatani	4A	PF1			
			3B				
			3A				
			2B				
Lower Miocene	18	Burdigalian	2A	PF1	Kinpokusan	Kyozukayama	
			20				Masaragawa
	Aquitanian	Mikawa	1	PF1	Aikawa	Aikawa	
			22				

Figure 2. Correlation of formations distributed on the Osado (a) and Kosado (b) mountain ranges containing whale fossil horizon with relevant systems of chronology. Wave lines indicate unconformities. After [6, 19, 10]. Diatom zones: [10, 37, 38]; Planktonic foraminiferal zones:[24, 32].

330x310mm (96 x 96 DPI)

Figure 3. Dorsal view of the cranium of *Berardius kobayashii*, sp. nov., holotype, SCM 5530-6. (Scale bar = 10 cm)

502x257mm (96 x 96 DPI)

Figure 4. Lateral view of the cranium of *Berardius kobayashii*, sp. nov., holotype, SCM 5530-6. (Scale bar = 10 cm)

506x252mm (96 x 96 DPI)

Figure 5. Anterior view of the cranium of *Berardius kobayashii*, sp. nov., holotype, SCM 5530-6. (Scale bar = 10 cm)

579x250mm (96 x 96 DPI)

Figure 6. Ventral view of the cranium of *Berardius kobayashii*, sp. nov., holotype, SCM 5530-6. (Scale bar = 10 cm)

521x270mm (96 x 96 DPI)

Figure 7. Posterior fragment of the left mandible of *Berardius kobayashii*, sp. nov., holotype, SCM 5530-6. A, lateral view; B, medial view. (Scale bar = 10 cm)

245x197mm (96 x 96 DPI)

Figure 8. Left periotic of *Berardius kobayashii*, sp. nov., holotype, SCM 5530-6. A, ventral view; B, corresponding line drawing with anatomical interpretations (gray hatches indicate parts of the tympanic bulla); C, dorsal view of the reconstructed CT image from 3D surface rendering, with anatomical interpretations; D, comparisons of the lateral view of anterior process of periotic among the genera of Ziphiidae (modified from Bianucci, 1997). (Scale bar = 1 cm)

486x300mm (96 x 96 DPI)

Figure 9. The left tympanic bulla A, dorsal view; B, anterior view; C, medial view; D, lateral view; E, posterior view; F, ventral view (Scale bar = 1 cm)

277x256mm (96 x 96 DPI)

Figure 10. Key features of the left tympanic bulla of *Berardius kobayashii*, sp. nov., holotype, SCM 5530-6. A, dorsal view; B, anterior view; C, medial view; D, lateral view; E, posterior view; F, ventral view. (Scale bar = 1 cm)

381x293mm (96 x 96 DPI)

Figure 11. Phylogenetic relationship and paleobiogeographic distribution of the Ziphiidae based on Bianucci et al. [3]

352x174mm (96 x 96 DPI)

Figure 12. Distribution and potential roots for the dispersal of the known extinct species of the Berardiinae in the middle Miocene.

705x522mm (96 x 96 DPI)

Appendix B**ROYAL SOCIETY
OPEN SCIENCE****The oldest fossil record of the extant genus *Berardius*
(Odontoceti, Ziphiidae) from the middle Miocene of the
western North Pacific**

Journal:	Royal Society Open Science
Manuscript ID	RSOS-201152
Article Type:	Research
Date Submitted by the Author:	02-Jul-2020
Complete List of Authors:	Kawatani, Ayako; University of Tsukuba Graduate School of Life and Environmental Sciences Kohno, Naoki; University of Tsukuba Graduate School of Life and Environmental Sciences; National Museum of Nature and Science, Geology and Paleontology
Subject:	palaeontology < BIOLOGY, Palaeontology < EARTH SCIENCES
Keywords:	middle Miocene, extant genus, Berardius kobayashii , sp. nov., western North Pacific
Subject Category:	Organismal and Evolutionary Biology

Author-supplied statements

Relevant information will appear here if provided.

Ethics

Does your article include research that required ethical approval or permits?:

This article does not present research with ethical considerations

Statement (if applicable):

CUST_IF_YES_ETHICS :No data available.

Data

It is a condition of publication that data, code and materials supporting your paper are made publicly available. Does your paper present new data?:

Yes

Statement (if applicable):

All the new data including materials, measurements photographs and necessary text data are provided in our manuscript.

Conflict of interest

I/We declare we have no competing interests

Statement (if applicable):

CUST_STATE_CONFLICT :No data available.

Authors' contributions

This paper has multiple authors and our individual contributions were as below

Statement (if applicable):

A.K. and N.K. conceived and designed the project, performed the experiment and analysed the data.

A.K. prepared SCM 5530-6. N.K. contributed reagents/materials/preparation tools/analysis tools.

A.K. and N.K. wrote the paper.

The oldest fossil record of the extant genus *Berardius* (Odontoceti, Ziphiidae) from the middle Miocene of the western North Pacific

Ayako Kawatani¹ and Naoki Kohno^{1,2}

¹Graduate School of Life and Environmental Sciences, University of Tsukuba, 1-1-1, Tennoudai, Tsukuba, Ibaraki, 305-8577, Japan

²National Museum of Nature and Science, 4-1-1, Amakubo, Tsukuba, Ibaraki, 305-0005, Japan

Keywords: middle Miocene, extant genus, *Berardius kobayashii*, sp. nov., western North Pacific.

1. Summary

A new species of a beaked whale that belongs to the extant genus *Berardius* is described from the late middle Miocene Tsurushi Formation (ca. 12 Ma) on the Sado Island, Niigata Prefecture, Japan. The new species, *Berardius kobayashii*, represents the oldest record of the genus and also of the extant ziphiid genera in the world. *Berardius kobayashii*, sp. nov., has following generic characters of *Berardius* such as the gently elevated vertex, the long and slender anterior process of the periotic. *Berardius kobayashii*, sp. nov., also has following unique specific characters: it is extremely small in size, the pieces of the left and right hamular process of the pterygoids contact medially, forming together a posteriorly directed medial point. *Berardius kobayashii*, sp. nov. fills the gap between the origin of the genus and later diversifications of the extant species in the genus. Furthermore, it suggests that *Berardius* species had originated in the North Pacific including the Sea of Japan at the time at latest 12 Ma. The discovery will also be a key to elucidate the process of their habitat expansion as the cosmopolitan genus among the Ziphiidae. Based on the distributional patterns of the fossil and extant species of the genus suggest that the western North Pacific including the Sea of Japan may have been one of the areas for the evolution and radiation of the extant genera among the crown Ziphiidae.

2. Introduction

The family Ziphiidae of the cetartiodactyl clade Odontoceti (Cetacea) are represented by 23 extant species [1, 2] and almost the same number of extinct species [3, 4]. Although the extant ziphiids have cosmopolitan distribution, they are one of the most mysterious groups of whales because of their deep-diving behavior. The Ziphiidae is thought to be originated and diversified in the Southern Ocean, and the ancestors of these deep-diving whales initially lived in the epipelagic and/or neritic oceans [5]. But, their adaptation to and diversification in the deep-sea have still been uncertain.

[revised manuscript text omitted]

The Aikawa, Sanze and Kyozyukasan Formations, which are stratigraphically lower units in the Neogene of the Kosado mountain range, are terrestrial formations and mainly consisted of andesites and dacite lavas [17, 20, 21]. The early Miocene Sanze and Kyozyukayama Formations are unconformably overlain by the Orito Formation [20]. The Orito Formation is the shallow marine sediments composed of conglomerates, sandstone and mudstone [10, 20]. The Orito Formation yields fossils of large foraminifers, echinoderms, mollusks and marine mammals [9, 22], and its age is estimated to be late early Miocene, ca. 17.5-17.3 Ma, based on marine diatom fossils [10, 19]. The Tsurushi Formation is deep sea sediments overlying the Orito Formation, and it consists of mainly hard siliceous mudstone. The Notayama and Yamadagawa Formations consist of diatomaceous mudstones. According to the diatom biostratigraphy, these two formations are late Miocene in age [19].

2.2 The Tsurushi Formation

The Middle Miocene Tsurushi Formation was originally defined by Watanabe [21] based on the hard siliceous mudstone exposed at Tsurushi in Sawada Town on the Osado mountain range. The same or similar rock facies has been distributed also on the Kosado mountain range and was recognized as the same Formation [23]. The Tsurushi Formation distributed on the Kosado mountain range is composed of hard siliceous mudstone and the contemporaneous basalt (i.e., Ogi Basalt Member) distributed regionally on the Ogi Peninsula [19]. The geologic age of the Ogi Basalt Member is estimated to be 14.1–11.7 Ma [10]. Watanabe [7] identified the foraminiferal fossils from the Tsurushi Formation on the Osado mountain range, which indicate the PF4 Zone of Maiya [24], and it corresponds to 12.3–11.5 Ma [10, 19]. Kawatani et al. [25] also confirmed the age of the Tsurushi Formation on the Kosado mountain range that produced whale fossils based on the radiolarian fossils, which corresponds to 15.3–9.1 Ma. Based on these chronological information, the geologic age of the Tsurushi Formation is estimated to be approximately 12.3–11.5 Ma. Studies of foraminiferal fossils have suggested that the Tsurushi Formation was deposited in the deep-sea environment [7, 9].

As mentioned above, various megafossils have been reported from the Tsurushi Formation; for example, mollusks, fish, a sea turtle, birds, and marine mammals [9, 11, 26]. In particular, a lot of marine mammal fossils such as whales, pinnipeds and desmostylians have been found in calcareous nodules from the Tsurushi Formation on the Kosado mountain range, and these fossils were briefly reported and summarized by Sado Research Group for Marine Mammalian Fossils [11], Tazaki et al. [13], Horikawa et al. [12], Takahashi et al. [14] and Barnes and Hirota [15].

4. Materials and Methods

The specimen described here was initially reported by Takahashi et al. [14]. At that time the specimen was temporarily deposited at Niigata University, and now it is permanently stored at the Sado Museum, Sado City, Niigata Prefecture, Japan, where it bears the catalog number SCM 5530-6. This specimen had been enclosed in calcareous hard mud stone matrices. Accordingly, it was prepared by acid etching technic using formic acid diluted with concentration of about 4.5–5.0% in water, which allowed the extraction of fragile bone elements of the skull, partial mandible and ear bones from hard matrices. However, the periotic has been fused to the skull loosely by its fan-shaped posterior process, and the dorsal surface of the periotic is invisible entirely from the ventral aspect. So we reconstructed the 3D image of the periotic by the micro-Computed Tomographic (μ CT) scanner of the Microfocus CT, TXS320-ACTIS, at the National Museum of Nature and Science, Tokyo, Japan.

We then compared them osteologically and taxonomically with both fossil and extant species of ziphiids stored at the following institutions and the literature cited. Definitions of measured portions and morphological terms follow Mead and Fordyce [27] and Kasuya [28]. The characters for comparisons and the phylogenetic analysis follow Bianucci et al. [3].

Institutional abbreviations are the following: NMNS-PV, Department of Geology and Paleontology, National Museum of Nature and Science, Tsukuba, Japan; NSMT-M, Department of Zoology, National Museum of Nature and Science, Tsukuba, Japan; SCM, Sado Museum, Sado City, Niigata, Japan; TMNHN, Togakushi Museum of Natural History, Nagano, Japan.

5. Systematics

Order CETARTIODACTYLA Montgelard, Catzefis and Dujzery, 1997
Infraorder CETACEA Brisson, 1762
Parvorder ODONTOCETI Flower, 1867
Family ZIPHIIDAE Gray, 1850
Subfamily BERARDIINAE Moore, 1968

Emended Diagnosis of subfamily: Two pairs of well-developed teeth on the mandible; the nodular protuberance formed by the interparietal or the frontals on the vertex; the narrow and thin premaxillary crest; the supraoccipital is lower than the frontals on the vertex; the part between dorsal

surfaces of the nasal bones and ends of the premaxillaries rises above the supraoccipital (modified from [29]).

Genus *Berardius* Duvernoy, 1851

Emended diagnosis of genus: The ratio between the rostral length and the condylobasal length is between 0.63–0.70; the mesorostral groove is empty; the mesorostral groove opens for the proximal third of the rostrum; the prenasal basin is absent; asymmetry of the premaxillary sac fossae absent or weak; the premaxillary sac fossa is not laterally overhanged on the maxilla; deep excavation of the premaxillary sac fossa is absent; the ascending process of the premaxilla in lateral view is slightly concave; constriction on the ascending process of the right premaxilla is roughly absent; the vertex is gently elevated; the premaxillary crest is transversely directed; the ratio between the width of the premaxillary crests and the width of the premaxillary sac fossae is 1.0 to 1.25; the distance between the left and right premaxillary crests is large; the nasal elongation ratio between the length of the medial suture of the nasals and the maximum width of the nasals is large; the anteromedial excavation of the dorsal surface of the nasal is absent; the premaxillary crest present on the vertex; inclusion of the nasal in the premaxillary crest is absent; contact between the nasal and the premaxillary crest is extended more than half the length of the nasal to almost the whole length; the premaxillary foramen is located roughly at the level of the antorbital notch; there is no excrescences on the dorsal surface of the maxilla at the posterior half of the rostrum; the interparietal or the frontal is present as an isolated rounded protuberance on the posterior part of the vertex; the posterolateral narrowing of the nasals and frontals is present at the vertex that is narrower than the nasals; the anteromedial margin of the supraoccipital is distinctly lower than the dorsal margin of the vertex; the extremely ossified trapezoidal vertex is absent; the angle formed by the basioccipital crests is larger than 50° in ventral view; the hamular fossa of the pterygoid sinus is wide, extending anteriorly on the palatal surface of the rostrum; the apices of the left and right hamular processes of the pterygoids contact medially; the apex of the hamular process is excavated by the fossa for the hamular lobe of the pterygoid sinus; the anteroventral wall of the tympanic bulla is narrow and triangular; the anterior spine of the tympanic with a more or less rectilinear anterior margin; the posteroventral corner of the sigmoid process of the tympanic is posteriorly projected in lateral view; the dorsal margin of the involucrum of the tympanic that is cut by an indentation is present and visible in medial and/or dorsal view; the periotic has long and slender anterior and posterior processes; the posterior bullar facet of the periotic is fan-shaped (modified from [3], [29], [30], [31]).

Berardius kobayashii, sp. nov.
(Skull: Figure 3-7; Ear bone: Figure 8-10)

Berardius sp. indet. Takahashi et al. 1989 
URN: lsid:zoobank.org:act: 317C1452-AE2E-4A08-A473-3B507B452D2C

Diagnosis of species: Extremely small in size among the four species of *Berardius* (60–70 % smaller than *B. bairdii* and *B. anuxii*, 50% smaller than *B. minimus*); the transverse diameter of the premaxilla is relatively wide; the lateral margin of the premaxilla is curved; the zygomatic process of the squamosal is anterodorsally tapered and triangle in shape; the nasals are short; there is a tiny anteromedial excavation on the dorsal surface of the nasals.; the apices of the left and right hamular process of the pterygoids contact medially, forming together a posteriorly directed medial point.

Holotype: SCM 5530-6, incomplete skull with a left tympanic bulla and periotic, lacking the rostral portion; a posterior one fourth of the horizontal ramus of the left dentary with the condyloid process and mandibular condyle. Collected by Kiyoei Kaneko in or before 1969.

Type Locality: Sobama Beach, Ogi Town, Sado Island, Niigata Prefecture, Japan. Geographic coordinates: 37° 84'50" N, 138° 26'58" E.

Formation and Age of Holotype: SCM 5530-6 was collected from the middle to late Miocene Tsurushi Formation. As discussed above, the geologic age of the Tsurushi Formation is estimated to be approximately 12.3–11.5 Ma.

Etymology: The species is named in honor of the late Dr. Iwao Kobayashi, a professor emeritus at Niigata University, for his longstanding contributions to the geology and paleontology of the northern Fossa Magna region including Niigata Prefecture, and in gratitude for his permission to re-study the marine mammal fossils collected by him and his colleagues from the Sado Island and encouragement to both of us throughout this study.

Description

Cranium (general morphology)

The cranium is relatively small (Table 1) and less asymmetrical (Figure 3,5). The ratio between the height and width of the skull is 1:1.19. The rostral and the occipital parts are almost broken and missing away. The small part of the left mandible had attached to the cranium. The vertex is moderately elevated and slightly skewed toward the left. The extremely ossified trapezoidal vertex is almost broken. The temporal fossa is isosceles triangle in shape that converges anteriorly, and the posterior margin is weakly rounded. The temporal fossa is shallow and concave. The sutures of the skull are still visible but tightly closed, indicating that the individual is an adult.

Premaxilla

The premaxilla is almost flat. The premaxillary crest is present on the vertex. The medial margin between the left and right premaxillae is very close. Contact between the nasal and the premaxillary crest is extended along the almost whole length of the nasal. The premaxillary sac fossae are weakly asymmetrical and do not overhang the maxillae laterally. The premaxillary sac fossa is almost flat and not deeply excavated. The ascending process of the premaxilla gradually rises toward the vertex. It is slightly concave in lateral view. The ascending process of the left premaxilla is somewhat damaged, but that of the right premaxilla is well preserved and moderately constricted just in front of the posterior end of the premaxillary crest. The vertex is gently elevated. In anterior view, lateral expansion of the premaxillae at the portion of the premaxillary sac fossae is prominent. The premaxillary crest is transversely oriented, and it is narrow in width. The distance between the premaxillary crests is wide. The anterolateral margin of the bony nares is V-shaped. The small premaxillary foramina are located roughly at the level of the antorbital notch and slightly posterior to the infraorbital foramina on the premaxillae. There are distinct grooves anterior to the foramina. The angle between the dorsal surface of the rostrum and the ascending process of the premaxilla is 150°.

Maxilla

The maxilla is broad and its margin is curved. The rostral maxillary crest is present and moderately elevated on the maxilla. The infraorbital foramina open anterodorsally on the maxillae near the base of the rostrum, and the largest foramen is located on the right maxilla. There are two maxillary foramina on the left maxilla and are 13 mm and 7 mm in diameter, respectively. There is one large maxillary foramen on the right maxilla and has a diameter reaching 16 mm. The maxilla is slightly elevated at the posterior side of these foramina. The excrescence on the dorsal surface of the maxilla is absent on the half of the rostrum. The prominent notch and related structure are low.

Nasals

The nasal is short, and its dorsal outline is trapezoidal. The shape of the bony nares is triangular in outline, and the anteroposterior diameter (67 mm) of the bony nares is wider than the transverse diameter (39 mm). Although the posterior margin of the right nasal is broken, it is slightly broader than the left one. The dorsal surface of the nasal is roughly flat, but there is a weak anteromedial excavation on its dorsal surface. The nasal is not included in the premaxillary crest. The nasal contacts the premaxilla laterally for almost the whole length. The posterior nasal-frontal suture is posteriorly convex.

Frontal

The frontal is fully covered by the maxilla in dorsal view and only visible its vertical section beneath
the overlapping maxilla at the portion of the orbital and temporal fossae in lateral view. It is
dorsoventrally thick and relatively dense but not osteosclerotic nor pachyosteosclerotic.

**Supraoccipital**

The supraoccipital is almost broken away. The anteromedial margin of the supraoccipital is
distinctly lower than the dorsal margin of the vertex. The supraoccipital protuberance protrudes
posteriorly.

**Pterygoid**

The pterygoid is dorsoventrally deep and excavated by the anteroposteriorly and dorsoventrally
wide hamular fossa. There are distinct transverse crests on the pterygoid. The hamular fossa of the
pterygoid sinus is wide, extending anteriorly on the palatal surface of the rostrum. Apices of the left
and right hamular processes are incomplete, but both processes contact medially and form together a
posteriorly directed medial point.

**Squamosal**

The left zygomatic process of the squamosal is preserved. It is short anteroposteriorly compared to
its dorsoventral height. The zygomatic process of the squamosal is well developed and anterodorsally
tapered with triangle outline, but its anterior margin is crescent in shape.

**Mandible**

The mandibles are almost broken away, but the ascending ramus of the left mandible with the
coronoid process and the mandibular condyle is partially preserved (Figure 7). The posterior margin
of the angular process is also broken away. The mandibular body is very thin and fragile at the
portion of the lateral wall of the mandibular foramen. Its outer or lateral surface is smooth and
slightly swelled laterally, forming the condyloid crest. Its inner or medial surface (or the lateral wall
of the mandibular foramen) is relatively rough. The mandibular foramen is anteroposteriorly long and
dorsoventrally wide. The coronoid crest is well developed as an anteroposteriorly extended ridge and
continues to the coronoid process at the posterodorsal end of the mandible. The ventral margin of the
coronoid crest is weakly concave along the crest. The mandibular condyle protrudes posteriorly and
locates almost the same position at the end of the coronoid process. It is dorsoventrally long and
laterally crescentic, and it is slightly swelled posteriorly. There is a notch between the coronoid
process and the mandibular condyle but is absent between the mandibular condyle and the angular
process, which is almost broken though.

**Tympanic Bulla**

The left tympanic bulla is preserved (Table 2). The anterior part of the ventral wall of the tympanic
bulla is cylindrical and is lacking its anterior spine. The involucrem is shorter than the lateral wall.
The dorsal margin of the involucrem is cut by an indentation. The inner and outer posterior
prominences are thick and short, and they swell well. The outer posterior prominence is much
swelled than the inner one. The median furrow runs anteroposteriorly on the ventral wall, and it is
transversely narrow and anteriorly shallow. The interprominental notch is relatively deep at the
posterior end of the median furrow. The lateral furrow is shallow, and its posterior ridge is prominent
and twisted on the lateral wall of the tympanic bulla. The sigmoid process is square and rather thin.
The lateral border of the sigmoid process is twisted posteriorly. A damaged base of the posterior
process lies on the posterolateral end of the sigmoidal process. The elliptical foramen is present
(Figure 9,10).

**Periotic**

The left periotic is preserved (Table 2). It is relatively slender in shape. The dorsal surface of the
body is smooth. The anterior and posterior processes are almost straight anteroposteriorly in ventral
view. The anterior process is long and slender, and its apex is pointed (Figure 8A). It is weakly bent
ventrally. The anterior bullar facet is weakly excavated anteroposteriorly. A rounded accessory

ossicle of the tympanic bulla is still articulated with the fovea epitubaria at the posteromedial base of the anterior process. The malleolar fossa is excavated posterior to the accessory ossicle (in the fovea epitubaria) and opens posteriorly. It is laterally delimited by a mediolaterally elongated lateral tuberosity. The cochlear portion (=pars cochlearis) is rounded in outline, relatively low, and it is slightly bent anteroventrally with a weak depression at the anteromedial surface, so the anterior incisure is narrow and slightly covered with the posterior edge of the cochlear portion. The epitympanic hiatus is wide. The internal acoustic meatus containing the central foramen (=spiral cribriform tract) and the foramen singulare is separated by a developed transverse crest from the proximal opening of the facial canal. The central foramen is relatively small and almost circular, slightly compressed transversely. Its posterolateral margin is elevated and reaches almost the top of the cochlear portion. The foramen singulare opens near the aperture for the vestibular aqueduct (Figure 8B). The proximal opening of the facial canal is relatively large and elliptical in shape. The aperture for the cochlear aqueduct is large and semicircular. It is located close to the posterior margin of the internal auditory meatus. It is also relatively close to the rounded aperture for the vestibular aqueduct (=endolymphatic duct) that is elliptical in shape. The fenestra rotunda is relatively large and semicircular in outline. The posterior process is relatively long posteriorly and inflated mediolaterally. It is fused to the skull by the mediolaterally narrow periotic fossa of the skull. The posterior bullar surface is covered by the fused fragment of the posterior process of the tympanic bulla. It is slightly bent posterolaterally at the base just posterior to the cochlear portion. Because of this fusion of the posterior process of the tympanic bulla and its posterior expansion, the portion of the posterior bullar facet is posteriorly flared and fan-shape in ventral view (Figure 8A). Although mostly covered by the remnant of the posterior process of the tympanic bulla, the surface of the posterior bullar facet is smooth. The epitympanic hiatus is relatively deep.

6. Comparisons

Takahashi et al. [14] identified SCM 5530-6 as belonging to the Ziphiidae based on the following characters: the bony nares are located posteriorly; the rostrum is moderately wide; the occipital slopes relatively steeply posterodorsally; and the cranium is not strongly asymmetrical. Furthermore, they compared SCM 5530-6 with extant genera of the Ziphiidae; *Ziphius*, *Berardius*, and *Mesoplodon*, and identified SCM 5530-6 as a species of the genus *Berardius* among them based on the following characteristics: the bony nares is triangular in outline; the vertex is elevated gently; and the median part of the palatine is convex on the ventral side ([14], p.104). But they also pointed out that the transverse diameter of the premaxilla of SCM 5530-6 is relatively wider than the then extant two species of *Berardius*, and the lateral margin of the same portion is curved in contrast to the straight condition of this portion in the extant species of *Berardius*. In addition, the morphology of the zygomatic process of the squamosal is shaped like an anterodorsally tapered triangle in contrast to an anteriorly squared rectangle in the extant species of *Berardius*.

We farther compared the characteristics of SCM 5530-6 with both extant and fossil Ziphiidae including 28 genera and 55 species based mainly on Bianucci et al. [3] and Lambert et al. [31]. Although the left side of the premaxilla of SCM 5530-6 is broken, the ratio between the width of the premaxillary crest and that of the premaxillary sac fossa of SCM 5530-6 is moderate, from 1.0 to 1.25 (Char. 11 of [3]). The infraorbital foramina open anterodorsally on the maxilla near the base of the rostrum. In general, a cluster of the infraorbital foramina is present on the base of the rostrum in the early diverging ziphiids [3]. On the other hand, only a large foramen is present in the crown ziphiids [3]. SCM 5530-6 has a large foramen on the right maxilla, and two relatively large foramina on the left maxilla, and therefore, the condition of these foramina is thought to be the condition seen in the later diverging, extant ziphiids.

The supraoccipital of SCM 5530-6 is lower than the frontals on the vertex, and the vertex is gently elevated (Chars. 9 and 18 of [3]). A similar low elevation of the ascending process is observed in *Berardius* and related fossil species such as *Archaeoziphius microglenoideus* and *Microberardius africanus* [3, 29, 33].

Archaeoziphius differs from *Berardius* by the following characters: the overall size of the skull is much smaller; the nasal is included in the premaxillary crest for a short distance along the posteromedial angle of the crest [33]; Chars. 15 and 51 of [3]. However, the inclusion of the nasal in the premaxillary crest is absent on SCM 5530-6 as in the species of *Berardius*.

Microberardius differs from *Berardius* by the following characters: the size of the skull is much smaller; the premaxillary crest is much narrower; the vomer at the anterior portion of the mesorostral groove is thickening; the rostrum is higher than wider along the most of its length; the base of the rostrum is narrower and the maxillary crest is not extended at the base of the rostrum [29]. By contrast, SCM 5530-6 does not have the narrow premaxillary crest.

Consequently, SCM 5530-6 differs from both *Archaeoziphius* and *Microberardius*. In addition, the anterior process of the periotic is slender, and its apex is long and pointed, and the posterior process is long and slender. These characters are potential synapomorphies of the genus *Berardius* (see also [28, 34]). Therefore, SCM 5530-6 is definitely identified as a species of the genus *Berardius* as was already pointed out by Takahashi et al. [14].

However, SCM 5530-6 has some additional characters differing from the extant, now three species of *Berardius* such as *Berardius bairdii*, *B. arnuxii* and *B. minimus*, which was not mentioned by Takahashi et al. [14]. For instance, the nasal of SCM 5530-6 is shorter than *B. bairdii*, *B. arnuxii* and *B. minimus*. Also, there is a tiny anteromedial excavation on the dorsal surface of the nasal, which is not observed in other three species of *Berardius*. The apices of the pterygoids contact medially in all the species of *Berardius*, and these are forming together a posteriorly directed medial point in SCM 5530-6, but these are less excavated and U-shaped posterior margin in other three species of *Berardius*. The last character is not seen in any species of both fossil nor extant ziphiids including the three extant species of *Berardius*. But, the hamular fossa of the pterygoid sinus is wide, extending anteriorly on the palatal surface of the rostrum as seen in the extant *Berardius* and deep diving whales [35, 36]. The height of the skull relative to the width of SCM 5530-6 is 84 % and slightly higher than the extant species of *Berardius*; i.e., 83 % in *B. minimus* and 77 % in *B. bairdii*. The skull of SCM 5530-6 has much tighter sutures compared with those of extant *Berardius bairdii* and *B. arnuxii*. As has pointed out by Yamada et al. [2], the skull of *B. minimus* also has much tighter sutures, but SCM 5530-6 has much tighter sutures than that of *B. minimus*. The temporal fossa of SCM 5530-6 is shallow like *B. minimus*, but the lateral wall of the fossa is concave like *B. bairdii* and *B. arnuxii*. The lateral expansion of the premaxillae at the posterior halves is prominent like *B. minimus*. The prominent notch and related structure are much lower, less distinct and less rugged than *B. minimus*. The greatest postorbital width of SCM 5530-6 is almost 60 to 70 % smaller than the extant *B. bairdii* and *B. arnuxii* and 50 % smaller than *B. minimus*. The tympanic bulla and the periotic of SCM 5530-6 closely resembles that of *B. bairdii*. But, the total length of the tympano-periotic bone is 40 to 50 % smaller than that of *B. bairdii*, the total length of tympanic bulla is 30 % smaller than that of *B. minimus*. Consequently, the length of the tympano-periotic bone relative to the skull is larger than other *Berardius* species. As results of the above morphological comparisons, the new ziphiid fossil SCM 5530-6 from the middle Miocene Tsurushi Formation is undoubtedly identifiable as a species of the genus *Berardius*. In addition, comparisons with all the known species within the genus; i.e., the extant three species that are presently distributed in the northern North Pacific (i.e., *B. bairdii* and *B. minimus*) and the Antarctic (i.e., *B. arnuxii*), SCM 5530-6 is distinguishable from all the three extant species based on the above mentioned cranial characters: i.e., it is extremely small in size; the nasal is short; and the apices of the left and right hamular process of the pterygoids contact medially, forming together a posteriorly directed medial point. Furthermore, the size of the new fossil is extreme among all the known species of the genus; i.e., 50% smaller than *B. minimus* and surprisingly 60 to 70% smaller than both *B. bairdii* and *B. arnuxii*. Consequently, identification of SCM 5530-6 as a new species within the genus *Berardius*, i.e., *Berardius kobayashii*, is warranted.

7. Discussion

~~Interestingly, despite its extremely small size in *B. kobayashii* that is smaller than the smallest extant species *B. minimus*, the cranial morphologies of *B. kobayashii* are much closer to the larger~~

species, *B. bairdii* and *B. arnuxii* as discussed in the “comparisons” section. It suggests that although *B. minimus* is represented only by the extant species, its emergence could have been much earlier than the emergence and diversification of the rest of three species including the northern North Pacific middle Miocene extinct *B. kobayashii* sp. nov. and the extant *B. bairdii* and also the Antarctic *B. arnuxii*. Consequently, their emergence and diversification must have occurred in or by the middle Miocene and potentially in the western North Pacific based on the new fossil from the middle Miocene Tsurushi Formation described here and the present distribution of two extant species of *Berardius* in the North Pacific (see also [2]).

For now, three genera are recognized in the subfamily Berardiinae; i.e., *Microberardius* from the late Miocene of South Africa [29], *Archaeoziphius* from the middle Miocene of Belgium [33] and *Berardius* from the middle Miocene to the Recent of the northern North Pacific and the Antarctic oceans [1, 2, this study]. Previously, there was no fossil record of the berardiines in the North Pacific Ocean. So, the new fossil described here reveals that the berardiine ziphiid was already existed at least in the western North Pacific at latest in the late middle Miocene (ca. 12 Ma). It also suggests that the emergence of the berardiine ziphiids goes back to the early middle Miocene or much earlier because their sister taxon *Archaeoziphius* in the North Atlantic was already existed in the early middle Miocene [33]. At the moment, it is difficult to consider more precise origin of the Berardiinae based on the present fossil evidence, but its potential origin might have been in the Southern Hemisphere on the basis of recent studies on their phylogenetic relationships and phylogeographic distributions of sister and out groups for the Berardiine ziphiids [3, 5, 33]. However, all these evidence suggest the rapid diversification of the subfamily Berardiinae in the middle Miocene in the Northern Hemisphere (Figure 11).

8. Conclusions

An extinct beaked whale from the late middle Miocene Tsurushi Formation (ca. 12 Ma) in the Sado Island, Niigata Prefecture, Japan, was described as a new species of the extant genus *Berardius*; i.e., *B. kobayashii*. *Berardius kobayashii*, sp. nov., is the fourth species of the genus. Namely, the lineage leading to the extant genus of the ziphiids had already lived in the western North Pacific at the time of at latest 12 Ma. ~~The western North Pacific including the Sea of Japan may, therefore, have been one of the areas for the evolution and radiation of *Berardius* species.~~ The existence of a species of the extant ziphiid genus that have been known as one of the deep-diving whales from the middle Miocene deep-sea deposits will become the key to elucidate the process of their habitat expansions as deep-sea dwellers among the Ziphiidae.

Acknowledgments

We would like to thank the late I. Kobayashi (Professor Emeritus at Niigata University), A. Matsuoka (Niigata University), M. Shimizu (Asahimachi Museum, Niigata University), M. Ikarashi (Sado Museum), M. Aida (Sado Geopark Promotion Office and Sado Museum), K. Narita (Nagano City Museum), M. Kato (Nagaoka Municipal Science Museum), Y. Tajima and T.K. Yamada (NMNS), for allowing access to the specimens and giving me the opportunities to observe specimens at respective museums; K. Takahashi (Biwako Museum), H. Ichishima (Fukui Prefectural Dinosaur Museum), K. Nagasawa (formerly Yamagata Prefectural Museum), T. Kimura (Gunma Museum of Natural History), for providing useful discussion and advice with the literature on fossil ziphiids.

Funding Statement

A.K. was funded by the travel fund for the 2017 fiscal year by the Regional Promotion Division of Sado City.

Data Accessibility

This study and the nomenclatural acts it contains have been registered in Zoobank. The LSID for this publication is: urn:lsid:zoobank.org:pub: 7BEDEBA0-46B9-4125-A7F8-2169EB016C29.

Competing Interests

We declare we have no competing interests.

Authors' Contributions

A.K. and N.K. conceived and designed the project, performed the experiment and analysed the data. A.K. prepared SCM 5530-6. N.K. contributed reagents/materials/preparation tools/analysis tools. A.K. and N.K. wrote the paper.

References

1. Rosel P E, Archer FI, Baker CS, Boness DJ, Brownell RL, Churchill M Jr., Costa AP, Domning DP, Fordyce RE, Jefferson TA, Kinze C, Oliveira LR, Perrin WF, Wang JY, Yamada TK. 2019 List of marine mammal species and subspecies. *Committee on Taxonomy, The Society for Marine Mammalogy*.
2. Yamada TK, Kitamura S, Abe S, Tajima Y, Matsuda A, Mead JG, Matsuishi TF. 2019 Description of a new species of beaked whale (*Berardius*) found in the North Pacific. *Scientific reports* **9(1)**, 1-14.
3. Bianucci G, Di Celma C, Urbina M, Lambert O. 2016 New beaked whales from the late Miocene of Peru and evidence for convergent evolution in stem and crown Ziphiidae (Cetacea, Odontoceti). *PeerJ*. **4**, e2479.
4. Lambert O, de Muizon C, Duhamel G, Van Der Plicht J. 2018 Neogene and Quaternary fossil remains of beaked whales (Cetacea, Odontoceti, Ziphiidae) from deep-sea deposits off Crozet and Kerguelen islands, Southern Ocean. *Geodiversitas* **40(2)**, 135-140.
5. Lambert O, Collareta A, Landini W, Post K, Ramassamy B, Di Celma C, Urbina M, Bianucci G. 2015 No deep diving: evidence of predation on epipelagic fish for a stem beaked whale from the Late Miocene of Peru. *Proc. R. Soc. B*, 282.
6. Niigata Foraminiferal Research Group. 1967 Some foraminiferal assemblages from the Sawane area, Sado Island (Preliminary report): Restudy on the stratigraphy of the "Sawane Formation" based on the foraminiferal fossils. *Commemorative Volume for Mr. Y. Hiramatsu*, 115-119.
7. Watanabe K. 1987 Tertiary Foraminifera and Radiolaria Fossils from Sado Island, Central Japan. *Publications from the Sado Museum* **9**, 127-156.
8. Okamoto Y, Satou M, Watanabe M, Yamamoto H. 1992 Inversion Tectonics in the southeastern part of the Japan Sea. *Jour. Struct. Geol. Japan* **38**, 47-58.
9. Kobayashi I. 2001 The Cenozoic of Niigata prefecture, the case of Sado Island (Niigata no shinseikai Sado-hen). *Niigata Geotechnology Society* **57**, 9-21.
10. Yanagisawa Y, Watanabe M. 2017 Revised lithostratigraphy of the Neogene sedimentary sequence in the southern part of the Osado Mountain area, Sado Island, Niigata Prefecture, Japan. *Bull. Geol. Surv. Japan* **68 (6)**, 259-285.
11. Sado Research Group for Marine Mammalian Fossils. 1987 Fossil Cetacean from Sado Island, Central Japan. *Publications from the Sado Museum* **9**, 211-217.
12. Horikawa H, Tazaki K, Kanno T. 1987 Fossil Ziphiidae from Koshiji-Syo off Sado Island, Central Japan. *Publications from the Sado Museum* **9**, 225-230.
13. Tazaki K, Horikawa H, Miyazaki S. 1987 Fossil Ziphiidae from Hyotan-Guri off Sado Island, Central Japan. *Publications from the Sado Museum* **9**, 219-223.
14. Takahashi K, Nomura M, Kobayashi I. 1989 A fossil cetacean skull (*Berardius* sp. indet.) from Dogama, Ogi-machi, Sado Island, Central Japan. *Earth Science; Chikyu Kagaku* **43**, 102-105.
15. Barnes LG, Hirota K. 1995 Miocene pinnipeds of the otariid subfamily Allodesminae in the North Pacific Ocean: Systematics and relationships. *The Island Arc* **3**, 329-360.
16. Utashiro T. 1950 The stratigraphy of Sawane-Aikawa region in Southwest Sado Island (Osado Mountain range). *Journal of the Geological Society of Japan* **56**, 302-303.
17. Shimazu M, Kanai Y, Toyama T, Ichihashi K, Minakawa J, Takahama N. 1973 Structural development and igneous activity in the Sado island. *Memoirs of the Geological Society of Japan* **9**, 147-157.
18. Shimazu M, Toyama T. 1982 Neogene volcanic rocks of the Sado islands. *Journal of the Geological Society of Japan* **88**, 381-400.
19. Yanagisawa Y. 2012 Late Miocene diatoms in the Hamochi area, Sado Island, Niigata Prefecture, Japan. *Open-File Report of the Geological Survey of Japan, AIST* **568**, 1-19.
20. Ogi Collaborative Research Group. 1986 Late Cenozoic Group in the Southern part of Kosado Mountains, Niigata prefecture. *Earth Science* **40**, 417-436.
21. Takeuchi K, Ozaki M, Komatsubara T. 2011 Explanatory notes of 1:200,000 geological map of the coastal zone around Niigata. Digital Geoscience Map, Seamless Geoinformation of Coastal Zone "Coastal Zone around Niigata".
22. Tokunaga S. 1939 A New Fossil Mammal Belonging to Desmostyliidae. *Jub. Pub. Com., Professor H. Yabe, M.I.A. Sixtieth Birthday*. Sendai: Institute of Geology and Paleontology, Tohoku Imperial University. 289-299.
23. Watanabe H. 1932 Neogene (Shin Daisankei). *Nihon Chishitsu kousan shi* 92-152.
24. Maiya S. 1978 Late Cenozoic planktonic foraminiferal biostratigraphy of the oil-field region of Northeast Japan. *Cenozoic Geology of Japan*, 35-60.
25. Kawatani A, Sashida K, Sachiko A, Kohno N. 2019 Radiolarian fossils and estimated depositional age of the Miocene Tsurushi Formation distributed in Sado Island, Niigata Prefecture, Japan. *Bull. Geol. Surv. Japan* **70(1-2)**, 91-99. (doi: <https://doi.org/10.9795/bullgsj.70.91>)
26. Ono K, Ueno T. 1985 Tertiary Vertebrates from Sado Island, Niigata Prefecture, Central Japan. *Memoirs of the National Science Museum* **18**, 65-71.
27. Mead JG, Fordyce RE. 2009 The therian skull: a lexicon with emphasis on the odontocetes. *Smithsonian Contributions to Zoology*, 1-249.
28. Kasuya T. 1973 Systematic consideration of recent toothed whales based on the morphology of tympano-periotic bone. *Scientific Reports of the Whales Research Institute* **25**, 1-103.
29. Bianucci G, Lambert O, Post K. 2007 A high diversity in fossil beaked whales (Mammalia, Odontoceti, Ziphiidae) recovered by trawling from the sea floor off South Africa. *Geodiversitas* **29(4)**, 561-618.
30. Bianucci G, Lambert O, Post K. 2010 High concentration of long - snouted beaked whales (genus *Messapicetus*) from the Miocene of Peru. *Palaeontology* **53(5)**, 1077-1098.
31. Lambert O, De Muizon C, Bianucci G. 2013 The most basal beaked whale *Ninziphius platyrostris* Muizon, 1983: clues on the evolutionary history of the family Ziphiidae (Cetacea: Odontoceti). *Zoological Journal of the Linnean Society* **167(4)**, 569-598.
32. Miwa M, Yanagisawa Y, Yamada K, Irizuki T, Shoji M, Tanaka Y. 2004 Planktonic foraminiferal biostratigraphy of the Pliocene Kuwae Formation in the Tainai River section, Niigata Prefecture and the age of the base of the No. 3 Globorotalia inflata bed. *Journal of the Japanese Association for Petroleum Technology* **69(3)**, 272-283.
33. Lambert O, Louwye S. 2006 *Archaeoziphius microglenoideus*, a new primitive beaked whale (Mammalia, Cetacea, Odontoceti) from the Middle Miocene of Belgium. *Journal of Vertebrate Paleontology* **26(1)**, 182-191.
34. Bianucci G. 1997 The Odontoceti (Mammalia Cetacea) from Italian Pliocene. The Ziphiidae. *Palaeontographia italica* **84**, 163-192.
35. Reidenberg JS, Laitman JT. 2008 Sisters of the sinuses: cetacean air sacs. *The Anatomical Record: Advances in Integrative Anatomy and Evolutionary Biology* **291(11)**, 1389-1396.
36. Ramassamy B, Lambert O, Collareta A, Urbina M, Bianucci G. 2018 Description of the skeleton of the fossil beaked whale *Messapicetus gregarius*: searching potential proxies for deep-diving abilities. *Mitteilungen aus dem Museum für Naturkunde in Berlin. Fossil Record* **21(1)**, 11-32.
37. Akiba F. 1986 Middle Miocene to Quaternary diatom biostratigraphy in the Nankai Trough and Japan Trench, and modified Lower Miocene through Quaternary diatom zones for middle-to-high latitudes of the North Pacific. In Kagami, H., Karig, D. E., Coulbourn, W. T., et al., *Initial Reports of the Deep Sea Drilling Project, U. S. Government*

Printing Office, Washington D. C. **87**,
 393–480.

38. Yanagisawa Y, Akiba F. 1998 Refined
 Neogene diatom biostratigraphy for the
 northwest Pacific around Japan, with an

introduction of code numbers for
 selected diatom biohorizons. *Jour. Geol.
 Soc. Japan* **104**, 395–414.

Tables

Measurements (in mm) of the skull of *Berardius kobayashii*

Skull

Condylobasal length from tip of rostrum to hindmost margin of occipital condyles.	236
Greatest postorbital width.	245
Greatest width of bony nares.	38
Maximum width of premaxillary sac fossae.	87
Maximum width of right premaxillary sac fossa.	43
Maximum width of left premaxillary sac fossa.	38
Maximum width of nasals.	40
Greatest length of left posttemporal fossa, measured to external margin of raised suture.	87
Greatest width of left posttemporal fossa at right angles to greatest length.	61
Major diameter of left temporal fossa proper.	38
Minor diameter of left temporal fossa proper.	33
Distance from foremost end of junction between nasals to hindmost point of margin of supraoccipital crest.	33
Length of left orbit from apex of preorbital process of frontal to apex of postorbital process.	51
Greatest width of internal flares.	91
Greatest length of left pterygoid.	109

Measurements (in mm) of the ear bone of *Berardius kobayashii*

Tympanic bulla

Standard length of tympanic bulla, distance from anterior tip to posterior end of outer posterior prominence.	37
Distance from anterior tip to posterior end of inner posterior prominence.	34
Distance postero-ventral tip of outer posterior prominence to tip of sigmoid process.	24
Distance postero-ventral tip of outer posterior prominence to tip of conical process.	17.5
Width of tympanic bulla at the level of sigmoid process.	20
Height of tympanic bulla, from tip of sigmoid process to ventral keel.	23
Width across inner and outer posterior prominence.	18
Greatest depth of interprominential notch.	5
Width of upper border of sigmoid process.	11

Periotic

Standard (maximum anteroposterior) length, from tip of anterior process to posterior end of posterior process, measured on a straight line parallel to cerebral border	39.7
Standard (maximum mediolateral) width, from medial edge of cochlear portion to apex of lateral tuberosity	23
Maximum dorsoventral depth of anterior process perpendicular to axis of periotic	13.6

Maximum mediolateral width of anterior process at base	17
Length of anterior process from anterior apex to posterior border of mallear fossa	16.2
Length of anterior process from anterior apex to anterior incisure	14.3
Anteroposterior length of cochlear potion	17.3
Dorsoventral depth of cochlear potion	18.7
Mediolateral width of cochlear potion from medial edge to fenestra ovalis	10.8
Maximum diameter of fenestra rotunda	2.7
Maximum diameter of internal acoustic meatus	4.7
Maximum diameter of cochlear aquiduct	1.9
Maximum diameter of aperture for the vestibular aqueduct	3.6
Anteroposterior diameter of proximal opening of facial canal	3.4
Anteroposterior diameter of distal opening of facial canal	1.9
Length of posterior process (posterior bullar facet between anteroposterior tips)	17.3
Width of posterior process (posterior bullar facet between mediolateral tips)	12.1

Figure and table captions

Figure 1. Index maps of the Pacific Northwest: A, locations of the Japanese Islands, showing Sado Island; B, map of Sado Island, showing some major place names mentioned in text, and location of enlarged area including the type locality for *Berardius kobayashii*, sp. nov.; C, detail of Sobama beach in Donokama, Ogi Town, showing the exact locality for the holotype of *Berardius kobayashii*, SCM 5530-6. Based on the GIS map (<https://maps.gsi.go.jp/#15/37.844139/138.272681/&base=std&ls=std&disp=1&vs=c0j0h0k0l0u0t0z0r0s0m0f1>) of the Geospatial Information Authority of Japan.

Figure 2. Correlation of formations distributed on the Osado (a) and Kosado (b) mountain ranges containing whale fossil horizon with relevant systems of chronology. Wave lines indicate unconformities. After [6, 19, 10]. Diatom zones: [10, 37, 38]; Planktonic foraminiferal zones:[24, 32].

Figure 3. Dorsal view of the cranium of *Berardius kobayashii*, sp. nov., holotype, SCM 5530-6. (Scale bar = 10 cm)

Figure 4. Lateral view of the cranium of *Berardius kobayashii*, sp. nov., holotype, SCM 5530-6. (Scale bar = 10 cm)

Figure 5. Anterior view of the cranium of *Berardius kobayashii*, sp. nov., holotype, SCM 5530-6. (Scale bar = 10 cm)

Figure 6. Ventral view of the cranium of *Berardius kobayashii*, sp. nov., holotype, SCM 5530-6. (Scale bar = 10 cm)

Figure 7. Posterior fragment of the left mandible of *Berardius kobayashii*, sp. nov., holotype, SCM 5530-6. A, lateral view; B, medial view. (Scale bar = 10 cm)

Figure 8. Left periotic of *Berardius kobayashii*, sp. nov., holotype, SCM 5530-6. A, ventral view; B, corresponding line drawing with anatomical interpretations (gray hatches indicate parts of the tympanic bulla); C, dorsal view of the reconstructed CT image from 3D surface rendering, with anatomical interpretations; D, comparisons of the lateral view of anterior process of periotic among the genera of Ziphiidae (modified from Bianucci, 1997). (Scale bar = 1 cm)

Figure 9. The left tympanic bulla A, dorsal view; B, anterior view; C, medial view; D, lateral view; E, posterior
view; F, ventral view (Scale bar = 1 cm)

Figure 10. Key features of the left tympanic bulla of *Berardius kobayashii*, sp. nov., holotype, SCM 5530-6. A,
dorsal view; B, anterior view; C, medial view; D, lateral view; E, posterior view; F, ventral view. (Scale bar = 1
8 cm)

Figure 11. Phylogenetic relationship and paleobiogeographic distribution of the Ziphiidae based on Bianucci
et al. [3]

Figure 12. Distribution and potential roots for the dispersal of the known extinct species of the Berardiinae in
the middle Miocene.

Table 1. Measurements (in mm) of the skull of *Berardius kobayashii*, sp. nov., holotype, SCM 5530-6.

Table 2. Measurements (in mm) of the ear bones of *Berardius kobayashii*, sp. nov., holotype, SCM 5530-6.

Figure 1. Index maps of the Pacific Northwest: A, locations of the Japanese Islands, showing Sado Island; B, map of Sado Island, showing some major place names mentioned in text, and location of enlarged area including the type locality for *Berardius kobayashii*, sp. nov.; C, detail of Sobama beach in Donokama, Ogi Town, showing the exact locality for the holotype of *Berardius kobayashii*, SCM 5530-6. Based on the GIS map (<https://maps.gsi.go.jp/#15/37.844139/138.272681/&base=std&ls=std&disp=1&vs=c0j0h0k0I0u0t0z0r0s0m0f1>) of the Geospatial Information Authority of Japan.

514x745mm (96 x 96 DPI)

Series	Age (Ma)	Stages	Diatom Zones	Planktonic foraminiferal zones	Osado (a) (Yanagisawa & Watanabe, 2017)	Kosado (b) (Niigata Foraminiferal Research Group, 1967; Yanagisawa, 2012; Kawatani et al., 2019)
Upper Miocene	6	Messinian	7B	PF5	Nozaka	Yamadagawa
		Tortonian	7A			
	6B					
	6A					
	5D					
	5C		PF3	Hanyugawa	Orito	
	5B					
	Langhian		5A	PF2	Hanyugawa	Orito
			4Bb			
		4Ba	PF1	Hanyugawa	Orito	
4A						
Middle Miocene	16	Serravallian	3B	PF1	Orito	Orito
			3A			
	14	Nanatani	2B	PF1	Kinpokusan	Kyozukayama
			2A			
	12	Serravallian	1	PF1	Masaragawa	Sanze
			1			
	10	Tortonian	1	PF1	Aikawa	Aikawa
			1			
	8	Tortonian	1	PF1	Aikawa	Aikawa
			1			
6	Messinian	1	PF1	Aikawa	Aikawa	
		1				
Lower Miocene	22	Aquitanian	1	PF1	Aikawa	Aikawa
			1			

Figure 2. Correlation of formations distributed on the Osado (a) and Kosado (b) mountain ranges containing whale fossil horizon with relevant systems of chronology. Wave lines indicate unconformities. After [6, 19, 10]. Diatom zones: [10, 37, 38]; Planktonic foraminiferal zones:[24, 32].

330x310mm (96 x 96 DPI)

Figure 3. Dorsal view of the cranium of *Berardius kobayashii*, sp. nov., holotype, SCM 5530-6. (Scale bar = 10 cm)

502x257mm (96 x 96 DPI)

Figure 4. Lateral view of the cranium of *Berardius kobayashii*, sp. nov., holotype, SCM 5530-6. (Scale bar = 10 cm)

506x252mm (96 x 96 DPI)

Figure 5. Anterior view of the cranium of *Berardius kobayashii*, sp. nov., holotype, SCM 5530-6. (Scale bar = 10 cm)

579x250mm (96 x 96 DPI)

Figure 6. Ventral view of the cranium of *Berardius kobayashii*, sp. nov., holotype, SCM 5530-6. (Scale bar = 10 cm)

521x270mm (96 x 96 DPI)

Figure 7. Posterior fragment of the left mandible of *Berardius kobayashii*, sp. nov., holotype, SCM 5530-6. A, lateral view; B, medial view. (Scale bar = 10 cm)

245x197mm (96 x 96 DPI)

Figure 8. Left periotic of *Berardius kobayashii*, sp. nov., holotype, SCM 5530-6. A, ventral view; B, corresponding line drawing with anatomical interpretations (gray hatches indicate parts of the tympanic bulla); C, dorsal view of the reconstructed CT image from 3D surface rendering, with anatomical interpretations; D, comparisons of the lateral view of anterior process of periotic among the genera of Ziphiidae (modified from Bianucci, 1997). (Scale bar = 1 cm)

486x300mm (96 x 96 DPI)

Figure 9. The left tympanic bulla A, dorsal view; B, anterior view; C, medial view; D, lateral view; E, posterior view; F, ventral view (Scale bar = 1 cm)

277x256mm (96 x 96 DPI)

Figure 10. Key features of the left tympanic bulla of *Berardius kobayashii*, sp. nov., holotype, SCM 5530-6. A, dorsal view; B, anterior view; C, medial view; D, lateral view; E, posterior view; F, ventral view. (Scale bar = 1 cm)

381x293mm (96 x 96 DPI)

Figure 11. Phylogenetic relationship and paleobiogeographic distribution of the Ziphiidae based on Bianucci et al. [3]

352x174mm (96 x 96 DPI)

Figure 12. Distribution and potential roots for the dispersal of the known extinct species of the Berardiinae in the middle Miocene

705x522mm (96 x 96 DPI)

Appendix C

21 November 2020

Dear Editors,

First of all, we deeply thank both reviewers for their generous comments and suggestions for our manuscript on the oldest *Berardius* from Japan, and we revised our manuscript to address their concerns. The reviewers' comments and suggestions greatly improved our manuscript. In spite of the constructive reviews, we have to apologize the delay to revise our manuscript because of the recent COVID19 crisis as a second wave in August and third wave in October and now. Although delayed, we thoroughly revised the manuscript and improved it with additional data following recommendations and suggestions from both reviewers. Changes are recorded with the track and change function in the revised manuscript. We followed all the reviewers' comments. Reviewer's comments are written in Black below. Our responses are written in Blue.

Reviewers' Comments to Author:

Reviewer: 1

Comments to the Author(s)

The present work deals with a very informative ziphiid fossil skull, including the highly informative ear bones. Those elements are only rarely found associated to other cranial material. Furthermore, the stratigraphic context is relatively well constrained, providing a geological age that is rather exciting (most extinct ziphiids are found in younger deposits). Finally, the specimen displays similarities with members of the extant family Berardiinae, potentially confirming the antiquity of this clade. All together, these elements make this study important and worth publishing.

The text is well organized and the bones are finely illustrated (see a few comments below and in the annotated pdf). I found the descriptions clear and concise (maybe a bit too short), although a few anatomical interpretations may ask for a second check.

I made a series of comments in the annotated pdf, and those mostly deal with English and typos,

which should be easily fixed. My main concerns and suggestions are listed below, and those may request some more work. I hope that those will prove useful, as this very fine, scientifically informative specimen definitely deserves to be published.

—>We corrected all the grammatical and typographic errors following suggestions on the marked pdf. We are ashamed of such problems, and we strongly appreciate to the reviewer's advice.

1- Concerning the stratigraphic context, you reach the conclusion that the formation from which the studied fossil originates can be dated from an interval spanning 12.3–11.5 Ma. This means that this unit may extend in the earliest Tortonian (starting at 11.63 Ma). If this is correct it may be difficult to exclude an early late Miocene age, and I would suggest modifying the text (and the title + abstract) accordingly. Maybe 'from the latest middle to earliest late Miocene', or 'from the middle to late Miocene boundary'. This would remove any possible confusion, I think. Indeed, many ziphiids have been dated from the Tortonian, whereas Serravallian records are much less frequent.

—>We changed the above-mentioned sentence in our revised MS from “from the middle Miocene” to “from the middle to early Miocene boundary” to avoid any possible confusion. The estimation of the geologic age for the lower part of the Tsurushi Formation that produced the holotype skull was based on the diatom fossils. So, the “minimum” age estimation is actually crossing the boundary between the middle and late Miocene. Again, we appreciate to the reviewer's advice.

2- The emended diagnosis of *Berardius* is long, and it is not easy to understand how it was compiled, for several reasons. First, some characters are also listed for the subfamily Berardiinae. Second, differences are not associated to other genera. It may be ok to keep such a list, but then it should start with something like 'The genus *Berardius* differs from all other ziphiids with the following unique combination of characters: ...'. However, because you provide a diagnosis for Berardiinae just above, it may be sufficient to provide here differences with other Berardiinae (and maybe expand somewhat the diagnosis for Berardiinae, including all characters cited here that also apply to other berardiines).

It would certainly be useful to list, in the diagnosis of *Berardius*, specific differences with *Archaeoziphius* and *Microberardius*.

—>We revised the diagnosis of the genus *Berardius* based partly on our own phylogenetic analysis, and changed the ‘style’ of descriptions to become it understandable to be able to

differentiate the genus from other genera within the subfamily.

3- For the very informative periotic I would suggest also providing lateral and medial views, based on the 3D model.

A minor issue: the photo of the ventral view in fig. 8a is not in the same orientation as the line drawing in fig. 8b. If not too complicated I would suggest correcting this. Another probably easier solution would be to provide a true ventral view based on the 3D model.

—>We added lateral and medial views of the periotic with additional anatomical interpretations. Also, we re-prepared the Figure 8a to fit the illustration of the same orientation as the Figure 8b. However, the orientations of figures 8a and 8b slightly differ from the exact ventral view, so we also added the picture of ‘correct’ orientation for the ventral view as the Figure 8c.

4- I fully agree that this specimen differs from *Archaeoziphius* and *Microberardius*. However, are these differences greater than the ones separating this specimen from extant *Berardius* species? You mention potential synapomorphies of *Berardius* at the level of the periotic, but unfortunately the periotic is unknown in *Archaeoziphius* and *Microberardius*, meaning that these characters could possibly represent synapomorphies for *Berardiinae* instead of *Berardius*, making the attribution of that genus not strongly supported. Furthermore, other similarities with *Berardius* may correspond to plesiomorphic features.

I would suggest reorganizing the comparison, in a way to first provide characters of the family *Ziphiidae* that can be observed in the new species, then provide characters of the subfamily *Berardiinae*, and finally derived characters of the genus *Berardius*. This last part may prove more difficult, as sister groups *Archaeoziphius* and *Microberardius* unfortunately lack ear bones.

—>This is somewhat difficult to formularize because of the incompleteness of fossil taxa including *Archaeoziphius* and *B. kobayashii* new species itself. The completeness of those three taxa including *Archaeoziphius* and *Microberardius* and *B. kobayashii* were: $25/51=49\%$ for *Microberardius*, $27/51=53\%$ for *Archaeoziphius* and $32/51=63\%$ for *B. kobayashii* new species. Five most parsimonious trees have suggested missing synapomorphies and/or autapomorphies for each node because of incompleteness of specimens. But three of those five consensus trees have suggested that the new species was the closest to *Berardius minimus* with synapomorphies, and other two were kinds of polytomy with *B. minimus* and *Archaeoziphius* without sufficient synapomorphies. So, we decided somewhat conservatively to describe a new species within *Berardius*. However, we revised the Comparisons section following the above advice based on

our analysis.

5- A phylogenetic tree is provided, but it is not made clear how the relationships of the new species with other ziphiids were investigated. My impression is that you used a tree from a previous analysis, before manually adding the new species. This should probably be explained. If this is the case, I would tend to think that testing the affinities of the new species with a proper phylogenetic analysis would provide a more convenient base for the systematic and taxonomic results, as well as for the discussion on paleobiogeography and the evolution of the genus *Berardius*. As currently presented, the discussion may sound a bit too speculative. In other words, I think that the attribution of the new species to the genus *Berardius* should be backed up by a phylogeny.

—>The reviewer's recommendation is very fundamental for our treatment regarding the new species described here with taxonomic considerations without the proper phylogenetic analysis. Accordingly, we did our own cladistic analysis based on characters and matrix prepared by Bianucci et al. (2016) with additions of three taxa that were not included in Bianucci et al. (2016): i.e., our new taxon, *Berardius minimus* Yamada et al., 2019 that was described after the publication of Bianucci et al. (2016), and *Berardius bairdii*, which was probably "represented" by *B. arnuxii* as the extant genus and was not included in their analysis. We located *B. kobayashii* in the cladogram more precisely based on the result of our own analysis and re-prepared the Figure 12 based on our result. Unfortunately, the consensus tree from five most parsimonious trees were a kind of unresolved polytomy, but we put additional discussion based on our analysis.

6- It may be interesting to note that divergence date estimates for the genus *Berardius* (at that time only including *B. bairdii* and *B. arnuxii*) are considerably younger (mean = 2.9 Ma) than the age of the new species (see McGowen et al. 2009).

McGowen, M.R., Spaulding, M., and Gatesy, J. 2009. Divergence date estimation and a comprehensive molecular tree of extant cetaceans. *Molecular Phylogenetics and Evolution* 53 (3): 891-906.

—>The divergence date estimation for the extant species has been established by the above paper (2009), and the latest paper (2020) also by the same authors has estimated them as the maximum and minimum time estimations of their divergences based on the hard and soft bounds methods

(McGowen et al., 2020). Based on that paper, the divergence time estimation for the subfamily Berardiinae is to be 21.98 Ma (95% CI is 17.11-29 Ma). They also estimated the split of the two species of *Berardius* is to be 2.9 Ma (95% CI is 0.68-5.81 Ma). So, the expected divergence time of the genus *Berardius* and its closest genus *Archaeoziphius* could still spans 15-5.81 Ma in minimum estimation. In addition, recently described new species of the genus, i.e., *B. minimus* is recognized as an earlier diverging species from *B. bairdii* and *B. arnuxii* clade (Kitamura et al., 2013; Morin et al., 2017; Yamada et al., 2019). We do not know how to estimate the “longevity” of the genus properly, but we can recognize it based on the relationship of the fossil taxon to other species. In the case of a new fossil described in our MS, it was recognized a sister taxon to the extant species within the extant genus; i.e., *Berardius minimus*, despite the phylogenetic position of *Archaeoziphius* was not stable in terms of the monophyly of the species within the genus *Berardius* because of the above reason (missing entries of characters in fossil taxa). In fact, they share quite a few characters used in Bianucci et al. (2016) and in our analysis. Accordingly, we preferred conservatively to keep the generic status for the new species based on our result until much better specimen(s) of *Microberardius* and especially more closely related *Archaeoziphius* are obtained.

Reviewer: 2

Comments to the Author(s)

Dear

Kawatani, Ayako; Kohno, Naoki

I have done an extensive revision of your work and found the overall purpose of the manuscript interesting and useful. Despite this, I do believe that the work in its current state will need a substantial revision in order to be publishable. Below you'll find my comments:

1) I found the manuscript inconsistently written and difficult to read. I would strongly advise to send the manuscript to a native English speaker to help with the drafting of the paper. As a non-native English speaker, I completely understand the fact that the authors could have made more mistakes than usual, as this is the case with me. There are several parts where the inconsistency of the manuscript (grammar, bad translations or changes between tenses) makes it mostly difficult

to read, or the idea in the sentences is lost.

—>We corrected all such grammatical and typographic errors following suggestions on the marked pdf from both reviewers. We are ashamed of such problems, and we strongly appreciate to the reviewer's kind advice and corrections.

2) Proposing a new species, especially for an extant genus, must have a strong phylogenetic framework in order to support this assignation. Since this is a work on Systematics it should be the center of the manuscript. I did not find major references to it, but a figure and with some lines in text:

- a. There is no description of the trees, statistical support, consensus, or even references to the methodology undertaken. A simple statement as 'it follows Bianucci et al. 2016' is not enough.
- b. No list of plesiomorphies, synapomorphies or apomorphies that support each of the Systematical assignations is specified, but some references in the 'Comparisons' section. If this is not provided, the support of the new species cannot be easily corroborated or compared, either in reviewing process or by other researchers interested in this work.

—>Another reviewer has also pointed out that it was very fundamental problem for our treatment regarding the description of a new species in the recent genus without sufficient phylogenetic analysis. So, we did it. Initially, we used a tree from the result of the phylogenetic analysis in Bianucci et al. (2016) and manually added the new species based on our observations of shared characters among species in the subfamily Berardiinae. In our revised MS, we did our own cladistic analysis based on characters and matrix prepared by Bianucci et al. (2016) with additions of three taxa that were not included in Bianucci et al. (2016): i.e., our new taxon from Japan, *Berardius minimus* that was described by Yamada et al., (2019) after the publication of Bianucci et al. (2016), and *Berardius bairdii* that was also not included in their analysis. We located *B. kobayashii*, a new species described in our MS, in the cladogram more precisely based on the result of our own analysis and re-prepared the Comparisons and Discussion sections with the new figures 12 and 13. Unfortunately, we obtained five most parsimonious trees, and the consensus tree from five most parsimonious trees were a kind of unresolved poritomy, but we added those results in our discussion.

3) A differential diagnosis of the new species is missing. This section is key to check both the apomorphies that support the new species, and the synapomorphies that it shares with other closely related taxa. Furthermore, only a single apomorphy (provided in the abstract) supports the

new species? (Being the ‘smallest’ is not a diagnostic feature; it could be important to identify it, but how can it be cladistical coded?). References to this should be in the abstract, some features can be found on the ‘diagnosis of the species’ section, but not clear.

—>We revised the diagnosis of the subfamily Berardiinae, genus *Berardius* and a new species *B. kobayashii* based mainly on our phylogenetic analysis, and changed the ‘style’ of descriptions to become it understandable to be able to differentiate the genus and species from other genera and species within the subfamily Berardiinae.

4) The description is mostly ambiguous and difficult to read. I would highly advise to rewrite the descriptions based on regions or views. Please see comments in text. Squamosal is almost 100% complete and has several major characters of odontocetes, a few lines to describe it is not enough.

—>Again, we appreciate to the reviewer’s critical and substantial advice (with corrections) and suggestion. We re-prepared descriptions fundamentally following comments and suggestions from both reviewers, and added descriptions of, for instance, the lateral and medial laminae, squamosal, frontal and so on, based on our additional observations to specimens. We also re-described such characters that were used in the cladistic analysis more clearly. The descriptions and discussion for morphological comparisons among the three extinct taxa; i.e., *Microberardius*, *Archaeoziphius* and a new taxon *B. kobayashii*. Unfortunately, the completeness of those three extinct taxa are at present $25/51=49\%$ for *Microberardius*, $27/51=53\%$ for *archaeoziphius* and $32/51=63\%$ for *B. kobayashii* new species, but we believe that the description became more informative following the advice from both reviewers. Once again, we appreciate to the reviewers’ critical and substantial advice.

5) The comparisons section is okay, as it emphasizes the comparison between taxa. However, I would encourage the authors to discuss the specific variations among taxa in the Discussion. A suitable way to reorder both the Comparisons and Discussion could be to briefly present the phylogenetic synapomorphies + how this support the assignation to: Ziphiida, Berardiinae, *Berardius* in the Comparison section, with references to other taxa; and emphasize the most important characters in the Discussion section. Furthermore, key characteristics that delimit the genus *Berardius* should be referenced in the Discussion.

—>We re-organized the Comparisons and Discussion sections following the advice from both reviewers with clade-based comparisons to emphasize important characters to support our result. Based on our cladistic analysis, we re-organized them based on our result. Although we have

obtained five most parsimonious trees resulting partially unresolved poritomy, we put those results in our comparisons and discussion, and reflected in the revised figures 12 and 13.

Finally, I wish to emphasize that the material presented in this manuscript could be easily It has a lot of potential, but several deficiencies still need to be corrected.

—>Once again, we appreciate to the reviewer's critical and substantial advice.

Sincerely,

Ayako Kawatani

Department of Life and Environmental Sciences, University of Tsukuba

kawatani@geol.tsukuba.ac.jp

Naoki Kohno

Department of Geology and Paleontology, National Museum of Nature and Science and

Department of Life and Environmental Sciences, University of Tsukuba

Kohno@kahaku.go.jp

Appendix D**ROYAL SOCIETY
OPEN SCIENCE****The oldest fossil record of the extant genus *Berardius*
(*Odontoceti*, *Ziphiidae*) from the middle to late Miocene
boundary of the western North Pacific**

Journal:	Royal Society Open Science
Manuscript ID	RSOS-201152.R1
Article Type:	Research
Date Submitted by the Author:	24-Nov-2020
Complete List of Authors:	Kawatani, Ayako; University of Tsukuba Graduate School of Life and Environmental Sciences Kohno, Naoki; University of Tsukuba Graduate School of Life and Environmental Sciences; National Museum of Nature and Science, Geology and Paleontology
Subject:	palaeontology < BIOLOGY, Palaeontology < EARTH SCIENCES
Keywords:	middle Miocene, extant genus, Berardius kobayashii , sp. nov., western North Pacific
Subject Category:	Organismal and Evolutionary Biology

Author-supplied statements

Relevant information will appear here if provided.

Ethics

Does your article include research that required ethical approval or permits?:

This article does not present research with ethical considerations

Statement (if applicable):

CUST_IF_YES_ETHICS :No data available.

Data

It is a condition of publication that data, code and materials supporting your paper are made publicly available. Does your paper present new data?:

Yes

Statement (if applicable):

Additional data are available as the electronic supplementary materials (Supplementary files S1-S7: S1, list of the taxa and specimens of the berardiine ziphiids used for this study; S2, list of characters used for our cladistic analysis; S3, character/taxon matrix; S4, resultant cladograms including the unweighted strict and 50% majority-rule consensus cradograms, five equally most parsimonious cladograms with implied weighting of K=3; S5, References for character collections; S6, Nexus file; S7, TNT file). The new species has been registered in Zoobank. The LSID for this publication is: urn:lsid:zoobank.org:act:317C1452-AE2E-4A08-A473-3B507B452D2C.

Conflict of interest

I/We declare we have no competing interests

Statement (if applicable):

CUST_STATE_CONFLICT :No data available.

Authors' contributions

This paper has multiple authors and our individual contributions were as below

Statement (if applicable):

A.K. and N.K. conceived and designed the project, performed the experiment and analysed the data.

A.K. prepared SCM 5530-6. N.K. contributed reagents/materials/preparation tools/analysis tools.

A.K. and N.K. wrote the paper.

21 November 2020

Dear Editors,

First of all, we deeply thank both reviewers for their generous comments and suggestions for our manuscript on the oldest *Berardius* from Japan, and we revised our manuscript to address their concerns. The reviewers' comments and suggestions greatly improved our manuscript. In spite of the constructive reviews, we have to apologize the delay to revise our manuscript because of the recent COVID19 crisis as a second wave in August and third wave in October and now. Although delayed, we thoroughly revised the manuscript and improved it with additional data following recommendations and suggestions from both reviewers. Changes are recorded with the track and change function in the revised manuscript. We followed all the reviewers' comments. Reviewer's comments are written in Black below. Our responses are written in Blue.

Reviewers' Comments to Author:

Reviewer: 1

Comments to the Author(s)

The present work deals with a very informative ziphiid fossil skull, including the highly informative ear bones. Those elements are only rarely found associated to other cranial material. Furthermore, the stratigraphic context is relatively well constrained, providing a geological age that is rather exciting (most extinct ziphiids are found in younger deposits). Finally, the specimen displays similarities with members of the extant family Berardiinae, potentially confirming the antiquity of this clade. All together, these elements make this study important and worth publishing.

The text is well organized and the bones are finely illustrated (see a few comments below and in the annotated pdf). I found the descriptions clear and concise (maybe a bit too short), although a few anatomical interpretations may ask for a second check.

I made a series of comments in the annotated pdf, and those mostly deal with English and typos,

which should be easily fixed. My main concerns and suggestions are listed below, and those may
request some more work. I hope that those will prove useful, as this very fine, scientifically
informative specimen definitely deserves to be published.

—>We corrected all the grammatical and typographic errors following suggestions on the marked
pdf. We are ashamed of such problems, and we strongly appreciate to the reviewer's advice.

1- Concerning the stratigraphic context, you reach the conclusion that the formation from which
the studied fossil originates can be dated from an interval spanning 12.3–11.5 Ma. This means
that this unit may extend in the earliest Tortonian (starting at 11.63 Ma). If this is correct it may
be difficult to exclude an early late Miocene age, and I would suggest modifying the text (and the
title + abstract) accordingly. Maybe 'from the latest middle to earliest late Miocene', or 'from the
middle to late Miocene boundary'. This would remove any possible confusion, I think. Indeed,
many ziphiids have been dated from the Tortonian, whereas Serravallian records are much less
frequent.

—>We changed the above-mentioned sentence in our revised MS from “from the middle Miocene”
to “from the middle to early Miocene boundary” to avoid any possible confusion. The
estimation of the geologic age for the lower part of the Tsurushi Formation that produced the
holotype skull was based on the diatom fossils. So, the “minimum” age estimation is actually
crossing the boundary between the middle and late Miocene. Again, we appreciate to the
reviewer's advice.

2- The emended diagnosis of *Berardius* is long, and it is not easy to understand how it was
compiled, for several reasons. First, some characters are also listed for the subfamily Berardiinae.
Second, differences are not associated to other genera. It may be ok to keep such a list, but then
it should start with something like 'The genus *Berardius* differs from all other ziphiids with the
following unique combination of characters: ...'. However, because you provide a diagnosis for
Berardiinae just above, it may be sufficient to provide here differences with other Berardiinae
(and maybe expand somewhat the diagnosis for Berardiinae, including all characters cited here
that also apply to other berardiines).

It would certainly be useful to list, in the diagnosis of *Berardius*, specific differences with
*Archaeoziphius* and *Microberardius*.

—>We revised the diagnosis of the genus *Berardius* based partly on our own phylogenetic
analysis, and changed the ‘style’ of descriptions to become it understandable to be able to

differentiate the genus from other genera within the subfamily.

3- For the very informative periotic I would suggest also providing lateral and medial views,
based on the 3D model.

A minor issue: the photo of the ventral view in fig. 8a is not in the same orientation as the line
drawing in fig. 8b. If not too complicated I would suggest correcting this. Another probably easier
solution would be to provide a true ventral view based on the 3D model.

—>We added lateral and medial views of the periotic with additional anatomical interpretations.
Also, we re-prepared the Figure 8a to fit the illustration of the same orientation as the Figure 8b.
However, the orientations of figures 8a and 8b slightly differ from the exact ventral view, so we
also added the picture of ‘correct’ orientation for the ventral view as the Figure 8c.

4- I fully agree that this specimen differs from *Archaeoziphius* and *Microberardius*. However, are
these differences greater than the ones separating this specimen from extant *Berardius* species?
You mention potential synapomorphies of *Berardius* at the level of the periotic, but unfortunately
the periotic is unknown in *Archaeoziphius* and *Microberardius*, meaning that these characters
could possibly represent synapomorphies for *Berardiinae* instead of *Berardius*, making the
attribution of that genus not strongly supported. Furthermore, other similarities with *Berardius*
may correspond to plesiomorphic features.

I would suggest reorganizing the comparison, in a way to first provide characters of the family
*Ziphiidae* that can be observed in the new species, then provide characters of the subfamily
*Berardiinae*, and finally derived characters of the genus *Berardius*. This last part may prove more
difficult, as sister groups *Archaeoziphius* and *Microberardius* unfortunately lack ear bones.

—>This is somewhat difficult to formularize because of the incompleteness of fossil taxa
including *Archaeoziphius* and *B. kobayashii* new species itself. The completeness of those three
taxa including *Archaeoziphius* and *Microberardius* and *B. kobayashii* were: $25/51=49\%$ for
*Microberardius*, $27/51=53\%$ for *Archaeoziphius* and $32/51=63\%$ for *B. kobayashii* new species.
Five most parsimonious trees have suggested missing synapomorphies and/or autapomorphies for
each node because of incompleteness of specimens. But three of those five consensus trees have
suggested that the new species was the closest to *Berardius minimus* with synapomorphies, and
other two were kinds of polytomy with *B. minimus* and *Archaeoziphius* without sufficient
synapomorphies. So, we decided somewhat conservatively to describe a new species within
*Berardius*. However, we revised the Comparisons section following the above advice based on

our analysis.

5- A phylogenetic tree is provided, but it is not made clear how the relationships of the new
species with other ziphiids were investigated. My impression is that you used a tree from a
previous analysis, before manually adding the new species. This should probably be explained. If
this is the case, I would tend to think that testing the affinities of the new species with a proper
phylogenetic analysis would provide a more convenient base for the systematic and taxonomic
results, as well as for the discussion on paleobiogeography and the evolution of the genus
Berardius. As currently presented, the discussion may sound a bit too speculative. In other words,
I think that the attribution of the new species to the genus Berardius should be backed up by a
phylogeny.

—>The reviewer's recommendation is very fundamental for our treatment regarding the new
species described here with taxonomic considerations without the proper phylogenetic analysis.
Accordingly, we did our own cladistic analysis based on characters and matrix prepared by
Bianucci et al. (2016) with additions of three taxa that were not included in Bianucci et al. (2016):
i.e., our new taxon, *Berardius minimus* Yamada et al., 2019 that was described after the
publication of Bianucci et al. (2016), and *Berardius bairdii*, which was probably "represented"
by *B. arnuxii* as the extant genus and was not included in their analysis. We located *B.*
*kobayashii* in the cladogram more precisely based on the result of our own analysis and re-
prepared the Figure 12 based on our result. Unfortunately, the consensus tree from five most
parsimonious trees were a kind of unresolved polytomy, but we put additional discussion based
on our analysis.

6- It may be interesting to note that divergence date estimates for the genus Berardius (at that time
only including *B. bairdii* and *B. arnuxii*) are considerably younger (mean = 2.9 Ma) than the age
of the new species (see McGowen et al. 2009).

McGowen, M.R., Spaulding, M., and Gatesy, J. 2009. Divergence date estimation and a
comprehensive molecular tree of extant cetaceans. *Molecular Phylogenetics and Evolution* 53 (3):
891-906.

—>The divergence date estimation for the extant species has been established by the above paper
(2009), and the latest paper (2020) also by the same authors has estimated them as the maximum
and minimum time estimations of their divergences based on the hard and soft bounds methods

(McGowen et al., 2020). Based on that paper, the divergence time estimation for the subfamily
Berardiinae is to be 21.98 Ma (95% CI is 17.11-29 Ma). They also estimated the split of the two
species of *Berardius* is to be 2.9 Ma (95% CI is 0.68-5.81 Ma). So, the expected divergence
time of the genus *Berardius* and its closest genus *Archaeoziphius* could still spans 15-5.81 Ma in
minimum estimation. In addition, recently described new species of the genus, i.e., *B. minimus*
is recognized as an earlier diverging species from *B. bairdii* and *B. arnuxii* clade (Kitamura et al.,
2013; Morin et al., 2017; Yamada et al., 2019). We do not know how to estimate the “longevity”
of the genus properly, but we can recognize it based on the relationship of the fossil taxon to other
species. In the case of a new fossil described in our MS, it was recognized a sister taxon to the
extant species within the extant genus; i.e., *Berardius minimus*, despite the phylogenetic position
of *Archaeoziphius* was not stable in terms of the monophyly of the species within the genus
*Berardius* because of the above reason (missing entries of characters in fossil taxa). In fact, they
share quite a few characters used in Bianucci et al. (2016) and in our analysis. Accordingly, we
preferred conservatively to keep the generic status for the new species based on our result until
much better specimen(s) of *Microberardius* and especially more closely related *Archaeoziphius*
are obtained.

Reviewer: 2

Comments to the Author(s)

Dear

Kawatani, Ayako; Kohno, Naoki

I have done an extensive revision of your work and found the overall purpose of the manuscript
interesting and useful. Despite this, I do believe that the work in its current state will need a
substantial revision in order to be publishable. Below you'll find my comments:

1) I found the manuscript inconsistently written and difficult to read. I would strongly advise to
send the manuscript to a native English speaker to help with the drafting of the paper. As a non-
native English speaker, I completely understand the fact that the authors could have made more
mistakes than usual, as this is the case with me. There are several parts where the inconsistency
of the manuscript (grammar, bad translations or changes between tenses) makes it mostly difficult

to read, or the idea in the sentences is lost.

—>We corrected all such grammatical and typographic errors following suggestions on the
marked pdf from both reviewers. We are ashamed of such problems, and we strongly appreciate
to the reviewer's kind advice and corrections.

2) Proposing a new species, especially for an extant genus, must have a strong phylogenetic
framework in order to support this assignation. Since this is a work on Systematics it should be
the center of the manuscript. I did not find major references to it, but a figure and with some lines
in text:

a. There is no description of the trees, statistical support, consensus, or even references to the
methodology undertaken. A simple statement as 'it follows Bianucci et al. 2016' is not enough.

b. No list of plesiomorphies, synapomorphies or apomorphies that support each of the
Systematical assignations is specified, but some references in the 'Comparisons' section. If this
is not provided, the support of the new species cannot be easily corroborated or compared, either
in reviewing process or by other researchers interested in this work.

—>Another reviewer has also pointed out that it was very fundamental problem for our treatment
regarding the description of a new species in the recent genus without sufficient phylogenetic
analysis. So, we did it. Initially, we used a tree from the result of the phylogenetic analysis in
Bianucci et al. (2016) and manually added the new species based on our observations of shared
characters among species in the subfamily Berardiinae. In our revised MS, we did our own
cladistic analysis based on characters and matrix prepared by Bianucci et al. (2016) with additions
of three taxa that were not included in Bianucci et al. (2016): i.e., our new taxon from Japan,
*Berardius minimus* that was described by Yamada et al., (2019) after the publication of Bianucci
et al. (2016), and *Berardius bairdii* that was also not included in their analysis. We located *B.*
*kobayashii*, a new species described in our MS, in the cladogram more precisely based on the
result of our own analysis and re-prepared the Comparisons and Discussion sections with the new
figures 12 and 13. Unfortunately, we obtained five most parsimonious trees, and the consensus
tree from five most parsimonious trees were a kind of unresolved poritomy, but we added those
results in our discussion.

3) A differential diagnosis of the new species is missing. This section is key to check both the
apomorphies that support the new species, and the synapomorphies that it shares with other
closely related taxa. Furthermore, only a single apomorphy (provided in the abstract) supports the

new species? (Being the ‘smallest’ is not a diagnostic feature; it could be important to identify it,
but how can it be cladistical coded?). References to this should be in the abstract, some features
can be found on the ‘diagnosis of the species’ section, but not clear.

—>We revised the diagnosis of the subfamily Berardiinae, genus *Berardius* and a new species *B.*
*kobayashii* based mainly on our phylogenetic analysis, and changed the ‘style’ of descriptions to
become it understandable to be able to differentiate the genus and species from other genera and
species within the subfamily Berardiinae.

4) The description is mostly ambiguous and difficult to read. I would highly advise to rewrite the
descriptions based on regions or views. Please see comments in text. Squamosal is almost 100%
complete and has several major characters of odontocetes, a few lines to describe it is not enough.

—>Again, we appreciate to the reviewer’s critical and substantial advice (with corrections) and
suggestion. We re-prepared descriptions fundamentally following comments and suggestions
from both reviewers, and added descriptions of, for instance, the lateral and medial laminae,
squamosal, frontal and so on, based on our additional observations to specimens. We also re-
described such characters that were used in the cladistic analysis more clearly. The descriptions
and discussion for morphological comparisons among the three extinct taxa; i.e., *Microberardius*,
*Archaeoziphius* and a new taxon *B. kobayashii*. Unfortunately, the completeness of those three
extinct taxa are at present $25/51=49\%$ for *Microberardius*, $27/51=53\%$ for *archaeoziphius* and
$32/51=63\%$ for *B. kobayashii* new species, but we believe that the description became more
informative following the advice from both reviewers. Once again, we appreciate to the
reviewers’ critical and substantial advice.

5) The comparisons section is okay, as it emphasizes the comparison between taxa. However, I
would encourage the authors to discuss the specific variations among taxa in the Discussion. A
suitable way to reorder both the Comparisons and Discussion could be to briefly present the
phylogenetic synapomorphies + how this support the assignation to: Ziphiida, Berardiinae,
*Berardius* in the Comparison section, with references to other taxa; and emphasize the most
important characters in the Discussion section. Furthermore, key characteristics that delimit the
genus *Berardius* should be referenced in the Discussion.

—>We re-organized the Comparisons and Discussion sections following the advice from both
reviewers with clade-based comparisons to emphasize important characters to support our result.
Based on our cladistic analysis, we re-organized them based on our result. Although we have

obtained five most parsimonious trees resulting partially unresolved poritomy, we put those
results in our comparisons and discussion, and reflected in the revised figures 12 and 13.

Finally, I wish to emphasize that the material presented in this manuscript could be easily It has
a lot of potential, but several deficiencies still need to be corrected.

—>Once again, we appreciate to the reviewer's critical and substantial advice.

-----

Thank you for your consideration.

Sincerely,

Ayako Kawatani

Department of Life and Environmental Sciences, University of Tsukuba

kawatani@geol.tsukuba.ac.jp

Naoki Kohno

Department of Geology and Paleontology, National Museum of Nature and Science and

Department of Life and Environmental Sciences, University of Tsukuba

Kohno@kahaku.go.jp

The oldest fossil record of the extant genus *Berardius* (Odontoceti, Ziphiidae) from the middle to late Miocene boundary of the western North Pacific

Ayako Kawatani¹ and Naoki Kohno^{1,2}

¹Graduate School of Life and Environmental Sciences, University of Tsukuba, 1-1-1, Tennoudai, Tsukuba, Ibaraki, 305-8577, Japan

²National Museum of Nature and Science, 4-1-1, Amakubo, Tsukuba, Ibaraki, 305-0005, Japan

Keywords: middle Miocene, extant genus, *Berardius kobayashii*, sp. nov., western North Pacific.

1. Summary

A new species of a beaked whale that belongs to the extant genus *Berardius* is described from the middle to late Miocene boundary age Tsurushi Formation (ca. 12.3–11.5 Ma) on the Sado Island, Niigata Prefecture, Japan. The new species, *Berardius kobayashii*, sp. nov., represents the oldest record of this genus and suggests the timing of the emergence of this extant genus during this age period. *Berardius kobayashii*, sp. nov., has the following generic characters: the ratio between the width of the premaxillary crests and the width of the premaxillary sac fossae is 1.0 to 1.25, nodular frontals make isolated protuberance on the posterior part of the vertex. Among the species within the genus, *B. kobayashii*, sp. nov., shares a unique character with *B. minimus* such as the apices of the left and right hamular processes of the pterygoids contact medially, forming together a posteriorly directed medial point. In addition, *B. kobayashii*, sp. nov., displays the unique combination of both primitive and derived characters: it is extremely small in size, and the nasals are short, ratio between the length of the medial suture of nasals on vertex and the maximum width of nasals is <0.4 . *Berardius kobayashii*, sp. nov. fills the gap between the origin of the genus and later diversifications of the extant species. This discovery is also a key to elucidate the process of their emergence and habitat expansion during the latest middle and earliest late Miocene in age. 
[revised manuscript text omitted]
. We also added *Berardius bairdii* based on our own observation into the data matrix, because it was also not included in the original analysis by Bianucci et al. [3]. Character and character states were managed using Mesquite 3.61 [29]. The phylogenetic analysis based on 31 ziphiids of both extinct and extant taxa with six outgroup taxa as OTUs and 51 morphological characters was performed with TNT 1.5 using the “New Technology Search” tasked to find minimum length trees 1,000 times [30]. As was mentioned by Bianucci et al. [3], our preliminary analyses also led to less-resolved trees. So we also downweighted as the same with Bianucci et al. [3] as the homoplastic characters using the default value of 3 for the constant K of the method formulated by

Goloboff [31]. Multi-state characters were treated as ordered for 18 characters (i.e., chars. 1, 3, 7, 8, 9, 11, 13, 14, 15, 16, 22, 23, 30, 34, 36, 39, 47, and 51) and unordered for six characters (i.e., chars. 2, 4, 10, 28, 34, and 45), and other binary characters were all treated as unordered, also following Bianucci et al. [3].

Institutional abbreviations are the following: IRSNB, Institut royal des Sciences naturelles de Belgique, Brussels, Belgium; MNHN: Muséum National d'Histoire Naturelle, Paris, France; NMNS-PV, Department of Geology and Paleontology, National Museum of Nature and Science, Tsukuba, Japan; NSMT-M, Department of Zoology, National Museum of Nature and Science, Tsukuba, Japan; SAM: Iziko South African Museum, Cape Town, South Africa; SCM, Sado Museum, Sado City, Niigata, Japan.

5. Systematics

CETARTIODACTYLA Montgelard, Catzefis and Douzery, 1997

CETACEA Brisson, 1762

ODONTOCETI Flower, 1867

ZIPHIIDAE Gray, 1850

BERARDIINAE Moore, 1968

Emended diagnosis of subfamily: The subfamily differs from all other ziphiids with the following unique combination of characters: the ascending process of the premaxilla in lateral view is slightly concave (Char. 7, reversal); constriction on the ascending process of the right premaxilla (between premaxillary sac fossa and the premaxillary crest) is weak, and the ratio between the minimal width of ascending process of premaxilla and the width of right premaxillary crest is >0.80 (Char. 8, reversal); no anteromedial excavation of the dorsal surface of the nasal (Char. 14, reversal); the nodular protuberance on the posterior part of the vertex is formed by the interparietal and/or the frontals (Char. 17, 0->1 & 2); and the anteromedial margin of the supraoccipital is distinctly lower than the dorsal margin of the vertex (Char. 18).

Genus *Berardius* Duvernoy, 1851

Emended diagnosis of genus: The genus differs from *Microberardius* with the following unique combination of characters: the ratio between the width of the premaxillary crests and the width of the premaxillary sac fossae is 1.0 to 1.25 (Char. 11); the nodular protuberance on the posterior part of the vertex is formed only by the frontals (Char. 17³). Following characters shared with *Berardius* species, but uncertain in *Microberardius* and *Archaeoziphius* because of lack of such portion, are potentially diagnostic for the genus: the anterior spine of the tympanic with a more or less rectilinear anterior margin (Char. 22); the posteroventral corner of the sigmoid process of the tympanic is posteriorly projected in lateral view (Char. 23); the dorsal margin of the involucrum of the tympanic that is cut by an indentation that is present and visible in medial and/or dorsal view (Char. 24).

Berardius kobayashii, sp. nov.

(Skull: Figure 3-8; Ear bone: Figure 9-11)

URN: lsid:zoobank.org:act: 317C1452-AE2E-4A08-A473-3B507B452D2C

Diagnosis of species: The species differs from all other *Berardius* species (i.e., *B. arnuxii*, *B. bairdii* and *B. minimus*) with the following unique characters: the nasals are short, ratio between the length of medial suture of nasals on the vertex and the maximum width of nasals <0.4 (Char. 13, reversal). Although it is potentially a shared primitive trait with *Archaeoziphius microglenoideus*, an extremely small size among the four species (60-70 % smaller than *B. bairdii* and *B. arnuxii*, 50% smaller than *B. minimus*) is potentially the specific trait within the genus.

Holotype: SCM 5530-6, incomplete skull with the left tympanic bulla and periotic, lacking the rostral portion; the posterior fourth of the horizontal ramus of the left dentary with the coronoid process and mandibular condyle. Collected by Kiyoei Kaneko in or before 1969.

Type Locality: Sobama Beach, Ogi Town, Sado Island, Niigata Prefecture, Japan. Geographic coordinates: 37° 84'50" N, 138° 26'58" E.

Formation and age of holotype: SCM 5530-6 was collected from the latest middle to earliest late Miocene age Tsurushi Formation. As discussed above, the geologic age of the Tsurushi Formation is estimated to be approximately 12.3–11.5 Ma.

Etymology: The species is named in honor of the late Dr. Iwao Kobayashi, a professor emeritus at Niigata University, for his longstanding contributions to the geology and paleontology of the northern Fossa Magna region including Niigata Prefecture, and in gratitude for his permission to re-study the marine mammal fossils collected by him and his colleagues from the Sado Island and encouragement to both of us throughout this study.

Description

Cranium (general morphology)

The cranium is relatively small (Table 1) and slightly asymmetric (Figure 3,5). The ratio between the height and width of the skull is 1:1.19. The rostral and the occipital parts are almost completely broken and missing. The small part of the left mandible was originally attached to the cranium. The vertex is moderately elevated and slightly skewed toward the left. The posterior part of the vertex is almost broken. The outline of the temporal fossa in lateral view is an isosceles triangle with long sides that converge anteriorly, and the posterior margin is weakly rounded. The temporal fossa is shallow and its lateral surface is convex. The sutures of the skull are still visible but tightly closed, indicating that the individual is an adult.

Premaxilla

The premaxilla is well preserved at the level posterior to the antorbital notch, and its dorsal surface is almost flat. The premaxillary crest is present on the vertex. The medial margins of the left and right premaxillae are very close to each other. Contact between the nasal and the premaxillary crest is extended along almost the whole length of the nasal. The premaxillary sac fossae are weakly asymmetrical and do not overhang the maxillae laterally. The premaxillary sac fossa is almost flat and not deeply excavated. The ascending process of the premaxilla gradually rises toward the vertex. The angle between the dorsal surface of the rostrum and the ascending process of the premaxilla is 150°. It is slightly concave in lateral view. The ascending process of the left premaxilla is somewhat damaged, but that of the right premaxilla is well preserved, and its lateral margin is moderately constricted at the level along the bony nares at the midpoint of the premaxillary crest. The vertex is gently elevated. In anterior view, lateral expansion of the premaxillae at the level of the premaxillary sac fossae is prominent. The premaxillary crest is transversely oriented, and it is narrow in transverse width. The distance between the premaxillary crests is wide. The anterior margin of the bony nares is V-shaped and the overall shape of the bony nares is triangular in outline. The anteroposterior diameter (67 mm) of the bony nares is greater than the transverse diameter (39 mm). The small premaxillary foramina on the premaxilla are located roughly at the level of the antorbital notch and slightly posterior to the infraorbital foramina on the maxillae. There are distinct grooves anterior to the foramina.

Maxilla

The maxilla is broad, but its lateral margin is damaged on both sides at the level posterior to the antorbital notch. In anterior view, the rostral maxillary crest is present and moderately elevated on the maxilla. The dorsal infraorbital foramina open anterodorsally on the maxillae near the base of the rostrum, and the largest foramen is located on the right maxilla. There are two dorsal infraorbital foramina on the left maxilla with diameters of 13 mm and 7 mm. There is one large dorsal infraorbital foramen on the right maxilla, with a diameter of 16 mm. The maxilla is slightly elevated at the posterior side of these foramina. There are no excrescences on the dorsal surface of the maxilla on the preserved proximal part of the rostrum. The portion of the prominent notch and related structure are somewhat weathered away, but it is low and flat.

Nasals

The nasal is short, and its dorsal outline is trapezoidal. Although the posterior margin of the right nasal is broken, it is slightly broader than the left one. The dorsal surface of the nasal is roughly flat, and there is a weak anteromedial excavation. The nasal is not included in the premaxillary crest. The posterior nasal-frontal suture is only preserved on the left nasal and is posteriorly convex.

Frontal

The frontal is fully covered by the maxilla in the supraorbital region and only visible as a vertical section beneath the overlapping maxilla in the orbital region and temporal fossae in lateral view. It is dorsoventrally thick and relatively dense but not osteosclerotic. It also makes a dorsal roof of the antorbital and fossa in ventral view. The pre- and supraorbital area is indicated by the slight flange at the level posterior to the antorbital notch. The antorbital fossa is shallow and flares posteriorly to the slightly thickening and anterolaterally extended postorbital ridge. The posterodorsal part of the frontal around the posterior part of the vertex is broken away.

Supraoccipital

The supraoccipital is almost completely broken away, but the anteromedial margin of the supraoccipital is distinctly lower than the dorsal margin of the vertex in posterior view (Fig. 6).

Pterygoid

The pterygoid is dorsoventrally deep and excavated by the anteroposteriorly and dorsoventrally wide hamular fossa. The lateral lamina is preserved on the right pterygoid. It is thin and low at its base. There is a similar structure like transverse crests on the surface of the medial lamina. There are distinct transverse crests on the medial lamina of the pterygoid. The hamular fossa of the pterygoid sinus is wide, extending anteriorly on the palatal surface of the rostrum. Apices of the left and right hamular processes are incomplete, but both processes contact medially and form together a posteriorly directed medial point. Also each process sends a posterolateral point, so the posterior margin of the pterygoid processes are rounded W in shape (Fig. 6).

Alisphenoid

The alisphenoid is well preserved on the right side, just posterior to the pterygoid sinus and ventrally exposed as a distinct concavity that is directed anterolaterally. A thin plate-like crest runs anterolateral-posteromedial direction, which is thought to be the alisphenoid-squamosal suture, and it separates this concavity posteriorly from the tympanosquamosal recess. This concavity on the alisphenoid continues posteromedially to the large foramen ovale.

Squamosal

The left squamosal is well preserved. The temporal fossa on the squamosal is anteroposteriorly long and dorsoventrally shallow, and its surface is slightly concave. The zygomatic process of the left squamosal is short anteroposteriorly compared to its dorsoventral height. It is well developed and anterodorsally tapered with a triangle outline, but its anterior margin is crescentic. The glenoid cavity is wide and shallow, but its medial part is deep just in front of the obliquely extended postglenoid process. It continues to the tympanosquamosal recess. The tympanosquamosal recess extends posteriorly to the fused periotic. The surface of the posterior part of the tympanosquamosal recess just in front of the periotic is rugose with numerous pits and crests. The falciform process presents at the anterior process of the periotic, and it is short and thin. The ventral margin of the postglenoid process is more dorsally located than the ventral margin of the paroccipital process of the exoccipital (Fig. 4).

Mandible

The mandibles are almost broken away, but the ascending ramus of the left mandible with the coronoid process and the mandibular condyle is partially preserved (Figure 7). The posterior margin of the angular process is also broken away. The mandibular body is very thin and fragile in the portion of the lateral wall of the mandibular foramen. Its outer or lateral surface is smooth and

slightly swelled laterally, forming the condyloid crest. Its inner or medial surface (or the lateral wall
of the mandibular foramen) is relatively rough. The mandibular foramen is anteroposteriorly long and
dorsoventrally wide. The coronoid crest is well developed as an anteroposteriorly extended ridge and
continues to the coronoid process at the posterodorsal end of the mandible. The ventral margin of the
coronoid crest is weakly concave along the crest. The mandibular condyle protrudes posteriorly and
is located almost at the same anteroposterior level as the end of the coronoid process. It is
dorsoventrally long and laterally crescentic, and it is slightly swelled posteriorly. There is a notch
between the coronoid process and the mandibular condyle.

**Periotic**

The left periotic is preserved (Table 2). It is relatively slender in shape. The dorsal surface of the
body is smooth. The anterior and posterior processes are almost straight anteroposteriorly in ventral
view. The anterior process is laterally thick, but its apex is slender and pointed (Figure 9A). It is
weakly bent ventrally. The anterior bullar facet is weakly excavated anteroposteriorly. A rounded
accessory ossicle of the tympanic bulla is still articulated with the fovea epitubaria at the
posteromedial base of the anterior process. The malleolar fossa is excavated posterior to the accessory
ossicle and opens posteriorly. It is laterally delimited by a mediolaterally elongated lateral tuberosity.
The cochlear portion (=pars cochlearis) is rounded in outline, relatively low, and it is slightly bent
anteroventrally with a weak depression at the anteromedial surface, so the anterior incisure is narrow
and slightly covered with the anterior edge of the cochlear portion. The epitympanic hiatus is
relatively deep and wide. The internal acoustic meatus containing the spiral cribriform tract and the
foramen singulare is separated by a developed transverse crest from the proximal opening of the
facial canal. The spiral cribriform tract is relatively small and almost circular, slightly compressed
transversely. Its posterolateral margin is elevated and reaches almost the top of the cochlear portion.
The foramen singulare opens in the same aperture for the facial canal (Figure 9B). The proximal
opening of the facial canal is relatively large and elliptical in shape. The aperture for the cochlear
aqueduct is large and semicircular. It is located close to the posterior margin of the internal auditory
meatus. It is also relatively close to the rounded aperture for the vestibular aqueduct (=endolymphatic
duct) that is elliptical in shape. The fenestra rotunda is relatively large and semicircular in outline.
The posterior process is relatively long posteriorly and inflated mediolaterally. It is fused to the skull
in the mediolaterally narrow periotic fossa of the skull. The posterior bullar surface is covered by the
fused fragment of the posterior process of the tympanic bulla. It is slightly bent posterolaterally at the
base just posterior to the cochlear portion. Because of this fusion of the posterior process of the
tympanic bulla and its posterior expansion, the portion of the posterior bullar facet is posteriorly
flared and fan-shape in ventral view (Figure 9A). Although mostly covered by the remnant of the
posterior process of the tympanic bulla, the surface of the posterior bullar facet is smooth.

**Tympanic Bulla**

The left tympanic bulla is preserved (Table 2). The anterior part of the ventral wall of the tympanic
bulla is cylindrical and is transversely rectilinear in outline. The involucrem is shorter anteriorly than
the lateral wall. The dorsal margin of the involucrem is cut by an indentation. The inner and outer
posterior prominences are thick and short, and they are ventrally bulging. The outer posterior
prominence is much bulged ventrally than the inner one. The median furrow runs anteroposteriorly
on the ventral wall, and it is transversely narrow and anteriorly shallow. The interprominential notch
is relatively deep at the posterior end of the median furrow. The lateral furrow is shallow, and its
posterior ridge is prominent and twisted on the lateral wall of the tympanic bulla. The sigmoid
process is square in anterior view, and it is anteroposteriorly thin. The posteroventral corner of the
sigmoid process is projected posteriorly. The lateral border of the sigmoid process is twisted
posteriorly. A damaged base of the posterior process lies on the posterolateral end of the sigmoidal
process. The elliptical foramen is present (Figure 10, 11).

6. Results of the Phylogenetic Analysis

We compared the characteristics of SCM 5530-6 with both extant and fossil Ziphiidae including 27 genera and 55 species based mainly on Bianucci et al. [3] and Lambert et al. [34]. Because SCM 5530-6 described here from the middle to upper Miocene boundary Tsurushi Formation was initially reported by Takahashi et al. [14] as a species of the Ziphiidae based on the posteriorly located bony nares with gently elevated ascending process of the premaxilla, and the posteroventrally steeply sloped occipital. In addition to the above characters, SCM 5530-6 has such diagnostic characters of the family Ziphiidae at least as the presence of the premaxillary crests on the gently elevated vertex, wide and anteriorly extended hamular fossa of the pterygoid sinus [3, 34], we analyzed its phylogenetic relationship based on the data matrix and parsimony settings prepared by Bianucci et al. [3] to locate SCM 5530-6 in the Ziphiidae with the addition of the North Pacific Baird's beaked whale, *Berardius bairdii* that was not included in their analysis and the recently described Sato's beaked whale, *Berardius minimus* of the northern North Pacific [2], as well as SCM 5530-6.

As a result of the analysis, five most parsimonious trees were obtained with a tree length of 177 steps in each, Goloboff fit ($k=3$) -58.44286, the consistency index (CI) of 0.475, and the retention index (RI) of 0.767. The 50% majority rule consensus tree is shown in Figure 12. The strict consensus tree is also shown in the Supplementary Data S4. Our result suggests that *Chavinziphius* from the late Miocene of Peru (the South Pacific) is the basal to the clade including all the extant subfamilies, i.e., the Berardiinae, Ziphiinae, and Hyperoodontinae. Instead, *Ninoziphius* also from the Miocene of Peru is recognized as the most basal to the Ziphiidae as a whole. As for SCM 5530-6, it was nested in the monophyletic subfamily Berardiinae consisting of the genera *Berardius*, *Archaeoziphius* and *Microberardius* ([3], [36] and Figure 12).

The Berardiinae is supported by five synapomorphies: slightly concave ascending process of the premaxilla (Char.7: as reversal); a weak constriction on the ascending process of the right premaxilla (between premaxillary sac fossa and premaxillary crest) and the ratio between the minimal width of the ascending process of premaxilla and the width of the right premaxillary crest is > 0.80 (Char.8: as reversal); no anteromedial excavation of the dorsal surface of the nasal (Char.14: as reversal); an isolated rounded protuberance formed by the interparietal and/or frontals on the posterior part of the vertex (Char.17), and the anteromedial margin of the supraoccipital that is lower than the frontals on the vertex (Chars. 18) (see also [3, 29, 33]). The following character is also considered to be a potentially unique character for the subfamily: the posteromedial narrowing nasals and frontals at the vertex, which is narrower than the anterior-most transverse width of nasals (Char. 50). Among those characters, quite a few synapomorphies of the subfamily are interpreted as reversals from derived to secondarily primitive character states. However, the extant ziphiids are deep-diving whales having adapted to deep sea with considerable ecomorphological specializations during their adaptation and diversification in each clade, and they might have yielded unique characters frequently to adapt such a harsh environment. Other examples are also interesting to be noted. A low elevation of the ascending process of the premaxilla is observed in *Berardius* and related fossil species such as *Archaeoziphius microglenoideus* and *Microberardius africanus* [3, 32, 36], though this condition is also seen in *Chavinziphius maxillocrestatus*, one of the basal ziphiids [3]. Another example is a cluster of dorsal infraorbital foramina, which is present on the base of the rostrum in the early diverging ziphiids [3]. On the other hand, only a large foramen is present in the crown ziphiids [3]. SCM 5530-6 has a large foramen on the right maxilla, and two relatively large foramina on the left maxilla, and therefore, the condition of these foramina is thought to be the condition seen in the later diverging, extant ziphiids. However, *Ninoziphius platyrostris* from the Miocene of Peru, which is considered to be the most basal ziphiid (Figure 12), has a relatively large dorsal infraorbital foramen [34], suggesting that this character state might have been multiply derived in two distantly related clades. This case is also suggestive that some characters seen in the berardiines are not appropriate to be considered synplesiomorphies conservatively retained in them as a subfamily having earlier-branching, deeper node.

Among the species within the Berardiinae, *Microberardius africanus* was recognized as the first split with the rest of genera and species including SCM 5530-6, but no autapomorphies were recognized to characterize *Microberardius* by our analysis. In this regard, *Microberardius* seems to be "ancestral" to the later diverging berardiines. However, Bianucci et al. [3] suggests that *Microberardius* is distinguishable from other berardiines by having the following characters: the

rostrum is higher than wide along most of its length, the rostral base is narrower, and the maxillary crest is not extended on the rostral base.

By contrast, the monophyly of the species belonging in the genera *Berardius* and *Archaeoziphius* including SCM 5530-6 is strongly supported by the following unique character combinations: the ratio between the width of the premaxillary crest and that of the premaxillary sac fossa of SCM 5530-6 is moderate, from 1.0 to 1.25 (Char. 11); an isolated rounded protuberance formed by the frontals, not the interparietal, on the posterior part of the vertex (Char. 17, 0->2).

Although, the basic framework of the relationships among the genera of the Berardiinae (i.e., [*Microberardius*, [*Archaeoziphius*, *Berardius*]]), which was originally demonstrated by Bianucci et al. [3], is also supported by our analysis, *Archaeoziphius microglenoideus* was bracketed in between the clade containing two other extant species of *Berardius* (i.e., *B. bairdii* and *B. arnuxii*) and the clade containing newly added *Berardius minimus* plus SCM 5530-6 in our analysis. However, 27 out of 51 (i.e., 53%) characters used in Bianucci et al. [3] and in our analysis are missing and uncertain in *A. microglenoideus*, and therefore, its phylogenetic position is still unstable (see also the Supplementary Data S4). Because of overall primitiveness of observable characters in *Archaeoziphius*, the position of this genus is still controversial and is potentially anticipated to be "outside" from the clade of the genus *Berardius* as a sister taxon to the latter. In this regard, the monophyly of the genus *Berardius* including SCM 5530-6 is highly considerable, and the position of SCM 5530-6 as a sister taxon to *B. minimus* makes SCM 5530-6 a species of *Berardius* in the Berardiinae. As for the specific relationship of SCM 5530-6 to other species in the genus, SCM 5530-6 is the closest to *B. minimus* as was mentioned above in having the apices of the left and right hamular process of the pterygoids that contact medially, forming together a posteriorly directed medial point (Char. 36). In addition, the lateral expansion of the premaxillae at the posterior halves in SCM 55306 is prominent as in *B. minimus*. Furthermore, the temporal fossa of SCM 5530-6 is shallow and its lateral surface is convex also as in *B. minimus*. By contrast, it is definitely concave in *B. bairdii* and *B. arnuxii*. As for skull sutures, *B. minimus* has much tighter sutures as has pointed out by Yamada et al. [2], SCM 5530-6 also has much tighter sutures compared with those in *Berardius bairdii* and *B. arnuxii*. In fact, three out of five most parsimonious trees support that SCM 5530-6 is the sister taxon of *B. minimus* (Supplementary Data S4). Finally, SCM 5530-6 is distinguishable from *B. minimus* in having the short nasal, the ratio between the length of medial suture of nasals on the vertex and the maximum width of nasals < 0.4 (Char.13, as reversal). Furthermore, the skull size, based on the bizygomatic width, of SCM 5530-6 is 50% smaller than *B. minimus* as was mentioned above. Consequently, identification of SCM 5530-6 as a distinctive species within the genus *Berardius* is warranted. Thus, we propose *Berardius kobayashii*, sp. nov. based on SCM 5530-6 from the middle to late Miocene boundary age Tsurushi Formation.

7. Discussion

In previous studies, Takahashi et al. [14] identified SCM 5530-6, now the holotype of *B. kobayashii* sp. nov., as "provisionally *Berardius* sp. indet." based on the following characteristics: the external bony nares are triangular in outline; the vertex is gently elevated; and the posterior part of the pterygoid (though it was misidentified as median part of the palate in [14, p.104]) is ventrally well developed. However, they compared SCM 5530-6 only with some extant genera of the Ziphiidae; i.e., *Ziphius*, *Berardius*, and *Mesoplodon*, and they also pointed out that the transverse width of the premaxilla of SCM 5530-6 is wider relative to the width of the maxilla at the level of the antorbital notch than that of then extant two species of *Berardius*, and that the lateral margin of the same portion is curved laterally in contrast to the relatively straight condition of this portion in both species of *Berardius*. In addition, they also pointed out that the zygomatic process of the squamosal is shaped like an anterodorsally tapered triangle in contrast to an anteriorly sub-squared rectangle in both species of *Berardius*. In this regard, both inter- and intra-generic comparisons were quite limited to their comparable taxa among the extant ziphiids, and the generality of their taxonomical differentiations of characters were somewhat arbitrary. In addition, two new genera of the Berardiinae were described as mentioned above [32, 36], and the third extant species of *Berardius* was also added

after Takahashi et al [14] as mentioned above [3]. Accordingly, the comprehensive comparisons
among both extinct and extant species of *Berardius* and of the Berardiinae in the Ziphiidae were
necessary to locate SCM 5530-6 more precisely in the phylogenetic framework. As a result of our
cladistic analysis modified from Bianucci et al. [32], *B. kobayashii* sp. nov. was recognized as the
closest to *Berardius minimus* as discussed above. It confirms that the new species based on SCM
5530-6 as the holotype actually belongs in the genus *Berardius*, and it also indicates that although *B.*
*minimus* is represented only by the extant individuals in the northern North Pacific, the generic
emergence could have been much earlier than the emergence and diversification of the other
previously known two antitropical species; i.e., the northern North Pacific *B. bairdii* and the
Antarctic *B. arnuxii*. Consequently, the emergence and diversification of the genus *Berardius* must
have occurred in the late middle Miocene and potentially in the northern North Pacific based on the
new fossil from the middle to late Miocene boundary age Tsurushi Formation.

For now, three genera are recognized in the subfamily Berardiinae; i.e., *Microberardius* from the
middle Miocene of off South Africa [32], *Archaeoziphius* from the middle Miocene of Belgium [36]
and *Berardius* from the middle to late Miocene boundary to the Recent of the northern North Pacific
and the Antarctic oceans [1, 2, this study]. Previously, there was no fossil record of the berardiines in
the North Pacific Ocean. So, the new fossil described here reveals that berardiine ziphiids already
diversified to the western North Pacific at latest in the early late Miocene (ca. 11 Ma) and that the
emergence of the extant genus *Berardius* goes back to the middle to late Miocene boundary age of
the North Pacific, then they quickly diversified with or just after the age of their closely related
*Archaeoziphius* in the early middle Miocene of the North Atlantic [36].

At the moment, it is difficult to consider the more precise origin of the subfamily Berardiinae and
the genus *Berardius* based on the present fossil evidence, but its potential origin of the subfamily
must have been in the Southern Hemisphere on the basis of recent studies on their phylogenetic
relationships and phylogeographic distributions of sister and out groups for the berardiine ziphiids
([3, 5, 32, 33, 36] and Figure 13). In addition, the close relationship between *B. kobayashii* and *B.*
*minimus* indicates an emergence of the genus in the North Pacific, and then the ancestor of *B. bairdii*
and *B. arnuxii* may have evolved somewhere from *B. kobayashii*-like berardiine in the Northern
Hemisphere or somewhere in between the northern and southern hemispheres by the end of Miocene
[37]. It could have been followed by a dispersal (with gigantism) from the Southern to or from the
Northern hemisphere [38]. In any case, the two currently sympatric form, smaller *B. minimus* and
larger *B. bairdii* in the northern North Pacific could have been secondary contact after the Pliocene in
age [38]. However, all these evidence suggest the rapid diversification of taxa in the subfamily
Berardiinae in the initial stage of their evolution during the middle and late Miocene in the Northern
Hemisphere (Figure 13).

It is difficult to establish the generic emergence of extant taxa based only on morphologies without
genetic information such as divergence time estimations. In the case of the genus *Berardius*, its
emergence time estimation depends on the divergence time of the Berardiinae and other ziphiid
subfamilies because the Berardiinae is at present monogeneric. In this regard, McGowen et al. [39],
for instance, estimated the divergence time of the Berardiinae from other subfamilies to be 21.98 Ma
(95%CI: 17.11-29 Ma). They also estimated the differentiation of the two species of *Berardius*
species (i.e., *B. arnuxii* and *B. bairdii*) to be 2.9 Ma (95%CI: 0.68-5.81 Ma). It means that the
expected divergence time of the genus *Berardius* still spans 17.11-5.81 Ma in minimum estimation or
15-5.81 Ma based on the geologic age of the closest genus *Archaeoziphius* [36]. Although the
divergence time of *B. minimus* from the ancestor of the two other species (i.e., *B. bairdii* and *B.*
*arnuxii*) within the genus is still uncertain [2, 38, 40], the minimum estimate of the divergence time
could be dated back to at latest the early late Miocene in age based on our phylogenetic analysis that
suggested the closest relationship of *B. minimus* to the middle/late Miocene boundary age *B.*
*kobayashii*, sp. nov.

Accordingly, it is possible to interpret that the genus *Berardius* began to diversify at this period,
which is known to be the period of global cooling and of coastline regression [41, 42, 43, 44]. It
should be important to be noted that somewhat speciose extant genus *Mesoplodon*, which is consisted
of at least 15 extant species, has already appeared and diversified in the late middle or early late
Miocene ([37, 39, 45]. It strongly suggests that the much earlier and less diversified appearance of
the genus *Berardius* might have proceeded their generic emergence, and *B. kobayashii*, sp. nov. could

be interpreted as one of such representatives of the earlier generic emergence and diversification. It will be further considered based on molecular studies for *B. minimus* and also morphological studies for other fossil berardiines to be described in the future.

8. Conclusions

An extinct beaked whale from the middle to late Miocene boundary age Tsurushi Formation (ca. 12.3 – 11.5 Ma) in the Sado Island, Niigata Prefecture, Japan, was described as a new species of the extant genus *Berardius*; i.e., *B. kobayashii*. It is distinguishable from all other species within *Berardius* by having the following characters: extremely small size among the four species of *Berardius* (60-70 % smaller than *B. bairdii* and *B. arnuxii*, 50% smaller than *B. minimus*); the transverse diameter of the premaxilla is relatively wide to the width of the rostrum at the level of the anterior notch; the zygomatic process of the squamosal is anterodorsally tapered and triangular in shape; the nasals are short. *Berardius kobayashii*, sp. nov., is the fourth species of the genus. Namely, the lineage leading to the extant genus of the ziphiids had already lived in the western North Pacific at the time of at latest 11 Ma. The western North Pacific including the Sea of Japan may have been one of the areas of the evolution and radiation of *Berardius* species.

Acknowledgments

We would like to thank the late I. Kobayashi (Professor Emeritus at Niigata University), A. Matsuoka (Niigata University), M. Shimizu (Asahimachi Museum, Niigata University), M. Ikarashi (Sado Museum), M. Aida (Sado Geopark Promotion Office and Sado Museum), K. Narita (Nagano City Museum), M. Kato (Nagaoka Municipal Science Museum), Y. Tajima and T.K. Yamada (NMNS), for allowing access to the specimens and giving me the opportunities to observe specimens at respective museums; Y. Yanagisawa (Geological Survey of Japan), K. Takahashi (Biwako Museum), H. Ichishima (Fukui Prefectural Dinosaur Museum), K. Nagasawa (formerly Yamagata Prefectural Museum), T. Kimura (Gunma Museum of Natural History), for providing useful discussion and advice with the literature on fossil ziphiids.

Funding Statement

A.K. was funded the travel grant for the 2017 fiscal year from the Regional Promotion Division of Sado City.

Data Accessibility

Additional data are available as the electronic supplementary materials (Supplementary files S1-S7: S1, list of the taxa and specimens of the berardiine ziphiids used for this study; S2, list of characters used for our cladistic analysis; S3, character/taxon matrix; S4, resultant cladograms including the unweighted strict and 50% majority-rule consensus cladograms, five equally most parsimonious cladograms with implied weighting of K=3; S5, References for character collections; S6, Nexus file; S7, TNT file). The new species has been registered in Zoobank. The LSID for this publication is: urn:lsid:zoobank.org:act:317C1452-AE2E-4A08-A473-3B507B452D2C.

Competing Interests

We declare we have no competing interests.

Authors' Contributions

A.K. and N.K. conceived and designed the project, performed the experiment and analysed the data. A.K. prepared SCM 5530-6. N.K. contributed reagents/materials/preparation tools/analysis tools. A.K. and N.K. wrote the paper.

References

1. Rosel P E, Archer FI, Baker CS, Boness DJ, Brownell RL, Churchill M Jr., Costa AP, Domning DP, Fordyce RE, Jefferson TA, Kinze C, Oliveira LR, Perrin WF, Wang JY, Yamada TK. 2019 List of marine mammal species and subspecies. *Committee on Taxonomy, The Society for Marine Mammalogy*.
2. Yamada TK, Kitamura S, Abe S, Tajima Y, Matsuda A, Mead JG, Matsuishi TF. 2019 Description of a new species of beaked whale (*Berardius*) found in the North Pacific. *Scientific reports* **9**(1), 1-14.
3. Bianucci G, Di Celma C, Urbina M, Lambert O. 2016 New beaked whales from the late Miocene of Peru and evidence for convergent evolution in stem and crown Ziphiidae (Cetacea, Odontoceti). *PeerJ*. **4**, e2479.
4. Lambert O, de Muizon C, Duhamel G, Van Der Plicht J. 2018 Neogene and Quaternary fossil remains of beaked whales (Cetacea, Odontoceti, Ziphiidae) from deep-sea deposits off Crozet and Kerguelen islands, Southern Ocean. *Geodiversitas* **40**(2), 135-140.
5. Lambert O, Collareta A, Landini W, Post K, Ramassamy B, Di Celma C, Urbina M, Bianucci G. 2015 No deep diving: evidence of predation on epipelagic fish for a stem beaked whale from the Late Miocene of Peru. *Proc. R. Soc. B*, 282.
6. Niigata Foraminiferal Research Group. 1967 Some foraminiferal assemblages from the Sawane area, Sado Island (Preliminary report): Restudy on the stratigraphy of the "Sawane Formation"

- based on the foraminiferal fossils. *Commemorative Volume for Mr. Y. Hiramatsu*, 115-119.
7. Watanabe K. 1987 Tertiary Foraminifera and Radiolaria Fossils from Sado Island, Central Japan. *Publications from the Sado Museum* **9**, 127-156.
8. Okamoto Y, Satou M, Watanabe M, Yamamoto H. 1992 Inversion Tectonics in the southeastern part of the Japan Sea. *Jour. Struct. Geol. Japan* **38**, 47-58.
9. Kobayashi I. 2001 The Cenozoic of Niigata Prefecture, the case of Sado Island (Niigata no shinseikai Sado-hen). *Niigata Geotechnology Society* **57**, 9-21.
10. Yanagisawa Y, Watanabe M. 2017 Revised lithostratigraphy of the Neogene sedimentary sequence in the southern part of the Osado Mountain area, Sado Island, Niigata Prefecture, Japan. *Bull. Geol. Surv. Japan* **68** (6), 259-285.
11. Sado Research Group for Marine Mammalian Fossils. 1987 Fossil Cetacean from Sado Island, Central Japan. *Publications from the Sado Museum* **9**, 211-217.
12. Horikawa H, Tazaki K, Kanno T. 1987 Fossil Ziphiidae from Koshiji-Syo off Sado Island, Central Japan. *Publications from the Sado Museum* **9**, 225-230.
13. Tazaki K, Horikawa H, Miyazaki S. 1987 Fossil Ziphiidae from Hyotan-Guri off Sado Island, Central Japan. *Publications from the Sado Museum* **9**, 219-223.
14. Takahashi K, Nomura M, Kobayashi I. 1989 A fossil cetacean skull (Berardius sp. indet.) from Dogama, Ogi-machi, Sado Island, Central Japan. *Earth Science; Chikyu Kagaku* **43**, 102-105.
15. Barnes LG, Hirota K. 1995 Miocene pinnipeds of the otariid subfamily Allodesminae in the North Pacific Ocean: Systematics and relationships. *The Island Arc* **3**, 329-360.
16. Utashiro T. 1950 The stratigraphy of Sawane-Aikawa region in Southwest Sado Island (Osado Mountain range). *Journal of the Geological Society of Japan* **56**, 302-303.
17. Shimazu M, Kanai Y, Toyama T, Ichihashi K, Minakawa J, Takahama N. 1973 Structural development and igneous activity in the Sado island. *Memoirs of the Geological Society of Japan* **9**, 147-157.
18. Shimazu M, Toyama T. 1982 Neogene volcanic rocks of the Sado islands. *Journal of the Geological Society of Japan* **88**, 381-400.
19. Yanagisawa Y. 2012 Late Miocene diatoms in the Hamochi area, Sado Island, Niigata Prefecture, Japan. *Open-File Report of the Geological Survey of Japan, AIST* **568**, 1-19.
20. Ogi Collaborative Research Group. 1986 Late Cenozoic Group in the Southern part of Kosado Mountains, Niigata prefecture. *Earth Science* **40**, 417-436.
21. Takeuchi K, Ozaki M, Komatsubara T. 2011 Explanatory notes of 1:200,000 geological map of the coastal zone around Niigata. Digital Geoscience Map, Seamless Geoinformation of Coastal Zone "Coastal Zone around Niigata".
22. Tokunaga S. 1939 A New Fossil Mammal Belonging to Desmostyliidae. Jubilee Publication in the Commemoration of Professor H. Yabe, M.I.A. Sixtieth Birthday. Sendai: Institute of Geology and Paleontology, Tohoku Imperial University. 289-299.
23. Watanabe H. 1932 Neogene (Shin Daisankei). *Nihon Chishitsu kousan shi* 92-152.
24. Maiya S. 1978 Late Cenozoic planktonic foraminiferal biostratigraphy of the oil-field region of Northeast Japan. *Cenozoic Geology of Japan*, 35-60.
25. Kawatani A, Sashida K, Sachiko A, Kohno N. 2019 Radiolarian fossils and estimated depositional age of the Miocene Tsurushi Formation distributed in Sado Island, Niigata Prefecture, Japan. *Bull. Geol. Surv. Japan* **70**(1-2), 91-99. (doi: <https://doi.org/10.9795/bullgsj.70.91>)
26. Ono K, Ueno T. 1985 Tertiary Vertebrates from Sado Island, Niigata Prefecture, Central Japan. *Memoirs of the National Science Museum* **18**, 65-71.
27. Mead JG, Fordyce RE. 2009 The therian skull: a lexicon with emphasis on the odontocetes. *Smithsonian Contributions to Zoology*, 1-249.
28. Kasuya T. 1973 Systematic consideration of recent toothed whales based on the morphology of tympanoperiotic bone. *Scientific Reports of the Whales Research Institute* **25**, 1-103.
29. Maddison WP and Maddison DR. 2018. Mesquite: a modular system for evolutionary analysis. Version 3.61. <http://www.mesquiteproject.org>.
30. Goloboff PA, Catalano SA. 2016. TNT version 1.5, including a full implementation of phylogenetic morphometrics. *Cladistics* **32**(3), 221-238.
31. Goloboff PA. 1993. Estimating character weights during tree search. *Cladistics* **9**, 83-91.
32. Bianucci G, Lambert O, Post K. 2007 A high diversity in fossil beaked whales (Mammalia, Odontoceti, Ziphiidae) recovered by trawling from the sea floor off South Africa. *Geodiversitas* **29**(4), 561-618.
33. Bianucci G, Lambert O, Post K. 2010 High concentration of long - snouted beaked whales (genus *Messapicetus*) from the Miocene of Peru. *Palaeontology* **53**(5), 1077-1098.
34. Lambert O, De Muizon C, Bianucci G. 2013 The most basal beaked whale *Ninziphius platyrostris* Muizon, 1983: clues on the evolutionary history of the family Ziphiidae (Cetacea: Odontoceti). *Zoological Journal of the Linnean Society* **167**(4), 569-598.
35. Miwa M, Yanagisawa Y, Yamada K, Irizuki T, Shoji M, Tanaka Y. 2004 Planktonic foraminiferal biostratigraphy of the Pliocene Kuwae Formation in the Tainai River section, Niigata Prefecture and the age of the base of the No. 3 Globorotalia inflata bed. *Journal of the Japanese Association for Petroleum Technology* **69**(3), 272-283.
36. Lambert O, Louwye S. 2006 *Archaeoziphius microglonoideus*, a new primitive beaked whale (Mammalia, Cetacea, Odontoceti) from the Middle Miocene of Belgium. *Journal of Vertebrate Paleontology* **26**(1), 182-191.
37. McGowen MR, Tsagkogeorga G, Álvarez-Carretero S, Dos Reis M, Struebig M, Deaville R, Jepson PD, Jarman S, Polanowski A, Morin PA, Rossiter SJ. 2020 Phylogenomic Resolution of the Cetacean Tree of Life Using Target Sequence Capture. *Systematic Biology* **69**, 479-501. DOI: 10.1093/sysbio/syz068.
38. Morin PA, Baker CS, Brewer RS, Burdin AM, Dalebout ML, Fedutin ID, Filatova OA, Jung JL, Lauf M, Potter CW, Richard G, Ridgway M, Robertson KM, Wade P. 2017 Genetic structure of the beaked whale genus *Berardius* in the North Pacific, with genetic evidence for a new species. *Marine Mammal Science* **33**, 96-111.
39. McGowen MR, Spaulding M, Gatesy J. 2009 Divergence date estimation and a comprehensive molecular tree of extant cetaceans. *Molecular Phylogenetics and Evolution* **53**, 891-906. DOI: 10.1016/j.ympev.2009.08.018.
40. Kitamura S, Matsuishi T, Yamada TK, Tajima Y, Ishikawa H, Tanabe S, Nakagawa H, Uni Y, Abe S. 2013 Two genetically distinct stocks in Baird's beaked whale (Cetacea: Ziphiidae). *Marine Mammal Science* **29**, 755-766.
41. Miller KG, Kominz MA, Browning JV, Wright JD, Mountain GS, Katz ME, Sugarman PJ, Cramer BS, Christie-Blick N, Pekar SF. 2005 The Phanerozoic record of global sea-level change. *Science* **310**, 1293-1298. <https://doi.org/10.1126/science.1116412> PMID: 16311326.
42. Zachos JC, Dickens GR, Zeebe RE. 2008 An early Cenozoic perspective on greenhouse warming and carbon-cycle dynamics. *Nature* **451**, 279-283. <https://doi.org/10.1038/nature06588> PMID: 18202643.
43. Hilgen FJ, Abels HA, Iaccarino S, Krijgsman W, Raffi I, Sprovieri R, Turco E, Zachariasse WJ. 2009 The global stratotype section and point (GSSP) of the Serravallian Stage (Middle Miocene). *Episodes* **32**, 152-166.
44. Betzler C, Eberli GP, Lüdmann T, Reolid J, Kroon D, Reijmer JGG, Swart PK, Wright J, Young JR, Alvarez-Zarikian C, Alonso-García M, Bialik OM, Blättler CL, Guo JA, Haffen S, Horozal S, Inoue M, Jovane L, Lanci L, Laya JC, Hui Mee AL, Nakakuni M, Nath BN, Niino K, Petruny LM, Pratiwi SD, Slagle AL, Sloss C R, Su X, Yao Z. 2018 Refinement of Miocene sea level and monsoon events from the sedimentary archive of the Maldives

- (Indian Ocean). *Progress in Earth and Planetary Science* **5**, 1-18. DOI 10.1186/s40645-018-0165-x.
45. Steeman ME, Hebsgaard MB, Fordyce RE, Ho SYW, Rabosky DL, Nielsen R, Rahbek C, Glenner H, Rensen MVS, Willerslev E. 2009 Radiation of extant cetaceans driven by restructuring of the oceans. *Systematic Biology* **58**, 573-585. DOI:10.1093/sysbio/syp060.
46. Akiba F. 1986 Middle Miocene to Quaternary diatom biostratigraphy in the Nankai Trough and Japan Trench, and modified Lower Miocene through Quaternary diatom zones for middle-to-high latitudes of the North Pacific. In Kagami, H., Karig, D. E., Coulbourn, W. T., et al., *Initial Reports of the Deep Sea Drilling Project, U. S. Government Printing Office, Washington D. C.* **87**, 393-480.
47. Yanagisawa Y, Akiba F. 1998 Refined Neogene diatom biostratigraphy for the northwest Pacific around Japan, with an introduction of code numbers for selected diatom biohorizons. *Jour. Geol. Soc. Japan* **104**, 395-414.

Tables

Measurements (in mm) of the skull of *Berardius kobayashii*

Skull

Preserved length from tip of broken rostrum to hindmost margin of occipital.	236
Greatest postorbital width.	*e219
Greatest zygomatic width	*e245
Greatest width of bony nares.	38
Maximum width of premaxillary sac fossae.	87
Maximum width of right premaxillary sac fossa.	43
Maximum width of left premaxillary sac fossa.	38
Maximum width of nasals.	40
Greatest length of left temporal fossa, measured to external margin of raised suture.	87
Greatest width of left temporal fossa at right angles to greatest length.	61
Major diameter of left temporal fossa proper.	38
Minor diameter of left temporal fossa proper.	33
Distance from foremost end of junction between nasals to hindmost point of margin of supraoccipital crest.	33
Length of left orbit from apex of preorbital process of frontal to apex of postorbital process.	51
Greatest width of internal nares.	91
Greatest length of left pterygoid.	109

Measurements (in mm) of the ear bones of *Berardius kobayashii*

Tympanic bulla

Standard length of tympanic bulla, distance from anterior tip to posterior end of outer posterior prominence.	37
Distance from anterior tip to posterior end of inner posterior prominence.	34
Distance postero-ventral tip of outer posterior prominence to tip of sigmoid process.	24
Distance postero-ventral tip of outer posterior prominence to tip of conical process.	17.5
Width of tympanic bulla at the level of sigmoid process.	20
Height of tympanic bulla, from tip of sigmoid process to ventral keel.	23
Width across inner and outer posterior prominence.	18
Greatest depth of interprominential notch.	5
Width of upper border of sigmoid process.	11

Periotic

Standard (maximum anteroposterior) length, from tip of anterior process to posterior end	39.7
of posterior process, measured on a straight line parallel to cerebral border
Standard (maximum mediolateral) width, from medial edge of cochlear portion to apex of	23
lateral tuberosity
Maximum dorsoventral depth of anterior process perpendicular to axis of periotic	13.6
Maximum mediolateral width of anterior process at base	17
Length of anterior process from anterior apex to posterior border of malleolar fossa	16.2
Length of anterior process from anterior apex to anterior incisure	14.3
Anteroposterior length of cochlear portion	17.3
Dorsoventral depth of cochlear portion	18.7
Mediolateral width of cochlear portion from medial edge to fenestra ovalis	10.8
Maximum diameter of fenestra rotunda	2.7
Maximum diameter of internal acoustic meatus	4.7
Maximum diameter of cochlear aqueduct	1.9
Maximum diameter of aperture for the vestibular aqueduct	3.6
Anteroposterior diameter of proximal opening of facial canal	3.4
Anteroposterior diameter of distal opening of facial canal	1.9
Length of posterior process (posterior bullar facet between anteroposterior tips)	17.3
Width of posterior process (posterior bullar facet between mediolateral tips)	12.1

Figure and table captions

Figure 1. Index maps of the Pacific Northwest: A, locations of the Japanese Islands, showing Sado Island; B, map of Sado Island, showing some major place names mentioned in text, and location of enlarged area including the type locality for *Berardius kobayashii*, sp. nov.; C, detail of Sobama beach in Donokama, Ogi Town, showing the exact locality for the holotype of *Berardius kobayashii*, SCM 5530-6. Based on the GIS map (<https://maps.gsi.go.jp/#15/37.844139/138.272681/&base=std&ls=std&disp=1&vs=c0j0h0k0l0u0t0z0r0s0m0f1>) of the Geospatial Information Authority of Japan.

Figure 2. Correlation of formations distributed on the Osado (a) and Kosado (b) mountain ranges containing whale fossil horizon with relevant systems of chronology. Wave lines indicate unconformities. After [6, 19, 10]. Diatom zones: [10, 46, 47]; Planktonic foraminiferal zones: [24, 35].

Figure 3. Dorsal view of the cranium of *Berardius kobayashii*, sp. nov., holotype, SCM 5530-6. (Scale bar = 10 cm)

Figure 4. Lateral view of the cranium of *Berardius kobayashii*, sp. nov., holotype, SCM 5530-6. A and B, left side; C and D, right side. (Scale bar = 10 cm)

Figure 5. Anterior view of the cranium of *Berardius kobayashii*, sp. nov., holotype, SCM 5530-6. (Scale bar = 10 cm)

Figure 6. Posterior view of the cranium of *Berardius kobayashii*, sp. nov., holotype, SCM 5530-6. (Scale bar = 10 cm)

Figure 7. Ventral view of the cranium of *Berardius kobayashii*, sp. nov., holotype, SCM 5530-6. (Scale bar = 10 cm)

Figure 8. Posterior fragment of the left mandible of *Berardius kobayashii*, sp. nov., holotype, SCM 5530-6. A, lateral view; B, medial view. (Scale bar = 10 cm)

Figure 9. Left periotic of *Berardius kobayashii*, sp. nov., holotype, SCM 5530-6. A, ventral view; B, corresponding line drawing with anatomical interpretations (gray hatches indicate parts of the tympanic bulla); C-F, the reconstructed CT image from 3D surface rendering, with anatomical interpretations (C, ventral view; D, lateral view; E, medial view; F, dorsal view). (Scale bar = 1 cm)

Figure 10. The left tympanic bulla A, dorsal view; B, anterior view; C, medial view; D, lateral view; E, posterior view; F, ventral view (Scale bar = 1 cm)

Figure 11. Key features of the left tympanic bulla of *Berardius kobayashii*, sp. nov., holotype, SCM 5530-6. A, dorsal view; B, anterior view; C, medial view; D, lateral view; E, posterior view; F, ventral view. (Scale bar = 1 cm)

Figure 12. Phylogenetic relationship and paleobiogeographic distribution of *Berardius kobayashii*, sp. nov., within the Ziphiidae based on Bianucci et al. [3] and our re-analysis. The strict consensus tree resulting from five most parsimonious trees, 177 steps long, with the consistency index = 0.475 and the retention index = 0.767.

Figure 13. Distribution and potential roots for the dispersal of the known extinct species of the Berardiinae in the middle Miocene inferred from the phylogenetic analysis.

Table 1. Measurements (in mm) of the skull of *Berardius kobayashii*, sp. nov., holotype, SCM 5530-6. '*' indicates estimated transverse measurements that are half-skull measurements multiplied by two, and 'e' indicates estimation.

Table 2. Measurements (in mm) of the ear bones of *Berardius kobayashii*, sp. nov., holotype, SCM 5530-6.

R. Soc. open sci. article template

ROYAL SOCIETY
OPEN SCIENCE

R. Soc. open sci.
doi:10.1098/not yet assigned

The oldest fossil record of the extant genus *Berardius* (Odontoceti, Ziphiidae) from the middle to late Miocene boundary of the western North Pacific

Ayako Kawatani¹ and Naoki Kohno^{1,2}

¹Graduate School of Life and Environmental Sciences, University of Tsukuba, 1-1-1, Tennoudai, Tsukuba, Ibaraki, 305-8577, Japan

²National Museum of Nature and Science, 4-1-1, Amakubo, Tsukuba, Ibaraki, 305-0005, Japan

Keywords: middle Miocene, extant genus, *Berardius kobayashii*, sp. nov., western North Pacific.

1. Summary

A new species of a beaked whale that belongs to the extant genus *Berardius* is described from the late-middle to late Miocene boundary age Tsurushi Formation (ca. 12.3-11.5 Ma) on the Sado Island, Niigata Prefecture, Japan. This new species, *Berardius kobayashii*, sp. nov., represents the oldest record of this genus and suggests the timing of the emergence of this extant genus during this age period also of the extant ziphiid genera in the world. *Berardius kobayashii*, sp. nov., has the following generic characters of *Berardius*: the ratio between the width of the premaxillary crests and the width of the premaxillary sac fossae is 1.0 to 1.25, nodular frontals make isolated protuberance on the posterior part of the gently elevated vertex of the skull, and anterior process of the periotic elongated and transversely slender such as the gently elevated vertex, the long and slender anterior process of the periotic. Among the species within the genus, *B. kobayashii*, sp. nov., shares a unique character with *B. minimus*, such as the apices of the left and right hamular processes of the pterygoids contact medially, forming together a posteriorly directed medial point. In addition, *B. erardius kobayashii*, sp. nov., displays also the a unique combination of both primitive and derived characters also has following unique specific characters: it is extremely small in size, and the nasals are short, ratio between the length of the medial suture of nasals on vertex and the maximum width of nasals is <0.4 it is extremely small in size, apices of the left and right hamular processes of the pterygoids contact medially, forming together a posteriorly directed medial point. *Berardius kobayashii*, sp. nov. fills the gap between the origin of the genus and later diversifications of the extant species in the genus. Furthermore, it suggests that *Berardius* species had originated in the North Pacific, including the Sea of Japan, at the time at latest 12 Ma. This discovery is will also be a key to elucidate the process of their emergence and habitat expansion during the latest middle and earliest late Miocene in age as the cosmopolitan genus among the Ziphiidae. Based on the distributional patterns of the fossil and extant species of the genus, suggest that the western North Pacific including the Sea of Japan may have been one of the areas for the evolution and radiation of this extant genus among the crown Ziphiidae at the time not later than 11 Ma.

2. Introduction

The family Ziphiidae is a of the cetartiodactyl clade among the Odontoceti (Cetartiodactyla, Cetacea) that is 
[revised manuscript text omitted]
 ~~that produced whale fossils~~ based on ~~the~~ radiolarians ~~fossils~~, which corresponds to yielding a 15.3-9.1 Ma ~~interval~~. Based on ~~these~~ these chronological information, the geologic age of the Tsurushi Formation is estimated to be approximately 12.3-11.5 Ma. Studies of foraminiferal fossils have suggested that the Tsurushi Formation was deposited in ~~the~~ deep-sea environment [7, 9].

As mentioned above, various megafossils have been reported from the Tsurushi Formation,; for example, mollusks, fish, a sea turtle, birds, and marine mammals [9, 11, 26]. In particular, a lot of marine mammal fossils ~~such as~~ including whales, pinnipeds and desmostylians have been found in calcareous nodules from the Tsurushi Formation on the Kosado mountain range, and these fossils were briefly reported and summarized by ~~the~~ Sado Research Group for Marine Mammalian Fossils [11], Tazaki et al. [13], Horikawa et al. [12], Takahashi et al. [14] and Barnes and Hirota [15].

4. Materials and Methods

The specimen described here was initially reported by Takahashi et al. [14]. At that time the specimen was temporarily deposited at Niigata University, ~~and~~ now it is permanently stored at the Sado Museum, Sado City, Niigata Prefecture, Japan, where it bears the catalog number SCM 5530-6. This specimen ~~was had been~~ enclosed in calcareous hard ~~mudstone matrix~~ ~~mud stone matrices~~. Accordingly, it was prepared by acid etching, ~~technie~~ using formic acid diluted ~~to a~~ with concentration of about 4.5-5.0% ~~in water~~, which allowed the extraction of fragile bone elements of the skull, partial mandible and ear bones from ~~the matrix~~ ~~hard matrices~~. However, the periotic ~~is has~~ been fused to the skull ~~loosely~~ by its fan-shaped posterior process, and the dorsal surface of the periotic is ~~entirely~~ invisible ~~entirely~~ from the ventral aspect. ~~So~~ we reconstructed the 3D image of the periotic ~~using~~ by the micro-Computed Tomographic (μ CT) scanner ~~of the~~ Microfocus CT, TXS320-ACTIS, at the National Museum of Nature and Science, Tokyo, Japan.

We then compared ~~the specimen~~ them osteologically and taxonomically ~~with~~ both ~~extinct~~ fossil and extant species of ziphiids stored at the ~~following~~ institutions ~~listed below~~ and ~~from~~ the literature ~~cited~~. Definitions of measurement ~~portion~~s and morphological terms follow Mead and Fordyce [27]

R. Soc. open sci.

and Kasuya [28]. ~~Then we performed a cladistic analysis to clarify the phylogenetic relationship of the new fossil described here to the previously named ziphiids using the characters and data matrix prepared by Bianucci et al. [3], and it was slightly modified with the addition of recently described *Berardius minimus* ([2]) and the new species described here. We also added *Berardius bairdii* based on our own observation into the data matrix, because it was also not included in the original analysis by Bianucci et al. [3].~~ Character and character states were managed using Mesquite 3.61 ([29]). The phylogenetic analysis based on 31 ziphiids of both extinct and extant taxa with six outgroup taxa as OTUs and 51 morphological characters was performed with TNT 1.5 using the “New Technology Search” tasked to find minimum length trees 1,000 times ([30]). ~~As was mentioned by Bianucci et al. [3], our preliminary analyses also led to less-resolved trees. So we also downweighted as the same with Bianucci et al. [3] as the homoplastic characters using the default value of 3 for the constant *K* of the method formulated by Goloboff [31]. Multi-state characters were treated as ordered for 18 characters (i.e., chars. 1, 3, 7, 8, 9, 11, 13, 14, 15, 16, 22, 23, 30, 34, 36, 39, 47, and 51) and unordered for six characters (i.e., chars. 2, 4, 10, 28, 34, and 45), and other binary characters were all treated as unordered, also following Bianucci et al. [3].~~ The characters for comparisons and the phylogenetic analysis follow Bianucci et al. [3].

Institutional abbreviations are the following: IRSNB, Institut royal des Sciences naturelles de Belgique, Brussels, Belgium; MNHN: Muséum National d’Histoire Naturelle, Paris, France; NMNS-PV, Department of Geology and Paleontology, National Museum of Nature and Science, Tsukuba, Japan; NSMT-M, Department of Zoology, National Museum of Nature and Science, Tsukuba, Japan; SAM: Iziko South African Museum, Cape Town, South Africa; SCM, Sado Museum, Sado City, Niigata, Japan; TMNH, Togakushi Museum of Natural History, Nagano, Japan.

5. Systematics

~~Order~~ CETARTIODACTYLA Montgelard, Catzefis and Douzery, 1997

~~Infraorder~~ CETACEA Brisson, 1762

~~Parvorder~~ ODONTOCETI Flower, 1867

~~Family~~ ZIPHIIDAE Gray, 1850

~~Subfamily~~ BERARDIINAE Moore, 1968

Emended diagnosis of subfamily: ~~The subfamily differs from all other ziphiids with the following unique combination of characters: the ascending process of the premaxilla in lateral view is slightly concave (Char. 7, reversal); constriction on the ascending process of the right premaxilla (between premaxillary sac fossa and the premaxillary crest) is weak, and the ratio between the minimal width of ascending process of premaxilla and the width of right premaxillary crest is >0.80 (Char. 8, reversal); no anteromedial excavation of the dorsal surface of the nasal (Char. 14, reversal); the nodular protuberance on the posterior part of the vertex is formed by the interparietal and/or the frontals (Char. 17, 0->1 & 2); and the anteromedial margin of the supraoccipital is distinctly lower than the dorsal margin of the vertex (Char. 18) mandible;; the narrow and thin premaxillary crest; the supraoccipital is lower than the frontals on the vertex (modified from [29]).~~

Genus *Berardius* Duvernoy, 1851

Emended diagnosis of genus: ~~The ratio between the rostral length and the condylobasal length is between 0.63 and 0.70; the mesorostral groove is empty;; The genus differs from *Microberardius* with the following unique combination of characters: the ratio between the width of the premaxillary crests and the width of the premaxillary sac fossae is 1.0 to 1.25 (Char. 11); the nodular protuberance on the posterior part of the vertex is formed only by the frontals (Char. 17’). Following characters shared with *Berardius* species, but uncertain in *Microberardius* and *Archaeoziphius* because of lack of such portion, are potentially diagnostic for the genus: the anterior spine of the tympanic with a more or less rectilinear anterior margin (Char. 22); the posteroventral corner of the sigmoid process of the tympanic is posteriorly projected in lateral view (Char. 23); the dorsal margin of the involucrum of the tympanic that is cut by an indentation that is present and visible in medial and/or dorsal view (Char. 24) the mesorostral groove opens for the proximal third of the rostrum; the~~

Commented [OP1]: the proximal part of the mesorostral groove is actually filled by bone in *Berardius*. this bone has been identified as the mesethmoid, but could possibly be re-identified as the presphenoid. see the following works:

Lambert, O., Buffrénil, V., de, and Muizon, C., de. 2011. Rostral densification in beaked whales: diverse processes for a similar pattern. *Comptes Rendus Palevol* 10: 453-468.

Ichishima, H. 2016. The ethmoid and presphenoid of cetaceans. *Journal of Morphology* 277: 1661-1674.

this is a condition that differs from most, but not all other ziphiids.

R. Soc. open sci. article template

5

preaural basin is absent; asymmetry of the premaxillary sac fossae absent or weak; the premaxillary sac fossa does not laterally overhang on the maxilla; deep excavation of the premaxillary sac fossa is absent; the ascending process of the premaxilla in lateral view is slightly concave; constriction on the ascending process of the right premaxilla is roughly absent; the vertex is gently elevated; the premaxillary crest is transversely directed; the distance between the left and right premaxillary crests is large; the nasal elongation ratio between the length of the medial suture of the nasals and the maximum width of the nasals is high; the anteromedial excavation of the dorsal surface of the nasal is absent; the premaxillary crest present on the vertex; inclusion of the nasal in the premaxillary crest is absent; contact between the nasal and the premaxillary crest is extended on more than half the length of the nasal to almost the whole length; the premaxillary foramen is located roughly at the level of the antorbital notch; there are no excrescences on the dorsal surface of the maxilla at the posterior half of the rostrum; the posterolateral narrowing of the nasals and frontals is present at the vertex that is narrower than the nasals; the anteromedial margin of the supraoccipital is distinctly lower than the dorsal margin of the vertex; the extremely ossified trapezoidal vertex is absent; the angle formed by the basioccipital crests is larger than 50° in ventral view; the hamular fossa of the pterygoid sinus is wide, extending anteriorly on the palatal surface of the rostrum; the apices of the left and right hamular processes of the pterygoids contact medially; the apex of the hamular process is excavated by the fossa for the hamular lobe of the pterygoid sinus; the anteroventral wall of the tympanic bulla is narrow and triangular; the periotic has long and slender anterior and posterior processes; the posterior bullar facet of the periotic is fan-shaped (modified from [3], [29], [30], [31]).

Berardius kobayashii, sp. nov.

(Skull: Figure 3-87; Ear bone: Figure 98-1140)

Berardius sp. indet. Takahashi et al. 1989

URN: lsid:zoobank.org:act:317C1452-AE2E-4A08-A473-3B507B452D2C

Diagnosis of species: The species differs from all other *Berardius* species (i.e., *B. arnuxii*, *B. bairdii* and *B. minimus*) with the following unique characters: the nasals are short, ratio between the length of medial suture of nasals on the vertex and the maximum width of nasals <0.4 (Char. 13, reversal); Although it is potentially a shared primitive trait with *Archaeoziphius microglenoides*, an extremely small in size among the four species of *Berardius* (60–70 % smaller than *B. bairdii* and *B. arnuxii*, 50% smaller than *B. minimus*) is potentially the specific trait within the genus; the transverse diameter of the premaxilla is relatively wide; the lateral margin of the premaxilla is curved; there is a tiny anteromedial excavation on the dorsal surface of the nasals; the apices of the left and right hamular processes of the pterygoids contact medially, forming together a posteriorly directed medial point.

Holotype: SCM 5530-6, incomplete skull with the left tympanic bulla and periotic, lacking the rostral portion; the posterior one-fourth of the horizontal ramus of the left dentary with the condyloid coronoid process and mandibular condyle. Collected by Kiyoei Kaneko in or before 1969.

Type Locality: Sobama Beach, Ogi Town, Sado Island, Niigata Prefecture, Japan. Geographic coordinates: 37° 84'50" N, 138° 26'58" E.

Formation and age of holotype: SCM 5530-6 was collected from the latest middle to earliest late Miocene age Tsurushi Formation. As discussed above, the geologic age of the Tsurushi Formation is estimated to be approximately 12.3–11.5 Ma.

Etymology: The species is named in honor of the late Dr. Iwao Kobayashi, a professor emeritus at Niigata University, for his longstanding contributions to the geology and paleontology of the northern Fossa Magna region including Niigata Prefecture, and in gratitude for his permission to re-study the marine mammal fossils collected by him and his colleagues from the Sado Island and encouragement to both of us throughout this study.

Description

R. Soc. open sci.

Cranium (general morphology)

The cranium is relatively small (Table 1) and ~~slightly less~~ asymmetry (Figure 3,5). The ratio between the height and width of the skull is 1:1.19. The rostral and the occipital parts are almost ~~completely~~ broken and missing ~~away~~. The small part of the left mandible ~~was originally had~~ attached to the cranium. The vertex is moderately elevated and slightly skewed toward the left. The ~~posterior part of the~~ vertex is ~~almost broken~~. The ~~outline of the~~ temporal fossa ~~in lateral view~~ is an isosceles triangle ~~with long sides in shape~~ that converges anteriorly, and the posterior margin is weakly rounded. The temporal fossa is shallow and ~~its lateral surface is convex~~ ~~on~~ ~~neave~~. The sutures of the skull are still visible but tightly closed, indicating that the individual is an adult.

Premaxilla

The premaxilla is ~~well preserved at the level posterior to the antorbital notch, and its dorsal surface~~ is almost flat. The premaxillary crest is present on the vertex. The medial margins ~~of between~~ the left and right premaxillae ~~are is~~ very close to each other. Contact between the nasal and the premaxillary crest is extended along ~~almost~~ the ~~almost~~ whole length of the nasal. The premaxillary sac fossae are ~~weakly asymmetrical~~ and do not overhang the maxillae laterally. The premaxillary sac fossa is almost flat and not deeply excavated. The ascending process of the premaxilla gradually rises toward the vertex. ~~The angle between the dorsal surface of the rostrum and the ascending process of the premaxilla is 150°. It~~ is slightly concave in lateral view. The ascending process of the left premaxilla is somewhat damaged, but that of the right premaxilla is well preserved, and ~~its lateral margin is~~ moderately constricted ~~at the level along the bony nares just~~ at the midpoint of the premaxillary crest. The vertex is gently elevated. In anterior view, lateral expansion of the premaxillae at the ~~portion~~ level of the premaxillary sac fossae is prominent. The premaxillary crest is transversely oriented, and it is narrow in ~~transverse~~ width. The distance between the premaxillary crests is wide. The anterior margin of the bony nares is V-shaped ~~and t-he overall shape of the bony nares is triangular in outline.~~ ~~The anteroposterior diameter (67 mm) of the bony nares is greater than the transverse diameter (39 mm).~~ The small premaxillary foramina ~~on the premaxilla are~~ located roughly at the level of the antorbital notch and slightly posterior to the infraorbital foramina on the premaxillae. There are distinct grooves anterior to the foramina. ~~The angle between the dorsal surface of the rostrum and the ascending process of the premaxilla is 150°.~~

Maxilla

The maxilla is broad, ~~but~~ and its lateral margin is ~~damaged~~ ~~curved on both sides at the level posterior to the antorbital notch.~~ In anterior view, ~~t~~he rostral maxillary crest is present and moderately elevated on the maxilla. The ~~dorsal~~ infraorbital foramina open anterodorsally on the maxillae near the base of the rostrum, and the largest foramen is located on the right maxilla. There are two ~~maxillary dorsal infraorbital~~ foramina on the left maxilla ~~with diameters of and are~~ 13 mm and 7 mm ~~in diameter, respectively.~~ There is one large ~~maxillary dorsal infraorbital~~ foramen on the right maxilla, ~~with and has~~ a diameter ~~of reaching~~ 16 mm. The maxilla is slightly elevated at the posterior side of these foramina. ~~There are no~~ ~~The~~ excrescences ~~on the dorsal surface of the maxilla is absent~~ on the ~~preserved proximal part~~ half of the rostrum. The ~~portion of the~~ prominent notch and related structure ~~are somewhat weathered away, but it is low and flat~~ low.

Nasals

The nasal is short, and its dorsal outline is trapezoidal. ~~The shape of the bony nares is triangular in outline, and the anteroposterior diameter (67 mm) of the bony nares is greater~~ ~~wider than the transverse diameter (39 mm).~~ Although the posterior margin of the right nasal is broken, it is slightly broader than the left one. The dorsal surface of the nasal is roughly flat, and there is a weak anteromedial excavation. The nasal is not included in the premaxillary crest. ~~The nasal contacts the premaxilla laterally for almost the whole length.~~ The posterior nasal-frontal suture is ~~only preserved on the left nasal and is~~ posteriorly convex.

Frontal

R. Soc. open sci.

R. Soc. open sci. article template

7

The frontal is fully covered by the maxilla in the supraorbital region dorsal view and only visible as its vertical section beneath the overlapping maxilla in at the portion of the orbital region and temporal fossae in lateral view. It is dorsoventrally thick and relatively dense but not osteosclerotic nor pachyosteosclerotic pachyosteosclerotic. It also makes a dorsal roof of the antorbital and fossa in ventral view. The pre- and supraorbital area is indicated by the slight flange at the level posterior to the antorbital notch. The antorbital fossa is shallow and flares posteriorly to the slightly thickening and anterolaterally extended postorbital ridge. The posterodorsal part of the frontal around the posterior part of the vertex is broken away.

Supraoccipital

The supraoccipital is almost completely broken away, but the anteromedial margin of the supraoccipital is distinctly lower than the dorsal margin of the vertex in posterior view (Fig. 6).

Pterygoid

The pterygoid is dorsoventrally deep and excavated by the anteroposteriorly and dorsoventrally wide hamular fossa. The lateral lamina is preserved on the right pterygoid. It is thin and low at its base. There is a similar structure like transverse crests on the surface of the medial lamina. There are distinct transverse crests on the medial lamina of the pterygoid. The hamular fossa of the pterygoid sinus is wide, extending anteriorly on the palatal surface of the rostrum. Apices of the left and right hamular processes are incomplete, but both processes contact medially and form together a posteriorly directed medial point. Also each process sends a posterolateral point, so the posterior margin of the pterygoid processes are rounded W in shape (Fig. 6).

Alisphenoid

The alisphenoid is well preserved on the right side, just posterior to the pterygoid sinus and ventrally exposed as a distinct concavity that is directed anterolaterally. A thin plate-like crest runs anterolateral-posteromedial direction, which is thought to be the alisphenoid-squamosal suture, and it separates this concavity posteriorly from the tympanosquamosal recess. This concavity on the alisphenoid continues posteromedially to the large foramen ovale. The supraoccipital protuberance protrudes posteriorly.

Squamosal

The left squamosal is well preserved. The temporal fossa on the squamosal is anteroposteriorly long and dorsoventrally shallow, and its surface is slightly concave. The zygomatic process of the left squamosal is preserved. It is short anteroposteriorly compared to its dorsoventral height. The zygomatic process of the squamosal is well developed and anterodorsally tapered with a triangle outline, but its anterior margin is crescentic in shape. The glenoid cavity is wide and shallow, but its medial part is deep just in front of the obliquely extended postglenoid process. It continues to the tympanosquamosal recess. The tympanosquamosal recess extends posteriorly to the fused periotic. The surface of the posterior part of the tympanosquamosal recess just in front of the periotic is rugose with numerous pits and crests. The falciform process presents at the anterior process of the periotic, and it is short and thin. The ventral margin of the postglenoid process is more dorsally located than the ventral margin of the paroccipital process of the exoccipital (Fig. 4).

Mandible

The mandibles are almost broken away, but the ascending ramus of the left mandible with the coronoid process and the mandibular condyle is partially preserved (Figure 7). The posterior margin of the angular process is also broken away. The mandibular body is very thin and fragile in at the portion of the lateral wall of the mandibular foramen. Its outer or lateral surface is smooth and slightly swelled laterally, forming the condyloid crest. Its inner or medial surface (or the lateral wall of the mandibular foramen) is relatively rough. The mandibular foramen is anteroposteriorly long and dorsoventrally wide. The coronoid crest is well developed as an anteroposteriorly extended ridge and continues to the coronoid process at the posterodorsal end of the mandible. The ventral margin of the

R. Soc. open sci.

coronoid crest is weakly concave along the crest. The mandibular condyle protrudes posteriorly and is located ~~locates~~ almost at the same anteroposterior level ~~asposition at~~ the end of the coronoid process. It is dorsoventrally long and laterally crescentic, and it is slightly swelled posteriorly. There is a notch between the coronoid process and the mandibular condyle ~~but is absent between the mandibular condyle and the angular process, which is almost broken though.~~

Periotic

The left periotic is preserved (Table 2). It is relatively slender in shape. The dorsal surface of the body is smooth. The anterior and posterior processes are almost straight anteroposteriorly in ventral view. The anterior process is laterally thick, but its apex is slender and pointed (Figure 9A). It is weakly bent ventrally. The anterior bullar facet is weakly excavated anteroposteriorly. A rounded accessory ossicle of the tympanic bulla is still articulated with the fovea epitubaria at the posteromedial base of the anterior process. The malleolar fossa is excavated posterior to the accessory ossicle (~~in the fovea epitubaria~~) and opens posteriorly. It is laterally delimited by a mediolaterally elongated lateral tuberosity. The cochlear portion (=pars cochlearis) is rounded in outline, relatively low, and it is slightly bent anteroventrally with a weak depression at the anteromedial surface, so the anterior incisure is narrow and slightly covered with the ~~anteriorposterior~~ edge of the cochlear portion. The epitympanic hiatus is relatively deep and wide. The internal acoustic meatus containing the ~~central foramen~~ (=spiral cribriform tract and the foramen singulare is separated by a developed transverse crest from the proximal opening of the facial canal. The ~~spiral cribriform tract~~ ~~central foramen~~ is relatively small and almost circular, slightly compressed transversely. Its posterolateral margin is elevated and reaches almost the top of the cochlear portion. The ~~foramen foreman~~ singulare opens ~~in the same~~ near the aperture for the ~~facial canal~~ vestibular aqueduct (Figure 9B). The proximal opening of the facial canal is relatively large and elliptical in shape. The aperture for the cochlear aqueduct is large and semicircular. It is located close to the posterior margin of the internal auditory meatus. It is also relatively close to the rounded aperture for the vestibular aqueduct (=endolymphatic duct) that is elliptical in shape. The fenestra rotunda is relatively large and semicircular in outline. The posterior process is relatively long posteriorly and inflated mediolaterally. It is fused to the skull ~~in~~ by the mediolaterally narrow periotic fossa of the skull. The posterior bullar ~~surface~~ is covered by the fused fragment of the posterior process of the tympanic bulla. It is slightly bent posterolaterally at the base just posterior to the cochlear portion. Because of this fusion ~~of~~ the posterior process of the tympanic bulla and its posterior expansion, the ~~portion of~~ the posterior bullar facet is posteriorly flared and fan-shaped in ventral view (Figure 9A). Although mostly covered by the remnant of the posterior process of the tympanic bulla, the surface of the posterior bullar facet is smooth. ~~The epitympanic hiatus is relatively deep.~~

Tympanic Bulla

The left tympanic bulla is preserved (Table 2). The anterior part of the ventral wall of the tympanic bulla is cylindrical and is ~~transversely rectilinear in outline~~ ~~lacking its anterior spine~~. The involucrem is shorter ~~anteriorly~~ than the lateral wall. The dorsal margin of the involucrem is cut by an indentation. The inner and outer posterior prominences are thick and short, and they ~~swell are~~ ventrally bulging ~~well~~. The outer posterior prominence is ~~much bulgedswelled~~ ventrally than the inner ~~one~~. The median furrow runs anteroposteriorly on the ventral ~~wall~~, and it is transversely narrow and anteriorly shallow. The interprominential notch is relatively deep at the posterior end of the median furrow. The lateral furrow is shallow, and its posterior ridge is prominent and ~~twisted~~ on the lateral wall of the tympanic bulla. The sigmoid process is square ~~in anterior view~~, and it is ~~anteroposteriorly~~ ~~rather~~ thin. ~~The posteroventral corner of the sigmoid process is projected posteriorly.~~ The lateral border of the sigmoid process is twisted posteriorly. A damaged base of the posterior process lies on the posterolateral end of the sigmoidal process. The elliptical foramen is present (Figure 10, 11).

6. Results of the Phylogenetic Analysis

We compared the characteristics of SCM 5530-6 with both extant and fossil Zhiphiidae including 27 genera and 55 species based mainly on Bianucci et al. [3] and Lambert et al. [34]. ~~Because SCM~~

R. Soc. open sci. article template

9

5530-6 described here from the middle to upper Miocene boundary Tsurushi Formation was initially reported by Takahashi et al. [14] as a species of the Ziphiidae based on the posteriorly located bony nares with gently elevated ascending process of the premaxilla, and the posteroventrally steeply sloped occipital. In addition to the above characters, SCM 5530-6 has such diagnostic characters of the family Ziphiidae at least as the presence of the premaxillary crests on the gently elevated vertex, wide and anteriorly extended hamular fossa of the pterygoid sinus [3, 34]. We analyzed its phylogenetic relationship based on the data matrix and parsimony settings prepared by Bianucci et al. [3] to locate SCM 5530-6 in the Ziphiidae with the addition of the North Pacific Baird's beaked whale, *Berardius bairdii* that was not included in their analysis and the recently described Sato's beaked whale, *Berardius minimus* of the northern North Pacific [2], as well as SCM 5530-6.

As a result of the analysis, five most parsimonious trees were obtained with a tree length of 177 steps in each, Goloboff fit ($k=3$) -58.44286, the consistency index (CI) of 0.475, and the retention index (RI) of 0.767. The 50% majority rule consensus tree is shown in Figure 12. The strict consensus tree is also shown in the Supplementary Data S4. Our result suggests that *Chavziphius* from the late Miocene of Peru (the South Pacific) is the basal to the clade including all the extant subfamilies, i.e., the Berardiinae, Ziphiinae, and Hyperoodontinae. Instead, *Ninoziphius* also from the Miocene of Peru is recognized as the most basal to the Ziphiidae as a whole. As for SCM 5530-6, it was nested in the monophyletic subfamily Berardiinae consisting of the genera *Berardius*, *Archaeoziphius* and *Microberardius* ([3], [36] and Figure 12).

The Berardiinae is supported by five synapomorphies: slightly concave ascending process of the premaxilla (Char.7: as reversal); a weak constriction on the ascending process of the right premaxilla (between premaxillary sac fossa and premaxillary crest) and the ratio between the minimal width of the ascending process of premaxilla and the width of the right premaxillary crest is > 0.80 (Char.8: as reversal); no anteromedial excavation of the dorsal surface of the nasal (Char.14: as reversal); an isolated rounded protuberance formed by the interparietal and/or frontals on the posterior part of the vertex (Char.17), and the anteromedial margin of the supraoccipital that of SCM 5530-6 is lower than the frontals on the vertex (Chars. 18 of [3]) (see also [3, 29, 33]). The following character is also considered to be a potentially unique character for the subfamily: the posteromedial narrowing nasals and frontals at the vertex, which is narrower than the anterior-most transverse width of nasals (Char. 50). Among those characters, quite a few synapomorphies of the subfamily are interpreted as reversals from derived to secondarily primitive character states. However, the extant ziphiids are deep-diving whales having adapted to deep sea with considerable ecomorphological specializations during their adaptation and diversification in each clade, and they might have yielded unique characters frequently to adapt such a harsh environment. Other examples are also interesting to be noted. A low elevation of the ascending process of the premaxilla is observed in *Berardius* and related fossil species such as *Archaeoziphius microglenoideus* and *Microberardius africanus* [3, 32, 36], though this condition is also seen in *Chavziphius maxillocrestatus*, one of the basal ziphiids [3]. Another example is a cluster of dorsal infraorbital foramina, which is present on the base of the rostrum in the early diverging ziphiids [3]. On the other hand, only a large foramen is present in the crown ziphiids [3]. SCM 5530-6 has a large foramen on the right maxilla, and two relatively large foramina on the left maxilla, and therefore, the condition of these foramina is thought to be the condition seen in the later diverging, extant ziphiids. However, *Ninoziphius platyrostris* from the Miocene of Peru, which is considered to be the most basal ziphiid (Figure 12), has a relatively large dorsal infraorbital foramen [34], suggesting that this character state might have been multiply derived in two distantly related clades. This case is also suggestive that some characters seen in the berardiines are not appropriate to be considered synplesiomorphies conservatively retained in them as a subfamily having earlier-branching, deeper node.

Among the species within the Berardiinae, *Microberardius africanus* was recognized as the first split with the rest of genera and species including SCM 5530-6, but no autapomorphies were recognized to characterize *Microberardius* by our analysis. In this regard, *Microberardius* seems to be "ancestral" to the later diverging berardiines. However, Bianucci et al. [3] suggests that *Microberardius* is distinguishable from other berardiines by having the following characters: the

R. Soc. open sci.

rostrum is higher than wide along most of its length, the rostral base is narrower, and the maxillary crest is not extended on the rostral base.

By contrast, the monophyly of the species belonging in the genera *Berardius* and *Archaeoziphius* including SCM 5530-6 is strongly supported by the following unique character combinations: Although the left side of the premaxilla of SCM 5530-6 is broken, the ratio between the width of the premaxillary crest and that of the premaxillary sac fossa of SCM 5530-6 is moderate, from 1.0 to 1.25 (Char. 11); an isolated rounded protuberance formed by the frontals, not the interparietal, on the posterior part of the vertex (Char. 17, 0->2).

Although, the basic framework of the relationships among the genera of the Berardiinae (i.e., [*Microberardius*, [*Archaeoziphius*, *Berardius*]]), which was originally demonstrated by Bianucci et al. [3], is also supported by our analysis, *Archaeoziphius microglenoideus* was bracketed in between the clade containing two other extant species of *Berardius* (i.e., *B. bairdii* and *B. arnuxii*) and the clade containing newly added *Berardius minimus* plus SCM 5530-6 in our analysis. However, 274 out of 51 (i.e., 5347%) characters used in Bianucci et al. [3] and in our analysis are missing and uncertain in *A. microglenoideus*, and therefore, its phylogenetic position is still unstable (see also the Supplementary Data S4). Because of overall primitiveness of observable characters in *Archaeoziphius*, the position of this genus is still controversial and is potentially anticipated to be "outside" from the clade of the genus *Berardius* as a sister taxon to the latter. In this regard, the monophyly of the genus *Berardius* including SCM 5530-6 is highly considerable, and the position of SCM 5530-6 as a sister taxon to *B. minimus* makes SCM 5530-6 a species of *Berardius* in the Berardiinae. As for the specific relationship of SCM 5530-6 to other species in the genus, SCM 5530-6 is the closest to *B. minimus* as was mentioned above in having the apices of the left and right hamular process of the pterygoids that contact medially, forming together a posteriorly directed medial point (Char. 36). In addition, the lateral expansion of the premaxillae at the posterior halves in SCM 5530-6 is prominent as in *B. minimus*. Furthermore, the temporal fossa of SCM 5530-6 is shallow as in *B. minimus*, and but its lateral surface wall of the fossa is convex also as in *B. minimus*. By contrast, it is definitely concave as in *B. bairdii* and *B. arnuxii*. As for skull sutures, *B. minimus* has much tighter sutures as has pointed out by Yamada et al. [2], the skull of SCM 5530-6 also has much tighter sutures compared with those in *Berardius bairdii* and *B. arnuxii*. The prominent notch and related structure are much lower, less distinct and less rugged than *B. minimus*. In fact, three out of five most parsimonious trees support that SCM 5530-6 is the sister taxon of *B. minimus* (Supplementary Data S4). Finally, SCM 5530-6 is distinguishable from *B. minimus* at the three e in having extant species based on the above-mentioned cranial characters: the short nasal is short, the ratio between the length of medial suture of nasals on the vertex and the maximum width of nasals < 0.4 (Char. 13, as reversal).

Accordingly, of the above morphological comparisons, SCM 5530-6 is identifiable as a species of the genus *Berardius*. Furthermore, the skull size, based on the bizygomatic width, of SCM 5530-6 is 50% smaller than *B. minimus* as was mentioned above. Consequently, identification of SCM 5530-6 as a new distinctive species within the genus *Berardius* is warranted. Thus, we propose *Berardius kobayashii*, sp. nov. based on SCM 5530-6 from the middle to late Miocene boundary age Tsurushi Formation. Although the left side of the premaxilla of SCM 5530-6 is broken, the ratio between the width of the premaxillary crest and that of the premaxillary sac fossa of SCM 5530-6 is moderate, from 1.0 to 1.25 (Char. 11 of [3]). The dorsal infraorbital foramina open anterodorsally on the maxilla near the base of the rostrum. In general, a cluster of dorsal infraorbital foramina is present on the base of the rostrum in the early diverging ziphiids [3]. On the other hand, only a large foramen is present in the crown ziphiids [3]. SCM 5530-6 has a large foramen on the right maxilla, and two relatively large foramina on the left maxilla, and therefore, the condition of these foramina is thought to be the condition seen in the later diverging, extant ziphiids.

The supraoccipital of SCM 5530-6 is lower than the frontals on the vertex, and the vertex is gently elevated (Chars. 9 and 18 of [3]). A similar low elevation of the ascending process is observed in *Berardius* and related fossil species such as *Archaeoziphius microglenoideus* and *Microberardius africanus* [3, 29, 33]. *Archaeoziphius* differs from *Berardius* by the following characters: the overall size of the skull is much smaller; the nasal is included in the premaxillary crest for a short distance along the posteromedial angle of the crest [33]; Chars. 15 and 51 of [3]. However, the inclusion of the nasal in the premaxillary crest is absent on SCM 5530-6 as in the species of *Berardius*.

Microberardius differs from *Berardius* by the following characters: the size of the skull is much smaller; the premaxillary crest is much narrower; the vomer at the anterior portion of the mesorostral groove is thickened; the rostrum is higher than wider along the most of its length; the base of the rostrum is narrower and the maxillary crest is not extended at the base of the rostrum [29]. By contrast, SCM 5530-6 does not have the narrow premaxillary crest.

Consequently, SCM 5530-6 differs from both *Archaeoziphius* and *Microberardius*. In addition, the anterior process of the periotic is slender, and its apex is long and pointed, and the posterior process is long and slender. These characters are potential synapomorphies of the genus *Berardius* (see also [28, 34]). Therefore, SCM 5530-6 is definitely identified as a species of the genus *Berardius* as was already pointed out by Takahashi et al. [14].

However, SCM 5530-6 has some additional characters differing from the extant, now three species of *Berardius*, such as *Berardius bairdii*, *B. arnuxii* and *B. minimus*, which were not mentioned by Takahashi et al. [14]. For instance, the nasal of SCM 5530-6 is shorter than in *B. bairdii*, *B. arnuxii* and *B. minimus*. Also, there is a tiny anteromedial excavation on the dorsal surface of the nasal, which is not observed in the other three species of *Berardius*. The apices of the pterygoids contact medially in all the species of *Berardius*, and these are forming together a posteriorly directed medial point in SCM 5530-6, but these are less excavated with a U-shaped posterior margin in the other three species of *Berardius*. The last character is not seen in any species of both fossil nor extant ziphiids including the three extant species of *Berardius*. But, the hamular fossa of the pterygoid sinus is wide, extending anteriorly on the palatal surface of the rostrum as seen in the extant *Berardius* and deep diving whales [35, 36]. The height of the skull relative to its width is 84% in SCM 5530-6 and slightly higher than in the extant species of *Berardius*; i.e., 83% in *B. minimus* and 77% in *B. bairdii*. The skull of SCM 5530-6 has much tighter sutures compared with those of extant *Berardius bairdii* and *B. arnuxii*. As has pointed out by Yamada et al. [2], the skull of *B. minimus* also has much tighter sutures, but SCM 5530-6 has much tighter sutures than that of *B. minimus*. The greatest postorbital width of SCM 5530-6 is almost 60 to 70% smaller than in the extant *B. bairdii* and *B. arnuxii* and 50% smaller than in *B. minimus*. The tympanic bulla and the periotic of SCM 5530-6 closely resembles those of *B. bairdii*. But, the total length of the tympano-periotic bone is 40 to 50% smaller than that of *B. bairdii*, and the total length of tympanic bulla is 30% smaller than that of *B. minimus*. In addition, based on comparisons with all the known species within the genus; i.e., the extant three species that are presently distributed in the northern North Pacific (i.e., *B. bairdii* and *B. minimus*) and the Antarctic (i.e., *B. arnuxii*), SCM 5530-6 is distinguishable from all the three extant species based on the above mentioned cranial characters: i.e., it is extremely small in size; the nasal is short; and the apices of the left and right hamular process of the pterygoids contact medially, forming together a posteriorly directed medial point. Furthermore, the size of the new fossil is extreme among all the known species of the genus; i.e., 50% smaller than *B. minimus* and surprisingly 60 to 70% smaller than both *B. bairdii* and *B. arnuxii*. Consequently, identification of SCM 5530-6 as a new species within the genus *Berardius*, i.e., *Berardius kobayashii*, is warranted.

7. Discussion

In previous studies, Takahashi et al. [14] identified SCM 5530-6, now the holotype of *B. kobayashii* sp. nov., as “provisionally *Berardius* sp. indet.” among them based on the following characteristics: the external bony nares are triangular in outline; the vertex is gently elevated; and the posterior median part of the pterygoid (though it was misidentified as median part of the palate in [14, p.104]) the palatine is ventrally well developed on convex on the ventral side. However, they compared identified SCM 5530-6 only as belonging to the Ziphiidae based on the following characters: the bony nares are located posteriorly; the rostrum is moderately wide; the occipital slopes relatively steeply posterodorsally; and the cranium is not strongly asymmetrical. Furthermore, they compared SCM 5530-6 with some extant genera of the Ziphiidae; i.e., *Ziphius*, *Berardius*, and *Mesoplodon*, and But they also pointed out that the transverse width/diameter of the premaxilla of SCM 5530-6 is relatively wider relative to the width of the maxilla at the level of the antorbital notch

than that of the extant two species of *Berardius*, and that the lateral margin of the same portion is curved laterally in contrast to the relatively straight condition of this portion in both the extant species of *Berardius*. In addition, they also pointed out that the morphology of the zygomatic process of the squamosal is shaped like an anterodorsally tapered triangle in contrast to an anteriorly sub-squared rectangle in both the extant species of *Berardius*. In this regard, both inter- and intra-generic comparisons were quite limited to their comparable taxa among the extant ziphiids, and the generality of their taxonomical differentiations of characters were somewhat arbitrary. In addition, two new genera of the Berardiinae were described as mentioned above ([32], [36]), and the third extant species of *Berardius* of the *Berardius*- was also added after Takahashi et al [14] as mentioned above ([3]). Accordingly, and, the comprehensive comparisons among both extinct and extant both extinct and extant species of *Berardius* and of the Berardiinae in the Ziphiidae were necessary to locate SCM 5530-6 more precisely in the phylogenetic framework. As a result of our cladistic analysis modified from Bianucci et al. [32], interestingly, despite its small size in *B. kobayashii* that is smaller than the smallest extant species *B. minimus*, the cranial morphologies of *B. kobayashii* sp. nov. were recognized as the most closely related to *Berardius minimus* as discussed above. It confirms that the new species based on SCM 5530-6 as the holotype actually belongs in the genus *Berardius*, and *B. bairdii* and *B. arnuxii* as discussed in the “comparisons” section. It suggests it also indicates that although *B. minimus* is represented only by the extant individuals in the northern North Pacific, the generic emergence could have been much earlier than the emergence and diversification of the other previously known rest of two free antitropical species, i.e., including the northern North Pacific middle Miocene extinct *B. kobayashii* sp. nov. and the *B. bairdii* and also the Antarctic *B. arnuxii*. Surprisingly, the cranial morphology of *B. kobayashii*, sp. nov. was the closest to that of *B. minimus* as mentioned in the “comparisons” section. It suggests that although *B. minimus* is represented only by the extant individuals from the northern North Pacific, its emergence could have been much earlier than the emergence and diversification of the two larger, antitropical or amphipolar species represented by them. Consequently, their emergence and diversification of the genus *Berardius* must have occurred in or by the late middle Miocene and potentially in the western North Pacific based on the new fossil from the middle to late Miocene boundary age Tsurushi Formation described here in the.

—For now, three genera are recognized in the subfamily Berardiinae; i.e., *Microberardius* from the middle late Miocene of South Africa [329], *Archaeoziphius* from the middle Miocene of Belgium [363] and *Berardius* from the middle to late Miocene boundary to the Recent of the northern North Pacific and the Antarctic oceans [1, 2, this study]. Previously, there was no fossil record of the berardiines in the North Pacific Ocean. So, the new fossil described here reveals that the berardiine ziphiids was already diversified existed at least in the western North Pacific at latest in the early late middle Miocene (ca. 11.2 Ma) and. It also suggests that the emergence of the extant genus berardiine *Berardius*-ziphiids goes back to the early middle to late Miocene boundary age of the North Pacific, or much then they quickly diversified with or just after the age of earlier because their closely related sister taxon *Archaeoziphius* in the early middle Miocene of the North Atlantic was already existed in the early middle Miocene [363].

At the moment, it is difficult to consider the more precise origin of the subfamily Berardiinae and the genus *Berardius* based on the present fossil evidence, but its potential origin of the subfamily might have been in the Southern Hemisphere on the basis of recent studies on their phylogenetic relationships and phylogeographic distributions of sister and out groups for the berardiine ziphiids ([3, 5, 32, 33, 36] and Figure 13). In addition, the close relationship between *B. kobayashii* and *B. minimus* indicates an emergence of the genus in the North Pacific, and then the ancestor of *B. bairdii* and *B. arnuxii* may have evolved somewhere from *B. kobayashii*-like berardiine in the Northern Hemisphere or somewhere in between the northern and southern hemispheres by the end of Miocene [377 McGowen]. It could have been followed by a dispersal (with gigantism) from the Southern to or from the Northern hemisphere [38 Morin et al., 2017]. In any case, the two currently sympatric form, smaller *B. minimus* and larger *B. bairdii* in the northern North Pacific could have been secondary contact after the Pliocene in age [38 Morin et al., 2017]. However, all these evidence suggest the rapid diversification of taxa in the subfamily Berardiinae in the initial stage of their evolution during in the middle and late Miocene in the Northern Hemisphere (Figure 13). It is difficult to establish the generic emergence of extant taxa based only on morphologies without

genetic information such as divergence time estimations. In the case of the genus *Berardius*, its emergence time estimation depends on the divergence time of the Berardiinae and other ziphiid subfamilies because the Berardiinae is at present monogeneric. In this regard, McGowen et al. [392009], for instance, estimated the divergence time of the Berardiinae from other subfamilies to be 21.98 Ma (95%CI: 17.11-29 Ma). They also estimated the differentiation of the two species of *Berardius* species (i.e., *B. arnuxii* and *B. bairdii*) to be 2.9 Ma (95%CI: 0.68-5.81 Ma). It means that the expected divergence time of the genus *Berardius* still spans 17.11-5.81 Ma in minimum estimation or 15-5.81 Ma based on the geologic age of the closest genus *Archaeoziphius* [36]. Although the divergence time of *B. minimus* from the ancestor of the two other species (i.e., *B. bairdii* and *B. arnuxii*) within the genus is still uncertain ([2, 38, 40], the minimum estimate of the divergence time could be dated back to at latest the early late Miocene in age based on our phylogenetic analysis that suggested the closest relationship of *B. minimus* to the middle/late Miocene boundary age *B. kobayashii*, sp. nov.

Accordingly, it is possible to interpret that the genus *Berardius* began to diversify at this period, which is known to be the period of global cooling and of coastline regression [41, 42, 43, 44]. It should be important to be noted that somewhat speciose extant genus *Mesoplodon*, which is consisted of at least 15 extant species, has already appeared and diversified in the late middle or early late Miocene ([37, 39, 45]. It strongly suggests that the much earlier and less diversified appearance of the genus *Berardius* might have proceeded their generic emergence, and *B. kobayashii*, sp. nov. could be interpreted as one of such representatives of the earlier generic emergence and diversification. It will be further considered based on molecular studies for *B. minimus* and also morphological studies for other fossil berardiines to be described in the future.

8. Conclusions

An extinct beaked whale from the middle to late Miocene boundary age Tsurushi Formation (ca. 12.3 – 11.5 Ma) in the Sado Island, Niigata Prefecture, Japan, was described as a new species of the extant genus *Berardius*; i.e., *B. kobayashii*. It is distinguishable from all other species within *Berardius* by having the following characters: extremely small size among the four species of *Berardius* (60-70 % smaller than *B. bairdii* and *B. arnuxii*, 50% smaller than *B. minimus*); the transverse diameter of the premaxilla is relatively wide to the width of the rostrum at the level of the anterior notch; the zygomatic process of the squamosal is anterodorsally tapered and triangular in shape; the nasals are short. *Berardius kobayashii*, sp. nov., is the fourth species of the genus. Namely, the lineage leading to the extant genus of the ziphiids had already lived in the western North Pacific at the time of at latest 11 Ma. The western North Pacific including the Sea of Japan may, therefore, have been one of the areas for of the evolution and radiation of *Berardius* species. The existence of a species of the extant ziphiid genus that have been known as one of the deep-diving whales from the middle Miocene deep-sea deposits will become the key to elucidate the process of their habitat expansions as deep-sea dwellers among the Ziphiidae.

Acknowledgments

We would like to thank the late I. Kobayashi (Professor Emeritus at Niigata University), A. Matsuoka (Niigata University), M. Shimizu (Asahimachi Museum, Niigata University), M. Ikarashi (Sado Museum), M. Aida (Sado Geopark Promotion Office and Sado Museum), K. Narita (Nagano City Museum), M. Kato (Nagaoka Municipal Science Museum), Y. Tajima and T.K. Yamada (NMNS), for allowing access to the specimens and giving me the opportunities to observe specimens at respective museums; Y. Yanagisawa (Geological Survey of Japan), K. Takahashi (Biwako Museum), H. Ichishima (Fukui Prefectural Dinosaur Museum), K. Nagasawa (formerly Yamagata Prefectural Museum), T. Kimura (Gunma Museum of Natural History), for providing useful discussion and advice with the literature on fossil ziphiids.

Funding Statement

A.K. was funded by the travel grant fund for the 2017 fiscal year from by the Regional Promotion Division of Sado City.

Data Accessibility

Additional data are available as the electronic supplementary materials (Supplementary files S1-S7: S1, list of the taxa and specimens of the berardiine ziphiids used for this study; S2, list of characters used for our cladistic analysis; S3,

character/taxon matrix; S4, resultant cladograms including the unweighted strict and 50% majority-rule consensus cladograms, five equally most parsimonious cladograms with implied weighting of K=3; S5, References for character collections; S6, Nexus file; S7, TNT file). The new species has been registered in Zoobank. The LSID for this publication is: urn:lsid:zoobank.org:act:317C1452-AE2E-4A08-A473-3B507B452D2C. This study and the nomenclatural acts it contains have been registered in Zoobank. The LSID for this publication is: urn:lsid:zoobank.org:pub:7BEDEBA0-46B9-4125-A7F8-2169EB016C29.

Competing Interests

We declare we have no competing interests.

Authors' Contributions

A.K. and N.K. conceived and designed the project, performed the experiment and analysed the data. A.K. prepared SCM 5530-6. N.K. contributed reagents/materials/preparation tools/analysis tools. A.K. and N.K. wrote the paper.

References

- Rosel P E, Archer FI, Baker CS, Boness DJ, Brownell RL, Churchill M Jr., Costa AP, Domning DP, Fordyce RE, Jefferson TA, Kinze C, Oliveira LR, Perrin WF, Wang JY, Yamada TK. 2019 List of marine mammal species and subspecies. *Committee on Taxonomy, The Society for Marine Mammalogy*.
- Yamada TK, Kitamura S, Abe S, Tajima Y, Matsuda A, Mead JG, Matsuishi TF. 2019 Description of a new species of beaked whale (*Berardius*) found in the North Pacific. *Scientific reports* 9(1), 1-14.
- Bianucci G, Di Celma C, Urbina M, Lambert O. 2016 New beaked whales in stem and crown Ziphiidae (Cetacea, Odontoceti). *PeerJ*, 4, e2479.
- Lambert O, de Muizon C, Duhamel G, Van Der Plicht J. 2018 Neogene and Quaternary fossil remains of beaked whales (Cetacea, Odontoceti, Ziphiidae) from deep-sea deposits off Crozet and Kerguelen islands, Southern Ocean. *Geodiversitas* 40(2), 135-140.
- Lambert O, Collareta A, Landini W, Post K, Ramassamy B, Di Celma C, Urbina M, Bianucci G. 2015 No deep diving: evidence of predation on epipelagic fish for a stem beaked whale from the Late Miocene of Peru. *Proc. R. Soc. B*, 282.
- Niigata Foraminiferal Research Group. 1967 Some foraminiferal assemblages from the Sawane area, Sado Island (Preliminary report): Restudy on the stratigraphy of the "Sawane Formation" based on the foraminiferal fossils. *Commemorative Volume for Mr. Y. Hiramatsu*, 115-119.
- Watanabe K. 1987 Tertiary Foraminifera and Radiolaria Fossils from Sado Island, Central Japan. *Publications from the Sado Museum* 9, 127-156.
- Okamoto Y, Satou M, Watanabe M, Yamamoto H. 1992 Inversion Tectonics in the southeastern part of the Japan Sea. *Jour. Struct. Geol.* Japan 38, 47-58.
- Kobayashi I. 2001 The Cenozoic of Niigata Prefecture, the case of Sado Island (Niigata no shinseikai Sado-hen). *Niigata Geotechnology Society* 57, 9-21.
- Yanagisawa Y, Watanabe M. 2017 Revised lithostratigraphy of the Neogene sedimentary sequence in the southern part of the Osado Mountain area, Sado Island, Niigata Prefecture, Japan. *Bull. Geol. Surv. Japan* 68 (6), 259-285.
- Sado Research Group for Marine Mammalian Fossils. 1987 Fossil Cetacean from Sado Island, Central Japan. *Publications from the Sado Museum* 9, 211-217.
- Horikawa H, Tazaki K, Kanno T. 1987 Fossil Ziphiidae from Koshiji-Syo off Sado Island, Central Japan. *Publications from the Sado Museum* 9, 225-230.
- Tazaki K, Horikawa H, Miyazaki S. 1987 Fossil Ziphiidae from Hyotan-Guri off Sado Island, Central Japan. *Publications from the Sado Museum* 9, 219-223.
- Takahashi K, Nomura M, Kobayashi I. 1989 A fossil cetacean skull (*Berardius* sp. indet.) from Dogama, Ogi-machi, Sado Island, Central Japan. *Earth Science: Chikyu Kagaku* 43, 102-105.
- Barnes LG, Hirota K. 1995 Miocene pinnipeds of the otariid subfamily Allodesminae in the North Pacific Ocean: Systematics and relationships. *The Island Arc* 3, 329-360.
- Utashiro T. 1950 The stratigraphy of Sawane-Aikawa region in Southwest Sado Island (Osado Mountain range). *Journal of the Geological Society of Japan* 56, 302-303.
- Shimazu M, Kanai Y, Toyama T, Ichihashi K, Minakawa J, Takahama N. 1973 Structural development and igneous activity in the Sado island. *Memoirs of the Geological Society of Japan* 9, 147-157.
- Shimazu M, Toyama T. 1982 Neogene volcanic rocks of the Sado islands. *Journal of the Geological Society of Japan* 88, 381-400.
- Yanagisawa Y. 2012 Late Miocene diatoms in the Hamochi area, Sado Island, Niigata Prefecture, Japan. *Open-File Report of the Geological Survey of Japan, AIST* 568, 1-19.
- Ogi Collaborative Research Group. 1986 Late Cenozoic Group in the Southern part of Kosado Mountains, Niigata prefecture. *Earth Science* 40, 417-436.
- Takeuchi K, Ozaki M, Komatsubara T. 2011 Explanatory notes of 1:200,000 geological map of the coastal zone around Niigata. Digital Geoscience Map, Seamless Geoinformation of Coastal Zone "Coastal Zone around Niigata".
- Tokunaga S. 1939 A New Fossil Mammal Belonging to Desmostylidae. *Jubilee Publication in the Commemoration of Professor H. Yabe, M.I.A. Sixtieth Birthday*. Sendai: Institute of Geology and Paleontology, Tohoku Imperial University. 289-299.
- Watanabe H. 1932 Neogene (Shin Daisankei). *Nihon Chishitsu kousan shi* 92-152.
- Maiya S. 1978 Late Cenozoic planktonic foraminiferal biostratigraphy of the oil-field region of Northeast Japan. *Cenozoic Geology of Japan*, 35-60.
- Kawatani A, Sashida K, Sachiko A, Kohno N. 2019 Radiolarian fossils and estimated depositional age of the Miocene Tsurushi Formation distributed in Sado Island, Niigata Prefecture, Japan. *Bull. Geol. Surv. Japan* 70(1-2), 91-99. (doi: <https://doi.org/10.9795/bullgsj.70.91>)
- Ono K, Ueno T. 1985 Tertiary Vertebrates from Sado Island, Niigata Prefecture, Central Japan. *Memoirs of the National Science Museum* 18, 65-71.
- Mead JG, Fordyce RE. 2009 The therian skull: a lexicon with emphasis on the odontocetes. *Smithsonian Contributions to Zoology*, 1-249.
- Kasuya T. 1973 Systematic consideration of recent toothed whales based on the morphology of tympano-periotic bone. *Scientific Reports of the Whales Research Institute* 25, 1-103.
- Maddison WP and Maddison DR. 2018. [Mesquite: a modular system for evolutionary analysis. Version 3.61.](http://www.mesquiteproject.org) <http://www.mesquiteproject.org>.
- Goloboff PA, Catalano SA. 2016. [TNT version 1.5, including a full implementation of phylogenetic morphometrics. Cladistics](https://doi.org/10.1111/tnt.12066) 32(3), 221-238.

R. Soc. open sci.

R. Soc. open sci. article template

15

- 29-31. Goloboff PA. 1993. Estimating character weights during tree search. *Cladistics* **9**, 83-91.
- 30-32. Bianucci G, Lambert O, Post K. 2007 A high diversity in fossil beaked whales (Mammalia, Odontoceti, Ziphiidae) recovered by trawling from the sea floor off South Africa. *Geodiversitas* **29**(4), 561-618.
- 31-33. Bianucci G, Lambert O, Post K. 2010 High concentration of long - snouted beaked whales (genus *Messapicetus*) from the Miocene of Peru. *Palaeontology* **53**(5), 1077-1098.
- 32-34. Lambert O, De Muizon C, Bianucci G. 2013 The most basal beaked whale *Ninziphius platyrostris* Muizon, 1983: clues on the evolutionary history of the family Ziphiidae (Cetacea: Odontoceti). *Zoological Journal of the Linnean Society* **167**(4), 569-598.
- 33-35. Miwa M, Yanagisawa Y, Yamada K, Irizuki T, Shoji M, Tanaka Y. 2004 Planktonic foraminiferal biostratigraphy of the Pliocene Kuwae Formation in the Tainai River section, Niigata Prefecture and the age of the base of the No. 3 Globorotalia inflata bed. *Journal of the Japanese Association for Petroleum Technology* **69**(3), 272-283.
- 34-36. Lambert O, Louwye S. 2006 *Archaeoziphius microglenoideus*, a new primitive beaked whale (Mammalia, Cetacea, Odontoceti) from the Middle Miocene of Belgium. *Journal of Vertebrate Paleontology* **26**(1), 182-191.
35. Bianucci G. 1997 The Odontoceti (Mammalia-Cetacea) from Italian Pliocene. The Ziphiidae. *Palaeontographia Italica* **84**, 163-192.
36. Reidenberg JS, Laitman JT. 2008 Sisters of the sinuses: cetacean air sacs. *The Anatomical Record: Advances in Integrative Anatomy and Evolutionary Biology* **291**(11), 1389-1396.
37. Ramassamy B, Lambert O, Collareta A, Urbina M, Bianucci G. 2018 Description of the skeleton of the fossil beaked whale *Messapicetus gregarius*: searching potential proxies for deep-diving abilities. *Mitteilungen aus dem Museum für Naturkunde in Berlin - Fossil Record* **21**(4), 11-32.
37. McGowen MR, Tsagkogeorga G, Álvarez-Carretero S, Dos Reis M, Struëbig M, Deaville R, Jepson PD, Jarman S, Polanowski A, Morin PA, Rossiter SJ. 2020 Phylogenomic Resolution of the Cetacean Tree of Life Using Target Sequence Capture. *Systematic Biology* **69**, 479-501. DOI: 10.1093/sysbio/svz068.
38. Morin PA, Baker CS, Brewer RS, Burdin AM, Dalebout ML, Fedutin ID, Filatova OA, Jung JL, Lauf M, Potter CW, Richard G, Ridgway M, Robertson KM, Wade P. 2017 Genetic structure of the beaked whale genus *Berardius* in the North Pacific, with genetic evidence for a new species. *Marine Mammal Science* **33**, 96-111.
39. McGowen MR, Spaulding M, Gates J. 2009 Divergence date estimation and a comprehensive molecular tree of extant cetaceans. *Molecular Phylogenetics and Evolution* **53**, 891-906. DOI: 10.1016/j.ympev.2009.08.018.
40. Kitamura S, Matsuishi T, Yamada TK, Tajima Y, Ishikawa H, Tanabe S, Nakagawa H, Uni Y, Abe S. 2013 Two genetically distinct stocks in Baird's beaked whale (Cetacea: Ziphiidae). *Marine Mammal Science* **29**, 755-766.
41. Miller KG, Kominz MA, Browning JV, Wright JD, Mountain GS, Katz ME, Sugarman PJ, Cramer BS, Christie-Blick N, Pekar SF. 2005 The Phanerozoic record of global sea-level change. *Science* **310**, 1293-1298. <https://doi.org/10.1126/science.1116412> PMID: 16311326.
42. Zachos JC, Dickens GR, Zeebe RE. 2008 An early Cenozoic perspective on greenhouse warming and carbon-cycle dynamics. *Nature* **451**, 279-283. <https://doi.org/10.1038/nature06588> PMID: 18202643.
43. Hilgen FJ, Abels HA, Iaccarino S, Krijgsman W, Raffi I, Sprovieri R, Turco E, Zachariasse WJ. 2009 The global stratotype section and point (GSSP) of the Serravallian Stage (Middle Miocene). *Episodes* **32**, 152-166.
44. Betzler C, Eberli GP, Lüdmann T, Reolid J, Kroon D, Reijmer JGG, Swart PK, Wright J, Young JR, Alvarez-Zarikian C, Alonso-García M, Bialik OM, Blättler CL, Guo JA, Haffen S, Horozal S, Inoue M, Jovane L, Lanci L, Laya JC, Hui Mee AL, Nakakuni M, Nath BN, Niino K, Petruny LM, Pratiwi SD, Slagle AL, Sloss CR, Su X, Yao Z. 2018 Refinement of Miocene sea level and monsoon events from the sedimentary archive of the Maldives (Indian Ocean). *Progress in Earth and Planetary Science* **5**, 1-18. DOI 10.1186/s40645-018-0165-x.
45. Steeman ME, Hebsgaard MB, Fordyce RE, Ho SYW, Rabosky DL, Nielsen R, Rahbek C, Glenner H, Rensen MVS, Willerslev E. 2009 Radiation of extant cetaceans driven by restructuring of the oceans. *Systematic Biology* **58**, 573-585. DOI:10.1093/sysbio/syp060.
- 38-46. Akiba F. 1986 Middle Miocene to Quaternary diatom biostratigraphy in the Nankai Trough and Japan Trench, and modified Lower Miocene through Quaternary diatom zones for middle-to-high latitudes of the North Pacific. In Kagami, H., Karig, D. E., Coullbourn, W. T., et al., *Initial Reports of the Deep Sea Drilling Project*, U. S. Government Printing Office, Washington D. C. **87**, 393-480.
- 39-47. Yanagisawa Y, Akiba F. 1998 Refined Neogene diatom biostratigraphy for the northwest Pacific around Japan, with an introduction of code numbers for selected diatom biohorizons. *Jour. Geol. Soc. Japan* **104**, 395-414.

Tables

Measurements (in mm) of the skull of *Berardius kobayashii*

Skull

Preserved Condyllobasal length from tip of broken rostrum to hindmost margin of occipital condyles.	236
Greatest postorbital width.	*e224519
Greatest zygomatic width	*e245
Greatest width of bony nares.	38
Maximum width of premaxillary sac fossae.	87
Maximum width of right premaxillary sac fossa.	43
Maximum width of left premaxillary sac fossa.	38
Maximum width of nasals.	40

R. Soc. open sci.

Greatest length of left post temporal fossa, measured to external margin of raised suture.	87
Greatest width of left post temporal fossa at right angles to greatest length.	61
Major diameter of left temporal fossa proper.	38
Minor diameter of left temporal fossa proper.	33
Distance from foremost end of junction between nasals to hindmost point of margin of supraoccipital crest.	33
Length of left orbit from apex of preorbital process of frontal to apex of postorbital process.	51
Greatest width of internal fl nares.	91
Greatest length of left pterygoid.	109

Measurements (in mm) of the ear bones of *Berardius kobayashii*

Tympanic bulla

Standard length of tympanic bulla, distance from anterior tip to posterior end of outer posterior prominence.	37
Distance from anterior tip to posterior end of inner posterior prominence.	34
Distance postero-ventral tip of outer posterior prominence to tip of sigmoid process.	24
Distance postero-ventral tip of outer posterior prominence to tip of conical process.	17.5
Width of tympanic bulla at the level of sigmoid process.	20
Height of tympanic bulla, from tip of sigmoid process to ventral keel.	23
Width across inner and outer posterior prominence.	18
Greatest depth of interprominential notch.	5
Width of upper border of sigmoid process.	11

Periotic

Standard (maximum anteroposterior) length, from tip of anterior process to posterior end of posterior process, measured on a straight line parallel to cerebral border	39.7
Standard (maximum mediolateral) width, from medial edge of cochlear portion to apex of lateral tuberosity	23
Maximum dorsoventral depth of anterior process perpendicular to axis of periotic	13.6
Maximum mediolateral width of anterior process at base	17
Length of anterior process from anterior apex to posterior border of malleolar fossa	16.2
Length of anterior process from anterior apex to anterior incisure	14.3
Anteroposterior length of cochlear portion	17.3
Dorsoventral depth of cochlear portion	18.7
Mediolateral width of cochlear portion from medial edge to fenestra ovalis	10.8
Maximum diameter of fenestra rotunda	2.7
Maximum diameter of internal acoustic meatus	4.7
Maximum diameter of cochlear aqueduct	1.9
Maximum diameter of aperture for the vestibular aqueduct	3.6
Anteroposterior diameter of proximal opening of facial canal	3.4
Anteroposterior diameter of distal opening of facial canal	1.9
Length of posterior process (posterior bullar facet between anteroposterior tips)	17.3

R. Soc. open sci.

R. Soc. open sci. article template

17

Width of posterior process (posterior bullar facet between mediolateral tips)	12.1
---	------

Figure and table captions

Figure 1. Index maps of the Pacific Northwest: A, locations of the Japanese Islands, showing Sado Island; B, map of Sado Island, showing some major place names mentioned in text, and location of enlarged area including the type locality for *Berardius kobayashii*, sp. nov.; C, detail of Sobama beach in Donokama, Ogi Town, showing the exact locality for the holotype of *Berardius kobayashii*, SCM 5530-6. Based on the GIS map (<https://maps.gsi.go.jp/#15/37.844139/138.272681/&base=std&ls=std&disp=1&vs=c0j0h0k0l0u0t0z0r0s0m0f1>) of the Geospatial Information Authority of Japan.

Figure 2. Correlation of formations distributed on the Osado (a) and Kosado (b) mountain ranges containing whale fossil horizon with relevant systems of chronology. Wave lines indicate unconformities. After [6, 19, 10]. Diatom zones: [10, 46374037, 47384138]; Planktonic foraminiferal zones: [24, 3532].

Figure 3. Dorsal view of the cranium of *Berardius kobayashii*, sp. nov., holotype, SCM 5530-6. (Scale bar = 10 cm)

Figure 4. Lateral view of the cranium of *Berardius kobayashii*, sp. nov., holotype, SCM 5530-6. A and B, left side; C and D, right side. (Scale bar = 10 cm)

Figure 5. Anterior view of the cranium of *Berardius kobayashii*, sp. nov., holotype, SCM 5530-6. (Scale bar = 10 cm)

Figure 6. Posterior view of the cranium of *Berardius kobayashii*, sp. nov., holotype, SCM 5530-6. (Scale bar = 10 cm)

Figure 76. Ventral view of the cranium of *Berardius kobayashii*, sp. nov., holotype, SCM 5530-6. (Scale bar = 10 cm)

Figure 87. Posterior fragment of the left mandible of *Berardius kobayashii*, sp. nov., holotype, SCM 5530-6. A, lateral view; B, medial view. (Scale bar = 10 cm)

Figure 98. Left periotic of *Berardius kobayashii*, sp. nov., holotype, SCM 5530-6. A, ventral view; B, corresponding line drawing with anatomical interpretations (gray hatches indicate parts of the tympanic bulla); C-F, the reconstructed CT image from 3D surface rendering, with anatomical interpretations (C, ventral view; D, lateral view; E, medial view; F, dorsal view); C, dorsal view of the reconstructed CT image from 3D surface rendering, with anatomical interpretations; D, comparisons of the lateral view of anterior process of periotic among the genera of Ziphiidae (modified from Bianucci, 1997). (Scale bar = 1 cm)

Figure 109. The left tympanic bulla A, dorsal view; B, anterior view; C, medial view; D, lateral view; E, posterior view; F, ventral view (Scale bar = 1 cm)

Figure 101. Key features of the left tympanic bulla of *Berardius kobayashii*, sp. nov., holotype, SCM 5530-6. A, dorsal view; B, anterior view; C, medial view; D, lateral view; E, posterior view; F, ventral view. (Scale bar = 1 cm)

Figure 121. Phylogenetic relationship and paleobiogeographic distribution of *Berardius kobayashii*, sp. nov., within the Ziphiidae based on Bianucci et al. [3] and our re-analysis. The strict consensus tree resulting from five most parsimonious trees, 17795 steps long, with the consistency index = 0.475523 and the retention index = 0.767.

R. Soc. open sci.

Figure 132. Distribution and potential roots for the dispersal of the known extinct species of the Berardiinae in
the middle Miocene inferred from the phylogenetic analysis.

Table 1. Measurements (in mm) of the skull of *Berardius kobayashii*, sp. nov., holotype, SCM 5530-6. ^{**}
indicates estimated transverse measurements that are half-skull measurements multiplied by two, and 'e'
indicates estimation.

Table 2. Measurements (in mm) of the ear bones of *Berardius kobayashii*, sp. nov., holotype, SCM 5530-6.

*R. Soc. open sci.*

Figure 1. Index maps of the Pacific Northwest: A, locations of the Japanese Islands, showing Sado Island; B, map of Sado Island, showing some major place names mentioned in text, and location of enlarged area including the type locality for *Berardius kobayashii*, sp. nov.; C, detail of Sobama beach in Donokama, Ogi Town, showing the exact locality for the holotype of *Berardius kobayashii*, SCM 5530-6. Based on the GIS map (<https://maps.gsi.go.jp/#15/37.844139/138.272681/&base=std&ls=std&disp=1&vs=c0j0h0k0l0u0t0z0r0s0m0f1>) of the Geospatial Information Authority of Japan.

318x487mm (96 x 96 DPI)

Series	Age (Ma)	Stages	Diatom Zones	Planktonic foraminiferal zones	Osado (a) (Yanagisawa & Watanabe, 2017)	Kosado (b) (Niigata Foraminiferal Research Group, 1967; Yanagisawa, 2012; Kawatani et al., 2019)			
Upper Miocene	6	Messinian	7B	PF5	Nozaka	Yamadagawa			
		Tortonian	7A				PF4	Nakayama	Notayama
	6B								
	6A								
	5D								
	10		5C	PF3	Hanyugawa	Orito			
	Middle Miocene		12				5B		
			14	Serravallian	5A				
				Nanatani	4Bb	PF2			
		4Ba	PF1		Orito	Orito			
16	4A								
Lower Miocene	18	Langhian	3B	PF1	Kinpokusan	Kyozukayama			
			3A						
	Burdigalian	Mikawa	2B	PF1	Masaragawa	Sanze			
			2A						
			Aquitania				1	Aikawa	Aikawa

Figure 2. Correlation of formations distributed on the Osado (a) and Kosado (b) mountain ranges containing whale fossil horizon with relevant systems of chronology. Wave lines indicate unconformities. After [6, 19, 10]. Diatom zones: [10, 46, 47]; Planktonic foraminiferal zones: [24, 35].

330x310mm (96 x 96 DPI)

Figure 3. Dorsal view of the cranium of *Berardius kobayashii*, sp. nov., holotype, SCM 5530-6. (Scale bar = 10 cm)

512x244mm (96 x 96 DPI)

Figure 4. Lateral view of the cranium of *Berardius kobayashii*, sp. nov., holotype, SCM 5530-6. A and B, left side; C and D, right side. (Scale bar = 10 cm)

789x720mm (96 x 96 DPI)

Figure 5. Anterior view of the cranium of *Berardius kobayashii*, sp. nov., holotype, SCM 5530-6. (Scale bar = 10 cm)

578x245mm (96 x 96 DPI)

Figure 6. Posterior view of the cranium of *Berardius kobayashii*, sp. nov., holotype, SCM 5530-6. (Scale bar = 10 cm)

463x247mm (96 x 96 DPI)

Figure 7. Ventral view of the cranium of *Berardius kobayashii*, sp. nov., holotype, SCM 5530-6. (Scale bar = 10 cm)

796x411mm (96 x 96 DPI)

Figure 8. Posterior fragment of the left mandible of *Berardius kobayashii*, sp. nov., holotype, SCM 5530-6. A, lateral view; B, medial view. (Scale bar = 10 cm)

245x197mm (96 x 96 DPI)

Figure 9. Left periotic of *Berardius kobayashii*, sp. nov., holotype, SCM 5530-6. A, ventral view; B, corresponding line drawing with anatomical interpretations (gray hatches indicate parts of the tympanic bulla); C-F, the reconstructed CT image from 3D surface rendering, with anatomical interpretations (C, ventral view; D, lateral view; E, medial view; F, dorsal view). (Scale bar = 1 cm)

924x469mm (96 x 96 DPI)

Figure 10. The left tympanic bulla A, dorsal view; B, anterior view; C, medial view; D, lateral view; E, posterior view; F, ventral view (Scale bar = 1 cm)

406x401mm (96 x 96 DPI)

Figure 11. Key features of the left tympanic bulla of *Berardius kobayashii*, sp. nov., holotype, SCM 5530-6. A, dorsal view; B, anterior view; C, medial view; D, lateral view; E, posterior view; F, ventral view. (Scale bar = 1 cm)

595x458mm (96 x 96 DPI)

Figure 12. Phylogenetic relationship and paleobiogeographic distribution of *Berardius kobayashii*, sp. nov, within the Ziphiidae based on Bianucci et al. [3] and our re-analysis. The strict consensus tree resulting from five most parsimonious trees, 177 steps long, with the consistency index = 0.475 and the retention index = 0.767.

397x252mm (96 x 96 DPI)

Figure 13. Distribution and potential roots for the dispersal of the known extinct species of the Berardiinae in the middle Miocene inferred from the phylogenetic analysis.

228x110mm (96 x 96 DPI)

Appendix E**ROYAL SOCIETY
OPEN SCIENCE****The oldest fossil record of the extant genus *Berardius*
(*Odontoceti*, *Ziphiidae*) from the middle to late Miocene
boundary of the western North Pacific**

Journal:	Royal Society Open Science
Manuscript ID	RSOS-201152.R1
Article Type:	Research
Date Submitted by the Author:	24-Nov-2020
Complete List of Authors:	Kawatani, Ayako; University of Tsukuba Graduate School of Life and Environmental Sciences Kohno, Naoki; University of Tsukuba Graduate School of Life and Environmental Sciences; National Museum of Nature and Science, Geology and Paleontology
Subject:	palaeontology < BIOLOGY, Palaeontology < EARTH SCIENCES
Keywords:	middle Miocene, extant genus, Berardius kobayashii , sp. nov., western North Pacific
Subject Category:	Organismal and Evolutionary Biology

Author-supplied statements

Relevant information will appear here if provided.

Ethics

Does your article include research that required ethical approval or permits?:

This article does not present research with ethical considerations

Statement (if applicable):

CUST_IF_YES_ETHICS :No data available.

Data

It is a condition of publication that data, code and materials supporting your paper are made publicly available. Does your paper present new data?:

Yes

Statement (if applicable):

Additional data are available as the electronic supplementary materials (Supplementary files S1-S7: S1, list of the taxa and specimens of the berardiine ziphiids used for this study; S2, list of characters used for our cladistic analysis; S3, character/taxon matrix; S4, resultant cladograms including the unweighted strict and 50% majority-rule consensus cladograms, five equally most parsimonious cladograms with implied weighting of K=3; S5, References for character collections; S6, Nexus file; S7, TNT file). The new species has been registered in Zoobank. The LSID for this publication is: urn:lsid:zoobank.org:act:317C1452-AE2E-4A08-A473-3B507B452D2C.

Conflict of interest

I/We declare we have no competing interests

Statement (if applicable):

CUST_STATE_CONFLICT :No data available.

Authors' contributions

This paper has multiple authors and our individual contributions were as below

Statement (if applicable):

A.K. and N.K. conceived and designed the project, performed the experiment and analysed the data.

A.K. prepared SCM 5530-6. N.K. contributed reagents/materials/preparation tools/analysis tools.

A.K. and N.K. wrote the paper.

21 November 2020

Dear Editors,

First of all, we deeply thank both reviewers for their generous comments and suggestions for our manuscript on the oldest *Berardius* from Japan, and we revised our manuscript to address their concerns. The reviewers' comments and suggestions greatly improved our manuscript. In spite of the constructive reviews, we have to apologize the delay to revise our manuscript because of the recent COVID19 crisis as a second wave in August and third wave in October and now. Although delayed, we thoroughly revised the manuscript and improved it with additional data following recommendations and suggestions from both reviewers. Changes are recorded with the track and change function in the revised manuscript. We followed all the reviewers' comments. Reviewer's comments are written in Black below. Our responses are written in Blue.

Reviewers' Comments to Author:

Reviewer: 1

Comments to the Author(s)

The present work deals with a very informative ziphiid fossil skull, including the highly informative ear bones. Those elements are only rarely found associated to other cranial material. Furthermore, the stratigraphic context is relatively well constrained, providing a geological age that is rather exciting (most extinct ziphiids are found in younger deposits). Finally, the specimen displays similarities with members of the extant family Berardiinae, potentially confirming the antiquity of this clade. All together, these elements make this study important and worth publishing.

The text is well organized and the bones are finely illustrated (see a few comments below and in the annotated pdf). I found the descriptions clear and concise (maybe a bit too short), although a few anatomical interpretations may ask for a second check.

I made a series of comments in the annotated pdf, and those mostly deal with English and typos,

which should be easily fixed. My main concerns and suggestions are listed below, and those may
request some more work. I hope that those will prove useful, as this very fine, scientifically
informative specimen definitely deserves to be published.

—>We corrected all the grammatical and typographic errors following suggestions on the marked
pdf. We are ashamed of such problems, and we strongly appreciate to the reviewer's advice.

1- Concerning the stratigraphic context, you reach the conclusion that the formation from which
the studied fossil originates can be dated from an interval spanning 12.3–11.5 Ma. This means
that this unit may extend in the earliest Tortonian (starting at 11.63 Ma). If this is correct it may
be difficult to exclude an early late Miocene age, and I would suggest modifying the text (and the
title + abstract) accordingly. Maybe 'from the latest middle to earliest late Miocene', or 'from the
middle to late Miocene boundary'. This would remove any possible confusion, I think. Indeed,
many ziphiids have been dated from the Tortonian, whereas Serravallian records are much less
frequent.

—>We changed the above-mentioned sentence in our revised MS from “from the middle Miocene”
to “from the middle to early Miocene boundary” to avoid any possible confusion. The
estimation of the geologic age for the lower part of the Tsurushi Formation that produced the
holotype skull was based on the diatom fossils. So, the “minimum” age estimation is actually
crossing the boundary between the middle and late Miocene. Again, we appreciate to the
reviewer's advice.

2- The emended diagnosis of *Berardius* is long, and it is not easy to understand how it was
compiled, for several reasons. First, some characters are also listed for the subfamily Berardiinae.
Second, differences are not associated to other genera. It may be ok to keep such a list, but then
it should start with something like 'The genus *Berardius* differs from all other ziphiids with the
following unique combination of characters: ...'. However, because you provide a diagnosis for
Berardiinae just above, it may be sufficient to provide here differences with other Berardiinae
(and maybe expand somewhat the diagnosis for Berardiinae, including all characters cited here
that also apply to other berardiines).

It would certainly be useful to list, in the diagnosis of *Berardius*, specific differences with
*Archaeoziphius* and *Microberardius*.

—>We revised the diagnosis of the genus *Berardius* based partly on our own phylogenetic
analysis, and changed the 'style' of descriptions to become it understandable to be able to

differentiate the genus from other genera within the subfamily.

3- For the very informative periotic I would suggest also providing lateral and medial views,
based on the 3D model.

A minor issue: the photo of the ventral view in fig. 8a is not in the same orientation as the line
drawing in fig. 8b. If not too complicated I would suggest correcting this. Another probably easier
solution would be to provide a true ventral view based on the 3D model.

—>We added lateral and medial views of the periotic with additional anatomical interpretations.
Also, we re-prepared the Figure 8a to fit the illustration of the same orientation as the Figure 8b.
However, the orientations of figures 8a and 8b slightly differ from the exact ventral view, so we
also added the picture of ‘correct’ orientation for the ventral view as the Figure 8c.

4- I fully agree that this specimen differs from *Archaeoziphius* and *Microberardius*. However, are
these differences greater than the ones separating this specimen from extant *Berardius* species?
You mention potential synapomorphies of *Berardius* at the level of the periotic, but unfortunately
the periotic is unknown in *Archaeoziphius* and *Microberardius*, meaning that these characters
could possibly represent synapomorphies for *Berardiinae* instead of *Berardius*, making the
attribution of that genus not strongly supported. Furthermore, other similarities with *Berardius*
may correspond to plesiomorphic features.

I would suggest reorganizing the comparison, in a way to first provide characters of the family
*Ziphiidae* that can be observed in the new species, then provide characters of the subfamily
*Berardiinae*, and finally derived characters of the genus *Berardius*. This last part may prove more
difficult, as sister groups *Archaeoziphius* and *Microberardius* unfortunately lack ear bones.

—>This is somewhat difficult to formularize because of the incompleteness of fossil taxa
including *Archaeoziphius* and *B. kobayashii* new species itself. The completeness of those three
taxa including *Archaeoziphius* and *Microberardius* and *B. kobayashii* were: $25/51=49\%$ for
*Microberardius*, $27/51=53\%$ for *Archaeoziphius* and $32/51=63\%$ for *B. kobayashii* new species.
Five most parsimonious trees have suggested missing synapomorphies and/or autapomorphies for
each node because of incompleteness of specimens. But three of those five consensus trees have
suggested that the new species was the closest to *Berardius minimus* with synapomorphies, and
other two were kinds of paritomy with *B. minimus* and *Archaeoziphius* without sufficient
synapomorphies. So, we decided somewhat conservatively to describe a new species within
*Berardius*. However, we revised the Comparisons section following the above advice based on

our analysis.

5- A phylogenetic tree is provided, but it is not made clear how the relationships of the new
species with other ziphiids were investigated. My impression is that you used a tree from a
previous analysis, before manually adding the new species. This should probably be explained. If
this is the case, I would tend to think that testing the affinities of the new species with a proper
phylogenetic analysis would provide a more convenient base for the systematic and taxonomic
results, as well as for the discussion on paleobiogeography and the evolution of the genus
Berardius. As currently presented, the discussion may sound a bit too speculative. In other words,
I think that the attribution of the new species to the genus Berardius should be backed up by a
phylogeny.

—>The reviewer's recommendation is very fundamental for our treatment regarding the new
species described here with taxonomic considerations without the proper phylogenetic analysis.
Accordingly, we did our own cladistic analysis based on characters and matrix prepared by
Bianucci et al. (2016) with additions of three taxa that were not included in Bianucci et al. (2016):
i.e., our new taxon, *Berardius minimus* Yamada et al., 2019 that was described after the
publication of Bianucci et al. (2016), and *Berardius bairdii*, which was probably "represented"
by *B. arnuxii* as the extant genus and was not included in their analysis. We located *B.*
*kobayashii* in the cladogram more precisely based on the result of our own analysis and re-
prepared the Figure 12 based on our result. Unfortunately, the consensus tree from five most
parsimonious trees were a kind of unresolved polytomy, but we put additional discussion based
on our analysis.

6- It may be interesting to note that divergence date estimates for the genus Berardius (at that time
only including *B. bairdii* and *B. arnuxii*) are considerably younger (mean = 2.9 Ma) than the age
of the new species (see McGowen et al. 2009).

McGowen, M.R., Spaulding, M., and Gatesy, J. 2009. Divergence date estimation and a
comprehensive molecular tree of extant cetaceans. *Molecular Phylogenetics and Evolution* 53 (3):
891-906.

—>The divergence date estimation for the extant species has been established by the above paper
(2009), and the latest paper (2020) also by the same authors has estimated them as the maximum
and minimum time estimations of their divergences based on the hard and soft bounds methods

(McGowen et al., 2020). Based on that paper, the divergence time estimation for the subfamily
Berardiinae is to be 21.98 Ma (95% CI is 17.11-29 Ma). They also estimated the split of the two
species of *Berardius* is to be 2.9 Ma (95% CI is 0.68-5.81 Ma). So, the expected divergence
time of the genus *Berardius* and its closest genus *Archaeoziphius* could still spans 15-5.81 Ma in
minimum estimation. In addition, recently described new species of the genus, i.e., *B. minimus*
is recognized as an earlier diverging species from *B. bairdii* and *B. arnuxii* clade (Kitamura et al.,
2013; Morin et al., 2017; Yamada et al., 2019). We do not know how to estimate the “longevity”
of the genus properly, but we can recognize it based on the relationship of the fossil taxon to other
species. In the case of a new fossil described in our MS, it was recognized a sister taxon to the
extant species within the extant genus; i.e., *Berardius minimus*, despite the phylogenetic position
of *Archaeoziphius* was not stable in terms of the monophyly of the species within the genus
*Berardius* because of the above reason (missing entries of characters in fossil taxa). In fact, they
share quite a few characters used in Bianucci et al. (2016) and in our analysis. Accordingly, we
preferred conservatively to keep the generic status for the new species based on our result until
much better specimen(s) of *Microberardius* and especially more closely related *Archaeoziphius*
are obtained.

Reviewer: 2

Comments to the Author(s)

Dear

Kawatani, Ayako; Kohno, Naoki

I have done an extensive revision of your work and found the overall purpose of the manuscript
interesting and useful. Despite this, I do believe that the work in its current state will need a
substantial revision in order to be publishable. Below you'll find my comments:

1) I found the manuscript inconsistently written and difficult to read. I would strongly advise to
send the manuscript to a native English speaker to help with the drafting of the paper. As a non-
native English speaker, I completely understand the fact that the authors could have made more
mistakes than usual, as this is the case with me. There are several parts where the inconsistency
of the manuscript (grammar, bad translations or changes between tenses) makes it mostly difficult

to read, or the idea in the sentences is lost.

—>We corrected all such grammatical and typographic errors following suggestions on the
marked pdf from both reviewers. We are ashamed of such problems, and we strongly appreciate
to the reviewer's kind advice and corrections.

2) Proposing a new species, especially for an extant genus, must have a strong phylogenetic
framework in order to support this assignation. Since this is a work on Systematics it should be
the center of the manuscript. I did not find major references to it, but a figure and with some lines
in text:

a. There is no description of the trees, statistical support, consensus, or even references to the
methodology undertaken. A simple statement as 'it follows Bianucci et al. 2016' is not enough.

b. No list of plesiomorphies, synapomorphies or apomorphies that support each of the
Systematical assignations is specified, but some references in the 'Comparisons' section. If this
is not provided, the support of the new species cannot be easily corroborated or compared, either
in reviewing process or by other researchers interested in this work.

—>Another reviewer has also pointed out that it was very fundamental problem for our treatment
regarding the description of a new species in the recent genus without sufficient phylogenetic
analysis. So, we did it. Initially, we used a tree from the result of the phylogenetic analysis in
Bianucci et al. (2016) and manually added the new species based on our observations of shared
characters among species in the subfamily Berardiinae. In our revised MS, we did our own
cladistic analysis based on characters and matrix prepared by Bianucci et al. (2016) with additions
of three taxa that were not included in Bianucci et al. (2016): i.e., our new taxon from Japan,
*Berardius minimus* that was described by Yamada et al., (2019) after the publication of Bianucci
et al. (2016), and *Berardius bairdii* that was also not included in their analysis. We located *B.*
*kobayashii*, a new species described in our MS, in the cladogram more precisely based on the
result of our own analysis and re-prepared the Comparisons and Discussion sections with the new
figures 12 and 13. Unfortunately, we obtained five most parsimonious trees, and the consensus
tree from five most parsimonious trees were a kind of unresolved poritomy, but we added those
results in our discussion.

3) A differential diagnosis of the new species is missing. This section is key to check both the
apomorphies that support the new species, and the synapomorphies that it shares with other
closely related taxa. Furthermore, only a single apomorphy (provided in the abstract) supports the

new species? (Being the ‘smallest’ is not a diagnostic feature; it could be important to identify it,
but how can it be cladistical coded?). References to this should be in the abstract, some features
can be found on the ‘diagnosis of the species’ section, but not clear.

—>We revised the diagnosis of the subfamily Berardiinae, genus *Berardius* and a new species *B.*
*kobayashii* based mainly on our phylogenetic analysis, and changed the ‘style’ of descriptions to
become it understandable to be able to differentiate the genus and species from other genera and
species within the subfamily Berardiinae.

4) The description is mostly ambiguous and difficult to read. I would highly advise to rewrite the
descriptions based on regions or views. Please see comments in text. Squamosal is almost 100%
complete and has several major characters of odontocetes, a few lines to describe it is not enough.

—>Again, we appreciate to the reviewer’s critical and substantial advice (with corrections) and
suggestion. We re-prepared descriptions fundamentally following comments and suggestions
from both reviewers, and added descriptions of, for instance, the lateral and medial laminae,
squamosal, frontal and so on, based on our additional observations to specimens. We also re-
described such characters that were used in the cladistic analysis more clearly. The descriptions
and discussion for morphological comparisons among the three extinct taxa; i.e., *Microberardius*,
*Archaeoziphius* and a new taxon *B. kobayashii*. Unfortunately, the completeness of those three
extinct taxa are at present $25/51=49\%$ for *Microberardius*, $27/51=53\%$ for *archaeoziphius* and
$32/51=63\%$ for *B. kobayashii* new species, but we believe that the description became more
informative following the advice from both reviewers. Once again, we appreciate to the
reviewers’ critical and substantial advice.

5) The comparisons section is okay, as it emphasizes the comparison between taxa. However, I
would encourage the authors to discuss the specific variations among taxa in the Discussion. A
suitable way to reorder both the Comparisons and Discussion could be to briefly present the
phylogenetic synapomorphies + how this support the assignation to: Ziphiida, Berardiinae,
*Berardius* in the Comparison section, with references to other taxa; and emphasize the most
important characters in the Discussion section. Furthermore, key characteristics that delimit the
genus *Berardius* should be referenced in the Discussion.

—>We re-organized the Comparisons and Discussion sections following the advice from both
reviewers with clade-based comparisons to emphasize important characters to support our result.
Based on our cladistic analysis, we re-organized them based on our result. Although we have

obtained five most parsimonious trees resulting partially unresolved poritomy, we put those
results in our comparisons and discussion, and reflected in the revised figures 12 and 13.

Finally, I wish to emphasize that the material presented in this manuscript could be easily It has
a lot of potential, but several deficiencies still need to be corrected.

—>Once again, we appreciate to the reviewer's critical and substantial advice.

-----

Thank you for your consideration.

Sincerely,

Ayako Kawatani

Department of Life and Environmental Sciences, University of Tsukuba

kawatani@geol.tsukuba.ac.jp

Naoki Kohno

Department of Geology and Paleontology, National Museum of Nature and Science and

Department of Life and Environmental Sciences, University of Tsukuba

Kohno@kahaku.go.jp

The oldest fossil record of the extant genus *Berardius* (Odontoceti, Ziphiidae) from the middle to late Miocene boundary of the western North Pacific

Ayako Kawatani¹ and Naoki Kohno^{1,2}

¹Graduate School of Life and Environmental Sciences, University of Tsukuba, 1-1-1, Tennoudai, Tsukuba, Ibaraki, 305-8577, Japan

²National Museum of Nature and Science, 4-1-1, Amakubo, Tsukuba, Ibaraki, 305-0005, Japan

Keywords: middle Miocene, extant genus, *Berardius kobayashii*, sp. nov., western North Pacific.

1. Summary

A new species of a beaked whale that belongs to the extant genus *Berardius* is described from the middle to late Miocene boundary age Tsurushi Formation (ca. 12.3–11.5 Ma) on the Sado Island, Niigata Prefecture, Japan. The new species, *Berardius kobayashii*, sp. nov., represents the oldest record of this genus and suggests the timing of the emergence of this extant genus during this age period. *Berardius kobayashii*, sp. nov., has the following generic characters: the ratio between the width of the premaxillary crests and the width of the premaxillary sac fossae is 1.0 to 1.25, nodular frontals make isolated protuberance on the posterior part of the vertex. Among the species within the genus, *B. kobayashii*, sp. nov., shares a unique character with *B. minimus* such as the apices of the left and right hamular processes of the pterygoids contact medially, forming together a posteriorly directed medial point. In addition, *B. kobayashii*, sp. nov., displays the unique combination of both primitive and derived characters: it is extremely small in size, and the nasals are short, ratio between the length of the medial suture of nasals on vertex and the maximum width of nasals is <0.4 . *Berardius kobayashii*, sp. nov. fills the gap between the origin of the genus and later diversifications of the extant species. This discovery is also a key to elucidate the process of their emergence and habitat expansion during the ~~latest middle and earliest late Miocene in age~~. 
[revised manuscript text omitted]
. We also added *Berardius bairdii* based on our own observation into the data matrix, because it was also not included in the original analysis by Bianucci et al. [3]. Character and character states were managed using Mesquite 3.61 [29]. The phylogenetic analysis based on 31 ziphiids of both extinct and extant taxa with six outgroup taxa as OTUs and 51 morphological characters was performed with TNT 1.5 using the “New Technology Search” tasked to find minimum length trees 1,000 times [30]. As was mentioned by Bianucci et al. [3], our preliminary analyses also led to less-resolved trees. So we also downweighted as the same with Bianucci et al. [3] as the homoplastic characters using the default value of 3 for the constant K of the method formulated by

Goloboff [31]. Multi-state characters were treated as ordered for 18 characters (i.e., chars. 1, 3, 7, 8, 9, 11, 13, 14, 15, 16, 22, 23, 30, 34, 36, 39, 47, and 51) and unordered for six characters (i.e., chars. 2, 4, 10, 28, 34, and 45), and other binary characters were all treated as unordered, also following Bianucci et al. [3].

Institutional abbreviations are the following: IRSNB, Institut royal des Sciences naturelles de Belgique, Brussels, Belgium; MNHN: Muséum National d'Histoire Naturelle, Paris, France; NMNS-PV, Department of Geology and Paleontology, National Museum of Nature and Science, Tsukuba, Japan; NSMT-M, Department of Zoology, National Museum of Nature and Science, Tsukuba, Japan; SAM: Iziko South African Museum, Cape Town, South Africa; SCM, Sado Museum, Sado City, Niigata, Japan.

5. Systematics

CETARTIODACTYLA Montgelard, Catzefis and Douzery, 1997

CETACEA Brisson, 1762

ODONTOCETI Flower, 1867

ZIPHIIDAE Gray, 1850

BERARDIINAE Moore, 1968

Emended diagnosis of subfamily: The subfamily differs from all other ziphiids with the following unique combination of characters: ~~the ascending process of the premaxilla in lateral view is slightly concave (Char. 7, reversal); constriction on the ascending process of the right premaxilla (between premaxillary sac fossa and the premaxillary crest) is weak, and the ratio between the minimal width of ascending process of premaxilla and the width of right premaxillary crest is >0.80 (Char. 8, reversal); no anteromedial excavation of the dorsal surface of the nasal (Char. 14, reversal); the nodular protuberance on the posterior part of the vertex is formed by the interparietal and/or the frontals (Char. 17, 0->1 & 2); and the anteromedial margin of the supraoccipital is distinctly lower than the dorsal margin of the vertex (Char. 18).~~

Genus *Berardius* Duvernoy, 1851

Emended diagnosis of genus: The genus differs from *Microberardius* with the following unique combination of characters: ~~the ratio between the width of the premaxillary crests and the width of the premaxillary sac fossae is 1.0 to 1.25 (Char. 11); the nodular protuberance on the posterior part of the vertex is formed only by the frontals (Char. 17').~~ Following characters shared with *Berardius* species, but uncertain in *Microberardius* and *Archaeoziphius* because of lack of such portion, are potentially diagnostic for the genus: ~~the anterior spine of the tympanic with a more or less rectilinear anterior margin (Char. 22); the posteroventral corner of the sigmoid process of the tympanic is posteriorly projected in lateral view (Char. 23); the dorsal margin of the involucrum of the tympanic that is cut by an indentation that is present and visible in medial and/or dorsal view (Char. 24).~~

Berardius kobayashii, sp. nov.

(Skull: Figure 3-8; Ear bone: Figure 9-11)

URN: lsid:zoobank.org:act: 317C1452-AE2E-4A08-A473-3B507B452D2C

Diagnosis of species: The species differs from all other *Berardius* species (i.e., *B. arnuxii*, *B. bairdii* and *B. minimus*) with the following unique characters: ~~the nasals are short~~, ratio between the length of medial suture of nasals on the vertex and the maximum width of nasals <0.4 (Char. 13, reversal). Although it is potentially a shared primitive trait with *Archaeoziphius microglenoideus*, an extremely small size among the four species (60-70 % smaller than *B. bairdii* and *B. arnuxii*, 50% smaller than *B. minimus*) is potentially the specific trait within the genus.

Holotype: SCM 5530-6, incomplete skull with the left tympanic bulla and periotic, lacking the rostral portion; the posterior fourth of the horizontal ramus of the left dentary with the coronoid process and mandibular condyle. Collected by Kiyoei Kaneko in or before 1969.

Type Locality: Sobama Beach, Ogi Town, Sado Island, Niigata Prefecture, Japan. Geographic coordinates: 37° 84'50" N, 138° 26'58" E.

Formation and age of holotype: SCM 5530-6 was collected from the latest middle to earliest late Miocene age Tsurushi Formation. As discussed above, the geologic age of the Tsurushi Formation is estimated to be approximately 12.3–11.5 Ma.

Etymology: The species is named in honor of the late Dr. Iwao Kobayashi, a professor emeritus at Niigata University, for his longstanding contributions to the geology and paleontology of the northern Fossa Magna region including Niigata Prefecture, and in gratitude for his permission to re-study the marine mammal fossils collected by him and his colleagues from the Sado Island and encouragement to both of us throughout this study.

Description

Cranium (general morphology)

The cranium is relatively small (Table 1) and slightly asymmetric (Figure 2,5). The ratio between the height and width of the skull is 1:1.19. The rostral and the occipital parts are almost completely broken and missing. The small part of the left mandible was originally attached to the cranium. The vertex is moderately elevated and slightly skewed toward the left. The posterior part of the vertex is almost broken. The outline of the temporal fossa in lateral view is an isosceles triangle with long sides that converge anteriorly, and the posterior margin is weakly rounded. The temporal fossa is shallow and its lateral surface is convex. The sutures of the skull are still visible but tightly closed, indicating that the individual is an adult.

Premaxilla

The premaxilla is well preserved at the level posterior to the antorbital notch, and its dorsal surface is almost flat. The premaxillary crest is present on the vertex. The medial margins of the left and right premaxillae are very close to each other. Contact between the nasal and the premaxillary crest is extended along almost the whole length of the nasal. The premaxillary sac fossae are weakly asymmetrical and do not overhang the maxillae laterally. The premaxillary sac fossa is almost flat and not deeply excavated. The ascending process of the premaxilla gradually rises toward the vertex. The angle between the dorsal surface of the rostrum and the ascending process of the premaxilla is 150°. It is slightly concave in lateral view. The ascending process of the left premaxilla is somewhat damaged, but that of the right premaxilla is well preserved, and its lateral margin is moderately constricted at the level along the bony nares at the midpoint of the premaxillary crest. The vertex is gently elevated. In anterior view, lateral expansion of the premaxillae at the level of the premaxillary sac fossae is prominent. The premaxillary crest is transversely oriented, and it is narrow in transverse width. The distance between the premaxillary crests is wide. The anterior margin of the bony nares is V-shaped and the overall shape of the bony nares is triangular in outline. The anteroposterior diameter (67 mm) of the bony nares is greater than the transverse diameter (39 mm). The small premaxillary foramina on the premaxilla are located roughly at the level of the antorbital notch and slightly posterior to the infraorbital foramina on the maxillae. There are distinct grooves anterior to the foramina.

Maxilla

The maxilla is broad, but its lateral margin is damaged on both sides at the level posterior to the antorbital notch. In anterior view, the rostral maxillary crest is present and moderately elevated on the maxilla. The dorsal infraorbital foramina open anterodorsally on the maxillae near the base of the rostrum, and the largest foramen is located on the right maxilla. There are two dorsal infraorbital foramina on the left maxilla with diameters of 13 mm and 7 mm. There is one large dorsal infraorbital foramen on the right maxilla, with a diameter of 16 mm. The maxilla is slightly elevated at the posterior side of these foramina. There are no excrescences on the dorsal surface of the maxilla on the preserved proximal part of the rostrum. The portion of the prominent notch and related structure are somewhat weathered away, but it is low and flat.

Nasals

The nasal is short, and its dorsal outline is trapezoidal. Although the posterior margin of the right nasal is broken, it is slightly broader than the left one. The dorsal surface of the nasal is roughly flat, and there is a weak anteromedial excavation. The nasal is not included in the premaxillary crest. The posterior nasal-frontal suture is only preserved on the left nasal and is posteriorly convex.

Frontal

The frontal is fully covered by the maxilla in the supraorbital region and only visible as a vertical section beneath the overlapping maxilla in the orbital region and temporal fossae in lateral view. It is dorsoventrally thick and relatively dense but not osteosclerotic. It also makes a dorsal roof of the antorbital and fossa in ventral view. The pre- and supraorbital area is indicated by the slight flange at the level posterior to the antorbital notch. The antorbital fossa is shallow and flares posteriorly to the slightly thickening and anterolaterally extended postorbital ridge. The posterodorsal part of the frontal around the posterior part of the vertex is broken away.

Supraoccipital

The supraoccipital is ~~almost completely broken away~~, but the anteromedial margin of the supraoccipital is distinctly lower than the dorsal margin of the vertex in posterior view (Fig. 6).

Pterygoid

The pterygoid is dorsoventrally deep and excavated by the anteroposteriorly and dorsoventrally wide hamular fossa. The lateral lamina is preserved on the right pterygoid. It is thin and low at its base. There is a similar structure like transverse crests on the surface of the medial lamina. There are distinct transverse crests on the medial lamina of the pterygoid. The hamular fossa of the pterygoid sinus is wide, extending anteriorly on the palatal surface of the rostrum. Apices of the left and right hamular processes are incomplete, but both processes contact medially and form together a posteriorly directed medial point. Also each process sends a posterolateral point, so the posterior margin of the pterygoid processes are rounded W in shape (Fig. 6).

Alisphenoid

The alisphenoid is well preserved on the right side, just posterior to the pterygoid sinus and ventrally exposed as a distinct concavity that is directed anterolaterally. A thin plate-like crest runs ~~anterolateral-posteromedial~~ direction, which is thought to be the alisphenoid-squamosal suture, and it separates this concavity posteriorly from the tympanosquamosal recess. This concavity on the alisphenoid continues posteromedially to the large foramen ovale.

Squamosal

The left squamosal is well preserved. The temporal fossa ~~on the squamosal~~ is anteroposteriorly long and dorsoventrally shallow, and its surface is slightly concave. The zygomatic process of the left squamosal is short anteroposteriorly compared to its dorsoventral height. It is well developed and anterodorsally tapered with a triangle outline, but its anterior margin is crescentic. The glenoid cavity is wide and shallow, but its medial part is deep just in front of the obliquely extended postglenoid process. It continues to the tympanosquamosal recess. The tympanosquamosal recess extends posteriorly to the fused periotic. The surface of the posterior part of the tympanosquamosal recess just in front of the periotic is rugose with numerous pits and crests. The falciform process presents at the anterior process of the periotic, and it is short and thin. The ventral margin of the postglenoid process is more dorsally located than the ventral margin of the paroccipital process of the exoccipital (Fig. 4).

Mandible

The mandibles are almost broken away, but the ascending ramus of the left mandible with the coronoid process and the mandibular condyle is partially preserved (Figure 7). The posterior margin of the angular process is also broken away. The mandibular body is very thin and fragile in the portion of the lateral wall of the mandibular foramen. Its outer or lateral surface is smooth and

slightly swelled laterally, forming the condyloid crest. Its inner or medial surface (or the lateral wall
of the mandibular foramen) is relatively rough. The mandibular foramen is anteroposteriorly long and
dorsoventrally wide. The coronoid crest is well developed as an anteroposteriorly extended ridge and
continues to the coronoid process at the posterodorsal end of the mandible. The ventral margin of the
coronoid crest is weakly concave along the crest. The mandibular condyle protrudes posteriorly and
is located almost at the same anteroposterior level as the end of the coronoid process. It is
dorsoventrally long and laterally crescentic, and it is slightly swelled posteriorly. There is a notch
between the coronoid process and the mandibular condyle.

**Periotic**

The left periotic is preserved (Table 2). It is relatively slender in shape. The dorsal surface of the
body is smooth. The anterior and posterior processes are almost straight anteroposteriorly in ventral
view. The anterior process is laterally thick, but its apex is slender and pointed (Figure 9A). It is
weakly bent ventrally. The anterior bullar facet is weakly excavated anteroposteriorly. A rounded
accessory ossicle of the tympanic bulla is still articulated with the fovea epitubaria at the
posteromedial base of the anterior process. The malleolar fossa is excavated posterior to the accessory
ossicle and opens posteriorly. It is laterally delimited by a mediolaterally elongated lateral tuberosity.
The cochlear portion (=pars cochlearis) is rounded in outline, relatively low, and it is slightly bent
anteroventrally with a weak depression at the anteromedial surface, so the anterior incisure is narrow
and slightly covered with the anterior edge of the cochlear portion. The epitympanic hiatus is
relatively deep and wide. The internal acoustic meatus containing the spiral cribriform tract and the
foramen singulare is separated by a developed transverse crest from the proximal opening of the
facial canal. The spiral cribriform tract is relatively small and almost circular, slightly compressed
transversely. Its posterolateral margin is elevated and reaches almost the top of the cochlear portion.
The foramen singulare opens in the same aperture for the facial canal (Figure 9B). The proximal
opening of the facial canal is relatively large and elliptical in shape. The aperture for the cochlear
aqueduct is large and semicircular. It is located close to the posterior margin of the internal auditory
meatus. It is also relatively close to the rounded aperture for the vestibular aqueduct (=endolymphatic
duct) that is elliptical in shape. The fenestra rotunda is relatively large and semicircular in outline.
The posterior process is relatively long posteriorly and inflated mediolaterally. It is fused to the skull
in the mediolaterally narrow periotic fossa of the skull. The posterior bullar surface is covered by the
fused fragment of the posterior process of the tympanic bulla. It is slightly bent posterolaterally at the
base just posterior to the cochlear portion. Because of this fusion of the posterior process of the
tympanic bulla and its posterior expansion, the portion of the posterior bullar facet is posteriorly
flared and fan-shape in ventral view (Figure 9A). Although mostly covered by the remnant of the
posterior process of the tympanic bulla, the surface of the posterior bullar facet is smooth.

**Tympanic Bulla**

The left tympanic bulla is preserved (Table 2). The anterior part of the ventral wall of the tympanic
bulla is cylindrical and is transversely rectilinear in outline. The involucrem is shorter anteriorly than
the lateral wall. The dorsal margin of the involucrem is cut by an indentation. The inner and outer
posterior prominences are thick and short, and they are ventrally bulging. The outer posterior
prominence is much bulged ventrally than the inner one. The median furrow runs anteroposteriorly
on the ventral wall, and it is transversely narrow and anteriorly shallow. The interprominential notch
is relatively deep at the posterior end of the median furrow. The lateral furrow is shallow, and its
posterior ridge is prominent and twisted on the lateral wall of the tympanic bulla. The sigmoid
process is square in anterior view, and it is anteroposteriorly thin. The posteroventral corner of the
sigmoid process is projected posteriorly. The lateral border of the sigmoid process is twisted
posteriorly. A damaged base of the posterior process lies on the posterolateral end of the sigmoidal
process. The elliptical foramen is present (Figure 10, 11).

6. Results of the Phylogenetic Analysis

We compared the characteristics of SCM 5530-6 with both extant and fossil Ziphiidae including 27 genera and 55 species based mainly on Bianucci et al. [3] and Lambert et al. [34]. ~~Because SCM 5530-6 described here from the middle to upper Miocene boundary Tsurushi Formation was initially reported by Takahashi et al. [14] as a species of the Ziphiidae based on the posteriorly located bony nares with gently elevated ascending process of the premaxilla, and the posteroventrally steeply sloped occipital. In addition to the above characters, SCM 5530-6 has such diagnostic characters of the family Ziphiidae at least as the presence of the premaxillary crests on the gently elevated vertex, wide and anteriorly extended hamular fossa of the pterygoid sinus [3, 34],~~ we analyzed its phylogenetic relationship based on the data matrix and parsimony settings prepared by Bianucci et al. [3] to locate SCM 5530-6 in the Ziphiidae with the addition of the North Pacific Baird's beaked whale, *Berardius bairdii* that was not included in their analysis and the recently described Sato's beaked whale, *Berardius minimus* of the northern North Pacific [2], as well as SCM 5530-6.

As a result of the analysis, five most parsimonious trees were obtained with a tree length of 177 steps in each, Goloboff fit (k=3) -58.44286, the consistency index (CI) of 0.475, and the retention index (RI) of 0.76. The 50% majority rule consensus tree is shown in Figure 12. The strict consensus tree is also shown in the Supplementary Data S4. Our result suggests that *Chavinziphius* from the late Miocene of Peru (the South Pacific) is the basal to the clade including all the extant subfamilies, i.e., the Berardiinae, Ziphiinae, and Hyperoodontinae. Instead, *Ninoziphius* also from the Miocene of Peru is recognized as the most basal to the Ziphiidae as a whole. As for SCM 5530-6, it was nested in the monophyletic subfamily Berardiinae consisting of the genera *Berardius*, *Archaeoziphius* and *Microberardius* ([3], [36] and Figure 12).

The Berardiinae is supported by five synapomorphies: slightly concave ascending process of the premaxilla (Char.7: as reversal); a weak constriction on the ascending process of the right premaxilla (between premaxillary sac fossa and premaxillary crest) and the ratio between the minimal width of the ascending process of premaxilla and the width of the right premaxillary crest is > 0.80 (Char.8: as reversal); no anteromedial excavation of the dorsal surface of the nasal (Char.14: as reversal); an isolated rounded protuberance formed by the interparietal and/or frontals on the posterior part of the vertex (Char.17), and the anteromedial margin of the supraoccipital that is lower than the frontals on the vertex (Chars. 18) (see also [3, 29, 33]). The following character is also considered to be a potentially unique character for the subfamily: the posteromedial narrowing nasals and frontals at the vertex, which is narrower than the anterior-most transverse width of nasals (Char. 50). Among those characters, quite a few synapomorphies of the subfamily are interpreted as reversals from derived to secondarily primitive character states. However, the extant ziphiids are deep-diving whales having adapted to deep sea with considerable ecomorphological specializations during their adaptation and diversification in each clade, and they might have yielded unique characters frequently to adapt such a harsh environment. Other examples are also interesting to be noted. A low elevation of the ascending process of the premaxilla is observed in *Berardius* and related fossil species such as *Archaeoziphius microglenoideus* and *Microberardius africanus* [3, 32, 36], though this condition is also seen in *Chavinziphius maxillocrestatus*, one of the basal ziphiids [3]. Another example is a cluster of dorsal infraorbital foramina, which is present on the base of the rostrum in the early diverging ziphiids [3]. On the other hand, only a large foramen is present in the crown ziphiids [3]. SCM 5530-6 has a large foramen on the right maxilla, and two relatively large foramina on the left maxilla, and therefore, the condition of these foramina is thought to be the condition seen in the later diverging, extant ziphiids. However, *Ninoziphius platyrostris* from the Miocene of Peru, which is considered to be the most basal ziphiid (Figure 12), has a relatively large dorsal infraorbital foramen [34], suggesting that this character state might have been multiply derived in two distantly related clades. This case is also suggestive that some characters seen in the berardiines are not appropriate to be considered synplesiomorphies conservatively retained in them as a subfamily having earlier-branching, deeper node.

Among the species within the Berardiinae, *Microberardius africanus* was recognized as the first split with the rest of genera and species including SCM 5530-6, but no autapomorphies were recognized to characterize *Microberardius* by our analysis. In this regard, *Microberardius* seems to be "ancestral" to the later diverging berardiines. However, Bianucci et al. [3] suggests that *Microberardius* is distinguishable from other berardiines by having the following characters: the

rostrum is higher than wide along most of its length, the rostral base is narrower, and the maxillary crest is not extended on the rostral base.

By contrast, the monophyly of the species belonging in the genera *Berardius* and *Archaeoziphius* including SCM 5530-6 is strongly supported by the following unique character combinations: the ratio between the width of the premaxillary crest and that of the premaxillary sac fossa of SCM 5530-6 is moderate, from 1.0 to 1.25 (Char. 11); an isolated rounded protuberance formed by the frontals, not the interparietal, on the posterior part of the vertex (Char. 17, 0->2).

Although, the basic framework of the relationships among the genera of the Berardiinae (i.e., [*Microberardius*, [*Archaeoziphius*, *Berardius*]]), which was originally demonstrated by Bianucci et al. [3], is also supported by our analysis, *Archaeoziphius microglenoideus* was bracketed in between the clade containing two other extant species of *Berardius* (i.e., *B. bairdii* and *B. arnuxii*) and the clade containing newly added *Berardius minimus* plus SCM 5530-6 in our analysis. However, 27 out of 51 (i.e., 53%) characters used in Bianucci et al. [3] and in our analysis are missing and uncertain in *A. microglenoideus*, and therefore, its phylogenetic position is still unstable (see also the Supplementary Data S4). Because of overall primitiveness of observable characters in *Archaeoziphius*, the position of this genus is still controversial and is potentially anticipated to be "outside" from the clade of the genus *Berardius* as a sister taxon to the latter. In this regard, the monophyly of the genus *Berardius* including SCM 5530-6 is highly considerable, and the position of SCM 5530-6 as a sister taxon to *B. minimus* makes SCM 5530-6 a species of *Berardius* in the Berardiinae. As for the specific relationship of SCM 5530-6 to other species in the genus, SCM 5530-6 is the closest to *B. minimus* as was mentioned above in having the apices of the left and right hamular process of the pterygoids that contact medially, forming together a posteriorly directed medial point (Char. 36). In addition, the lateral expansion of the premaxillae at the posterior halves in SCM 55306 is prominent as in *B. minimus*. Furthermore, the temporal fossa of SCM 5530-6 is shallow and its lateral surface is convex also as in *B. minimus*. By contrast, it is definitely concave in *B. bairdii* and *B. arnuxii*. As for skull sutures, *B. minimus* has much tighter sutures as has pointed out by Yamada et al. [2], SCM 5530-6 also has much tighter sutures compared with those in *Berardius bairdii* and *B. arnuxii*. In fact, three out of five most parsimonious trees support that SCM 5530-6 is the sister taxon of *B. minimus* (Supplementary Data S4). Finally, SCM 5530-6 is distinguishable from *B. minimus* in having the short nasal, the ratio between the length of medial suture of nasals on the vertex and the maximum width of nasals < 0.4 (Char.13, as reversal). Furthermore, the skull size, based on the bizygomatic width, of SCM 5530-6 is 50% smaller than *B. minimus* as was mentioned above. Consequently, identification of SCM 5530-6 as a distinctive species within the genus *Berardius* is warranted. Thus, we propose *Berardius kobayashii*, sp. nov. based on SCM 5530-6 from the middle to late Miocene boundary age Tsurushi Formation.

7. Discussion

In previous studies, Takahashi et al. [14] identified SCM 5530-6, now the holotype of *B. kobayashii* sp. nov., as "provisionally *Berardius* sp. indet." based on the following characteristics: the external bony nares are triangular in outline; the vertex is gently elevated; and the posterior part of the pterygoid (though it was misidentified as median part of the palate in [14, p.104]) is ventrally well developed. However, they compared SCM 5530-6 only with some extant genera of the Ziphiidae; i.e., *Ziphius*, *Berardius*, and *Mesoplodon*, and they also pointed out that the transverse width of the premaxilla of SCM 5530-6 is wider relative to the width of the maxilla at the level of the antorbital notch than that of then extant two species of *Berardius*, and that the lateral margin of the same portion is curved laterally in contrast to the relatively straight condition of this portion in both species of *Berardius*. In addition, they also pointed out that the zygomatic process of the squamosal is shaped like an anterodorsally tapered triangle in contrast to an anteriorly sub-squared rectangle in both species of *Berardius*. In this regard, both inter- and intra-generic comparisons were quite limited to their comparable taxa among the extant ziphiids, and the generality of their taxonomical differentiations of characters were somewhat arbitrary. In addition, two new genera of the Berardiinae were described as mentioned above [32, 36], and the third extant species of *Berardius* was also added

after Takahashi et al [14] as mentioned above [3]. Accordingly, the comprehensive comparisons among both extinct and extant species of *Berardius* and of the Berardiinae in the Ziphiidae were necessary to locate SCM 5530-6 more precisely in the phylogenetic framework. As a result of our cladistic analysis modified from Bianucci et al. [32], *B. kobayashii* sp. nov. was recognized as the closest to *Berardius minimus* as discussed above. It confirms that the new species based on SCM 5530-6 as the holotype actually belongs in the genus *Berardius*, and it also indicates that although *B. minimus* is represented only by the extant individuals in the northern North Pacific, the generic emergence could have been much earlier than the emergence and diversification of the other previously known two antitropical species; i.e., the northern North Pacific *B. bairdii* and the Antarctic *B. arnuxii*. Consequently, the emergence and diversification of the genus *Berardius* must have occurred in the late middle Miocene and potentially in the northern North Pacific based on the new fossil from the middle to late Miocene boundary age Tsurushi Formation.

For now, three genera are recognized in the subfamily Berardiinae; i.e., *Microberardius* from the middle Miocene of off South Africa [32], *Archaeoziphius* from the middle Miocene of Belgium [36] and *Berardius* from the middle to late Miocene boundary to the Recent of the northern North Pacific and the Antarctic oceans [1, 2, this study]. Previously, there was no fossil record of the berardiines in the North Pacific Ocean. So, the new fossil described here reveals that berardiine ziphiids already diversified to the western North Pacific at latest in the early late Miocene (ca. 11 Ma) and that the emergence of the extant genus *Berardius* goes back to the middle to late Miocene boundary age of the North Pacific, then they quickly diversified with or just after the age of their closely related *Archaeoziphius* in the early middle Miocene of the North Atlantic [36].

At the moment, it is difficult to consider the more precise origin of the subfamily Berardiinae and the genus *Berardius* based on the present fossil evidence, but its potential origin of the subfamily must have been in the Southern Hemisphere on the basis of recent studies on their phylogenetic relationships and phylogeographic distributions of sister and out groups for the berardiine ziphiids ([3, 5, 32, 33, 36] and Figure 13). In addition, the close relationship between *B. kobayashii* and *B. minimus* indicates an emergence of the genus in the North Pacific, and then the ancestor of *B. bairdii* and *B. arnuxii* may have evolved somewhere from *B. kobayashii*-like berardiine in the Northern Hemisphere or somewhere in between the northern and southern hemispheres by the end of Miocene [37]. It could have been followed by a dispersal (with gigantism) from the Southern to or from the Northern hemisphere [38]. In any case, the two currently sympatric form, smaller *B. minimus* and larger *B. bairdii* in the northern North Pacific could have been secondary contact after the Pliocene in age [38]. However, all these evidence suggest the rapid diversification of taxa in the subfamily Berardiinae in the initial stage of their evolution during the middle and late Miocene in the Northern Hemisphere (Figure 13).

It is difficult to establish the generic emergence of extant taxa based only on morphologies without genetic information such as divergence time estimations. In the case of the genus *Berardius*, its emergence time estimation depends on the divergence time of the Berardiinae and other ziphiid subfamilies because the Berardiinae is at present monogeneric. In this regard, McGowen et al. [39], for instance, estimated the divergence time of the Berardiinae from other subfamilies to be 21.98 Ma (95%CI: 17.11-29 Ma). They also estimated the differentiation of the two species of *Berardius* species (i.e., *B. arnuxii* and *B. bairdii*) to be 2.9 Ma (95%CI: 0.68-5.81 Ma). It means that the expected divergence time of the genus *Berardius* still spans 17.11-5.81 Ma in minimum estimation or 15-5.81 Ma based on the geologic age of the closest genus *Archaeoziphius* [36]. Although the divergence time of *B. minimus* from the ancestor of the two other species (i.e., *B. bairdii* and *B. arnuxii*) within the genus is still uncertain [2, 38, 40], the minimum estimate of the divergence time could be dated back to at latest the early late Miocene in age based on our phylogenetic analysis that suggested the closest relationship of *B. minimus* to the middle/late Miocene boundary age *B. kobayashii*, sp. nov.

Accordingly, it is possible to interpret that the genus *Berardius* began to diversify at this period, which is known to be the period of global cooling and of coastline regression [41, 42, 43, 44]. It should be important to be noted that somewhat speciose extant genus *Mesoplodon*, which is consisted of at least 15 extant species, has already appeared and diversified in the late middle or early late Miocene ([37, 39, 45]. It strongly suggests that the much earlier and less diversified appearance of the genus *Berardius* might have proceeded their generic emergence, and *B. kobayashii*, sp. nov. could

be interpreted as one of such representatives of the earlier generic emergence and diversification. It will be further considered based on molecular studies for *B. minimus* and also morphological studies for other fossil berardiines to be described in the future. 
8. Conclusions

~~An extinct beaked whale from the middle to late Miocene boundary age Tsurushi Formation (ca. 12.3 – 11.5 Ma) in the Sado Island, Niigata Prefecture, Japan, was described as a new species of the extant genus *Berardius*; i.e., *B. kobayashii*. It is distinguishable from all other species within *Berardius* by having the following characters: extremely small size among the four species of *Berardius* (60-70 % smaller than *B. bairdii* and *B. arnuxii*, 50% smaller than *B. minimus*); the transverse diameter of the premaxilla is relatively wide to the width of the rostrum at the level of the anterior notch; the zygomatic process of the squamosal is anterodorsally tapered and triangular in shape; the nasals are short. *Berardius kobayashii*, sp. nov., is the fourth species of the genus. Namely, the lineage leading to the extant genus of the ziphiids had already lived in the western North Pacific at the time of at latest 11 Ma. The western North Pacific including the Sea of Japan may have been one of the areas of the evolution and radiation of *Berardius* species.~~

Acknowledgments

We would like to thank the late I. Kobayashi (Professor Emeritus at Niigata University), A. Matsuoka (Niigata University), M. Shimizu (Asahimachi Museum, Niigata University), M. Ikarashi (Sado Museum), M. Aida (Sado Geopark Promotion Office and Sado Museum), K. Narita (Nagano City Museum), M. Kato (Nagaoka Municipal Science Museum), Y. Tajima and T.K. Yamada (NMNS), for allowing access to the specimens and giving me the opportunities to observe specimens at respective museums; Y. Yanagisawa (Geological Survey of Japan), K. Takahashi (Biwako Museum), H. Ichishima (Fukui Prefectural Dinosaur Museum), K. Nagasawa (formerly Yamagata Prefectural Museum), T. Kimura (Gunma Museum of Natural History), for providing useful discussion and advice with the literature on fossil ziphiids.

Funding Statement

A.K. was funded the travel grant for the 2017 fiscal year from the Regional Promotion Division of Sado City.

Data Accessibility

Additional data are available as the electronic supplementary materials (Supplementary files S1-S7: S1, list of the taxa and specimens of the berardiine ziphiids used for this study; S2, list of characters used for our cladistic analysis; S3, character/taxon matrix; S4, resultant cladograms including the unweighted strict and 50% majority-rule consensus cladograms, five equally most parsimonious cladograms with implied weighting of K=3; S5, References for character collections; S6, Nexus file; S7, TNT file). The new species has been registered in Zoobank. The LSID for this publication is: urn:lsid:zoobank.org:act:317C1452-AE2E-4A08-A473-3B507B452D2C.

Competing Interests

We declare we have no competing interests.

Authors' Contributions

A.K. and N.K. conceived and designed the project, performed the experiment and analysed the data. A.K. prepared SCM 5530-6. N.K. contributed reagents/materials/preparation tools/analysis tools. A.K. and N.K. wrote the paper.

References

1. Rosel P E, Archer FI, Baker CS, Boness DJ, Brownell RL, Churchill M Jr., Costa AP, Domning DP, Fordyce RE, Jefferson TA, Kinze C, Oliveira LR, Perrin WF, Wang JY, Yamada TK. 2019 List of marine mammal species and subspecies. *Committee on Taxonomy, The Society for Marine Mammalogy*.
2. Yamada TK, Kitamura S, Abe S, Tajima Y, Matsuda A, Mead JG, Matsuishi TF. 2019 Description of a new species of beaked whale (*Berardius*) found in the North Pacific. *Scientific reports* **9**(1), 1-14.
3. Bianucci G, Di Celma C, Urbina M, Lambert O. 2016 New beaked whales from the late Miocene of Peru and evidence for convergent evolution in stem and crown Ziphiidae (Cetacea, Odontoceti). *PeerJ*. **4**, e2479.
4. Lambert O, de Muizon C, Duhamel G, Van Der Plicht J. 2018 Neogene and Quaternary fossil remains of beaked whales (Cetacea, Odontoceti, Ziphiidae) from deep-sea deposits off Crozet and Kerguelen islands, Southern Ocean. *Geodiversitas* **40**(2), 135-140.
5. Lambert O, Collareta A, Landini W, Post K, Ramassamy B, Di Celma C, Urbina M, Bianucci G. 2015 No deep diving: evidence of predation on epipelagic fish for a stem beaked whale from the Late Miocene of Peru. *Proc. R. Soc. B*, 282.
6. Niigata Foraminiferal Research Group. 1967 Some foraminiferal assemblages from the Sawane area, Sado Island (Preliminary report): Restudy on the stratigraphy of the "Sawane Formation"

- based on the foraminiferal fossils. *Commemorative Volume for Mr. Y, Hiramatsu*, 115-119.
- Watanabe K. 1987 Tertiary Foraminifera and Radiolaria Fossils from Sado Island, Central Japan. *Publications from the Sado Museum* **9**, 127-156.
 - Okamoto Y, Satou M, Watanabe M, Yamamoto H. 1992 Inversion Tectonics in the southeastern part of the Japan Sea. *Jour. Struct. Geol. Japan* **38**, 47-58.
 - Kobayashi I. 2001 The Cenozoic of Niigata Prefecture, the case of Sado Island (Niigata no shinseikai Sado-hen). *Niigata Geotechnology Society* **57**, 9-21.
 - Yanagisawa Y, Watanabe M. 2017 Revised lithostratigraphy of the Neogene sedimentary sequence in the southern part of the Osado Mountain area, Sado Island, Niigata Prefecture, Japan. *Bull. Geol. Surv. Japan* **68 (6)**, 259-285.
 - Sado Research Group for Marine Mammalian Fossils. 1987 Fossil Cetacean from Sado Island, Central Japan. *Publications from the Sado Museum* **9**, 211-217.
 - Horikawa H, Tazaki K, Kanno T. 1987 Fossil Ziphiidae from Koshiji-Syo off Sado Island, Central Japan. *Publications from the Sado Museum* **9**, 225-230.
 - Tazaki K, Horikawa H, Miyazaki S. 1987 Fossil Ziphiidae from Hyotan-Guri off Sado Island, Central Japan. *Publications from the Sado Museum* **9**, 219-223.
 - Takahashi K, Nomura M, Kobayashi I. 1989 A fossil cetacean skull (Berardius sp. indet.) from Dogama, Ogi-machi, Sado Island, Central Japan. *Earth Science; Chikyū Kagaku* **43**, 102-105.
 - Barnes LG, Hirota K. 1995 Miocene pinnipeds of the otariid subfamily Allodesminae in the North Pacific Ocean: Systematics and relationships. *The Island Arc* **3**, 329-360.
 - Utashiro T. 1950 The stratigraphy of Sawane-Aikawa region in Southwest Sado Island (Osado Mountain range). *Journal of the Geological Society of Japan* **56**, 302-303.
 - Shimazu M, Kanai Y, Toyama T, Ichihashi K, Minakawa J, Takahama N. 1973 Structural development and igneous activity in the Sado island. *Memoirs of the Geological Society of Japan* **9**, 147-157.
 - Shimazu M, Toyama T. 1982 Neogene volcanic rocks of the Sado islands. *Journal of the Geological Society of Japan* **88**, 381-400.
 - Yanagisawa Y. 2012 Late Miocene diatoms in the Hamochi area, Sado Island, Niigata Prefecture, Japan. *Open-File Report of the Geological Survey of Japan, AIST* **568**, 1-19.
 - Ogi Collaborative Research Group. 1986 Late Cenozoic Group in the Southern part of Kosado Mountains, Niigata prefecture. *Earth Science* **40**, 417-436.
 - Takeuchi K, Ozaki M, Komatsubara T. 2011 Explanatory notes of 1:200,000 geological map of the coastal zone around Niigata. Digital Geoscience Map, Seamless Geoinformation of Coastal Zone "Coastal Zone around Niigata".
 - Tokunaga S. 1939 A New Fossil Mammal Belonging to Desmostyliidae. Jubilee Publication in the Commemoration of Professor H. Yabe, M.I.A. Sixtieth Birthday. Sendai: Institute of Geology and Paleontology, Tohoku Imperial University. 289-299.
 - Watanabe H. 1932 Neogene (Shin Daisankei). *Nihon Chishitsu kousan shi* **92-152**.
 - Maiya S. 1978 Late Cenozoic planktonic foraminiferal biostratigraphy of the oil-field region of Northeast Japan. *Cenozoic Geology of Japan*, 35-60.
 - Kawatani A, Sashida K, Sachiko A, Kohno N. 2019 Radiolarian fossils and estimated depositional age of the Miocene Tsurushi Formation distributed in Sado Island, Niigata Prefecture, Japan. *Bull. Geol. Surv. Japan* **70(1-2)**, 91-99. (doi: <https://doi.org/10.9795/bullgsj.70.91>)
 - Ono K, Ueno T. 1985 Tertiary Vertebrates from Sado Island, Niigata Prefecture, Central Japan. *Memoirs of the National Science Museum* **18**, 65-71.
 - Mead JG, Fordyce RE. 2009 The therian skull: a lexicon with emphasis on the odontocetes. *Smithsonian Contributions to Zoology*, 1-249.
 - Kasuya T. 1973 Systematic consideration of recent toothed whales based on the morphology of tympanoperiotic bone. *Scientific Reports of the Whales Research Institute* **25**, 1-103.
 - Maddison WP and Maddison DR. 2018. Mesquite: a modular system for evolutionary analysis. Version 3.61. <http://www.mesquiteproject.org>.
 - Goloboff PA, Catalano SA. 2016. TNT version 1.5, including a full implementation of phylogenetic morphometrics. *Cladistics* **32(3)**, 221-238.
 - Goloboff PA. 1993. Estimating character weights during tree search. *Cladistics* **9**, 83-91.
 - Bianucci G, Lambert O, Post K. 2007 A high diversity in fossil beaked whales (Mammalia, Odontoceti, Ziphiidae) recovered by trawling from the sea floor off South Africa. *Geodiversitas* **29(4)**, 561-618.
 - Bianucci G, Lambert O, Post K. 2010 High concentration of long-snouted beaked whales (genus *Messapicetus*) from the Miocene of Peru. *Palaeontology* **53(5)**, 1077-1098.
 - Lambert O, De Muizon C, Bianucci G. 2013 The most basal beaked whale *Ninziphius platyrostris* Muizon, 1983: clues on the evolutionary history of the family Ziphiidae (Cetacea: Odontoceti). *Zoological Journal of the Linnean Society* **167(4)**, 569-598.
 - Miwa M, Yanagisawa Y, Yamada K, Irizuki T, Shoji M, Tanaka Y. 2004 Planktonic foraminiferal biostratigraphy of the Pliocene Kuwae Formation in the Tainai River section, Niigata Prefecture and the age of the base of the No. 3 Globorotalia inflata bed. *Journal of the Japanese Association for Petroleum Technology* **69(3)**, 272-283.
 - Lambert O, Louwye S. 2006 *Archaeoziphius microglenoideus*, a new primitive beaked whale (Mammalia, Cetacea, Odontoceti) from the Middle Miocene of Belgium. *Journal of Vertebrate Paleontology* **26(1)**, 182-191.
 - McGowen MR, Tsagkogeorga G, Álvarez-Carretero S, Dos Reis M, Struebig M, Deaville R, Jepson PD, Jarman S, Polanowski A, Morin PA, Rossiter SJ. 2020 Phylogenomic Resolution of the Cetacean Tree of Life Using Target Sequence Capture. *Systematic Biology* **69**, 479-501. DOI: 10.1093/sysbio/syz068.
 - Morin PA, Baker CS, Brewer RS, Burdin AM, Dalebout ML, Fedutin ID, Filatova OA, Jung JL, Lauf M, Potter CW, Richard G, Ridgway M, Robertson KM, Wade P. 2017 Genetic structure of the beaked whale genus *Berardius* in the North Pacific, with genetic evidence for a new species. *Marine Mammal Science* **33**, 96-111.
 - McGowen MR, Spaulding M, Gatesy J. 2009 Divergence date estimation and a comprehensive molecular tree of extant cetaceans. *Molecular Phylogenetics and Evolution* **53**, 891-906. DOI: 10.1016/j.ympev.2009.08.018.
 - Kitamura S, Matsuishi T, Yamada TK, Tajima Y, Ishikawa H, Tanabe S, Nakagawa H, Uni Y, Abe S. 2013 Two genetically distinct stocks in Baird's beaked whale (Cetacea: Ziphiidae). *Marine Mammal Science* **29**, 755-766.
 - Miller KG, Kominz MA, Browning JV, Wright JD, Mountain GS, Katz ME, Sugarman PJ, Cramer BS, Christie-Blick N, Pekar SF. 2005 The Phanerozoic record of global sea-level change. *Science* **310**, 1293-1298. <https://doi.org/10.1126/science.1116412> PMID: 16311326.
 - Zachos JC, Dickens GR, Zeebe RE. 2008 An early Cenozoic perspective on greenhouse warming and carbon-cycle dynamics. *Nature* **451**, 279-283. <https://doi.org/10.1038/nature06588> PMID: 18202643.
 - Hilgen FJ, Abels HA, Iaccarino S, Krijgsman W, Raffi I, Sprovieri R, Turco E, Zachariasse WJ. 2009 The global stratotype section and point (GSSP) of the Serravallian Stage (Middle Miocene). *Episodes* **32**, 152-166.
 - Betzler C, Eberli GP, Lüdmann T, Reolid J, Kroon D, Reijmer JGG, Swart PK, Wright J, Young JR, Alvarez-Zarikian C, Alonso-García M, Bialik OM, Blättler CL, Guo JA, Hafften S, Horozal S, Inoue M, Jovane L, Lanci L, Laya JC, Hui Mee AL, Nakakuni M, Nath BN, Niino K, Petruny LM, Pratiwi SD, Slagle AL, Sloss C R, Su X, Yao Z. 2018 Refinement of Miocene sea level and monsoon events from the sedimentary archive of the Maldives

- (Indian Ocean). *Progress in Earth and Planetary Science* **5**, 1-18. DOI 10.1186/s40645-018-0165-x.
45. Steeman ME, Hebsgaard MB, Fordyce RE, Ho SYW, Rabosky DL, Nielsen R, Rahbek C, Glenner H, Rensen MVS, Willerslev E. 2009 Radiation of extant cetaceans driven by restructuring of the oceans. *Systematic Biology* **58**, 573-585. DOI:10.1093/sysbio/syp060.
46. Akiba F. 1986 Middle Miocene to Quaternary diatom biostratigraphy in the Nankai Trough and Japan Trench, and modified Lower Miocene through Quaternary diatom zones for middle-to-high latitudes of the North Pacific. In Kagami, H., Karig, D. E., Coulbourn, W. T., et al., *Initial Reports of the Deep Sea Drilling Project, U. S. Government Printing Office, Washington D. C.* **87**, 393-480.
47. Yanagisawa Y, Akiba F. 1998 Refined Neogene diatom biostratigraphy for the northwest Pacific around Japan, with an introduction of code numbers for selected diatom biohorizons. *Jour. Geol. Soc. Japan* **104**, 395-414.

Tables

Measurements (in mm) of the skull of *Berardius kobayashii*

Skull

Preserved length from tip of broken rostrum to hindmost margin of occipital.	236
Greatest postorbital width.	*e219
Greatest zygomatic width	*e245
Greatest width of bony nares.	38
Maximum width of premaxillary sac fossae.	87
Maximum width of right premaxillary sac fossa.	43
Maximum width of left premaxillary sac fossa.	38
Maximum width of nasals.	40
Greatest length of left temporal fossa, measured to external margin of raised suture.	87
Greatest width of left temporal fossa at right angles to greatest length.	61
Major diameter of left temporal fossa proper.	38
Minor diameter of left temporal fossa proper.	33
Distance from foremost end of junction between nasals to hindmost point of margin of supraoccipital crest.	33
Length of left orbit from apex of preorbital process of frontal to apex of postorbital process.	51
Greatest width of internal nares.	91
Greatest length of left pterygoid.	109

Measurements (in mm) of the ear bones of *Berardius kobayashii*

Tympanic bulla

Standard length of tympanic bulla, distance from anterior tip to posterior end of outer posterior prominence.	37
Distance from anterior tip to posterior end of inner posterior prominence.	34
Distance postero-ventral tip of outer posterior prominence to tip of sigmoid process.	24
Distance postero-ventral tip of outer posterior prominence to tip of conical process.	17.5
Width of tympanic bulla at the level of sigmoid process.	20
Height of tympanic bulla, from tip of sigmoid process to ventral keel.	23
Width across inner and outer posterior prominence.	18
Greatest depth of interprominential notch.	5
Width of upper border of sigmoid process.	11

Periotic

Standard (maximum anteroposterior) length, from tip of anterior process to posterior end	39.7
of posterior process, measured on a straight line parallel to cerebral border
Standard (maximum mediolateral) width, from medial edge of cochlear portion to apex of	23
lateral tuberosity
Maximum dorsoventral depth of anterior process perpendicular to axis of periotic	13.6
Maximum mediolateral width of anterior process at base	17
Length of anterior process from anterior apex to posterior border of malleolar fossa	16.2
Length of anterior process from anterior apex to anterior incisure	14.3
Anteroposterior length of cochlear portion	17.3
Dorsoventral depth of cochlear portion	18.7
Mediolateral width of cochlear portion from medial edge to fenestra ovalis	10.8
Maximum diameter of fenestra rotunda	2.7
Maximum diameter of internal acoustic meatus	4.7
Maximum diameter of cochlear aqueduct	1.9
Maximum diameter of aperture for the vestibular aqueduct	3.6
Anteroposterior diameter of proximal opening of facial canal	3.4
Anteroposterior diameter of distal opening of facial canal	1.9
Length of posterior process (posterior bullar facet between anteroposterior tips)	17.3
Width of posterior process (posterior bullar facet between mediolateral tips)	12.1

Figure and table captions

Figure 1. Index maps of the Pacific Northwest: A, locations of the Japanese Islands, showing Sado Island; B, map of Sado Island, showing some major place names mentioned in text, and location of enlarged area including the type locality for *Berardius kobayashii*, sp. nov.; C, detail of Sobama beach in Donokama, Ogi Town, showing the exact locality for the holotype of *Berardius kobayashii*, SCM 5530-6. Based on the GIS map (<https://maps.gsi.go.jp/#15/37.844139/138.272681/&base=std&ls=std&disp=1&vs=c0j0h0k0l0u0t0z0r0s0m0f1>) of the Geospatial Information Authority of Japan.

Figure 2. Correlation of formations distributed on the Osado (a) and Kosado (b) mountain ranges containing whale fossil horizon with relevant systems of chronology. Wave lines indicate unconformities. After [6, 19, 10]. Diatom zones: [10, 46, 47]; Planktonic foraminiferal zones: [24, 35].

Figure 3. Dorsal view of the cranium of *Berardius kobayashii*, sp. nov., holotype, SCM 5530-6. (Scale bar = 10 cm)

Figure 4. Lateral view of the cranium of *Berardius kobayashii*, sp. nov., holotype, SCM 5530-6. A and B, left side; C and D, right side. (Scale bar = 10 cm)

Figure 5. Anterior view of the cranium of *Berardius kobayashii*, sp. nov., holotype, SCM 5530-6. (Scale bar = 10 cm)

Figure 6. Posterior view of the cranium of *Berardius kobayashii*, sp. nov., holotype, SCM 5530-6. (Scale bar = 10 cm)

Figure 7. Ventral view of the cranium of *Berardius kobayashii*, sp. nov., holotype, SCM 5530-6. (Scale bar = 10 cm)

Figure 8. Posterior fragment of the left mandible of *Berardius kobayashii*, sp. nov., holotype, SCM 5530-6. A, lateral view; B, medial view. (Scale bar = 10 cm)

Figure 9. Left periotic of *Berardius kobayashii*, sp. nov., holotype, SCM 5530-6. A, ventral view; B, corresponding line drawing with anatomical interpretations (gray hatches indicate parts of the tympanic bulla); C-F, the reconstructed CT image from 3D surface rendering, with anatomical interpretations (C, ventral view; D, lateral view; E, medial view; F, dorsal view). (Scale bar = 1 cm)

Figure 10. The left tympanic bulla A, dorsal view; B, anterior view; C, medial view; D, lateral view; E, posterior view; F, ventral view (Scale bar = 1 cm)

Figure 11. Key features of the left tympanic bulla of *Berardius kobayashii*, sp. nov., holotype, SCM 5530-6. A, dorsal view; B, anterior view; C, medial view; D, lateral view; E, posterior view; F, ventral view. (Scale bar = 1 cm)

Figure 12. Phylogenetic relationship and paleobiogeographic distribution of *Berardius kobayashii*, sp. nov., within the Ziphiidae based on Bianucci et al. [3] and our re-analysis. The strict consensus tree resulting from five most parsimonious trees, 177 steps long, with the consistency index = 0.475 and the retention index = 0.767.

Figure 13. Distribution and potential roots for the dispersal of the known extinct species of the Berardiinae in the middle Miocene inferred from the phylogenetic analysis.

Table 1. Measurements (in mm) of the skull of *Berardius kobayashii*, sp. nov., holotype, SCM 5530-6. '*' indicates estimated transverse measurements that are half-skull measurements multiplied by two, and 'e' indicates estimation.

Table 2. Measurements (in mm) of the ear bones of *Berardius kobayashii*, sp. nov., holotype, SCM 5530-6.

R. Soc. open sci. article template

ROYAL SOCIETY
OPEN SCIENCE

R. Soc. open sci.
doi:10.1098/not yet assigned

The oldest fossil record of the extant genus *Berardius* (Odontoceti, Ziphiidae) from the middle to late Miocene boundary of the western North Pacific

Ayako Kawatani¹ and Naoki Kohno^{1,2}

¹Graduate School of Life and Environmental Sciences, University of Tsukuba, 1-1-1, Tennoudai, Tsukuba, Ibaraki, 305-8577, Japan

²National Museum of Nature and Science, 4-1-1, Amakubo, Tsukuba, Ibaraki, 305-0005, Japan

Keywords: middle Miocene, extant genus, *Berardius kobayashii*, sp. nov., western North Pacific.

1. Summary

A new species of a beaked whale that belongs to the extant genus *Berardius* is described from the late-middle to late Miocene boundary age Tsurushi Formation (ca. 12.3-11.5 Ma) on the Sado Island, Niigata Prefecture, Japan. The new species, *Berardius kobayashii*, sp. nov., represents the oldest record of this genus and suggests the timing of the emergence of this extant genus during this age period also of the extant ziphiid genera in the world. *Berardius kobayashii*, sp. nov., has the following generic characters of *Berardius*: the ratio between the width of the premaxillary crests and the width of the premaxillary sac fossae is 1.0 to 1.25, nodular frontals make isolated protuberance on the posterior part of the gently elevated vertex of the skull, and anterior process of the periotic elongated and transversely slender such as the gently elevated vertex, the long and slender anterior process of the periotic. Among the species within the genus, *B. kobayashii*, sp. nov., shares a unique character with *B. minimus* such as the apices of the left and right hamular processes of the pterygoids contact medially, forming together a posteriorly directed medial point. In addition, *B. erardius kobayashii*, sp. nov., displays also the a unique combination of both primitive and derived characters also has following unique specific characters: it is extremely small in size, and the nasals are short, ratio between the length of the medial suture of nasals on vertex and the maximum width of nasals is <0.4 it is extremely small in size, apices of the left and right hamular processes of the pterygoids contact medially, forming together a posteriorly directed medial point. *Berardius kobayashii*, sp. nov. fills the gap between the origin of the genus and later diversifications of the extant species in the genus. Furthermore, it suggests that *Berardius* species had originated in the North Pacific, including the Sea of Japan, at the time at latest 12 Ma. This discovery is will also be a key to elucidate the process of their emergence and habitat expansion during the latest middle and earliest late Miocene in age as the cosmopolitan genus among the Ziphiidae. Based on the distributional patterns of the fossil and extant species of the genus, suggest that the western North Pacific including the Sea of Japan may have been one of the areas for the evolution and radiation of this extant genus among the crown Ziphiidae at the time not later than 11 Ma.

2. Introduction

The family Ziphiidae is a of the cetartiodactyl clade among the Odontoceti (Cetartiodactyla, Cetacea) that is 
[revised manuscript text omitted]
 ~~that produced whale fossils~~ based on ~~the~~ radiolarians ~~fossils~~, ~~which corresponds to yielding a~~ 15.3–9.1 Ma ~~interval~~. Based on ~~these~~ these chronological information, the geologic age of the Tsurushi Formation is estimated to be approximately 12.3–11.5 Ma. Studies of foraminiferal fossils have suggested that the Tsurushi Formation was deposited in the deep-sea environment [7, 9].

As mentioned above, various megafossils have been reported from the Tsurushi Formation, ~~;~~ for example, mollusks, fish, a sea turtle, birds, and marine mammals [9, 11, 26]. In particular, a lot of marine mammal fossils ~~such as~~ ~~including~~ whales, pinnipeds and desmostylians have been found in calcareous nodules from the Tsurushi Formation on the Kosado mountain range, and these fossils were briefly reported and summarized by ~~the~~ Sado Research Group for Marine Mammalian Fossils [11], Tazaki et al. [13], Horikawa et al. [12], Takahashi et al. [14] and Barnes and Hirota [15].

4. Materials and Methods

The specimen described here was initially reported by Takahashi et al. [14]. At that time the specimen was temporarily deposited at Niigata University, ~~;~~ and now it is permanently stored at the Sado Museum, Sado City, Niigata Prefecture, Japan, where it bears the catalog number SCM 5530-6. This specimen ~~was had been~~ enclosed in calcareous hard ~~mudstone matrix~~ ~~mud stone matrices~~. Accordingly, it was prepared by acid etching, ~~technie~~ using formic acid diluted ~~to a~~ with concentration of about 4.5–5.0% ~~in water~~, which allowed the extraction of fragile bone elements of the skull, partial mandible and ear bones from ~~the matrix~~ ~~hard matrices~~. However, the periotic ~~is has~~ been fused to the skull ~~loosely~~ by its fan-shaped posterior process, and the dorsal surface of the periotic is ~~entirely~~ invisible ~~entirely~~ from the ventral aspect. ~~So~~ we reconstructed the 3D image of the periotic ~~using by~~ the micro-Computed Tomographic (μ CT) scanner ~~of the~~ Microfocus CT, TXS320-ACTIS, at the National Museum of Nature and Science, Tokyo, Japan.

We then compared ~~the specimen~~ them osteologically and taxonomically with both ~~extinct fossil~~ and extant species of ziphiids stored at the ~~following~~ institutions ~~listed below~~ and ~~from~~ the literature ~~cited~~. Definitions of measurement ~~ment~~ ~~portions~~ and morphological terms follow Mead and Fordyce [27]

R. Soc. open sci.

and Kasuya [28]. Then we performed a cladistic analysis to clarify the phylogenetic relationship of the new fossil described here to the previously named ziphiids using the characters and data matrix prepared by Bianucci et al. [3], and it was slightly modified with the addition of recently described *Berardius minimus* ([2]) and the new species described here. We also added *Berardius bairdii* based on our own observation into the data matrix, because it was also not included in the original analysis by Bianucci et al. [3]. Character and character states were managed using Mesquite 3.61 ([29]). The phylogenetic analysis based on 31 ziphiids of both extinct and extant taxa with six outgroup taxa as OTUs and 51 morphological characters was performed with TNT 1.5 using the “New Technology Search” tasked to find minimum length trees 1,000 times ([30]). As was mentioned by Bianucci et al. [3], our preliminary analyses also led to less-resolved trees. So we also downweighted as the same with Bianucci et al. [3] as the homoplastic characters using the default value of 3 for the constant *K* of the method formulated by Goloboff [31]. Multi-state characters were treated as ordered for 18 characters (i.e., chars. 1, 3, 7, 8, 9, 11, 13, 14, 15, 16, 22, 23, 30, 34, 36, 39, 47, and 51) and unordered for six characters (i.e., chars. 2, 4, 10, 28, 34, and 45), and other binary characters were all treated as unordered, also following Bianucci et al. [3]. The characters for comparisons and the phylogenetic analysis follow Bianucci et al. [3].

Institutional abbreviations are the following: IRSNB, Institut royal des Sciences naturelles de Belgique, Brussels, Belgium; MNHN: Muséum National d’Histoire Naturelle, Paris, France; NMNS-PV, Department of Geology and Paleontology, National Museum of Nature and Science, Tsukuba, Japan; NSMT-M, Department of Zoology, National Museum of Nature and Science, Tsukuba, Japan; SAM: Iziko South African Museum, Cape Town, South Africa; SCM, Sado Museum, Sado City, Niigata, Japan; TMNH, Togakushi Museum of Natural History, Nagano, Japan.

5. Systematics

Order CETARTIODACTYLA Montgelard, Catzefis and Douzery, 1997

Infraorder CETACEA Brisson, 1762

Parvorder ODONTOCETI Flower, 1867

Family ZIPHIIDAE Gray, 1850

Subfamily BERARDIINAE Moore, 1968

Emended diagnosis of subfamily: The subfamily differs from all other ziphiids with the following unique combination of characters: the ascending process of the premaxilla in lateral view is slightly concave (Char. 7, reversal); constriction on the ascending process of the right premaxilla (between premaxillary sac fossa and the premaxillary crest) is weak, and the ratio between the minimal width of ascending process of premaxilla and the width of right premaxillary crest is >0.80 (Char. 8, reversal); no anteromedial excavation of the dorsal surface of the nasal (Char. 14, reversal); the nodular protuberance on the posterior part of the vertex is formed by the interparietal and/or the frontals (Char. 17, 0->1 & 2); and the anteromedial margin of the supraoccipital is distinctly lower than the dorsal margin of the vertex (Char. 18) mandible; the narrow and thin premaxillary crest; the supraoccipital is lower than the frontals on the vertex (modified from [29]).

Genus *Berardius* Duvernoy, 1851

Emended diagnosis of genus: The ratio between the rostral length and the condylobasal length is between 0.63 and 0.70; the mesorostral groove is empty; The genus differs from *Microberardius* with the following unique combination of characters: the ratio between the width of the premaxillary crests and the width of the premaxillary sac fossae is 1.0 to 1.25 (Char. 11); the nodular protuberance on the posterior part of the vertex is formed only by the frontals (Char. 17). Following characters shared with *Berardius* species, but uncertain in *Microberardius* and *Archaeoziphius* because of lack of such portion, are potentially diagnostic for the genus: the anterior spine of the tympanic with a more or less rectilinear anterior margin (Char. 22); the posteroventral corner of the sigmoid process of the tympanic is posteriorly projected in lateral view (Char. 23); the dorsal margin of the involucrum of the tympanic that is cut by an indentation that is present and visible in medial and/or dorsal view (Char. 24) the mesorostral groove opens for the proximal third of the rostrum; the

Commented [OP1]: the proximal part of the mesorostral groove is actually filled by bone in *Berardius*. this bone has been identified as the mesethmoid, but could possibly be re-identified as the presphenoid. see the following works:

Lambert, O., Buffrénil, V., de, and Muizon, C., de. 2011. Rostral densification in beaked whales: diverse processes for a similar pattern. *Comptes Rendus Palevol* 10: 453-468.

Ichishima, H. 2016. The ethmoid and presphenoid of cetaceans. *Journal of Morphology* 277: 1661-1674.

this is a condition that differs from most, but not all other ziphiids.

R. Soc. open sci. article template

5

preaural basin is absent; asymmetry of the premaxillary sac fossae absent or weak; the premaxillary sac fossa does not laterally overhang on the maxilla; deep excavation of the premaxillary sac fossa is absent; the ascending process of the premaxilla in lateral view is slightly concave; constriction on the ascending process of the right premaxilla is roughly absent; the vertex is gently elevated; the premaxillary crest is transversely directed; the distance between the left and right premaxillary crests is large; the nasal elongation ratio between the length of the medial suture of the nasals and the maximum width of the nasals is high; the anteromedial excavation of the dorsal surface of the nasal is absent; the premaxillary crest present on the vertex; inclusion of the nasal in the premaxillary crest is absent; contact between the nasal and the premaxillary crest is extended on more than half the length of the nasal to almost the whole length; the premaxillary foramen is located roughly at the level of the antorbital notch; there are no excrescences on the dorsal surface of the maxilla at the posterior half of the rostrum; the posterolateral narrowing of the nasals and frontals is present at the vertex that is narrower than the nasals; the anteromedial margin of the supraoccipital is distinctly lower than the dorsal margin of the vertex; the extremely ossified trapezoidal vertex is absent; the angle formed by the basioccipital crests is larger than 50° in ventral view; the hamular fossa of the pterygoid sinus is wide, extending anteriorly on the palatal surface of the rostrum; the apices of the left and right hamular processes of the pterygoids contact medially; the apex of the hamular process is excavated by the fossa for the hamular lobe of the pterygoid sinus; the anteroventral wall of the tympanic bulla is narrow and triangular; the periotic has long and slender anterior and posterior processes; the posterior bullar facet of the periotic is fan-shaped (modified from [3], [29], [30], [31]).

Berardius kobayashii, sp. nov.

(Skull: Figure 3-87; Ear bone: Figure 98-1140)

Berardius sp. indet. Takahashi et al. 1989

URN: lsid:zoobank.org:act:317C1452-AE2E-4A08-A473-3B507B452D2C

Diagnosis of species: The species differs from all other *Berardius* species (i.e., *B. arnuxii*, *B. bairdii* and *B. minimus*) with the following unique characters: the nasals are short, ratio between the length of medial suture of nasals on the vertex and the maximum width of nasals <0.4 (Char. 13, reversal); Although it is potentially a shared primitive trait with *Archaeoziphius microglenoides*, an extremely small in-size among the four species of *Berardius* (60-70 % smaller than *B. bairdii* and *B. arnuxii*, 50% smaller than *B. minimus*) is potentially the specific trait within the genus; the transverse diameter of the premaxilla is relatively wide; the lateral margin of the premaxilla is curved; there is a tiny anteromedial excavation on the dorsal surface of the nasals; the apices of the left and right hamular processes of the pterygoids contact medially, forming together a posteriorly directed medial point.

Holotype: SCM 5530-6, incomplete skull with the left tympanic bulla and periotic, lacking the rostral portion; the posterior one-fourth of the horizontal ramus of the left dentary with the condyloid coronoid process and mandibular condyle. Collected by Kiyoei Kaneko in or before 1969.

Type Locality: Sobama Beach, Ogi Town, Sado Island, Niigata Prefecture, Japan. Geographic coordinates: 37° 84'50" N, 138° 26'58" E.

Formation and age of holotype: SCM 5530-6 was collected from the latest middle to earliest late Miocene age Tsurushi Formation. As discussed above, the geologic age of the Tsurushi Formation is estimated to be approximately 12.3–11.5 Ma.

Etymology: The species is named in honor of the late Dr. Iwao Kobayashi, a professor emeritus at Niigata University, for his longstanding contributions to the geology and paleontology of the northern Fossa Magna region including Niigata Prefecture, and in gratitude for his permission to re-study the marine mammal fossils collected by him and his colleagues from the Sado Island and encouragement to both of us throughout this study.

Description

R. Soc. open sci.

Cranium (general morphology)

The cranium is relatively small (Table 1) and ~~slightly less~~ asymmetry (Figure 3,5). The ratio between the height and width of the skull is 1:1.19. The rostral and the occipital parts are almost ~~completely~~ broken and missing ~~away~~. The small part of the left mandible ~~was originally had~~ attached to the cranium. The vertex is moderately elevated and slightly skewed toward the left. The ~~posterior part of the~~ vertex is almost broken. The ~~outline of the~~ temporal fossa ~~in lateral view~~ is an isosceles triangle ~~with long sides in shape~~ that converges anteriorly, and the posterior margin is weakly rounded. The temporal fossa is shallow and ~~its lateral surface is convex on a~~ ~~ve~~. The sutures of the skull are still visible but tightly closed, indicating that the individual is an adult.

Premaxilla

The premaxilla is ~~well preserved at the level posterior to the antorbital notch, and its dorsal surface~~ ~~is~~ almost flat. The premaxillary crest is present on the vertex. The medial margins ~~of between~~ the left and right premaxillae ~~are~~ very close ~~to each other~~. Contact between the nasal and the premaxillary crest is extended along ~~almost~~ the ~~almost~~ whole length of the nasal. The premaxillary sac fossae are weakly asymmetrical and do not overhang the maxillae laterally. The premaxillary sac fossa is almost flat and not deeply excavated. The ascending process of the premaxilla gradually rises toward the vertex. ~~The angle between the dorsal surface of the rostrum and the ascending process of the premaxilla is 150°.~~ It is slightly concave in lateral view. The ascending process of the left premaxilla is somewhat damaged, but that of the right premaxilla is well preserved, and ~~its lateral margin is~~ moderately constricted ~~at the level along the bony nares just~~ at the midpoint of the premaxillary crest. The vertex is gently elevated. In anterior view, lateral expansion of the premaxillae at the ~~portion~~ ~~level~~ of the premaxillary sac fossae is prominent. The premaxillary crest is transversely oriented, and it is narrow in ~~transverse~~ width. The distance between the premaxillary crests is wide. The anterior margin of the bony nares is V-shaped ~~and t-he overall shape of the bony nares is triangular in outline.~~ ~~The anteroposterior diameter (67 mm) of the bony nares is greater than the transverse diameter (39 mm).~~ The small premaxillary foramina ~~on the premaxilla~~ are located roughly at the level of the antorbital notch and slightly posterior to the infraorbital foramina on the ~~premaxillae~~. There are distinct grooves anterior to the foramina. ~~The angle between the dorsal surface of the rostrum and the ascending process of the premaxilla is 150°.~~

Maxilla

The maxilla is broad, ~~but~~ and its lateral margin is ~~damaged~~ ~~curved on both sides at the level posterior to the antorbital notch.~~ ~~In anterior view,~~ the rostral maxillary crest is present and moderately elevated on the maxilla. The ~~dorsal~~ infraorbital foramina open anterodorsally on the maxillae near the base of the rostrum, and the largest foramen is located on the right maxilla. There are two ~~maxillary dorsal infraorbital~~ foramina on the left maxilla ~~with diameters of and are~~ 13 mm and 7 mm ~~in diameter, respectively.~~ There is one large ~~maxillary dorsal infraorbital~~ foramen on the right maxilla, ~~with and has~~ a diameter ~~of reaching~~ 16 mm. The maxilla is slightly elevated at the posterior side of these foramina. ~~There are no~~ ~~The~~ excrescences ~~on the dorsal surface of the maxilla is absent~~ on the ~~preserved proximal part~~ ~~half~~ of the rostrum. The ~~portion of the~~ prominent notch and related structure are ~~somewhat weathered away, but it is low and flat~~ ~~low~~.

Nasals

The nasal is short, and its dorsal outline is trapezoidal. ~~The shape of the bony nares is triangular in outline, and the anteroposterior diameter (67 mm) of the bony nares is greater~~ ~~wider than the transverse diameter (39 mm).~~ Although the posterior margin of the right nasal is broken, it is slightly broader than the left one. The dorsal surface of the nasal is roughly flat, and there is a weak anteromedial excavation. The nasal is not included in the premaxillary crest. ~~The nasal contacts the premaxilla laterally for almost the whole length.~~ The posterior nasal-frontal suture is ~~only preserved on the left nasal and is~~ posteriorly convex.

Frontal

R. Soc. open sci.

R. Soc. open sci. article template

7

The frontal is fully covered by the maxilla in the supraorbital region dorsal view and only visible as its vertical section beneath the overlapping maxilla in at the portion of the orbital at region and temporal fossae in lateral view. It is dorsoventrally thick and relatively dense but not osteosclerotic nor pachyosteosclerotic pachyosteosclerotic. It also makes a dorsal roof of the antorbital and fossa in ventral view. The pre- and supraorbital area is indicated by the slight flange at the level posterior to the antorbital notch. The antorbital fossa is shallow and flares posteriorly to the slightly thickening and anterolaterally extended postorbital ridge. The posterodorsal part of the frontal around the posterior part of the vertex is broken away.

Supraoccipital

The supraoccipital is almost completely broken away, but t—The anteromedial margin of the supraoccipital is distinctly lower than the dorsal margin of the vertex in posterior view (Fig. 6).

Pterygoid

The pterygoid is dorsoventrally deep and excavated by the anteroposteriorly and dorsoventrally wide hamular fossa. The lateral lamina is preserved on the right pterygoid. It is thin and low at its base. There is a similar structure like transverse crests on the surface of the medial lamina. There are distinct transverse crests on the medial lamina of the pterygoid. The hamular fossa of the pterygoid sinus is wide, extending anteriorly on the palatal surface of the rostrum. Apices of the left and right hamular processes are incomplete, but both processes contact medially and form together a posteriorly directed medial point. Also each process sends a posterolateral point, so the posterior margin of the pterygoid processes are rounded W in shape (Fig. 6).

Alisphenoid

The alisphenoid is well preserved on the right side, just posterior to the pterygoid sinus and ventrally exposed as a distinct concavity that is directed anterolaterally. A thin plate-like crest runs anterolateral-posteromedial direction, which is thought to be the alisphenoid-squamosal suture, and it separates this concavity posteriorly from the tympanosquamosal recess. This concavity on the alisphenoid continues posteromedially to the large foramen ovale. The supraoccipital protuberance protrudes posteriorly.

Squamosal

The left squamosal is well preserved. The temporal fossa on the squamosal is anteroposteriorly long and dorsoventrally shallow, and its surface is slightly concave. The zygomatic process of the left squamosal is preserved. It is short anteroposteriorly compared to its dorsoventral height. It~~The zygomatic process of the squamosal~~ is well developed and anterodorsally tapered with a triangle outline, but its anterior margin is crescentic in shape. The glenoid cavity is wide and shallow, but its medial part is deep just in front of the obliquely extended postglenoid process. It continues to the tympanosquamosal recess. The tympanosquamosal recess extends posteriorly to the fused periotic. The surface of the posterior part of the tympanosquamosal recess just in front of the periotic is rugose with numerous pits and crests. The falciform process presents at the anterior process of the periotic, and it is short and thin. The ventral margin of the postglenoid process is more dorsally located than the ventral margin of the paroccipital process of the exoccipital (Fig. 4).

Mandible

The mandibles are almost broken away, but the ascending ramus of the left mandible with the coronoid process and the mandibular condyle is partially preserved (Figure 7). The posterior margin of the angular process is also broken away. The mandibular body is very thin and fragile in at the portion of the lateral wall of the mandibular foramen. Its outer or lateral surface is smooth and slightly swelled laterally, forming the condyloid crest. Its inner or medial surface (or the lateral wall of the mandibular foramen) is relatively rough. The mandibular foramen is anteroposteriorly long and dorsoventrally wide. The coronoid crest is well developed as an anteroposteriorly extended ridge and continues to the coronoid process at the posterodorsal end of the mandible. The ventral margin of the

R. Soc. open sci.

coronoid crest is weakly concave along the crest. The mandibular condyle protrudes posteriorly and is located ~~locates~~ almost at the same ~~anteroposterior level as position at~~ the end of the coronoid process. It is dorsoventrally long and laterally crescentic, and it is slightly swelled posteriorly. There is a notch between the coronoid process and the mandibular condyle ~~but is absent between the mandibular condyle and the angular process, which is almost broken though.~~

Periotic

The left periotic is preserved (Table 2). It is relatively slender in shape. The dorsal surface of the body is smooth. The anterior and posterior processes are almost straight anteroposteriorly in ventral view. The anterior process is laterally thick, but its apex is slender and pointed (Figure 9A). It is weakly bent ventrally. The anterior bullar facet is weakly excavated anteroposteriorly. A rounded accessory ossicle of the tympanic bulla is still articulated with the fovea epitubaria at the posteromedial base of the anterior process. The malleolar fossa is excavated posterior to the accessory ossicle (~~in the fovea epitubaria~~) and opens posteriorly. It is laterally delimited by a mediolaterally elongated lateral tuberosity. The cochlear portion (=pars cochlearis) is rounded in outline, relatively low, and it is slightly bent anteroventrally with a weak depression at the anteromedial surface, so the anterior incisure is narrow and slightly covered with the ~~anterior posterior~~ edge of the cochlear portion. The epitympanic hiatus is relatively deep and wide. The internal acoustic meatus containing the ~~central foramen~~ (=spiral cribriform tract and the foramen singulare) is separated by a developed transverse crest from the proximal opening of the facial canal. The ~~spiral cribriform tract~~ ~~central foramen~~ is relatively small and almost circular, slightly compressed transversely. Its posterolateral margin is elevated and reaches almost the top of the cochlear portion. The ~~foramen foreman~~ singulare opens ~~in the same~~ near the aperture for the ~~facial canal~~ vestibular aqueduct (Figure 9B). The proximal opening of the facial canal is relatively large and elliptical in shape. The aperture for the cochlear aqueduct is large and semicircular. It is located close to the posterior margin of the internal auditory meatus. It is also relatively close to the rounded aperture for the vestibular aqueduct (=endolymphatic duct) that is elliptical in shape. The fenestra rotunda is relatively large and semicircular in outline. The posterior process is relatively long posteriorly and inflated mediolaterally. It is fused to the skull ~~in by~~ the mediolaterally narrow periotic fossa of the skull. The posterior bullar surface is covered by the fused fragment of the posterior process of the tympanic bulla. It is slightly bent posterolaterally at the base just posterior to the cochlear portion. Because of this fusion of the posterior process of the tympanic bulla and its posterior expansion, the portion of the posterior bullar facet is posteriorly flared and fan-shape in ventral view (Figure 9A). Although mostly covered by the remnant of the posterior process of the tympanic bulla, the surface of the posterior bullar facet is smooth. ~~The epitympanic hiatus is relatively deep.~~

Tympanic Bulla

The left tympanic bulla is preserved (Table 2). The anterior part of the ventral wall of the tympanic bulla is cylindrical and is ~~transversely rectilinear in outline~~ ~~lacking its anterior spine~~. The involucrem is shorter ~~anteriorly~~ than the lateral wall. The dorsal margin of the involucrem is cut by an indentation. The inner and outer posterior prominences are thick and short, and they ~~swell are~~ ~~ventrally bulging~~ ~~well~~. The outer posterior prominence is much ~~bulged~~ ~~swelled~~ ~~ventrally~~ than the inner one. The median furrow runs anteroposteriorly on the ventral wall, and it is transversely narrow and anteriorly shallow. The interprominential notch is relatively deep at the posterior end of the median furrow. The lateral furrow is shallow, and its posterior ridge is prominent and twisted on the lateral wall of the tympanic bulla. The sigmoid process is square ~~in anterior view~~, and ~~it is~~ ~~anteroposteriorly~~ ~~rather~~ thin. ~~The posteroventral corner of the sigmoid process is projected posteriorly~~. The lateral border of the sigmoid process is twisted posteriorly. A damaged base of the posterior process lies on the posterolateral end of the sigmoidal process. The elliptical foramen is present (Figure 10, 11).

6. Results of the Phylogenetic Analysis

We compared the characteristics of SCM 5530-6 with both extant and fossil Zhiphiidae including 27 genera and 55 species based mainly on Bianucci et al. [3] and Lambert et al. [34]. ~~Because SCM~~

R. Soc. open sci. article template

9

5530-6 described here from the middle to upper Miocene boundary Tsurushi Formation was initially reported by Takahashi et al. [14] as a species of the Ziphiidae based on the posteriorly located bony nares with gently elevated ascending process of the premaxilla, and the posteroventrally steeply sloped occipital. In addition to the above characters, SCM 5530-6 has such diagnostic characters of the family Ziphiidae at least as the presence of the premaxillary crests on the gently elevated vertex, wide and anteriorly extended hamular fossa of the pterygoid sinus [3, 34], we analyzed its phylogenetic relationship based on the data matrix and parsimony settings prepared by Bianucci et al. [3] to locate SCM 5530-6 in the Ziphiidae with the addition of the North Pacific Baird's beaked whale, *Berardius bairdii* that was not included in their analysis and the recently described Sato's beaked whale, *Berardius minimus* of the northern North Pacific [2], as well as SCM 5530-6.

As a result of the analysis, five most parsimonious trees were obtained with a tree length of 177 steps in each, Goloboff fit (k=3) -58.44286, the consistency index (CI) of 0.475, and the retention index (RI) of 0.767. The 50% majority rule consensus tree is shown in Figure 12. The strict consensus tree is also shown in the Supplementary Data S4. Our result suggests that *Chavziphius* from the late Miocene of Peru (the South Pacific) is the basal to the clade including all the extant subfamilies, i.e., the Berardiinae, Ziphiinae, and Hyperoodontinae. Instead, *Ninoziphius* also from the Miocene of Peru is recognized as the most basal to the Ziphiidae as a whole. As for SCM 5530-6, it was nested in the monophyletic subfamily Berardiinae consisting of the genera *Berardius*, *Archaeoziphius* and *Microberardius* ([3], [36] and Figure 12).

The Berardiinae is supported by five synapomorphies: slightly concave ascending process of the premaxilla (Char.7: as reversal); a weak constriction on the ascending process of the right premaxilla (between premaxillary sac fossa and premaxillary crest) and the ratio between the minimal width of the ascending process of premaxilla and the width of the right premaxillary crest is > 0.80 (Char.8: as reversal); no anteromedial excavation of the dorsal surface of the nasal (Char.14: as reversal); an isolated rounded protuberance formed by the interparietal and/or frontals on the posterior part of the vertex (Char.17), and the anteromedial margin of the supraoccipital that of SCM 5530-6 is lower than the frontals on the vertex (Chars. 18 of [3]) (see also [3, 29, 33]). The following character is also considered to be a potentially unique character for the subfamily: the posteromedial narrowing nasals and frontals at the vertex, which is narrower than the anterior-most transverse width of nasals (Char. 50). Among those characters, quite a few synapomorphies of the subfamily are interpreted as reversals from derived to secondarily primitive character states. However, the extant ziphiids are deep-diving whales having adapted to deep sea with considerable ecomorphological specializations during their adaptation and diversification in each clade, and they might have yielded unique characters frequently to adapt such a harsh environment. Other examples are also interesting to be noted. A low elevation of the ascending process of the premaxilla is observed in *Berardius* and related fossil species such as *Archaeoziphius microglenoideus* and *Microberardius africanus* [3, 32, 36], though this condition is also seen in *Chavziphius maxillocrestatus*, one of the basal ziphiids [3]. Another example is a cluster of dorsal infraorbital foramina, which is present on the base of the rostrum in the early diverging ziphiids [3]. On the other hand, only a large foramen is present in the crown ziphiids [3]. SCM 5530-6 has a large foramen on the right maxilla, and two relatively large foramina on the left maxilla, and therefore, the condition of these foramina is thought to be the condition seen in the later diverging, extant ziphiids. However, *Ninoziphius platyrostris* from the Miocene of Peru, which is considered to be the most basal ziphiid (Figure 12), has a relatively large dorsal infraorbital foramen [34], suggesting that this character state might have been multiply derived in two distantly related clades. This case is also suggestive that some characters seen in the berardiines are not appropriate to be considered synplesiomorphies conservatively retained in them as a subfamily having earlier-branching, deeper node.

Among the species within the Berardiinae, *Microberardius africanus* was recognized as the first split with the rest of genera and species including SCM 5530-6, but no autapomorphies were recognized to characterize *Microberardius* by our analysis. In this regard, *Microberardius* seems to be "ancestral" to the later diverging berardiines. However, Bianucci et al. [3] suggests that *Microberardius* is distinguishable from other berardiines by having the following characters: the

R. Soc. open sci.

rostrum is higher than wide along most of its length, the rostral base is narrower, and the maxillary crest is not extended on the rostral base.

By contrast, the monophyly of the species belonging in the genera *Berardius* and *Archaeoziphius* including SCM 5530-6 is strongly supported by the following unique character combinations: Although the left side of the premaxilla of SCM 5530-6 is broken, the ratio between the width of the premaxillary crest and that of the premaxillary sac fossa of SCM 5530-6 is moderate, from 1.0 to 1.25 (Char. 11); an isolated rounded protuberance formed by the frontals, not the interparietal, on the posterior part of the vertex (Char. 17, 0->2).

Although, the basic framework of the relationships among the genera of the Berardiinae (i.e., [*Microberardius*, [*Archaeoziphius*, *Berardius*]]), which was originally demonstrated by Bianucci et al. [3], is also supported by our analysis, *Archaeoziphius microglenoideus* was bracketed in between the clade containing two other extant species of *Berardius* (i.e., *B. bairdii* and *B. arnuxii*) and the clade containing newly added *Berardius minimus* plus SCM 5530-6 in our analysis. However, 274 out of 51 (i.e., 5347%) characters used in Bianucci et al. [3] and in our analysis are missing and uncertain in *A. microglenoideus*, and therefore, its phylogenetic position is still unstable (see also the Supplementary Data S4). Because of overall primitiveness of observable characters in *Archaeoziphius*, the position of this genus is still controversial and is potentially anticipated to be "outside" from the clade of the genus *Berardius* as a sister taxon to the latter. In this regard, the monophyly of the genus *Berardius* including SCM 5530-6 is highly considerable, and the position of SCM 5530-6 as a sister taxon to *B. minimus* makes SCM 5530-6 a species of *Berardius* in the Berardiinae. As for the specific relationship of SCM 5530-6 to other species in the genus, SCM 5530-6 is the closest to *B. minimus* as was mentioned above in having the apices of the left and right hamular process of the pterygoids that contact medially, forming together a posteriorly directed medial point (Char. 36). In addition, the lateral expansion of the premaxillae at the posterior halves in SCM 55306 is prominent as in *B. minimus*. Furthermore, the temporal fossa of SCM 5530-6 is shallow as in *B. minimus*, and but its lateral surface wall of the fossa is convex also as in *B. minimus*. By contrast, it is definitely concave as in *B. bairdii* and *B. arnuxii*. As for skull sutures, *B. minimus* has much tighter sutures as has pointed out by Yamada et al. [2], the skull of SCM 5530-6 also has much tighter sutures compared with those in *Berardius bairdii* and *B. arnuxii*. The prominent notch and related structure are much lower, less distinct and less rugged than *B. minimus*. In fact, three out of five most parsimonious trees support that SCM 5530-6 is the sister taxon of *B. minimus* (Supplementary Data S4). Finally, SCM 5530-6 is distinguishable from *B. minimus* at the three e in having extant species based on the above mentioned cranial characters: the short nasal is short, the ratio between the length of medial suture of nasals on the vertex and the maximum width of nasals < 0.4 (Char. 13, as reversal).

Accordingly of the above morphological comparisons, SCM 5530-6 is identifiable as a species of the genus *Berardius*. Furthermore, the skull size, based on the bizygomatic width, of SCM 5530-6 is 50% smaller than *B. minimus* as was mentioned above. Consequently, identification of SCM 5530-6 as a new distinctive species within the genus *Berardius* is warranted. Thus, we propose *Berardius kobayashii*, sp. nov. based on SCM 5530-6 from the middle to late Miocene boundary age Tsurushi Formation. Although the left side of the premaxilla of SCM 5530-6 is broken, the ratio between the width of the premaxillary crest and that of the premaxillary sac fossa of SCM 5530-6 is moderate, from 1.0 to 1.25 (Char. 11 of [3]). The dorsal infraorbital foramina open anterodorsally on the maxilla near the base of the rostrum. In general, a cluster of dorsal infraorbital foramina is present on the base of the rostrum in the early diverging ziphiids [3]. On the other hand, only a large foramen is present in the crown ziphiids [3]. SCM 5530-6 has a large foramen on the right maxilla, and two relatively large foramina on the left maxilla, and therefore, the condition of these foramina is thought to be the condition seen in the later diverging, extant ziphiids.

The supraoccipital of SCM 5530-6 is lower than the frontals on the vertex, and the vertex is gently elevated (Chars. 9 and 18 of [3]). A similar low elevation of the ascending process is observed in *Berardius* and related fossil species such as *Archaeoziphius microglenoideus* and *Microberardius africanus* [3, 29, 33]. *Archaeoziphius* differs from *Berardius* by the following characters: the overall size of the skull is much smaller; the nasal is included in the premaxillary crest for a short distance along the posteromedial angle of the crest [33]; Chars. 15 and 51 of [3]. However, the inclusion of the nasal in the premaxillary crest is absent on SCM 5530-6 as in the species of *Berardius*.

R. Soc. open sci.

Microberardius differs from *Berardius* by the following characters: the size of the skull is much smaller; the premaxillary crest is much narrower; the vomer at the anterior portion of the mesorostral groove is thickened; the rostrum is higher than wider along the most of its length; the base of the rostrum is narrower and the maxillary crest is not extended at the base of the rostrum [29]. By contrast, SCM 5530-6 does not have the narrow premaxillary crest.

Consequently, SCM 5530-6 differs from both *Archaeoziphius* and *Microberardius*. In addition, the anterior process of the periotic is slender, and its apex is long and pointed, and the posterior process is long and slender. These characters are potential synapomorphies of the genus *Berardius* (see also [28, 34]). Therefore, SCM 5530-6 is definitely identified as a species of the genus *Berardius* as was already pointed out by Takahashi et al. [14].

However, SCM 5530-6 has some additional characters differing from the extant, now three species of *Berardius*, such as *Berardius bairdii*, *B. arnuxii* and *B. minimus*, which were not mentioned by Takahashi et al. [14]. For instance, the nasal of SCM 5530-6 is shorter than in *B. bairdii*, *B. arnuxii* and *B. minimus*. Also, there is a tiny anteromedial excavation on the dorsal surface of the nasal, which is not observed in the other three species of *Berardius*. The apices of the pterygoids contact medially in all the species of *Berardius*, and these are forming together a posteriorly directed medial point in SCM 5530-6, but these are less excavated with a U-shaped posterior margin in the other three species of *Berardius*. The last character is not seen in any species of both fossil nor extant ziphiids including the three extant species of *Berardius*. But, the hamular fossa of the pterygoid sinus is wide, extending anteriorly on the palatal surface of the rostrum as seen in the extant *Berardius* and deep diving whales [35, 36]. The height of the skull relative to its width is 84% in SCM 5530-6 and slightly higher than in the extant species of *Berardius*; i.e., 83% in *B. minimus* and 77% in *B. bairdii*. The skull of SCM 5530-6 has much tighter sutures compared with those of extant *Berardius bairdii* and *B. arnuxii*. As has pointed out by Yamada et al. [2], the skull of *B. minimus* also has much tighter sutures, but SCM 5530-6 has much tighter sutures than that of *B. minimus*. The greatest postorbital width of SCM 5530-6 is almost 60 to 70% smaller than in the extant *B. bairdii* and *B. arnuxii* and 50% smaller than in *B. minimus*. The tympanic bulla and the periotic of SCM 5530-6 closely resembles those of *B. bairdii*. But, the total length of the tympano-periotic bone is 40 to 50% smaller than that of *B. bairdii*, and the total length of tympanic bulla is 30% smaller than that of *B. minimus*. In addition, based on comparisons with all the known species within the genus; i.e., the extant three species that are presently distributed in the northern North Pacific (i.e., *B. bairdii* and *B. minimus*) and the Antarctic (i.e., *B. arnuxii*), SCM 5530-6 is distinguishable from all the three extant species based on the above mentioned cranial characters: i.e., it is extremely small in size; the nasal is short; and the apices of the left and right hamular process of the pterygoids contact medially, forming together a posteriorly directed medial point. Furthermore, the size of the new fossil is extreme among all the known species of the genus; i.e., 50% smaller than *B. minimus* and surprisingly 60 to 70% smaller than both *B. bairdii* and *B. arnuxii*. Consequently, identification of SCM 5530-6 as a new species within the genus *Berardius*, i.e., *Berardius kobayashii*, is warranted.

7. Discussion

In previous studies, Takahashi et al. [14] identified SCM 5530-6, now the holotype of *B. kobayashii* sp. nov., as “provisionally *Berardius* sp. indet.” among them based on the following characteristics: the external bony nares are triangular in outline; the vertex is gently elevated; and the posterior median part of the pterygoid (though it was misidentified as median part of the palate in [14, p.104]) the palatine is ventrally well developed on convex on the ventral side. However, they compared and identified SCM 5530-6 only as belonging to the Ziphiidae based on the following characters: the bony nares are located posteriorly; the rostrum is moderately wide; the occipital slopes relatively steeply posterodorsally; and the cranium is not strongly asymmetrical. Furthermore, they compared SCM 5530-6 with some extant genera of the Ziphiidae; i.e., *Ziphius*, *Berardius*, and *Mesoplodon*, and But they also pointed out that the transverse width/diameter of the premaxilla of SCM 5530-6 is relatively wider relative to the width of the maxilla at the level of the antorbital notch

than ~~that of~~ then extant two species of *Berardius*, and ~~that~~ the lateral margin of the same portion is curved laterally in contrast to the relatively straight condition of this portion in both the extant species of *Berardius*. In addition, they also pointed out that the morphology of the zygomatic process of the squamosal is shaped like an anterodorsally tapered triangle in contrast to an anteriorly sub-squared rectangle in both the extant species of *Berardius*. In this regard, both inter- and intra-generic comparisons were quite limited to their comparable taxa among the extant ziphiids, and the generality of their taxonomical differentiations of characters were somewhat arbitral. - In addition, two new genera of the Berardiinae were described as mentioned above ([32], [36]), and the third extant species of *Berardius* of the *Berardius*- was also added after Takahashi et al [14] as mentioned above ([3]). Accordingly, and, the comprehensive comparisons among both extinct and extant both extinct and extant species of *Berardius* and of the Berardiinae in the Ziphiidae were necessary to locate SCM 5530-6 more precisely in the phylogenetic framework. As a result of our cladistic analysis modified from Bianucci et al. [32], Interestingly, despite its small size in *B. kobayashii* that is smaller than the smallest extant species *B. minimus*, the cranial morphologies of *B. kobayashii* sp. nov. was recognized as the much closest to *Berardius minimus*! as discussed above. It confirms that the new species based on SCM 5530-6 as the holotype actually belongs in the genus *Berardius*, and *B. bairdii* and *B. arnuxii* as discussed in the “comparisons” section. It suggests it also indicates that although *B. minimus* is represented only by the extant individuals species in the northern North Pacific, the generic emergence could have been much earlier than the emergence and diversification of the other previously known rest of (two) three antitropical species, i.e., including the northern North Pacific middle Miocene extinct *B. kobayashii* sp. nov. and the *B. bairdii* and also the Antarctic *B. arnuxii*. Surprisingly, the cranial morphology of *B. kobayashii*, sp. nov. was the closest to that of *B. minimus* as mentioned in the “comparisons” section. It suggests that although *B. minimus* is represented only by the extant individuals from the northern North Pacific, its emergence could have been much earlier than the emergence and diversification of the two larger, antitropical or amphipolar species represented by ~~t~~ Consequently, their emergence and diversification of the genus *Berardius* must have occurred in ~~or by~~ the late middle Miocene and potentially in the western northern North Pacific based on the new fossil from the middle to late Miocene boundary age Tsurushi Formation ~~described here in the~~.

—For now, three genera are recognized in the subfamily Berardiinae; i.e., *Microberardius* from the middle late Miocene of off South Africa [3229], *Archaeoziphius* from the middle Miocene of Belgium [3633] and *Berardius* from the middle to late Miocene boundary to the Recent of the northern North Pacific and the Antarctic oceans [1, 2, this study]. Previously, there was no fossil record of the berardiines in the North Pacific Ocean. So, the new fossil described here reveals that ~~the berardiine ziphiids~~ was already diversified existed at least to in the western North Pacific at latest in the early late late middle Miocene (ca. 11.2 Ma) and. It also suggests that the emergence of the extant genus berardiine *Berardiine* ziphiids goes back to the early middle to late Miocene boundary age of the North Pacific, or much then they quickly diversified with or just after the age of earlier because their closely related sister taxon *Archaeoziphius* in the early middle Miocene of the North Atlantic was already existed in the early middle Miocene [3633].

-At the moment, it is difficult to consider the more precise origin of the subfamily Berardiinae and the genus *Berardius* based on the present fossil evidence, but its potential origin of the subfamily must might have been in the Southern Hemisphere on the basis of recent studies on their phylogenetic relationships and phylogeographic distributions of sister and out groups for the berardiine ziphiids ([3, 5, 32, 33, 36] and Figure 13). In addition, the close relationship between *B. kobayashii* and *B. minimus* indicates an emergence of the genus in the North Pacific, and then the ancestor of *B. bairdii* and *B. arnuxii* may have evolved somewhere from *B. kobayashii*-like berardiine in the Northern Hemisphere or somewhere in between the northern and southern hemispheres by the end of Miocene [3737 McGowen]. It could have been followed by a dispersal (with gigantism) from the Southern to or from the Northern hemisphere [38 Morin et al., 2017]. In any case, the two currently sympatric form, smaller *B. minimus* and larger *B. bairdii* in the northern North Pacific could have been secondary contact after the Pliocene in age [38 Morin et al., 2017]. However, all these evidence suggest the rapid diversification of taxa in the subfamily Berardiinae in the initial stage of their evolution during in the middle and late Miocene in the Northern Hemisphere (Figure 1344). It is difficult to establish the generic emergence of extant taxa based only on morphologies without

genetic information such as divergence time estimations. In the case of the genus *Berardius*, its emergence time estimation depends on the divergence time of the Berardiinae and other ziphiid subfamilies because the Berardiinae is at present monogeneric. In this regard, McGowen et al. [392009], for instance, estimated the divergence time of the Berardiinae from other subfamilies to be 21.98 Ma (95%CI: 17.11-29 Ma). They also estimated the differentiation of the two species of *Berardius* species (i.e., *B. arnuxii* and *B. bairdii*) to be 2.9 Ma (95%CI: 0.68-5.81 Ma). It means that the expected divergence time of the genus *Berardius* still spans 17.11-5.81 Ma in minimum estimation or 15-5.81 Ma based on the geologic age of the closest genus *Archaeoziphius* [36]. Although the divergence time of *B. minimus* from the ancestor of the two other species (i.e., *B. bairdii* and *B. arnuxii*) within the genus is still uncertain ([2, 38, 40], the minimum estimate of the divergence time could be dated back to at latest the early late Miocene in age based on our phylogenetic analysis that suggested the closest relationship of *B. minimus* to the middle/late Miocene boundary age *B. kobayashii*, sp. nov.

Accordingly, it is possible to interpret that the genus *Berardius* began to diversify at this period, which is known to be the period of global cooling and of coastline regression [41, 42, 43, 44]. It should be important to be noted that somewhat speciose extant genus *Mesoplodon*, which is consisted of at least 15 extant species, has already appeared and diversified in the late middle or early late Miocene ([37, 39, 45]. It strongly suggests that the much earlier and less diversified appearance of the genus *Berardius* might have proceeded their generic emergence, and *B. kobayashii*, sp. nov. could be interpreted as one of such representatives of the earlier generic emergence and diversification. It will be further considered based on molecular studies for *B. minimus* and also morphological studies for other fossil berardiines to be described in the future.

8. Conclusions

An extinct beaked whale from the middle to late Miocene boundary age Tsurushi Formation (ca. 12.3 – 11.5 Ma) in the Sado Island, Niigata Prefecture, Japan, was described as a new species of the extant genus *Berardius*; i.e., *B. kobayashii*. It is distinguishable from all other species within *Berardius* by having the following characters: extremely small size among the four species of *Berardius* (60-70 % smaller than *B. bairdii* and *B. arnuxii*, 50% smaller than *B. minimus*); the transverse diameter of the premaxilla is relatively wide to the width of the rostrum at the level of the anterior notch; the zygomatic process of the squamosal is anterodorsally tapered and triangular in shape; the nasals are short. *Berardius kobayashii*, sp. nov., is the fourth species of the genus. Namely, the lineage leading to the extant genus of the ziphiids had already lived in the western North Pacific at the time of at latest 11 Ma. The western North Pacific including the Sea of Japan may, therefore, have been one of the areas for of the evolution and radiation of *Berardius* species. The existence of a species of the extant ziphiid genus that have been known as one of the deep-diving whales from the middle Miocene deep-sea deposits will become the key to elucidate the process of their habitat expansions as deep-sea dwellers among the Ziphiidae.

Acknowledgments

We would like to thank the late I. Kobayashi (Professor Emeritus at Niigata University), A. Matsuoka (Niigata University), M. Shimizu (Asahimachi Museum, Niigata University), M. Ikarashi (Sado Museum), M. Aida (Sado Geopark Promotion Office and Sado Museum), K. Narita (Nagano City Museum), M. Kato (Nagaoka Municipal Science Museum), Y. Tajima and T.K. Yamada (NMNS), for allowing access to the specimens and giving me the opportunities to observe specimens at respective museums; Y. Yanagisawa (Geological Survey of Japan), K. Takahashi (Biwako Museum), H. Ichishima (Fukui Prefectural Dinosaur Museum), K. Nagasawa (formerly Yamagata Prefectural Museum), T. Kimura (Gunma Museum of Natural History), for providing useful discussion and advice with the literature on fossil ziphiids.

Funding Statement

A.K. was funded by the travel grant fund for the 2017 fiscal year from by the Regional Promotion Division of Sado City.

Data Accessibility

Additional data are available as the electronic supplementary materials (Supplementary files S1-S7: S1, list of the taxa and specimens of the berardiine ziphiids used for this study; S2, list of characters used for our cladistic analysis; S3,

character/taxon matrix; S4, resultant cladograms including the unweighted strict and 50% majority-rule consensus cladograms, five equally most parsimonious cladograms with implied weighting of K=3; S5, References for character collections; S6, Nexus file; S7, TNT file). The new species has been registered in Zoobank. The LSID for this publication is: urn:lsid:zoobank.org:act:317C1452-AE2E-4A08-A473-3B507B452D2C. This study and the nomenclatural acts it contains have been registered in Zoobank. The LSID for this publication is: urn:lsid:zoobank.org:pub:7BEDEBA0-46B9-4125-A7F8-2169EB016C29.

Competing Interests

We declare we have no competing interests.

Authors' Contributions

A.K. and N.K. conceived and designed the project, performed the experiment and analysed the data. A.K. prepared SCM 5530-6. N.K. contributed reagents/materials/preparation tools/analysis tools. A.K. and N.K. wrote the paper.

References

- Rosell P E, Archer FI, Baker CS, Boness DJ, Brownell RL, Churchill M Jr., Costa AP, Domning DP, Fordyce RE, Jefferson TA, Kinze C, Oliveira LR, Perrin WF, Wang JY, Yamada TK. 2019 List of marine mammal species and subspecies. *Committee on Taxonomy, The Society for Marine Mammalogy*.
- Yamada TK, Kitamura S, Abe S, Tajima Y, Matsuda A, Mead JG, Matsuishi TF. 2019 Description of a new species of beaked whale (*Berardius*) found in the North Pacific. *Scientific reports* **9**(1), 1-14.
- Bianucci G, Di Celma C, Urbina M, Lambert O. 2016 New beaked whales in stem and crown Ziphiidae (Cetacea, Odontoceti). *PeerJ*. **4**, e2479.
- Lambert O, de Muizon C, Duhamel G, Van Der Plicht J. 2018 Neogene and Quaternary fossil remains of beaked whales (Cetacea, Odontoceti, Ziphiidae) from deep-sea deposits off Crozet and Kerguelen islands, Southern Ocean. *Geodiversitas* **40**(2), 135-140.
- Lambert O, Collareta A, Landini W, Post K, Ramassamy B, Di Celma C, Urbina M, Bianucci G. 2015 No deep diving: evidence of predation on epipelagic fish for a stem beaked whale from the Late Miocene of Peru. *Proc. R. Soc. B*, 282.
- Niigata Foraminiferal Research Group. 1967 Some foraminiferal assemblages from the Sawane area, Sado Island (Preliminary report): Restudy on the stratigraphy of the "Sawane Formation" based on the foraminiferal fossils. *Commemorative Volume for Mr. Y. Hiramatsu*, 115-119.
- Watanabe K. 1987 Tertiary Foraminifera and Radiolaria Fossils from Sado Island, Central Japan. *Publications from the Sado Museum* **9**, 127-156.
- Okamoto Y, Satou M, Watanabe M, Yamamoto H. 1992 Inversion Tectonics in the southeastern part of the Japan Sea. *Jour. Struct. Geol.* **38**, 47-58.
- Kobayashi I. 2001 The Cenozoic of Niigata Prefecture, the case of Sado Island (Niigata no shinseikai Sado-hen). *Niigata Geotechnology Society* **57**, 9-21.
- Yanagisawa Y, Watanabe M. 2017 Revised lithostratigraphy of the Neogene sedimentary sequence in the southern part of the Osado Mountain area, Sado Island, Niigata Prefecture, Japan. *Bull. Geol. Surv. Japan* **68** (6), 259-285.
- Sado Research Group for Marine Mammalian Fossils. 1987 Fossil Cetacean from Sado Island, Central Japan. *Publications from the Sado Museum* **9**, 211-217.
- Horikawa H, Tazaki K, Kanno T. 1987 Fossil Ziphiidae from Koshiji-Syo off Sado Island, Central Japan. *Publications from the Sado Museum* **9**, 225-230.
- Tazaki K, Horikawa H, Miyazaki S. 1987 Fossil Ziphiidae from Hyotan-Guri off Sado Island, Central Japan. *Publications from the Sado Museum* **9**, 219-223.
- Takahashi K, Nomura M, Kobayashi I. 1989 A fossil cetacean skull (*Berardius* sp. indet.) from Dogama, Ogi-machi, Sado Island, Central Japan. *Earth Science: Chikyu Kagaku* **43**, 102-105.
- Barnes LG, Hirota K. 1995 Miocene pinnipeds of the otariid subfamily Allodesminae in the North Pacific Ocean: Systematics and relationships. *The Island Arc* **3**, 329-360.
- Utashiro T. 1950 The stratigraphy of Sawane-Aikawa region in Southwest Sado Island (Osado Mountain range). *Journal of the Geological Society of Japan* **56**, 302-303.
- Shimazu M, Kanai Y, Toyama T, Ichihashi K, Minakawa J, Takahama N. 1973 Structural development and igneous activity in the Sado island. *Memoirs of the Geological Society of Japan* **9**, 147-157.
- Shimazu M, Toyama T. 1982 Neogene volcanic rocks of the Sado islands. *Journal of the Geological Society of Japan* **88**, 381-400.
- Yanagisawa Y. 2012 Late Miocene diatoms in the Hamochi area, Sado Island, Niigata Prefecture, Japan. *Open-File Report of the Geological Survey of Japan, AIST* **568**, 1-19.
- Ogi Collaborative Research Group. 1986 Late Cenozoic Group in the Southern part of Kosado Mountains, Niigata prefecture. *Earth Science* **40**, 417-436.
- Takeuchi K, Ozaki M, Komatsubara T. 2011 Explanatory notes of 1:200,000 geological map of the coastal zone around Niigata. Digital Geoscience Map, Seamless Geoinformation of Coastal Zone "Coastal Zone around Niigata".
- Tokunaga S. 1939 A New Fossil Mammal Belonging to Desmostylidae. *Jubilee Publication in the Commemoration of Professor H. Yabe, M.I.A. Sixtieth Birthday*. Sendai: Institute of Geology and Paleontology, Tohoku Imperial University. 289-299.
- Watanabe H. 1932 Neogene (Shin Daisankei). *Nihon Chishitsu kousan shi* 92-152.
- Maiya S. 1978 Late Cenozoic planktonic foraminiferal biostratigraphy of the oil-field region of Northeast Japan. *Cenozoic Geology of Japan*, 35-60.
- Kawatani A, Sashida K, Sachiko A, Kohno N. 2019 Radiolarian fossils and estimated depositional age of the Miocene Tsurushi Formation distributed in Sado Island, Niigata Prefecture, Japan. *Bull. Geol. Surv. Japan* **70**(1-2), 91-99. (doi: <https://doi.org/10.9795/bullgsj.70.91>)
- Ono K, Ueno T. 1985 Tertiary Vertebrates from Sado Island, Niigata Prefecture, Central Japan. *Memoirs of the National Science Museum* **18**, 65-71.
- Mead JG, Fordyce RE. 2009 The therian skull: a lexicon with emphasis on the odontocetes. *Smithsonian Contributions to Zoology*, 1-249.
- Kasuya T. 1973 Systematic consideration of recent toothed whales based on the morphology of tympanoperiotic bone. *Scientific Reports of the Whales Research Institute* **25**, 1-103.
- Maddison WP and Maddison DR. 2018. [Mesquite: a modular system for evolutionary analysis. Version 3.61. http://www.mesquiteproject.org.](https://www.mesquiteproject.org)
- Goloboff PA, Catalano SA. 2016. [TNT version 1.5, including a full implementation of phylogenetic morphometrics. Cladistics](https://doi.org/10.1111/cup.12861) **32**(3), 221-238.

R. Soc. open sci.

R. Soc. open sci. article template

15

- 29-31. Goloboff PA. 1993. Estimating character weights during tree search. *Cladistics* **9**, 83-91.
- 30-32. Bianucci G, Lambert O, Post K. 2007 A high diversity in fossil beaked whales (Mammalia, Odontoceti, Ziphiidae) recovered by trawling from the sea floor off South Africa. *Geodiversitas* **29**(4), 561-618.
- 31-33. Bianucci G, Lambert O, Post K. 2010 High concentration of long - snouted beaked whales (genus *Messapicetus*) from the Miocene of Peru. *Palaeontology* **53**(5), 1077-1098.
- 32-34. Lambert O, De Muizon C, Bianucci G. 2013 The most basal beaked whale *Ninziphius platyrostris* Muizon, 1983: clues on the evolutionary history of the family Ziphiidae (Cetacea: Odontoceti). *Zoological Journal of the Linnean Society* **167**(4), 569-598.
- 33-35. Miwa M, Yanagisawa Y, Yamada K, Irizuki T, Shoji M, Tanaka Y. 2004 Planktonic foraminiferal biostratigraphy of the Pliocene Kuwae Formation in the Tainai River section, Niigata Prefecture and the age of the base of the No. 3 Globorotalia inflata bed. *Journal of the Japanese Association for Petroleum Technology* **69**(3), 272-283.
- 34-36. Lambert O, Louwye S. 2006 *Archaeoziphius microglenoideus*, a new primitive beaked whale (Mammalia, Cetacea, Odontoceti) from the Middle Miocene of Belgium. *Journal of Vertebrate Paleontology* **26**(1), 182-191.
35. Bianucci G. 1997 The Odontoceti (Mammalia-Cetacea) from Italian Pliocene. The Ziphiidae. *Palaeontographia Italica* **84**, 163-192.
36. Reidenberg JS, Laitman JT. 2008 Sisters of the sinuses: cetacean air sacs. *The Anatomical Record: Advances in Integrative Anatomy and Evolutionary Biology* **291**(11), 1389-1396.
37. Ramassamy B, Lambert O, Collareta A, Urbina M, Bianucci G. 2018 Description of the skeleton of the fossil beaked whale *Messapicetus gregarius*: searching potential proxies for deep-diving abilities. *Mitteilungen aus dem Museum für Naturkunde in Berlin - Fossil Record* **21**(4), 11-32.
37. McGowen MR, Tsagkogeorga G, Álvarez-Carretero S, Dos Reis M, Struëbig M, Deaville R, Jepson PD, Jarman S, Polanowski A, Morin PA, Rossiter SJ. 2020 Phylogenomic Resolution of the Cetacean Tree of Life Using Target Sequence Capture. *Systematic Biology* **69**, 479-501. DOI: 10.1093/sysbio/svz068.
38. Morin PA, Baker CS, Brewer RS, Burdin AM, Dalebout ML, Fedutin ID, Filatova OA, Jung JL, Lauf M, Potter CW, Richard G, Ridgway M, Robertson KM, Wade P. 2017 Genetic structure of the beaked whale genus *Berardius* in the North Pacific, with genetic evidence for a new species. *Marine Mammal Science* **33**, 96-111.
39. McGowen MR, Spaulding M, Gatesy J. 2009 Divergence date estimation and a comprehensive molecular tree of extant cetaceans. *Molecular Phylogenetics and Evolution* **53**, 891-906. DOI: 10.1016/j.ympev.2009.08.018.
40. Kitamura S, Matsuishi T, Yamada TK, Tajima Y, Ishikawa H, Tanabe S, Nakagawa H, Uni Y, Abe S. 2013 Two genetically distinct stocks in Baird's beaked whale (Cetacea: Ziphiidae). *Marine Mammal Science* **29**, 755-766.
41. Miller KG, Kominz MA, Browning JV, Wright JD, Mountain GS, Katz ME, Sugarman PJ, Cramer BS, Christie-Blick N, Pekar SF. 2005 The Phanerozoic record of global sea-level change. *Science* **310**, 1293-1298. <https://doi.org/10.1126/science.1116412> PMID: 16311326.
42. Zachos JC, Dickens GR, Zeebe RE. 2008 An early Cenozoic perspective on greenhouse warming and carbon-cycle dynamics. *Nature* **451**, 279-283. <https://doi.org/10.1038/nature06588> PMID: 18202643.
43. Hilgen FJ, Abels HA, Iaccarino S, Krijgsman W, Raffi I, Sprovieri R, Turco E, Zachariasse WJ. 2009 The global stratotype section and point (GSSP) of the Serravallian Stage (Middle Miocene). *Episodes* **32**, 152-166.
44. Betzler C, Eberli GP, Lüdmann T, Reolid J, Kroon D, Reijmer JGG, Swart PK, Wright J, Young JR, Alvarez-Zarikian C, Alonso-García M, Bialik OM, Blättler CL, Guo JA, Haffen S, Horozal S, Inoue M, Jovane L, Lanci L, Laya JC, Hui Mee AL, Nakakuni M, Nath BN, Niino K, Petruny LM, Pratiwi SD, Slagle AL, Sloss CR, Su X, Yao Z. 2018 Refinement of Miocene sea level and monsoon events from the sedimentary archive of the Maldives (Indian Ocean). *Progress in Earth and Planetary Science* **5**, 1-18. DOI 10.1186/s40645-018-0165-x.
45. Steeman ME, Hebsgaard MB, Fordyce RE, Ho SYW, Rabosky DL, Nielsen R, Rahbek C, Glenner H, Rensen MVS, Willerslev E. 2009 Radiation of extant cetaceans driven by restructuring of the oceans. *Systematic Biology* **58**, 573-585. DOI:10.1093/sysbio/syp060.
- 38-46. Akiba F. 1986 Middle Miocene to Quaternary diatom biostratigraphy in the Nankai Trough and Japan Trench, and modified Lower Miocene through Quaternary diatom zones for middle-to-high latitudes of the North Pacific. In Kagami, H., Karig, D. E., Coullbourn, W. T., et al., *Initial Reports of the Deep Sea Drilling Project*, U. S. Government Printing Office, Washington D. C. **87**, 393-480.
- 39-47. Yanagisawa Y, Akiba F. 1998 Refined Neogene diatom biostratigraphy for the northwest Pacific around Japan, with an introduction of code numbers for selected diatom biohorizons. *Jour. Geol. Soc. Japan* **104**, 395-414.

Tables

Measurements (in mm) of the skull of *Berardius kobayashii*

Skull	
Preserved Condyllobasal length from tip of broken rostrum to hindmost margin of occipital condyles.	236
Greatest postorbital width.	*e224519
Greatest zygomatic width	*e245
Greatest width of bony nares.	38
Maximum width of premaxillary sac fossae.	87
Maximum width of right premaxillary sac fossa.	43
Maximum width of left premaxillary sac fossa.	38
Maximum width of nasals.	40

R. Soc. open sci.

Greatest length of left post temporal fossa, measured to external margin of raised suture.	87
Greatest width of left post temporal fossa at right angles to greatest length.	61
Major diameter of left temporal fossa proper.	38
Minor diameter of left temporal fossa proper.	33
Distance from foremost end of junction between nasals to hindmost point of margin of supraoccipital crest.	33
Length of left orbit from apex of preorbital process of frontal to apex of postorbital process.	51
Greatest width of internal fl nares.	91
Greatest length of left pterygoid.	109

Measurements (in mm) of the ear bones of *Berardius kobayashii*

Tympanic bulla

Standard length of tympanic bulla, distance from anterior tip to posterior end of outer posterior prominence.	37
Distance from anterior tip to posterior end of inner posterior prominence.	34
Distance postero-ventral tip of outer posterior prominence to tip of sigmoid process.	24
Distance postero-ventral tip of outer posterior prominence to tip of conical process.	17.5
Width of tympanic bulla at the level of sigmoid process.	20
Height of tympanic bulla, from tip of sigmoid process to ventral keel.	23
Width across inner and outer posterior prominence.	18
Greatest depth of interprominential notch.	5
Width of upper border of sigmoid process.	11

Periotic

Standard (maximum anteroposterior) length, from tip of anterior process to posterior end of posterior process, measured on a straight line parallel to cerebral border	39.7
Standard (maximum mediolateral) width, from medial edge of cochlear portion to apex of lateral tuberosity	23
Maximum dorsoventral depth of anterior process perpendicular to axis of periotic	13.6
Maximum mediolateral width of anterior process at base	17
Length of anterior process from anterior apex to posterior border of malleolar fossa	16.2
Length of anterior process from anterior apex to anterior incisure	14.3
Anteroposterior length of cochlear portion	17.3
Dorsoventral depth of cochlear portion	18.7
Mediolateral width of cochlear portion from medial edge to fenestra ovalis	10.8
Maximum diameter of fenestra rotunda	2.7
Maximum diameter of internal acoustic meatus	4.7
Maximum diameter of cochlear aqueduct	1.9
Maximum diameter of aperture for the vestibular aqueduct	3.6
Anteroposterior diameter of proximal opening of facial canal	3.4
Anteroposterior diameter of distal opening of facial canal	1.9
Length of posterior process (posterior bullar facet between anteroposterior tips)	17.3

R. Soc. open sci.

R. Soc. open sci. article template

17

Width of posterior process (posterior bullar facet between mediolateral tips)	12.1
---	------

Figure and table captions

Figure 1. Index maps of the Pacific Northwest: A, locations of the Japanese Islands, showing Sado Island; B, map of Sado Island, showing some major place names mentioned in text, and location of enlarged area including the type locality for *Berardius kobayashii*, sp. nov.; C, detail of Sobama beach in Donokama, Ogi Town, showing the exact locality for the holotype of *Berardius kobayashii*, SCM 5530-6. Based on the GIS map (<https://maps.gsi.go.jp/#15/37.844139/138.272681/&base=std&ls=std&disp=1&vs=c0j0h0k0l0u0t0z0r0s0m0f1>) of the Geospatial Information Authority of Japan.

Figure 2. Correlation of formations distributed on the Osado (a) and Kosado (b) mountain ranges containing whale fossil horizon with relevant systems of chronology. Wave lines indicate unconformities. After [6, 19, 10]. Diatom zones: [10, 46374037, 47384138]; Planktonic foraminiferal zones: [24, 3532].

Figure 3. Dorsal view of the cranium of *Berardius kobayashii*, sp. nov., holotype, SCM 5530-6. (Scale bar = 10 cm)

Figure 4. Lateral view of the cranium of *Berardius kobayashii*, sp. nov., holotype, SCM 5530-6. A and B, left side; C and D, right side. (Scale bar = 10 cm)

Figure 5. Anterior view of the cranium of *Berardius kobayashii*, sp. nov., holotype, SCM 5530-6. (Scale bar = 10 cm)

Figure 6. Posterior view of the cranium of *Berardius kobayashii*, sp. nov., holotype, SCM 5530-6. (Scale bar = 10 cm)

Figure 76. Ventral view of the cranium of *Berardius kobayashii*, sp. nov., holotype, SCM 5530-6. (Scale bar = 10 cm)

Figure 87. Posterior fragment of the left mandible of *Berardius kobayashii*, sp. nov., holotype, SCM 5530-6. A, lateral view; B, medial view. (Scale bar = 10 cm)

Figure 98. Left periotic of *Berardius kobayashii*, sp. nov., holotype, SCM 5530-6. A, ventral view; B, corresponding line drawing with anatomical interpretations (gray hatches indicate parts of the tympanic bulla); C-F, the reconstructed CT image from 3D surface rendering, with anatomical interpretations (C, ventral view; D, lateral view; E, medial view; F, dorsal view); C, dorsal view of the reconstructed CT image from 3D surface rendering, with anatomical interpretations; D, comparisons of the lateral view of anterior process of periotic among the genera of Ziphiidae (modified from Bianucci, 1997). (Scale bar = 1 cm)

Figure 109. The left tympanic bulla A, dorsal view; B, anterior view; C, medial view; D, lateral view; E, posterior view; F, ventral view (Scale bar = 1 cm)

Figure 101. Key features of the left tympanic bulla of *Berardius kobayashii*, sp. nov., holotype, SCM 5530-6. A, dorsal view; B, anterior view; C, medial view; D, lateral view; E, posterior view; F, ventral view. (Scale bar = 1 cm)

Figure 121. Phylogenetic relationship and paleobiogeographic distribution of *Berardius kobayashii*, sp. nov., within the Ziphiidae based on Bianucci et al. [3] and our re-analysis. The strict consensus tree resulting from five most parsimonious trees, 17795 steps long, with the consistency index = 0.475523 and the retention index = 0.767.

R. Soc. open sci.

Figure 132. Distribution and potential roots for the dispersal of the known extinct species of the Berardiinae in
the middle Miocene inferred from the phylogenetic analysis.

Table 1. Measurements (in mm) of the skull of *Berardius kobayashii*, sp. nov., holotype, SCM 5530-6. ******
indicates estimated transverse measurements that are half-skull measurements multiplied by two, and 'e'
indicates estimation.

Table 2. Measurements (in mm) of the ear bones of *Berardius kobayashii*, sp. nov., holotype, SCM 5530-6.

*R. Soc. open sci.*

Figure 1. Index maps of the Pacific Northwest: A, locations of the Japanese Islands, showing Sado Island; B, map of Sado Island, showing some major place names mentioned in text, and location of enlarged area including the type locality for *Berardius kobayashii*, sp. nov.; C, detail of Sobama beach in Donokama, Ogi Town, showing the exact locality for the holotype of *Berardius kobayashii*, SCM 5530-6. Based on the GIS map (<https://maps.gsi.go.jp/#15/37.844139/138.272681/&base=std&ls=std&disp=1&vs=c0j0h0k0l0u0t0z0r0s0m0f1>) of the Geospatial Information Authority of Japan.

318x487mm (96 x 96 DPI)

Series	Age (Ma)	Stages	Diatom Zones	Planktonic foraminiferal zones	Osado (a) (Yanagisawa & Watanabe, 2017)	Kosado (b) (Niigata Foraminiferal Research Group, 1967; Yanagisawa, 2012; Kawatani et al., 2019)
Upper Miocene	6	Messinian	7B	PF5	Nozaka	Yamadagawa
		Tortonian	7A			
	6B					
	6A					
	5D					
	10		5C	PF3	Hanyugawa	Orito
	Middle Miocene		12			
			14	Serravallian	5A	
				Nanatani	4Bb	PF2
		4Ba				
Lower Miocene	16	Langhian	4A	PF1	Orito	Orito
			3B			
	18	Burdigalian	2B	PF1	Kinpokusan	Kyozukayama
			2A			
20	Aquitanian	Mikawa	1	PF1	Masaragawa	Sanze
22					Aikawa	Aikawa

Figure 2. Correlation of formations distributed on the Osado (a) and Kosado (b) mountain ranges containing whale fossil horizon with relevant systems of chronology. Wave lines indicate unconformities. After [6, 19, 10]. Diatom zones: [10, 46, 47]; Planktonic foraminiferal zones: [24, 35].

330x310mm (96 x 96 DPI)

Figure 3. Dorsal view of the ~~cranium~~ of *Berardius kobayashii*, sp. nov., holotype, SCM 5530-6. (Scale bar = 10 cm)

512x244mm (96 x 96 DPI)

Figure 4. Lateral view of the cranium of *Berardius kobayashii*, sp. nov., holotype, SCM 5530-6. A and B, left side; C and D, right side. (Scale bar = 10 cm)

789x720mm (96 x 96 DPI)

Figure 5. Anterior view of the cranium of *Berardius kobayashii*, sp. nov., holotype, SCM 5530-6. (Scale bar = 10 cm)

578x245mm (96 x 96 DPI)

Figure 6. Posterior view of the cranium of *Berardius kobayashii*, sp. nov., holotype, SCM 5530-6. (Scale bar = 10 cm)

463x247mm (96 x 96 DPI)

Figure 7. Ventral view of the cranium of *Berardius kobayashii*, sp. nov., holotype, SCM 5530-6. (Scale bar = 10 cm)

796x411mm (96 x 96 DPI)

Figure 8. Posterior fragment of the left mandible of *Berardius kobayashii*, sp. nov., holotype, SCM 5530-6. A, lateral view; B, medial view. (Scale bar = 10 cm)

245x197mm (96 x 96 DPI)

Figure 9. Left periotic of *Berardius kobayashii*, sp. nov., holotype, SCM 5530-6. A, ventral view; B, corresponding line drawing with anatomical interpretations (gray hatches indicate parts of the tympanic bulla); C-F, the reconstructed CT image from 3D surface rendering, with anatomical interpretations (C, ventral view; D, lateral view; E, medial view; F, dorsal view). (Scale bar = 1 cm)

924x469mm (96 x 96 DPI)

Figure 10. The left tympanic bulla A, dorsal view; B, anterior view; C, medial view; D, lateral view; E, posterior view; F, ventral view (Scale bar = 1 cm)

406x401mm (96 x 96 DPI)

Figure 11. Key features of the left tympanic bulla of *Berardius kobayashii*, sp. nov., holotype, SCM 5530-6. A, dorsal view; B, anterior view; C, medial view; D, lateral view; E, posterior view; F, ventral view. (Scale bar = 1 cm)

595x458mm (96 x 96 DPI)

Figure 12. Phylogenetic relationship and paleobiogeographic distribution of *Berardius kobayashii*, sp. nov, within the Ziphiidae based on Bianucci et al. [3] and our re-analysis. The strict consensus tree resulting from five most parsimonious trees, 177 steps long, with the consistency index = 0.475 and the retention index = 0.767.

397x252mm (96 x 96 DPI)

Figure 13. Distribution and potential roots for the dispersal of the known extinct species of the Berardiinae in the middle Miocene inferred from the phylogenetic analysis.

228x110mm (96 x 96 DPI)

Appendix F

February 2, 2021

Dear Editors,

We deeply thank both reviewers for their generous comments and suggestions for our revised manuscript on the oldest *Berardius* from Japan, and we revised again our manuscript to address their concerns. The reviewers' comments and suggestions greatly improved our manuscript. We thoroughly revised the manuscript and improved it with additional data following recommendations and suggestions from both reviewers. Changes are recorded with the track and change function in the revised manuscript. **We uploaded the clean version of the revised manuscript first, and then the tracked change copy.** We followed almost all the reviewers' comments. Reviewer's comments are written in Black below. **Our responses are written in Blue.** We hope that our revision is acceptable for publication.

Reviewer: 1

Comments to the Author(s)

I would like to congratulate the authors for the many improvements they made to their interesting work. The addition of a phylogenetic analysis is certainly an asset, further supporting the referral of the new species to the subfamily Berardiinae. At this point, I still noted a number of minor and moderate issues that should, in my opinion, be addressed before publication. I noted most of my minor questions directly in the pdf (text with highlighted changes, figures and figure captions). My main concerns are listed below:

—>**We corrected all the gramatical and typographic errors following suggestions on the mrked pdf. We appreciate to the reviewer's careful review and continuous advice.**

- Diagnosis of the genus *Berardius*: As it is, there is no difference with *Archaeoziphius* in this diagnosis, meaning that one won't be able to distinguish members of these two genera. To better support the attribution to the genus *Berardius*, characters shared with extant *Berardius* species and not with *Archaeoziphius* should be provided.

Considering the greater age proximity with *Archaeoziphius* and the relatively poorly resolved phylogenetic relationships among berardiines (see below), the reader may find it more appropriate to exclude the new species from the genus *Berardius*. Placing a new species in a pre-existing genus or in a new genus is at least partly arbitrary, and although I would personally not refer the new species to the genus *Berardius* (due to the relatively limited support) I understand that the authors may prefer this solution. If they keep that hypothesis, I strongly recommend to somewhat rework the diagnosis, in a way to make the case more convincing.

—>We appreciate to the reviewer's advice. Because we did not include characters that were not used for the phylogenetic analysis (i.e., characters not used for the phylogenetic analysis by Bianucci et al., 2016) into our diagnosis of genus, the generic distinction was somewhat ambiguous in our discussion. To resolve this problem, we followed the reviewer's advice and added the characters that are distinguishable the genus *Berardius* from *Archaeoziphius* (i.e., anterodorsally well-developed robust zygomatic process of squamosal, dorsoventrally long and laterally wide glenoid fossa, anteroventrally projected large postglenoid process, and anteroposteriorly well developed post-tympanic process) in the 'Diagnosis of Genus' of the Systematics and Discussion sections. These characters are shared derived characters with three extant species of *Berardius* and *B. kobayashii* sp. nov. (although not only for the species of this genus, but distinguishable from *Archaeoziphius* at least in the Berardiinae), and therefore, we have retained our opinion that the new species belongs to the genus *Berardius*.

- Another way to make your hypothesis better supported would be to prepare a separate section, before or after the phylogeny, where a detailed morphological comparison is made between the new species and all the other berardiines. Providing additional similarities and differences in a clearly organized section will facilitate the reading, and I suspect that other differences, for example with *Archaeoziphius*, could be found (for example at the level of the glenoid fossa).

—>We appreciate to the reviewer's advice. As mentioned above, we added some characters observable on the holotype such as the portion of the zygomatic process of the squamosal, and we reprepared the Discussion section to include similarities and differences among the species of the berardiines.

- The tree that is illustrated in figure 12 is not the consensus tree, contrary to what is mentioned in the figure caption. I guess that this is the 50% consensus tree. Though you may have a favoured topology, represented by this tree, I would recommend figuring the strict consensus too. This will better inform on the support of the proposed relationships.

—>We corrected the caption for the figure 12 and clearly mentioned that the strict consensus tree was also shown in the Supplementary Data S4.

- As a consequence of the relatively low support, I would suggest somewhat toning down the conclusions regarding the placement of the new species in the genus *Berardius*. Maybe something like: 'we tentatively refer the new species to the genus *Berardius*, a hypothesis that should be further tested in the future with the discovery of more complete specimens'. This is just a suggestion.

—>Once again, we appreciate to the reviewer's substantial advice. We added the characters that are distinguishable the genus *Berardius* from *Archaeoziphius* more clearly, and thus, we have retained our opinion that the new species belongs in the extant genus *Berardius*. However, we have slightly toned down the conclusions following the above advice and added the sentence 'These hypotheses could be further tested in the future with the discovery of more complete specimens' in our Discussion section.

- I found some last problematic interpretations of morphological features in the description (for example labels for the posterolateral sinus fossa in several figures), but I think those will be easily solved using recent literature on the group.

—>We corrected them: posterolateral sinus fossa was replaced with external auditory meatus, and some other labels including the glenoid fossa, postglenoid and post-tympanic processes of squamosal were also added in Figures 4, 5 and 7.

- In the comparison, you note an interesting similarity of the new species with *Berardius minimus*, at the level of the hamular processes. Looking at the illustrations of the holotype of the latter species in Yamada et al. (2019), I could not find that feature. If you observed other specimens, then this could be mentioned. Another possibility is that I did not

understand well the morphological feature, and then this part could be slightly revised.

—>Once again, we appreciate to the reviewer’s substantial advice. The condition of the hamular processes of the holotype of *Berardius minimus* is only slightly but distinctly projected posteriorly, but other individuals including NSMT-M 35206 and 42000 have relatively strong posterior projection of this portion as shown in the following photographs:

In this regard, the degree of the posterior projection of the hamular processes is individually or ontogenetically variable, but it is distinctly present and projected posteriorly. To avoid ambiguity, we added some comments on this character state in the ‘Results of phylogenetic analysis’ section.

- In some places the use of geographic concepts is not made consistently (see comments in the text and figures about the northern North Pacific, the Southern Ocean...). The geographic distribution of the extant species of *Berardius* could also be somewhat more precisely described.

—>We corrected the geographic categories for the oceans in the Southern Hemisphere following the reviewer’s advice, and we discarded the Figure 13 to avoid such inconsistency.

O Lambert

Reviewer: 2

Comments to the Author(s)

Dear Ayako Kawatani and Naoki Kohno,

I found really nice the improvements that you performed into this research piece. Because

the specimen will not only be of interest to palaeontologists, but also to modern marine mammalogists. I think that some minor improvements could help doing so. Nevertheless, these improvements are just small corrections that deal mostly with structure or phrasing some not so clear statements in the manuscript. Please refer to the attached PDF to see in detail what I do refer to: as there are sometimes really long sentences of four or even five lines which are difficult to follow, or other time some sentences really short.

Hope you can cope with these small corrections, which should not be a problem, and I'm really hoping to see this paper published soon.

—>We corrected all the grammatical and typographic errors following suggestions on the marked pdf. We also added a sentence referring to the environment in which this new species live, including coexisting ziphiids with references in the 'Discussion' section. We strongly appreciate to the reviewer's substantial advice.

All the best,

Aldo Benites-Palomino
Palaeontological Museum & Institute
University of Zurich

Sincerely,
Ayako Kawatani
Department of Life and Environmental Sciences, University of Tsukuba
kawatani@geol.tsukuba.ac.jp

Naoki Kohno
Department of Geology and Paleontology, National Museum of Nature and Science and
Department of Life and Environmental Sciences, University of Tsukuba
Kohno@kahaku.go.jp